# FoMo-0D: A Foundation Model for Zero-shot Tabular Outlier Detection

**Yuchen Shen**                                     *yuchens3@andrew.cmu.edu*
*Carnegie Mellon University*

**Haomin Wen**                                      *haominwe@andrew.cmu.edu*
*Carnegie Mellon University*

**Leman Akoglu**                                    *lakoglu@andrew.cmu.edu*
*Carnegie Mellon University*

**Reviewed on OpenReview:** *https://openreview.net/forum?id=XCQzwpR9jE*

## Abstract

Outlier detection (OD) has a vast literature as it finds numerous real-world applications. Being an unsupervised task, model selection is a key bottleneck for OD without label supervision. Despite a long list of available OD algorithms with tunable hyperparameters, the lack of systematic approaches for unsupervised algorithm and hyperparameter selection limits their effective use in practice. In this paper, we present FoMo-0D, a pre-trained Foundation Model for zero/0-shot OD on tabular data, which bypasses the hurdle of model selection altogether. Capitalizing on synthetic pre-training, FoMo-0D can directly predict the (outlier/inlier) label of test samples without parameter fine-tuning. Even more importantly, it *requires no labeled data, and no additional training or hyperparameter tuning when given a new task.* Extensive experiments on **57** real-world datasets against **26** baselines show that FoMo-0D is highly competitive; outperforming the majority of the baselines with no statistically significant difference from the 2*nd* best method. Further, FoMo-0D is efficient in inference time requiring only **7.7 ms** per sample on average, with at least 7**x** speed-up compared to previous methods. To facilitate future research, our implementations for data synthesis and pre-training as well as model checkpoints are openly available at `https://github.com/A-Chicharito-S/FoMo-0D`.

## 1 Introduction

Outlier detection (OD) in tabular data finds numerous applications in critical domains such as security, environmental monitoring, finance, and medicine, to name a few. This popularity brings along a large literature with plethora of detection algorithms to choose from given a new OD task. These algorithms, however, exhibit several hyperparameters (HPs) that need careful tuning (Ma et al., 2023). Since most OD tasks are unsupervised[1], what makes effective detection notoriously difficult is unsupervised model selection (both algorithm and HP selection) in the absence of labels.

While deep learning has revolutionized many areas of machine learning (ML), it is not quite the case for OD. This is mainly because compared to classical methods, deep OD models (Pang et al., 2021) have many more HPs to which the detection performance is sensitive (Ding et al., 2022), making model selection even more challenging. While the recent success of large foundation models, pre-trained on massive amounts of data, offers new opportunities for zero-shot OD, thus far the most notable progress has been in NLP and computer vision (Brown et al., 2020; Touvron et al., 2023; Radford et al., 2021). This is thanks to the admirable

---

[1]While supervised OD exists, unsupervised setting is preferred in most domains to detect novel, emergent anomalies.

Table 1: $p$-values of the one-sided Wilcoxon signed rank test, comparing FoMo-0D (with $D = 100$) to **top 10 baselines** with default hyperparameters (HPs), and **top $4^{\text{avg}}$** baselines[6] with **avg.** performance over varying HPs (denoted w/ $^{\text{avg}}$) over All (57) datasets, those (42) w/ $d \leq 100$ and (46) w/ $d \leq 500$ dimensions, and those (47) excluding NLP, CV datasets. FoMo-0D shows **no statistically significant difference from the $2nd$ best model** ($k$NN, w/ $p = 0.106$) over All datasets, and none of the differences are significant when embedded, i.e., NLP and CV datasets are excluded, while it is comparable to ($p > \alpha$) or significantly better than ($p > 1 - \alpha$) all 26 original + $4^{\text{avg}}$ baselines over datasets w/ $d \leq 100$ (aligned w/ pretraining where $D = 100$) as well as on datasets w/ $d \leq 500$ (generalizing beyond pretraining). We use underline and **bold** to indicate $p < \alpha$ and $p > 1 - \alpha$. Rank, avg.'ed over all 57 datasets by AUROC. (setting: $D = 100$, $R = 500$, train/inference context size=5K, w/ quantile transform, $\alpha = 0.05$) (See Tables 17.1&17.2 for full results.)

| | FoMo-0D | DTE-NP | $k$NN | ICL | DTE-C | LOF | CBLOF | Feat.Bag. | SLAD | DDPM | OCSVM | DTE-NP$^{\text{avg}}$ | $k$NN$^{\text{avg}}$ | ICL$^{\text{avg}}$ | DTE-C$^{\text{avg}}$ |
|---|---|---|---|---|---|---|---|---|---|---|---|---|---|---|---|
| $d \leq 100$ | - | 0.415 | 0.700 | 0.949 | **0.953** | **0.970** | **0.971** | **0.996** | 0.876 | **0.980** | **0.978** | 0.752 | 0.860 | **0.958** | **1.000** |
| $d \leq 500$ | - | 0.220 | 0.569 | 0.827 | 0.894 | **0.960** | **0.968** | **0.994** | 0.910 | **0.960** | **0.979** | 0.607 | 0.756 | 0.846 | **1.000** |
| All | - | 0.016 | 0.106 | 0.462 | 0.454 | 0.585 | 0.750 | 0.823 | 0.759 | 0.901 | 0.895 | 0.112 | 0.315 | 0.670 | **1.000** |
| All\\{NLP, CV} | - | 0.164 | 0.476 | 0.757 | 0.832 | 0.934 | 0.945 | 0.988 | 0.867 | 0.938 | **0.965** | 0.515 | 0.683 | 0.777 | **1.000** |
| Rank(avg) | 11.886 | 7.553 | 9.018 | 10.851 | 11.36 | 12.316 | 13.342 | 13.386 | 12.982 | 14.061 | 13.851 | 9.079 | 11.105 | 12.991 | 22.263 |

quantity and quality of public text and image datasets. In comparison, public tabular OD benchmarks remain minuscule (Han et al., 2022; Zhao et al., 2021; Steinbuss & Böhm, 2021).

Recently, Prior-data Fitted Networks (PFNs) has marked a milestone in ML as a new approach to learning on tabular data (Müller et al., 2022). The core idea is to compute a posterior predictive distribution (PPD) for a test point given training data as context. To approximate the PPD, a Transformer (Vaswani et al., 2017) is pre-trained on a large set of synthetic datasets drawn from pre-defined data priors. At inference, the pre-trained PFN is fed with test samples along with some training samples as context for zero-shot prediction, requiring no parameter fine-tuning or model selection on new datasets. Variants of PFN are shown to match tree-based models in performance on small classification datasets (Hollmann et al., 2023) as well as time series forecasting (Dooley et al., 2023).

In this paper, we capitalize on these ideas and introduce FoMo-0D: a prior-data fitted Foundation Model for zero/0-shot OD. Once pre-trained on synthetic datasets, FoMo-0D unlocks zero-shot OD on a new dataset where the (unlabeled) input data is fed only as context. As such, FoMo-0D bypasses not only model (parameter) training, but more importantly, the nontrivial task of unsupervised model (algorithm and HP) selection without labeled data. Figure 1 illustrates the new FoMo-0D paradigm versus the typical OD setting. To our knowledge, FoMo-0D is the first pre-trained foundation model for tabular OD.

In designing FoMo-0D, we use Gaussian mixture models as a simple yet effective tabular data prior for inlier data distributions (Hollmann et al., 2023; Zhao et al., 2021), which are also employed to simulate outlier types common in the real world; namely, local and global subspace outliers (Steinbuss & Böhm, 2021). While the data prior can be extended to comprise more complex data distributions (Hollmann et al., 2023) (e.g. Bayesian Neural Networks (Neal, 2012) and Structural Causal Models (Pearl, 2009)), and additional outlier types can be included (e.g. dependency, contextual, etc.), as we show in the experiments, even with its relatively straightforward prior, FoMo-0D achieves remarkable performance: As shown in Table 1 FoMo-0D, which is pre-trained on datasets with $d \leq 100$ dimensions, shows no statistically significant difference from all 26 state-of-the-art baselines (all $p$-values $> 0.4$) on 42 benchmark datasets with dimensionality $d \leq 100$ (aligned with pre-training), while our method consistently ranks among the top and outperforms a majority of the baselines with $p$-value $> 0.95$. (See Appendix Tables 17.1&17.2 for full results.) The results remain consistent on 46 benchmarks with $d \leq 500$ dimensions. FoMo-0D is also competitive across all 57 datasets, effectively generalizing beyond its pre-training distributions, with no statistically significant difference from the $2nd$ best baseline. When intrinsically non-tabular datasets are removed (i.e., datasets from NLP, CV domains embedded with pre-trained encoders), there is no statistical evidence to suggest a performance difference between FoMo-0D and any baseline on the remaining 47 datasets. Further, FoMo-0D takes a mere 7.7 ms to infer a test sample on average with no extra training or tuning overhead on the new dataset. We summarize the main contributions of our work as follows.

- **A Foundation Model for Tabular OD:** We present FoMo-0D, *the first foundation model for zero-shot OD* on unseen tabular datasets, with no additional training or hyperparameter tuning, backed by Transformer-based in-context learning (ICL), synthetic data pre-training, and feed-forward inference.

- **Model Selection Made Obsolete:** FoMo-0D is designed for zero-shot inference given a new dataset, fully abolishing not only model training on the new dataset, but also the notorious task of algorithm selection and hyperparameter tuning in the absence of labeled data.

- **Scalable Pre-training:** To enable pre-training on many large datasets, we propose (*i*) a new mechanism to reduce sample-to-sample attention from quadratic to linear time—enabling larger datasets, as well as (*ii*) on-the-fly data synthesis through data transformations—enabling more diverse datasets in less time.

- **Fast OD at Inference:** Given a new dataset, FoMo-0D bypasses both model training and selection, both of which can be slow for modern deep OD models with many hyper/parameters. Rather, it takes fraction of a second to label a test point through a single forward pass. Such speedy inference also unlocks the potential for deploying FoMo-0D in real time on data streams.

- **Effectiveness:** On a large benchmark of **57** datasets (Han et al., 2022) from diverse domains and against **26** baselines ranging from classical to modern OD models (Livernoche et al., 2024), FoMo-0D outperforms the majority of the baselines, with no statistical evidence for performance difference from the 2*nd* best baseline, while operating fully zero-shot on real-world datasets out-of-the-box.

- **New directions:** As the first foundation model for OD, FoMo-0D presents a paradigm shift in how to perform OD in practice, while offering new directions to explore as well as open questions to investigate. From the algorithmic perspective, how can we understand what (OD) algorithm, if any, has the Transformer- and ICL-based FoMo-0D learned? How can we interpret (mechanistically or otherwise) how the pretraining achieves zero-shot and out-of-distribution (OOD) generalization? From a data perspective, what prior distributions for pretraining are suitable for downstream real-world tasks? What is the role of prior data diversity and complexity in the generalization ability of the model? In summary, FoMo-0D paves the path towards a better understanding of ICL as well as building more powerful future foundation models for OD.

## 2 Problem and Preliminaries

### 2.1 Outlier Detection Problem and Setting

Outlier detection (OD) methods can be categorized based on the availability of labeled data. In supervised OD, the task is similar to binary classification with imbalanced classes (as outliers typically make up only a small portion of the overall data). The more difficult unsupervised setting assumes that the "contaminated" training data contains both inliers and outliers, but without any labels. This is also the transductive setting where training and test data are the same. The one-class setting lies between these two extremes, where the "clean" training data consists only of inliers, while the unlabeled test data contains the outliers to be detected. Here, training and test sets are disjoint and thus the setting is inductive. Note that there exist **no labeled outliers** in both settings, making model selection challenging.

We remark that some OD literature refers to the latter setting as semi-supervised OD, which is a misnomer from the supervised ML perspective where semi-supervised classification assumes the presence of some labeled instances from **all** classes in the training data. In the rest of text, we adopt the terminology unsupervised OD for both settings, and specify "clean" inlier-only versus "contaminated" mixed training data to differentiate them. Then, our work considers the unsupervised OD problem under "clean" inlier-only training data.

Formally, let $\mathcal{D}_{\text{in}} = \{(\mathbf{x}_1, y_1) \ldots, (\mathbf{x}_n, y_n)\}$ denote the inlier-only input data $\mathbf{x}_i \in \mathbb{R}^d$, where $y_i = 0 \; \forall i \in [n]$, and $\mathcal{D}_{\text{test}}$ depicts the unlabeled test data comprising both inliers and outliers. Note that $\mathcal{D}_{\text{in}} \cap \mathcal{D}_{\text{test}} = \emptyset$, i.e. train/test split is disjoint. The task is to assign labels to $\mathbf{x}_i \in \mathcal{D}_{\text{test}}$ given the inlier-only input $\mathcal{D}_{\text{in}}$.

### 2.2 Background on Prior-data Fitted Networks

**Posterior Predictive Distribution (PPD):** In the Bayesian framework for supervised learning, the prior defines a hypothesis space $\Phi$ which expresses our beliefs about the data distribution before seeing any data.

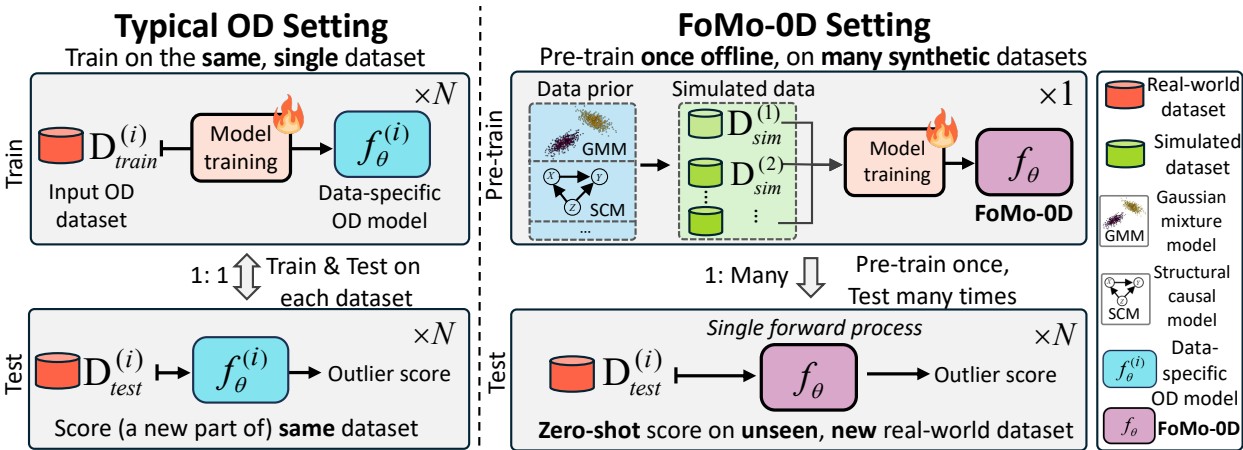

Figure 1: (best in color) Comparison of typical OD vs. the FoMo-0D settings. Given a new unlabeled OD dataset, FoMo-0D not only eliminates the need for model (parameter) training, but most importantly, also abolishes the onerous task of unsupervised model selection (algorithm and hyperparameters).

Each hypothesis $\phi \in \Phi$ describes a mechanism by which the data is generated. The posterior predictive distribution $p(\cdot|\mathbf{x}_{\text{test}}, \mathcal{D}_{\text{train}})$ provides a framework for making prediction on new, unseen test data $\mathbf{x}_{\text{test}}$, conditioned on observed training data $\mathcal{D}_{\text{train}} = \{(\mathbf{x}_1, y_1), \ldots, (\mathbf{x}_n, y_n)\}$. Based on Bayes' Theorem, the PPD can be derived by the integration over the space of hypotheses $\Phi$:

$$p(y_{\text{test}}|\mathbf{x}_{\text{test}}, \mathcal{D}_{\text{train}}) = \int_{\Phi} p(y_{\text{test}}|\mathbf{x}_{\text{test}}, \phi)p(\mathcal{D}_{\text{train}}|\phi)p(\phi)d\phi, \tag{1}$$

where $p(\phi)$ denotes the prior probability and $p(\mathcal{D}|\phi)$ is the likelihood of the data $\mathcal{D}$ given $\phi$.

**PFNs and PPD Approximation:** As obtaining the above PPD is generally intractable, Prior-data Fitted Networks (PFNs) are proposed to approximate the PPD (Müller et al., 2022). Unlike traditional machine learning models that are trained directly on observed datasets, PFNs are pre-trained on simulated datasets that are generated according to a prior distribution. Specifically, it contains the pre-training and inference stages described as the following.

*Pre-training on synthetic data.* Massive synthetic datasets are generated for the pre-training stage, by first sampling a hypothesis (i.e., the generating mechanism) $\phi \sim p(\phi)$, and then sampling a dataset $\mathcal{D} \sim p(\mathcal{D}|\phi)$. For training, each dataset $\mathcal{D}$ can be split as $\mathcal{D}_{\text{test}} \subset \mathcal{D}$ and $\mathcal{D}_{\text{train}} = \mathcal{D} \setminus \mathcal{D}_{\text{test}}$. Thus, the PFN with parameters $\theta$ can be optimized by making predictions on data points in $D_{\text{test}}$. For a test point $(\mathbf{x}_{\text{test}}, y_{\text{test}}) \in \mathcal{D}_{\text{test}}$, the training loss is formulated as follows.

$$\mathcal{L} = \mathbb{E}_{(\{(\mathbf{x}_{\text{test}}, y_{\text{test}})\} \cup \mathcal{D}_{\text{train}}) \sim p(\mathcal{D})}[-\log q_{\theta}(y_{\text{test}}|\mathbf{x}_{\text{test}}, \mathcal{D}_{\text{train}})]. \tag{2}$$

The above loss can also be interpreted as minimizing the expected KL divergence between $p(\cdot|\mathbf{x}, \mathcal{D})$ and $q_{\theta}(\cdot|\mathbf{x}, \mathcal{D})$ (Müller et al., 2022). In practice, a PFN model $q_{\theta}$ is typically implemented by a Transformer-based architecture (Vaswani et al., 2017), which takes $(\mathbf{x}_{\text{test}}, \mathcal{D}_{\text{train}})$ as input, where $\mathbf{x}_{\text{test}} \in \mathcal{D}_{\text{test}}$ and $\mathcal{D}_{\text{train}}$ contains an arbitrary number of instances. The output is the conditional class probabilities for $\mathbf{x}_{\text{test}}$. As the whole training set $\mathcal{D}_{\text{train}}$ is passed as input/context to the Transformer, it learns to predict class labels through sample-to-sample attention.

*Inference on real-world data.* In the inference stage, a fresh real-world dataset $\mathcal{D}_{\text{train}}$ and some test instance $\mathbf{x}_{\text{test}}$ are fed into the (frozen) pre-trained model, which computes the PPD $q_{\theta}(\cdot|\mathbf{x}_{\text{test}}, \mathcal{D}_{\text{train}})$ in a single forward pass. Importantly, PFNs do not require gradient-based parameter tuning on new datasets, where prediction is delivered *in less than a second* (Hollmann et al., 2023).

In summary, PFNs are trained once, and can be used many times for zero-shot inference on new datasets with different characteristics. The main benefit is that **no training or tuning** is required at the inference stage. This type of learning ability is also termed as in-context learning (ICL) (Xie et al., 2021), which was

shown to be effective for various tasks in languages (Brown et al., 2020). In fact, ICL with PFNs is recently shown to be a promising paradigm for supervised classification on tabular datasets (Hollmann et al., 2023).

## 3 FoMo-0D: A Foundation Model for Zero-shot Tabular Outlier Detection

Inspired by PFNs (Müller et al., 2022; Hollmann et al., 2023; Dooley et al., 2023), we propose FoMo-0D for zero-shot OD, which is pre-trained on large-scale synthetic OD datasets for zero-shot detection at inference time. FoMo-0D eliminates the need for model training on a new dataset and for model selection (both algorithm and HPs), which is difficult without any labeled data. The new FoMo-0D paradigm (right) versus the typical OD setting (left) is illustrated in Figure 1.

In the following we describe our OD data prior, training on prior-simulated datasets, inference on new datasets, and model architecture and improvements for scalability.

### 3.1 Designing a Data Prior for Outlier Detection

Foundation models benefit from massive amounts of datasets available for pre-training, along with high-capacity model architectures, however, the quantity (and quality) of publicly available tabular OD datasets is minuscule, compared to the massive size of open-domain text corpora. Even with large quantities of data, Ansari et al. (2024) show that using synthetic data in combination with real-world data improves the overall zero-shot performance for time-series foundation models. Hence, we design a new data prior from which we simulate numerous OD datasets for pre-training FoMo-0D.

Ideally, the data prior should reflect distributions as general and diverse as seen in the real world, however, "finding a prior supporting a large enough subset of possible [data generating] functions isn't trivial" (Nagler, 2023). Surprisingly, our results show that a straightforward, simple-to-implement data prior is sufficient to achieve remarkable performance.

**Inlier synthesis:** We simulate inliers by drawing from a Gaussian Mixture Model (GMM) with $m$-clusters in $d$-dimensions, with centers $\boldsymbol{\mu}_{jk} \in [-5,5]$, $j \in [m]$, $k \in [d]$ and $diagonal^2$ $\boldsymbol{\Sigma}_j$ with entries in $(0,5]$. We create different GMMs with varying $m \leq M$ and $d \leq D$ chosen uniformly at random from $[M]$ and $[D]$, respectively. From each GMM, we draw a set of $S$ inliers, defined as instances within the $90th$ percentile of the GMM.

**Outlier synthesis:** Following Han et al. (2022), we generate subspace outliers by first drawing a subset of dimensions $\mathcal{K}$ at random, for $|\mathcal{K}| \leq d$, and then generate $S$ points from the "inflated" GMM, which shares the same centers $\boldsymbol{\mu}_j$'s with the original GMM but with the inflated (diagonal) covariances $5 \times \boldsymbol{\Sigma}_{j,kk}$'s for $k \in \mathcal{K}$. Outliers are defined as points sampled outside the $90th$ percentile of the original GMM, which are labeled based on their Mahalanobis distances (see Property B.6 in the Appendix).

Specifically, we simulate datasets containing $2S = 10,000$ samples (half inlier, half outlier) from the two corresponding GMMs (original and inflated) with up to $M = 5$ clusters and up to $D = 100$ dimensions. Example 2-$d$ synthetic datasets are illustrated in Appendix A.

**Remarks:** Our model is not trained on **any** real-world data but rather, on purely synthetic data (although future work can combine existing benchmark OD datasets with synthesized data, as was done by Ansari et al. (2024) for time series). While we have intended to extend our preliminary attempt toward designing a sophisticated data prior for OD, we found (to our surprise) that even with a basic, GMM-based prior, FoMo-0D generalizes remarkably well to real-world OD datasets downstream[3], outperforming numerous SOTA baselines. Therefore, we present FoMo-0D with this simple prior to showcase the prowess of PFNs for OD. We leave as future work the exploration of other priors (Hollmann et al., 2023) and other outlier types (contextual, dependency, etc. (Steinbuss & Böhm, 2021)), the impact of different priors on performance, as well as prior mixture composition to further improve performance.

---

[2]In early experiments, we found no difference in test performance on synthetic datasets between using diagonal vs. non-diagonal $\boldsymbol{\Sigma}$, yet, it is easier to invert diagonal $\boldsymbol{\Sigma}$ for data synthesis.

[3]We refer to Appendix G.2 and Figure 16 for an exploration of FoMo-0D performance and GMM goodness of fit on real-world OD datasets.

### 3.2 (Pre)Training and Inference

**Model (Pre)Training (Once, Offline):** FoMo-0D is a Prior-data Fitted Network (PFN, see Section 2.2) based on the Transformer architecture. In the synthetic prior-data fitting phase, it is trained on datasets drawn from our OD data prior for tabular data introduced in Section 3.1. Each dataset is simulated from a different GMM configuration based on randomly drawn parameters, and consists of varying number of training samples and dimensions to capture the diversity in real-world tabular datasets. Details are outlined in Algorithm 1 in Appendix C, and described as follows.

At each time, we first draw a hypothesis (i.e. GMM configuration) uniformly at random, that is, $\phi = \{d \in [D], m \in [M], \{\boldsymbol{\mu}_j\}_{j=1}^m \in [-5,5]^d, \{\boldsymbol{\Sigma}_j\}_{j=1}^m; diag(\boldsymbol{\Sigma}_j) \in [-5,5]^d\}$, and then generate a synthetic dataset $\mathcal{D} = \{\mathcal{D}_{\text{in}}, \mathcal{D}_{\text{out}}\}$ containing synthetic inlier and outlier samples from the drawn hypothesis and its variance-inflated variant, respectively.

We optimize FoMo-0D's parameters $\theta$ to make predictions on $\mathcal{D}_{\text{test}} = \{\mathcal{D}_{\text{test}}^{\text{in}}, \mathcal{D}_{\text{test}}^{\text{out}}\}$, conditioned on the inlier-only training data $\mathcal{D}_{\text{train}} \subset \mathcal{D}_{\text{in}}$ based on the cross-entropy loss (see Eq. (2)). During training, $\mathcal{D}_{\text{test}}$ contains a *balanced* number of inlier and outlier samples, where $\mathcal{D}_{\text{test}}^{\text{in}} = \mathcal{D}_{\text{in}} \backslash \mathcal{D}_{\text{train}}$, and $\mathcal{D}_{\text{test}}^{\text{out}} \subset \mathcal{D}_{\text{out}}$ contains an equal number of samples as $\mathcal{D}_{\text{test}}^{\text{in}}$. To vary the training data size, we subsample $\mathcal{D}_{\text{train}}$ of randomly drawn size $n \in [n_L, n_U]$, where $n_L$ and $n_U$ denote the lower and upper bounds. In our implementation, we use $n_L = 500$, and $n_U = 5,000$.

FoMo-0D is trained on $200,000$ batches ($200$ epochs $\times 1,000$ steps/epoch) of $B = 8$ generated datasets in each batch. While this pre-training phase can be expensive, it is done *only once, offline.* Moreover, we introduce several scalability improvements to speed up pre-training, as discussed later in Section 3.3. Full details on the training and implementation of FoMo-0D are given in Appendix C.

**Zero-shot Inference (on Unseen/New Dataset):** At inference, the pre-trained FoMo-0D can be employed on any unseen real-world dataset. Specifically, for a new unsupervised OD task with inlier-only training data $\mathcal{D}_{\text{train}}$ and mixed test data $\mathcal{D}_{\text{test}}$, feeding $\langle \mathcal{D}_{\text{train}}, \mathbf{x}_{\text{test}} \rangle$ as input to FoMo-0D (for each $\mathbf{x}_{\text{test}} \in \mathcal{D}_{\text{test}}$ separately) yields the PPD $q_\theta(y|\mathbf{x}_{\text{test}}, \mathcal{D}_{\text{train}})$ in a *single forward pass.* As such, FoMo-0D performs model "training" and prediction simultaneously at test time. In fact, as the training data is passed as context, FoMo-0D leverages in-context learning (ICL) (Xie et al., 2021; Garg et al., 2022) for inference.

**Remarks:** The **key** contribution of FoMo-0D goes beyond eliminating the need for model training for a new dataset, it *renders model selection an obsolete concern for OD.* In other words, a practitioner with a new detection task no longer needs to choose an OD model to train or grapple with tuning any hyperparameters of the said model. Further, the speedy, easily parallelizable inference (for *less-than-a-second* per test sample) is the "icing on the cake". Figure 1 (right) illustrates (top) pre-train and (bottom) test phases of FoMo-0D, where the pre-trained FoMo-0D is reused during inference on new datasets directly, unlocking zero-shot OD.

### 3.3 Architecture and Scalability

**Architecture and sample-to-sample attention:** Like existing PFNs, FoMo-0D is based on the Transformer (Vaswani et al., 2017), encoding each sample's feature vector as a fixed size token through a linear embedding layer (see Appendix C.2), and allowing token representations to attend to each other, hence enabling sample-to-sample attention. We also adopt the three customizations from TabPFN (Hollmann et al., 2023), which (1) computes self-attention among all the training samples and only cross-attention from test samples to the training samples, (2) enables varying feature dimensionality by zero-padding, and (3) randomly permutes input samples while omitting positional encodings to achieve model invariance in the dataset.

Given $\mathcal{D}_{\text{train}} = \{\mathbf{x}_1, \ldots, \mathbf{x}_n\}$, each self-attention layer outputs $n$ embeddings $\{\mathbf{z}_i\}_{i=1}^n$; where the $i$-th token is mapped via linear transformations to a key $\mathbf{k}_i$, query $\mathbf{q}_i$ and value $\mathbf{v}_i$, where the $i$-th output is computed as

$$\mathbf{z}_i = \sum_{j=1}^n \texttt{softmax}(\ \{\langle \mathbf{q}_i, \mathbf{k}_{j'} \rangle\}_{j'=1}^n\ )_j \cdot \mathbf{v}_j\ . \tag{3}$$

The sample-to-sample attention is intriguing from the perspective of OD: many classical OD algorithms (Aggarwal, 2013) are based on nonparametrics; in particular, they leverage the distances to the $k$ *nearest* neighbors ($k$NNs) of a point to compute its outlierness, where $k$ is a critical hyperparameter. One can think

of `FoMo-0D` as mimicking non-parametric models but by using parametric attention mechanisms. Interestingly, PFNs are much more robust and flexible than $k$NN based OD approaches, for (1) sample-to-sample relations are not pre-specified but rather learned through attention weights, and thus (2) they are not limited to just the nearest neighbors but rather can *learn which* training points are worth attending to, and (3) as attention is dataset-wide across all points, there is no need for specifying a cut-off HP value like $k$, to which most $k$NN based OD techniques are sensitive to (Aggarwal & Sathe, 2015; Campos et al., 2016; Goldstein & Uchida, 2016; Ding et al., 2022). We present analyses on sample-to-sample attention in Appendix E.

To seize the power of scale, we incorporate a scalable architecture and data synthesis into our design to benefit pre-training and inference, as we describe next. The scale-up unlocks a larger context size for `FoMo-0D`, enabling pre-training and inference on larger datasets with fast speed.

**Scaling up attention with "routers":** The $\mathcal{O}(n^2)$ quadratic sample complexity at pre-training presents an obstacle for achieving high performance at inference, as it limits pre-training to relatively small training datasets, and degenerates in-context learning that typically benefits from longer context (Xie et al., 2021).

Toward a high-performance model, we scale up `FoMo-0D`'s attention via the "router mechanism" of Zhang & Yan (2023). As shown in Figure 2, the main idea is to learn a small number ($R \ll n$) of "routers", which gather information from all $n$ samples and then distribute the information back to the $n$ output embeddings, in effect, reducing complexity from $\mathcal{O}(n^2)$ to $\mathcal{O}(2Rn) = \mathcal{O}(n)$. This design allows `FoMo-0D` to **scale linearly** with respect to both dimensionality $d$ and dataset size $n$ in pre-training as well as during inference.

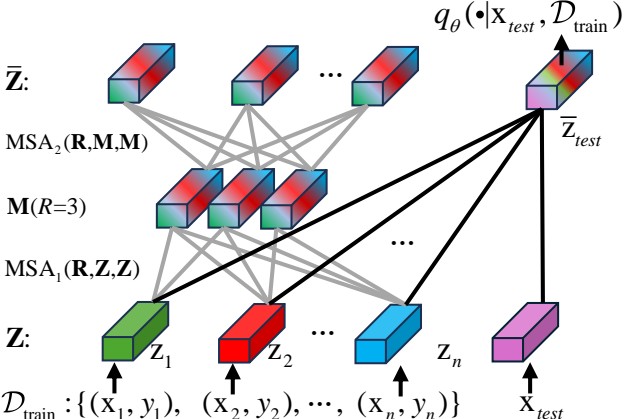

Figure 2: `FoMo-0D` architecture employs the "router mechanism" for scalable attention.

Concretely, the representatives first aggregate information from all samples by serving as the query in the multi-head self-attention (`MSA`):

$$\mathcal{M} = \texttt{MSA}_1(\mathbf{R}, \mathbf{Z}, \mathbf{Z}) , \tag{4}$$

where $\mathbf{R} \in \mathbb{R}^{R \times d}$ depicts the *learnable* vector array of representatives and $\mathcal{M}$ denotes the aggregated messages. Then, the routers distribute the received information among samples by using the sample embeddings as query and the aggregated messages as both key and value:

$$\hat{\boldsymbol{Z}} = \texttt{MSA}_2(\mathbf{Z}, \mathcal{M}, \mathcal{M}) . \tag{5}$$

Finally, we obtain $\bar{\boldsymbol{Z}} = \texttt{LayerNorm}(\hat{\boldsymbol{Z}} + \mathbf{Z})$ after layer normalization. Note that the test samples only attend to the training samples' embeddings, computed in the described manner across layers, and are finally fed into the prediction head to estimate the PPD at the output layer.

**Scaling up (pre)training data synthesis with linear transforms:** Besides the scalability challenge associated with architecture/attention, another computational challenge in pre-training `FoMo-0D` arises from drawing samples from the data prior, which requires considerable time, especially in high dimensions[4],

---

[4]This is because the inverse of the $(d \times d)$ covariance matrix plays a crucial role in the process of drawing samples from GMMs, which has $\mathcal{O}(d^3)$ time complexity. (It is also the reason why diagonal $\boldsymbol{\Sigma}_j$'s are favored in our data prior.) In addition, Mahalanobis distance for labeling inliers/outliers also requires the inverse.

provided the large number of datasets we sample (specifically, we utilize a batch size of 8 datasets over 1,000 steps each for 200 epochs).

To give an idea, sampling a dataset with $n = 10,000$ points in $d = 100$ dimensions using 10 CPUs in parallel takes $\approx 0.4$ seconds (see Appendix Figure 7). Across 200 training epochs with 1,000 steps each, it adds up to more than 177 hours just to generate 1,6 million datasets on-the-fly. Of course, one can trade storage with compute-time by generating all these datasets apriori via massive parallelism. Nevertheless, synthetic data generation demands considerable time (and/or storage).

To scale up data synthesis, FoMo-0D employs two distinct strategies. **First**, we propose *reuse at epoch level*: that is, one can reuse the same 8K ($8 \times 1000$) unique datasets at every epoch, or in general, the same 8K$\times P$ datasets periodically at every $P$ epochs. A larger $P$ would lead to more diversity in terms of the overall pre-training data used.

**Second**, we propose *reuse at dataset level via transformation*: that is, having generated one unique dataset $\mathbf{X} \in \mathbb{R}^{n \times d}$ from a GMM, we propose a linear transform $T(\mathbf{x})$ of the form $\mathbf{Wx} + \mathbf{b}$ for randomly drawn parameters $\mathbf{W} \in \mathbb{R}^{d \times d}$ and $\mathbf{b} \in \mathbb{R}^d$ (see Appendix B.1).[5] This simple yet efficient transformation creates a new dataset, akin to one being drawn from another GMM with centers $T(\boldsymbol{\mu}_j) = \mathbf{W}\boldsymbol{\mu}_j + \mathbf{b}$ and covariance $T(\boldsymbol{\Sigma}_j) = \mathbf{W}\boldsymbol{\Sigma}_j\mathbf{W}^T, \forall j \in [m]$. Note that we do not actually materialize these parameters but only transform the dataset. As we show in the following, such transformations preserve the Mahalanobis distances as well as the percentile thresholds for labeling points as inlier/outlier. Details and proofs are given in Appendix B.

**Lemma 3.1.** *Linear transform $T$ with invertible $\mathbf{W}$ on $\mathcal{G}_m^d$ preserves Mahalanobis distances.*

**Lemma 3.2.** *Linear transform $T$ with invertible $\mathbf{W}$ on $\mathcal{G}_m^d$ preserves the percentiles of the GMM.*

The implication of these lemmas is that a linear transformation of a dataset from a GMM retains the identity of the inliers and outliers, i.e. no relabeling is required. Moreover, notice that as a byproduct we obtain a transformed dataset as though it is drawn from a GMM with a *non-diagonal* covariance matrix which, besides the time savings, offers a slightly more complex data prior.

To reach 8K unique datasets for each epoch, we first generate 500 datasets from different GMMs (with varying configurations), then employ 15 different linear transformations to each dataset by varying $\mathbf{W}$ and $\mathbf{b}$. Drawing each $(\mathbf{W}, \mathbf{b})$ takes $\approx 0.02$ seconds, while the matrix-matrix product of $\mathbf{X}$ ($n \times d$) and $\mathbf{W}$ ($d \times d$) takes negligible time (for $d \le 100$). Thus, obtaining a transformed dataset offers $20\times$ speed-up compared to generating one (0.02 vs. 0.4 seconds).

## 4 Experiments

### 4.1 Setup

We present the experiment setup briefly, including data synthesis, real-world datasets, baselines, metrics and HPs. For more details, we refer to Appendix D.

**Pre-training Dataset Synthesis:** During pre-training, we generate unique GMM datasets by first drawing a configuration, including dimensionality $d \in [D]$, number of components $m \in [M]$, centers $\{\boldsymbol{\mu}_j\}_{j=1}^m$ (each $\boldsymbol{\mu}_j \in [-5, 5]^d$) and covariances $\{\boldsymbol{\Sigma}_j\}_{j=1}^m$ ($diag(\boldsymbol{\Sigma}_j) \in [-5, 5]^d$). We set $M = 5$ and vary $D \in \{20, 100\}$ to study pre-training with relatively small and high dimensional datasets, respectively. We synthesize inliers and outliers described in Section 3.1.

**Real-world Benchmark Datasets:** While pre-training is purely on synthetic datasets, we evaluate FoMo-0D on **57** real-world datasets from ADBench (Han et al., 2022) (see Table 20 in Appendix J). Following Livernoche et al. (2024), we use 5 train/test splits of each dataset via different seeds and report mean performance and standard deviation. Note that the baselines require model re-training and inference for each $\mathcal{D}_{\text{train}}/\mathcal{D}_{\text{test}}$ split, while FoMo-0D uses the splits only for inference as $\mathcal{D}_{\text{train}}$ is passed as context.

---

[5]In practice, we apply the linear transform on the subspace of inflated features only, wherein inliers and outliers are defined, which remains to be a multi-variate GMM.

**Baselines:** We compare FoMo-0D against **26** baselines, from classical/shallow methods to modern/deep models. The baselines are imported from one of the latest papers that proposed the SOTA diffusion-based OD model, DTE (Livernoche et al., 2024), and three variants; DTE-C, DTE-IG, DTE-NP. As such, the long list of baselines we compare to constitutes one of the most comprehensive in the literature. We refer to the original paper for more details.

**Model Implementation:** We train our final model for 200,000 steps with a batch size of 8 datasets. That is, FoMo-0D is trained on 1,600,000 synthetically generated datasets. This training takes about 25 hours on 1 GPU (Nvidia RTX A6000). Each dataset had a fixed size of 10,000 samples, with $|\mathcal{D}_{\text{train}}| \in [n_L = 500, n_U = 5000]$, and the rest as $\mathcal{D}_{\text{test}}$ with balanced number of inliers and outliers. Other details of FoMo-0D, including the training algorithm, model architecture, data synthesis and reuse, and hardware are in Appendix C.

**Metrics and Hypothesis Testing:** Detection performance is w.r.t. 3 widely-used metrics for OD: AUROC; area under ROC curve, AUPR; area under Precision-Recall curve, and F1 score; using threshold at the true number of outliers in the test data (varies by dataset) Livernoche et al. (2024).

To compare different methods on ADBench, we compute their rank on each dataset (lower is better), and present average rank across datasets. This is an alternative to the average metric (e.g. AUROC), which is not meaningful when tasks vary widely in terms of their difficulties.

In addition, we perform significance tests to compare two methods statistically, using the one-sided paired Wilcoxon signed rank test (Demšar, 2006) between FoMo-0D and a baseline based on the performances across all datasets, with the alternative hypothesis suggesting the "baseline-minus-FoMo-0D" performance gap is greater than zero. We consider results to be significant at $\alpha = 0.05$ following convention.

**Hyperparameters (HPs):** Importantly, Livernoche et al. (2024) picked for each baseline the best-performing set of HPs as recommended by the authors in their original paper. As for their own DTE, which behaves similarly to $k$NN, they use $k = 5$ and set the *same k* for the $k$NN baseline (Ramaswamy et al., 2000) to be consistent. However, it is well known that $k$NN is sensitive to the value of $k$ (Aggarwal & Sathe, 2015), and so are many other OD models to their respective HPs (Campos et al., 2016; Goldstein & Uchida, 2016; Zhao et al., 2021; Ding et al., 2022).

Therefore, besides comparing FoMo-0D with the 26 baselines in Livernoche et al. (2024), respectively for AUROC, F1, and AUPR (Livernoche et al., 2024), we also compare to the **top-4**[6] best-performing baselines (in order: DTE-NP, $k$NN, ICL, and DTE-C) on their *average* performance across a list of different HP settings. Such an approach reflects their *expected* performance under HP values selected at random, in the absence of any other prior knowledge, as recommended by Goldstein & Uchida (2016) "*to get a fair evaluation when comparing [OD] algorithms*". We annotate the method name with avg for the version with performance averaged over varying HPs. The detailed list of HP values for each top baseline is given in Appendix D.4. Overall, we compare FoMo-0D to 30 baselines; 26 from Livernoche et al. (2024) and avg of the top-4.

## 4.2 Results

**Detection performance:** Table 1 presented the comparison of FoMo-0D w/ $D = 100$ to all baselines w.r.t. average rank across datasets as well as pairwise Wilcoxon signed rank tests based on AUROC, and **we present full results on all datasets and all metrics in Appendix I**. We find that among 30 baselines and 2 variants of FoMo-0D (w/ $D = 100$ and $D = 20$), FoMo-0D w/ $D = 100$ performs as well as the 2*nd* best model ($k$NN with default HP; $k = 5$) on all datasets. While DTE-NP outperforms FoMo-0D with author-recommended $k = 5$, we find that DTE-NPavg is on par with FoMo-0D.

In our tests, $p > \alpha = 0.05$ implies no statistical evidence for performance difference between two methods. FoMo-0D w/ $D = 100$ performs statistically no different from **all** baselines on datasets with $d \leq 100$ (i.e., "at its own game" when pre-training data dimensions align with real-world datasets), while it outperforms the majority of baselines (where $p > 1 - \alpha$). These results continue to hold on datasets with $d \leq 500$.

---

[6] To rank the 26 baselines, we compute the $26{\times}26$ $p$-values of the pairwise Wilcoxon signed rank test (see Appendix Figure 24), and order them by their mean $p$-value against other baselines.

Table 2: $p$-values of the one-sided Wilcoxon signed rank test, comparing FoMo-0D (w/ $D = 20$) to **top 10** baselines with default HPs, and **top 4$^{\text{avg}}$** baselines[6] with **avg.** performance over varying HPs (denoted w/ $^{\text{avg}}$) over All (57) datasets, those (24) w/ $d \le 20$ and (38) datasets w/ $d \le 50$ dimensions. Although pretrained on datasets w/ small $D = 20$, FoMo-0D shows **no statistically significant difference from the top 3$rd$ baseline** (ICL, w/ $p = 0.089$) over All datasets, while it outperforms (w/ $p > 1 - \alpha$) the top 5$th$ (LOF) and onward baselines over datasets w/ $d \le 20$ (aligned w/ pretraining where $D = 20$) and on datasets w/ $d \le 50$ (generalizing beyond pretraining). We use underline and **bold** to indicate $p < \alpha$ and $p > 1 - \alpha$. Rank is avg.'ed over all 57 datasets, where methods are ranked on each dataset w.r.t. AUROC. (experiment setting: $D = 20$, $P = 50$, $R = 500$, train/inference context size=5K, no data transformation, $\alpha = 0.05$)

| | FoMo-0D | DTE-NP | $k$NN | ICL | DTE-C | LOF | CBLOF | Feat.Bag. | SLAD | DDPM | OCSVM | DTE-NP$^{\text{avg}}$ | $k$NN$^{\text{avg}}$ | ICL$^{\text{avg}}$ | DTE-C$^{\text{avg}}$ |
|---|---|---|---|---|---|---|---|---|---|---|---|---|---|---|---|
| $d \le 20$ | - | 0.572 | 0.789 | **0.968** | 0.616 | **0.993** | **0.989** | **1.000** | **0.978** | 0.906 | **0.992** | 0.813 | 0.924 | **0.999** | **1.000** |
| $d \le 50$ | - | 0.347 | 0.794 | 0.893 | 0.946 | **0.997** | **0.988** | **1.000** | **0.963** | **0.994** | **0.986** | 0.574 | 0.847 | **0.995** | **1.000** |
| All | - | 0.001 | 0.019 | 0.089 | 0.159 | 0.394 | 0.434 | 0.703 | 0.516 | 0.752 | 0.679 | 0.007 | 0.062 | 0.437 | **1.000** |
| Rank(avg) | 12.59 | 7.19 | 8.57 | 10.34 | 10.79 | 11.82 | 12.81 | 12.8 | 12.52 | 13.50 | 13.34 | 8.60 | 10.63 | 12.44 | 21.43 |

Table 2 shows similar results for FoMo-0D w/ $D = 20$, which is pre-trained on datasets with considerably fewer dimensions. Even in this limited setting, it performs on par with the 3$rd$ best baseline (ICL, with default HP) against 30 baselines, with an increased $p$-value (0.437) when compared to ICL$^{\text{avg}}$. On datasets with $d \le 20$ which align with its pre-training data, it outperforms the top 5$th$ baseline and the majority of others. With FoMo-0D pre-trained purely on synthetic datasets from a simple prior in small dimensions, these results showcase the prowess of PFNs for OD.

Figure 3 shows the distribution of AUCROC across datasets for all models (See rank distribution in Appendix Figure 17). FoMo-0D achieves a competitively small average rank with notably higher AUCROC across datasets compared to the majority of the baselines. Appendix H presents another comparison between detectors through performance profile plots (Dolan & Moré, 2002).

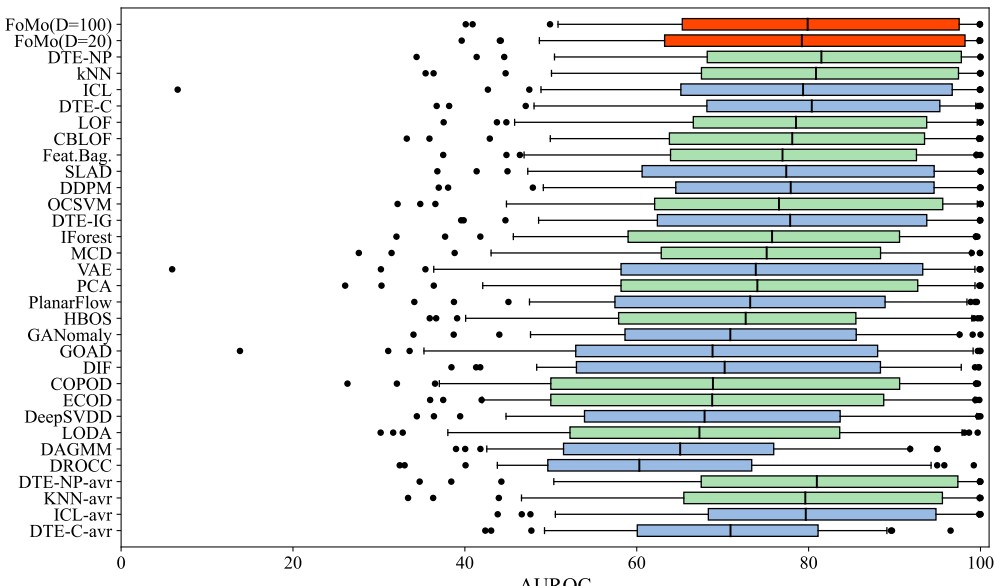

Figure 3: (best in color) AUROC (higher is better) distribution across all **57** real-world datasets shown via boxplots for (from top to bottom) FoMo-0D in red, all **26** baselines ordered by mean $p$-value[6] (shallow and deep baselines in green and blue), and **top 4** baselines' $^{\text{avg}}$ variants. The vertical line depicts the mean, the box shows the 25-75%, bars range 5-95%, and circles show the datasets at the tails.

**Running time:** Table 3 presents the total training time and the average inference time per test sample for FoMo-0D compared with the top-3 baselines as measured on ADBench's largest dataset. Given a new dataset, FoMo-0D bypasses model training (and HP tuning) and directly performs inference, within 7.7 ms per sample

Table 3: Train-time and Inference-time (in milliseconds) of FoMo-0D and top-3[6] baselines (w/ *default* HPs, *excluding* time for hyperparameter optimization) on our largest dataset donors (see Appendix Table 20). FoMo-0D skips any model training or fine-tuning and takes a mere forward pass for inference out-of-the-box.

| Method | FoMo-0D | DTE-NP | kNN | ICL |
|---|---|---|---|---|
| Train-time (total) | none/0-shot | 1,170.29 | 174,157.38 | 130,412.86 |
| Infer-time (per sample) | 7.7 | 1.38 | 0.65 | 0.02 |

on average (see Appendix Figure 6). In comparison, all baselines need to train on each individual dataset before inference. Training time can be high for deep learning based models like ICL, further compounded by hyperparameter tuning that requires training multiple models. Even for non-parametric and/or shallow models like $k$NN and DTE-NP (which query $k$ nearest neighbors), training involves various data pre-processing steps such as constructing a tree-like data structure[7] for fast (often approximate) $k$NN distance querying.

### 4.3   Ablation Analyses

Extensive ablations in Appendix F analyze **(1)** the effect of $D$ in F.1, **(2)** cost and performance by varying $R$ in F.2 and F.3, **(3)** context size in F.4, **(4)** reuse periodicity $P$ in F.5, **(5)** effect of data transformation $T$ on performance and speed-up in F.6 and F.7, **(6)** data diversity and prolonged training in F.8, **(7)** quantile transformation in F.9, and **(8)** model size in F.10.

### 4.4   Generalization Analyses

Our results that synthetic pretraining at large, and GMM data prior in particular, enables FoMo-0D to reach remarkable performance on real world tasks call for deeper investigation on its generalization. To this end, Appendix G provides an extensive analysis on FoMo-0D's generalization to out-of-distribution (OOD) synthetic datasets in G.1, GMM statistical goodness-of-fit analyses of real-world datasets in ADBench in G.2, as well as generalization to real-world OOD detection tasks in G.3.

## 5   Related Work

**Outlier Detection (OD):**  Thanks to diverse applications in numerous fields, such as security, finance, manufacturing, to name a few, OD on tabular (or point-cloud) datasets has a vast literature with a long list of techniques. For earlier, shallow approaches preceding the advances in deep learning, we refer to the books by Aggarwal (2013) and Aggarwal & Sathe (2017). The modern, deep learning based techniques are surveyed in Chalapathy & Chawla (2019); Pang et al. (2021); Ruff et al. (2021). Most recent deep OD techniques take advantage of newly emerging paradigms, including self-supervised learning (Hojjati et al., 2022; Yoo et al., 2023) as well as the most recently popularized diffusion-based models (Yoon et al., 2023; Livernoche et al., 2024; Du et al., 2024; He et al., 2024).

**Unsupervised Model Selection for OD:**  It is typical of models to exhibit various hyperparameters (HPs) that play a role in the bias-variance trade-off and hence the generalization performance, and OD models are no exception. Many earlier work on OD showed the sensitivity of classical (i.e. shallow) OD methods to the choice of their HP(s) (Aggarwal & Sathe, 2015; Campos et al., 2016; Goldstein & Uchida, 2016). Similarly, sensitivity to HPs has also been shown for deep OD models more recently (Zhao et al., 2021; Ding et al., 2022), as well as for those relying on self-supervised learning/data augmentation (Yoo et al., 2023).

While critical, work on unsupervised outlier model selection (UOMS) is slim as compared to the vast literature on detection methods. A handful of existing, mostly heuristic strategies has been studied by Ma et al. (2023) reporting discouraging results; they have shown that existing heuristics are either not significantly different from random selection, or do not outperform iForest (Liu et al., 2008) with its default HPs.

---

[7]Both kNN (from PyOD (Zhao et al., 2019)) and DTE-NP use BallTree from sklearn (Pedregosa et al., 2011) for nearest neighbor search; however, kNN has extra computations after calling the BallTree, which might lead to the time differences for training and inference compared to DTE-NP. We encourage the readers to refer to the official implementations for details.

Recent UOMS approaches go beyond heuristic measures and design scalable hyperensembles (Ding et al., 2022; 2024), or leverage meta-learning on historical real-world OD datasets (Zhao et al., 2021; 2022; Zhao & Akoglu, 2024). These approaches show the value of (meta)learning and transferring from historical tasks to a new task. Though grounded in the same idea of learning from a large set of (in our case, simulated) tasks, we differ in one key aspect: FoMo-0D is *not* a model selection technique, but rather, a foundation model that abolishes model training and selection altogether. As such, it unlocks zero(0)-shot inference on a new task.

**Prior-data Fitted Networks:** Based on the seminal work by Müller et al. (2022), Prior-data-fitted Networks (PFNs) establish a new paradigm for machine learning, where a PFN is pretrained on synthetic datasets generated from a data prior, and the pretrained PFN can then infer the posterior predictive distribution (PPD) for test points in a new dataset in a single forward pass, through in-context learning (Xie et al., 2021; Garg et al., 2022). It is shown that PFNs provably approximate Bayesian inference (Müller et al., 2022). Follow-up TabPFN (Hollmann et al., 2023) and its v2 Hollmann et al. (2025) achieved SOTA classification performance on small tabular datasets of size up to 1024. Other subsequent works designed LC-PFN (Adriaensen et al., 2024) and ForecastPFN (Dooley et al., 2023), respectively zero-shot learning curve extrapolation and zero-shot time-series forecasting models, trained purely on synthetic data. PFN4BO (Müller et al., 2023) employed PFNs for Bayesian optimization, while Nagler (2023) studied the statistical foundations of PFNs. Others proposed scaling the context size to enable training on larger datasets toward better generalization (Ma et al., 2024; Feuer et al., 2023; 2024; Qu et al., 2025).

Our proposed FoMo-0D differs from these in being the first PFN for OD, using a novel inlier/outlier data prior, employing linear transform for fast data synthesis, and incorporating the "router" attention mechanism for linear-time scalability w.r.t. context size. See Appendix K for additional details.

**Zero-Shot Outlier Detection:** Foundation models pretrained on massive text and image corpora, such as large language and/or vision models (L(V)LMs) like OpenAI's GPT-series (Achiam et al., 2023), DALL-E (Ramesh et al., 2021) and Flamingo (Alayrac et al., 2022), CLIP (Radford et al., 2021), and LLaVA (Liu et al., 2024) to name a few, have demonstrated remarkable success on several zero-shot tasks in CV and NLP. Follow-up work extended these models for zero-shot out-of-distribution detection (Esmaeilpour et al., 2022), zero-shot image OD (Liznerski et al., 2022; Jeong et al., 2023; Zhou et al., 2024) as well as dialogue-based industrial image anomaly detection (Gu et al., 2024).

Foundation models, however, do not exist for tabular data which is widespread across OD applications in the real world, such as detecting credit card fraud, network intrusion, medical anomalies, and any sensor measurement abnormalities, to name a few. The recent ACR model by Li et al. (2023) on zero-shot OD does *not* rely on a pretrained foundation model, but rather is meta-trained on each specific domain using inlier-only datasets from the *same domain*. Concurrent to our work, Li et al. (2024) apply pretrained LLMs for prompt-based OD on tabular data which they serialize to text. Similar to our work, they also use *simulated* labeled OD datasets to fine-tune several existing LLMs to improve their performance. Their work, however, is quite preliminary in several fronts; a key limitation is that they assume independent features and query the LLM one-feature-at-a-time to reach an outlier score. Further, they fine-tune using only 5,000 data batches with up to 100 samples each, subsample 150 points and the first 10 columns of each dataset for evaluation (due to GPU memory constraint), and their testbed includes only two baseline methods. In contrast, FoMo-0D employs and pretrains PFNs at a much larger scale with rigorous evaluation on a much larger testbed.

## 6   Conclusion

This work introduced FoMo-0D, **the first foundation model for outlier detection** (OD) on tabular data. It capitalizes on the in-context learning of a Transformer model pre-trained on a large number of synthetic datasets that can then perform **zero-shot** inference on a new dataset, without *any* hyper/parameter tuning/training. FoMo-0D breaks new ground by fully abolishing the notoriously-hard model selection task for unsupervised OD (see Impact Statement). Further, FoMo-0D offers extremely fast inference thanks to a mere single forward pass. Against a long list of **26** SOTA baselines on **57** public real-world datasets, FoMo-0D performs on par with the 2*nd* best baseline, while outperforming the majority of the baselines. Future work could expand our data prior and explore similar directions for zero-shot OD beyond tabular data. For a detailed discussion on limitations and future directions, we refer to Appendix L.

## Broader Impact Statement

FoMo-0D offers zero-shot outlier detection (OD), abolishing not only parameter training but also model selection given a new dataset. This is a radical paradigm shift for the OD literature, which historically focused on designing new models and recently unsupervised model selection. Obviating the need for either, we expect FoMo-0D to route attention of the community from new model design and selection to designing better data priors and gathering datasets for pre-training, along with better and more scalable architectures for PFN.

From the applied perspective, a zero-shot OD model like FoMo-0D is a game changer for practitioners! Given the plethora of OD algorithms to choose from, each with a list of hyperparameters to set, not having the tools for effective and efficient model selection leaves the practitioners with a "choice paralysis". With FoMo-0D, practitioners can not only bypass such dilemmas on one dataset, but thanks to the "train once, use many times" nature of pre-trained models, they can do so for any dataset, including those arriving over time. FoMo-0D is lightweight and optimized for fast inference (without any additional training or tuning), making it especially attractive for real-time or resource-constrained applications. While attractive from a performance viewpoint, we remark that FoMo-0D currently does not factor into account metrics beyond detection performance, such as fairness, biases, or other potential blindspots. In sensitive real-world applications, social and ethical costs of incorrect detections should be taken into consideration.

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

## Appendix

## Table of Contents

We detail the contents in the appendix below.

## A Illustration of synthetic data in 2-$d$

We visualize our synthetic data in Figure 4, with 3 randomly created 2-$d$ GMMs with the number of clusters ($N = 1, 2, 3$). We choose the $80th$ percentile as the criterion, such that inliers are samples drawn from the GMM and within the $80th$ percentile, and outliers are samples drawn from the inflated GMMs and outside of the $80th$ percentile.

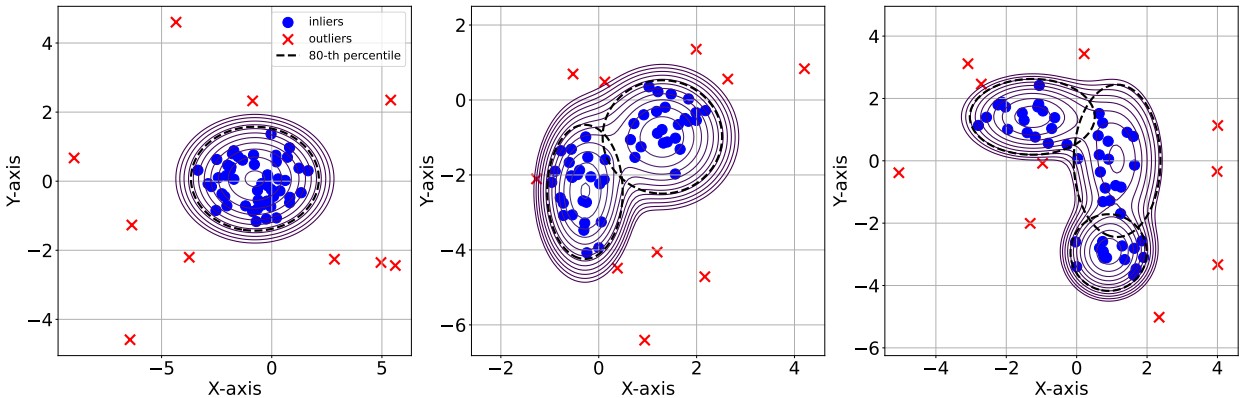

Figure 4: Illustration of synthetic data in 2D with $80th$ percentile as the criterion.

## B Linear Transform for Scalable GMM Data Synthesis

### B.1 Definitions

**Definition B.1** (Gaussian Mixture Model). We denote an $m$-cluster $d$-dimension Gaussian Mixture Model as $\mathcal{G}_m^d = \{(w_j, \boldsymbol{\mu}_j, \boldsymbol{\Sigma}_j)\}_{j=1}^m$, which is the weighted sum of $m$ Gaussian distributions:

$$p(\mathbf{x}) = \sum_{j=1}^m w_i \cdot g(\mathbf{x}|\boldsymbol{\mu}_j, \boldsymbol{\Sigma}_j) , \tag{6}$$

where $w_j \in \mathbb{R}^+$ is the weight for the $j$-th Gaussian $\mathcal{N}(\boldsymbol{\mu}_j, \boldsymbol{\Sigma}_j)$ with $\sum_{j=1}^m w_j = 1$, and $g(\cdot|\boldsymbol{\mu}_j, \boldsymbol{\Sigma}_j)$ is the density of the $j$-th component/cluster, with mean/center $\boldsymbol{\mu}_j \in \mathbb{R}^d$ and covariance $\boldsymbol{\Sigma}_j \in \mathbb{R}^{d \times d}$ being positive semi-definite, such that $\mathbf{x}^T \boldsymbol{\Sigma}_i \mathbf{x} \geq 0$, for all $\mathbf{x} \in \mathbb{R}^d$.

**Definition B.2** (Linear Transform). We denote a linear transformation $T$ in $\mathbb{R}^d$ as:

$$T(\mathbf{x}) = \mathbf{W}\mathbf{x} + \mathbf{b} , \tag{7}$$

where $\mathbf{x} \in \mathbb{R}^d$, and $\mathbf{W} \in \mathbb{R}^{d \times d}, \mathbf{b} \in \mathbb{R}^d$ are the parameters of $T$.

**Definition B.3** (Mahalanobis Distance). The Mahalanobis distance $\text{dist}_M$ between a point $\mathbf{x} \in \mathbb{R}^d$ and a Gaussian distribution $\mathcal{N}(\boldsymbol{\mu}, \boldsymbol{\Sigma})$ is defined as:

$$\text{dist}_M(\mathbf{x}) = \sqrt{(\mathbf{x} - \boldsymbol{\mu})^T \boldsymbol{\Sigma}^{-1} (\mathbf{x} - \boldsymbol{\mu})} . \tag{8}$$

**Definition B.4** ($\chi_d^2$-distribution). The Chi-squared distribution $\chi_d^2$ with $d$ degrees of freedom is the distribution of the sum of squares of $d$ independent standard Normal random variables.

### B.2 Properties

**Property B.5** (Lemma 5.3.2 (Casella & Berger, 2024)). If $Z \sim \mathcal{N}(0,1)$, then $Z^2 \sim \chi_1^2$; If $X_1, ..., X_d$ are independent and $X_i \sim \chi_1^2$, then $\sum_{i=1}^d X_i \sim \chi_d^2$.

**Property B.6.** The squared Mahalanobis distance $\text{dist}_M^2(\mathbf{x}) \sim \chi_d^2$, with $\mathbf{x} \sim \mathcal{N}(\boldsymbol{\mu}, \boldsymbol{\Sigma})$.

*Proof*: If $\mathbf{x} \sim \mathcal{N}(\boldsymbol{\mu}, \boldsymbol{\Sigma})$, then we have $\mathbf{z} = \boldsymbol{\Sigma}^{-\frac{1}{2}}(\mathbf{x} - \boldsymbol{\mu}) \sim \mathcal{N}(\mathbf{0}, \mathbf{I}_d)$ (Gut, 2009), such that:

$$\text{dist}_M^2(\mathbf{x}) = \mathbf{z}^T\mathbf{z} = \sum_{i=1}^{d} z_i^2 \tag{9}$$

where $z_i$ are independent standard Normal random variables. We have $\sum_{i=1}^{d} z_i^2 \sim \chi_d^2$ from Property B.5, which completes the proof.

### B.3  Lemmas

**Lemma B.7.** *Linear transform $T$ with invertible $\mathbf{W}$ on $\mathcal{G}_m^d$ preserves Mahalanobis distances.*

*Proof*: We denote the transformed GMM as $T(\mathcal{G}_m^d) = \{(w_j, \mathbf{W}\boldsymbol{\mu}_j + \mathbf{b}, \mathbf{W}\boldsymbol{\Sigma}_j\mathbf{W}^T)\}_{j=1}^{m}$, then with $\mathbf{x} \sim \mathcal{N}(\boldsymbol{\mu}_j, \boldsymbol{\Sigma}_j)$, for the transformed point $T(\mathbf{x})$ we have:

$$\text{dist}_M(T(\mathbf{x})) = \sqrt{(T(\mathbf{x}) - (\mathbf{W}\boldsymbol{\mu}_j + \mathbf{b}))^T(\mathbf{W}\boldsymbol{\Sigma}\mathbf{W}^T)^{-1}(T(\mathbf{x}) - (\mathbf{W}\boldsymbol{\mu}_j + \mathbf{b}))} \tag{10}$$

$$= \sqrt{(\mathbf{W}(\mathbf{x} - \boldsymbol{\mu}_j))^T(\mathbf{W}\boldsymbol{\Sigma}\mathbf{W}^T)^{-1}(\mathbf{W}(\mathbf{x} - \boldsymbol{\mu}_j))} \tag{11}$$

$$= \sqrt{(\mathbf{x} - \boldsymbol{\mu}_j)^T\mathbf{W}^T(\mathbf{W}^T)^{-1}\boldsymbol{\Sigma}^{-1}\mathbf{W}^{-1}\mathbf{W}(\mathbf{x} - \boldsymbol{\mu}_j)} \tag{12}$$

$$= \sqrt{(\mathbf{x} - \boldsymbol{\mu}_j)^T\boldsymbol{\Sigma}^{-1}(\mathbf{x} - \boldsymbol{\mu}_j)} = \text{dist}_M(\mathbf{x}) \ . \tag{13}$$

$\square$

**Lemma B.8.** *Linear transform $T$ with invertible $\mathbf{W}$ on $\mathcal{G}_m^d$ preserves the percentiles of the GMM.*

*Proof*: Let $\chi_d^2(\alpha)$ denote the $\alpha$-th percentile of $\chi_d^2$, such that for $X \sim \chi_d^2$:

$$\text{Prob}(X \le \chi_d^2(n)) = \frac{\alpha}{100} \ . \tag{14}$$

Based on Property B.6, we have $\text{Prob}(\text{dist}_M^2(\mathbf{x}) \le \chi_d^2(\alpha)) = \frac{\alpha}{100}$.

Let $\mathbf{x} \sim \mathcal{G}_m^d$, such that $\text{dist}_M^2(\mathbf{x}) > \chi_d^2(\alpha)$ for all $\mathcal{N}_j(\boldsymbol{\mu}_j, \boldsymbol{\Sigma}_j)$, which indicates that $\mathbf{x}$ is outside the $\alpha$-th percentile of $\mathcal{G}_m^d$. Since $\text{dist}_M(\mathbf{x})$ is preserved under $T$ (see Lemma B.7), then we conclude that the linear transform $T$ with invertible $\mathbf{W}$ preserves the percentiles of the GMM. $\square$

## C  Implementation details

### C.1  Hardware

We base our experiments on a NVIDIA RTX A6000 GPU with AMD EPYC 7742 64-Core Processors.

### C.2  Training and inference

We train our models for 200 epochs with the Adam optimizer (Kingma & Ba, 2017) and a `learning_rate` = 0.001, and test with the model corresponding to the lowest training loss. The size of our $D = \{20, 100\}$ model is 4.87M and 4.89M parameters, respectively. We show the training process of PFNs and our model in Algorithm 1.

**Dealing with varying dimensions and dataset size**  To handle input with varying number of $d$ features, we follow Müller et al. (2022). Specifically for $d < D$, we rescale the input with $\frac{D}{d}$ and pad the features to size $D$ with 0s; and for $d > D$, we randomly sample $D$ features out of $d$. In addition, FoMo-0D uses context size of up to 5K at inference, where for each test sample $\mathbf{x} \in \mathcal{D}_{\text{test}}$, we randomly sample (5K$-1$) points as $\mathcal{D}_{\text{train}}$ from datasets with $n > $ 5K.

---

**Algorithm 1:** Prior-fitting of a PFN (Müller et al., 2022) and ours

---

**Input** : A prior distribution over datasets $p(\mathcal{D})$, from which samples can be drawn and the number of datasets $Q$ to draw for one epoch, the number of training epochs $E$, the periodicity $P$, the number of unique datasets $q$, linear transformation $T$.

**Output** : A model $q_\theta$ that will approximate the PPD

**1** Initialize the neural network $q_\theta$;
**2** Initialize the epoch-level collection $\mathcal{C}_E = [\ ]$;
**3** **for** $i \leftarrow 1$ **to** $E$ **do**
**4**    **if** $i \leq P$ **then**
**5**       Initialize an empty buffer $\mathcal{B}_i = [\ ]$;
**6**       Initialize the dataset-level collection $\mathcal{C}_q = [\ ]$;
**7**       **for** $j \leftarrow 1$ **to** $Q$ **do**
**8**          **if** $j \leq q$ **then**
**9**             **Step 1**: sample $D_j := \mathcal{D}_{\text{train}} \cup \{(\mathbf{x}_k, y_k)\}_{i=k}^{|\mathcal{D}_{\text{test}}|} \sim p(\mathcal{D})$;
**10**             $\mathcal{C}_q \leftarrow \mathcal{C}_q + [D_j]$
**11**          **end**
**12**          **else**
**13**             $j \leftarrow j \mod q$
**14**             $D_j \leftarrow T(\mathcal{C}_q[j])$
**15**          **end**
**16**          **Step 2**: compute stochastic loss approximation $\bar{\ell}_\theta = \sum_{k=1}^{|\mathcal{D}_{\text{test}}|}(-\log q_\theta(y_k|\mathbf{x}_k, \mathcal{D}_{\text{train}}))$;
**17**          **Step 3**: update parameters $\theta$ with stochastic gradient descent on $\nabla_\theta \bar{\ell}_\theta$;
**18**          $\mathcal{B}_i \leftarrow \mathcal{B}_i + [D_j]$
**19**       **end**
**20**       $\mathcal{C}_E \leftarrow \mathcal{C}_E + [\mathcal{B}_i]$
**21**    **end**
**22**    **else**
**23**       $i \leftarrow i \mod P$
**24**       $\mathcal{B}_i \leftarrow \mathcal{C}_E[i]$
**25**       **for** $j \leftarrow 1$ **to** $Q$ **do**
**26**          $D_j \leftarrow T(\mathcal{B}_i[j])$
**27**          Perform **Step 2** and **Step 3**
**28**       **end**
**29**    **end**
**30** **end**

---

**Model architecture**   We use a 4-layer Transformer with hidden dimension `h_dim` = 256, a linear embedding layer at the input ($\mathbb{R}^D \rightarrow \mathbb{R}^{\text{h-dim}}$), and a 2-layer MLP layer at the output ($\mathbb{R}^{\text{h-dim}} \rightarrow \mathbb{R}^2$) for inlier vs. outlier binary classification. For each Transformer layer, we use `num_head` = 4 for each attention module and $R = 500$ for the router-based attention (Figure 2).

**Training loss**   In Figure 5, we plot the training loss of our $D = 100$ model trained with 8K unique datasets/epoch (denoted as "8K") versus 0.5K unique + 7.5K transformed datasets/epoch (denoted as "0.5K+T"), together with the $D = 20$ model trained with reuse periodicity $P = 1$ (denoted as "P=1", reusing the same 8K datasets across epochs) and $P = 1$ with transformation (denoted as "P=1+T", transforming the 8K datasets across epochs). Notice that the loss with transformation is slightly higher than no transformation (i.e., $D = 100$, "0.5K+T" vs. "8K", and $D = 20$, "P=1+T" vs. "P=1") across all 200 epochs, which is reasonable since the transformed datasets have non-diagonal covariances that make the learning task harder and thus result in a higher training loss. The training losses of FoMo-0D with $D = 100$ are also higher than with $D = 20$ since the subspace OD tasks are harder in higher dimensions.

**Inference time**   Figure 10 (left) showed the inference time of FoMo-0D on CPU, comparing typical attention versus the router-based attention (with $R = 500$ routers) under varying context sizes from 1K to 10K. The

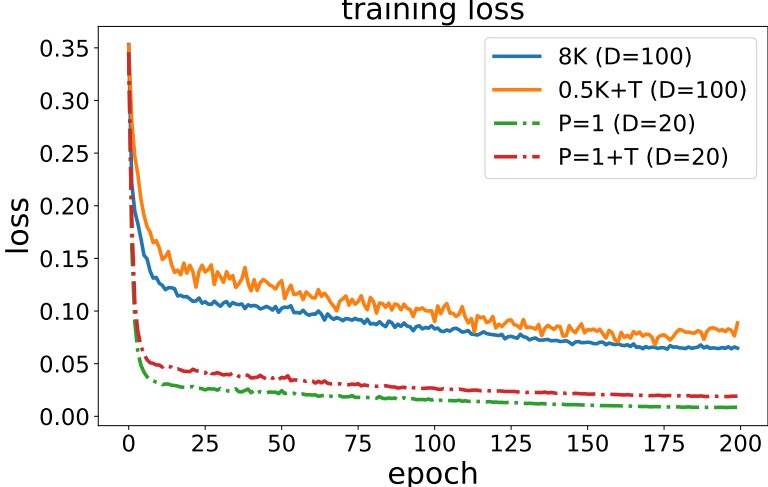

Figure 5: (best in color) Training loss of FoMo-0D ($D = 100$) with 8K unique datasets/epoch (in blue) and using 0.5K unique + 7.5K transformed datasets/epoch (in orange), and FoMo-0D ($D = 20$) with $P = 1$ (in green) and $P = 1$ with transformation (in red) over 200 epochs.

time is measured on CPU to clearly showcase the scalability trends; *quadratic* without routers and *linear* with routers.

Figure 6 shows the inference time on GPU. Notice that the time is much lower (in milliseconds), thanks to the Transformer architecture taking advantage of GPU parallelism, while the compute time for attention without routers continues to grow faster than that with routers.

In implementation, FoMo-0D (with $R = 500$ routers) uses inference context size of 5K by default, which takes about 7.7 ms per test sample on average.

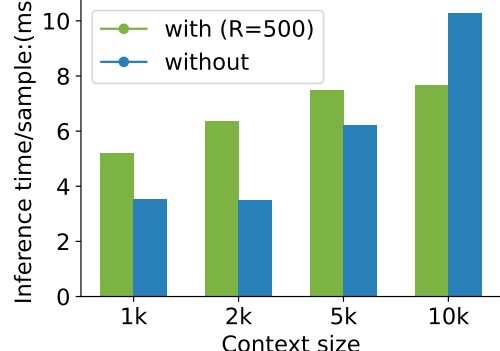

Figure 6: Inference time of FoMo-0D on *GPU* with vs. w/out router-based attention under varying context size.

## D   Detailed Experiment Setup

### D.1   Pre-training Dataset Synthesis

During pretraining, we generate unique GMM datasets by first drawing a configuration, including dimensionality $d \in [D]$, number of components $m \in [M]$, centers $\{\boldsymbol{\mu}_j\}_{j=1}^m$ (each $\boldsymbol{\mu}_j \in [-5, 5]^d$) and covariances $\{\boldsymbol{\Sigma}_j\}_{j=1}^m$ ($diag(\boldsymbol{\Sigma}_j) \in [-5, 5]^d$). We set $M = 5$ and vary $D \in \{20, 100\}$ to study pretraining with relatively small and high dimensional datasets, respectively. We synthesize inliers and outliers as described in Section 3.1.

We then sample $S = 5,000$ points that are within the $90th$ percentile of the GMM. To synthesize outliers, we "inflate" a *subset* of dimensions by randomly choosing $|\mathcal{K}| \in [D]$ dimensions and multiplying the corresponding variances by $\times 5$ (following (Han et al., 2022)), i.e. $5 \times \boldsymbol{\Sigma}_{j,kk}$'s for $k \in \mathcal{K}$, and then draw $S = 5,000$ samples from the inflated GMM that are outside the $90th$ percentile of the original GMM.

To speed up data synthesis via linear transformations, we first draw 500 unique datasets using $m \in [5]$ and $d \in \{1, 2, \ldots, 100\}$ (i.e. $5 \times 100$) and transform each one $15 \times$ using varying parameters $(\mathbf{W}, \mathbf{b})$ as described in Section 3.3.[8] This yields 8K unique datasets (500 original and 7,500 transformed) to use at one training epoch (over 1,000 steps with batch size $B = 8$). We repeat this process at each epoch, drawing 500 new datasets and transforming them to reach 8K datasets per epoch.

---

[8]It is important to ensure that the eigenvalues of $\mathbf{W}$ (i.e. variances) are not too small such that the dataset does not flatten in any direction. To this end, we draw a random orthonormal basis $\mathbf{U} \in [-1, 1]^{d \times d}$ and a diagonal $\boldsymbol{\Lambda}$ with eigenvalues $\lambda_{kk} \in ([-1, -0.1] \cup [0.1, 1])^d$, and obtain $\mathbf{W} = \mathbf{U}\boldsymbol{\Lambda}\mathbf{U}^T$. We also use $\mathbf{b} \in [-1, 1]^d$.

### D.2 Real-world Benchmark Datasets

While pretraining is purely on synthetic datasets, we evaluate FoMo-0D on **57** real-world datasets from the ADBench benchmark (Han et al., 2022) (see Table 20). They consist of 47 popular tabular outlier detection datasets, as well as 10 newly-constructed tabular datasets created from images and natural language tasks by using pretrained models to extract embeddings. We defer to the original paper for the details on these benchmark datasets.

We compare to DTE (Livernoche et al., 2024) and baselines therein as described next, thus, following their OD setting with inlier-only $\mathcal{D}_{\text{train}}$, we split each dataset five times into train/test using five different seeds and report the mean performance and its standard deviation. In particular, each random split designates 50% of the inliers as $\mathcal{D}_{\text{train}}$, while $\mathcal{D}_{\text{test}}$ contains the rest of the inliers and all the outlier samples. Note that while the baseline methods require model re-training and inference for each $\mathcal{D}_{\text{train}}/\mathcal{D}_{\text{test}}$ split, FoMo-0D uses the splits only for inference as $\mathcal{D}_{\text{train}}$ is merely passed as context.

Before passing the datasets as input to FoMo-0D, we perform a quantile transform such that the features follow a Normal distribution, to better align with the pretraining data from GMMs.

### D.3 Baselines

We compare FoMo-0D against **26** baselines, from classical/shallow methods to modern/deep models. Our baselines include all the baselines imported from one of the latest papers that proposed the SOTA diffusion-based model DTE (Livernoche et al., 2024), and its three variants; DTE-C, DTE-IG, and DTE-NP. Their baselines comprise all those in ADBench (Han et al., 2022); both classical ones ($k$NN (Ramaswamy et al., 2000), LOF (Breunig et al., 2000), iForest (Liu et al., 2008), HBOS (Goldstein & Dengel, 2012), etc.) and deep models (DeepSVDD (Ruff et al., 2018), DAGMM (Zong et al., 2018), DROCC (Goyal et al., 2020), etc.). They also include more recent approaches based on self-supervised learning (GOAD (Bergman & Hoshen, 2020), ICL (Shenkar & Wolf, 2022), SLAD (Xu et al., 2023), etc.), besides the four additional generative baselines: normalizing planar flows (Rezende & Mohamed, 2015), DDPM (Ho et al., 2020), VAE (Kingma, 2013) and GANomaly (Akcay et al., 2019). We defer to the original paper for additional details. Overall, our 26 baselines consist of the most recent, SOTA approaches for OD that span a diverse family (nonparametric, self-supervised, generative, etc.).

### D.4 Hyperparameters for Baselines

Table 4 gives the list of HP values we used to study the HP sensitivity/performance variability of the (from top to bottom) top-4 baselines.

Table 4: Top-4 baselines (from top to bottom) and hyperparameter (HP) configurations.

| Baseline | Hyperparameters |
|----------|-----------------|
| DTE-NP | $k \in \{5, 10, 20, 40, 50\}$ |
| $k$NN | $k \in \{5, 10, 20, 40, 50\}$ |
| ICL | `learning_rate` $\in \{10^{-1}, 10^{-2}, 10^{-3}, 10^{-4}, 10^{-5}\}$ |
| DTE-C | $k \in \{5, 10, 20, 40, 50\}$ |

### D.5 Ranking the 26 baselines

Figure 24 presents the visualization of the $p$-values of the pairwise Wilcoxon signed rank test w.r.t. AUROC among the baseline methods used by Livernoche et al. (2024). We rank these 26 baselines based on their mean $p$-value (i.e., row-wise average) against the other baselines.

### D.6 Comparison of top-4 baseline variants with varying HP configurations

Figure 25, 26, 27, 28 give the $p$-values, respectively comparing the variants of the top-4 baselines (DTE-NP, $k$NN, ICL, DTE-C) among themselves using different HP configurations, as well as the $^{\mathrm{avg}}$ model with the average performance across HPs. (Specifically for ICL, `learning_rate (lr)` $\in \{10^{-1}, 10^{-2}, 10^{-3}, 10^{-4}, 10^{-5}\}$; and for others, #nearest-neighbors $k \in \{5, 10, 20, 40, 50\}$). We find that for ICL, $\mathtt{lr} = 10^{-3}$ or $10^{-4}$ are preferable while those that are too small or too large perform poorly. For others, small $k \in \{5, 10\}$ tend to outperform larger $k \in \{40, 50\}$. Note that Livernoche et al. (2024) used $k = 5$ in their paper that proposed DTE (and variants) as well as the $k$NN baseline for fair comparison, while the DTE$^{\mathrm{avg}}$ and $k$NN$^{\mathrm{avg}}$ models across HP configurations perform subpar.

### D.7 Sampling time of $d$-dimensional GMM

Figure 7 shows the sampling time of drawing 10,000 points from different GMMs with increasing dimensionality $d = \{10, 20, ..., 200\}$. We parallelize the sampling process over 10 CPUs, where each CPU draws 1000 samples.

We observe that the sampling time grows nonlinearly as the number of dimensions increases, which suggests that it may incur considerable computational overhead to directly draw from the data prior over hundreds of thousands of training steps, motivating the use of our proposed on-the-fly linear transformation $T$ for scalability.

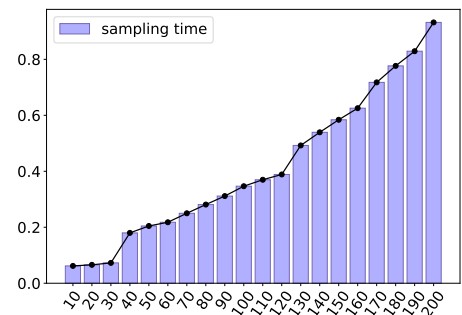

Figure 7: Sampling time (in seconds) of 10,000 points from GMMs with varying number of dimensions.

## E Qualitative Analysis on Sample-to-Sample Attention

We sample 50 inliers as context and 100 outliers from a 2-$d$ GMM using the $80th$ percentile as the labeling threshold, and visualize the top 5 inliers most attended by the 100 outliers based on the average (cross) attention weights over 4 heads from the last layer of `FoMo-0D` ($D = 100$), which accurately labeled all the 100 outliers. In Figure 8, the most frequently attended inliers are close to either the center of a Gaussian (e.g., $1st, 5th$) or the criterion (e.g., $3rd, 4th$), suggesting `FoMo-0D` tends to learn decision boundaries that reflect the prior data generation process.

For each outlier, we compute the sum of L2 distances to its top-5 attended inliers (`att`), the sum of L2 distances to 5 randomly chosen inliers (`rdm`), and the sum of L2 distances to top-5 inliers with highest likelihood under the GMM (`prob`). We perform Wilcoxon signed rank test between `att` and `rdm` (alternative: "less"), `att` and `prob` (alternative: "greater") over all the outliers, with a $p$-value of $4.4 \times 10^{-4}$ and 0.99, respectively, suggesting the distances based on attention weights

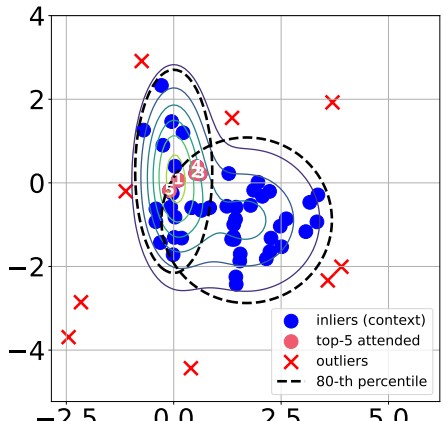

Figure 8: Top-5 attended inliers (all 50 inliers and only part of the outliers are shown for better visualization).

are significantly less than the random distances, and **not** significantly greater than the distances to inliers in high probability region.

We visualize the top-5 attended inliers for 3 outliers at different position of the 2-$d$ GMM in Figure 9. For a specific outlier, there is a similar trend of attending to the center of a Gaussian (as shown in Figure 8), besides, inliers that reflect the criterion boundary or are close to the outlier are actively attended (e.g., $3rd, 4th$ in the left, $1st$ in the middle, $2nd, 5th$ in the right), suggesting `FoMo-0D` is incorporating both boundary and nearest neighbor information dynamically for each outlier.

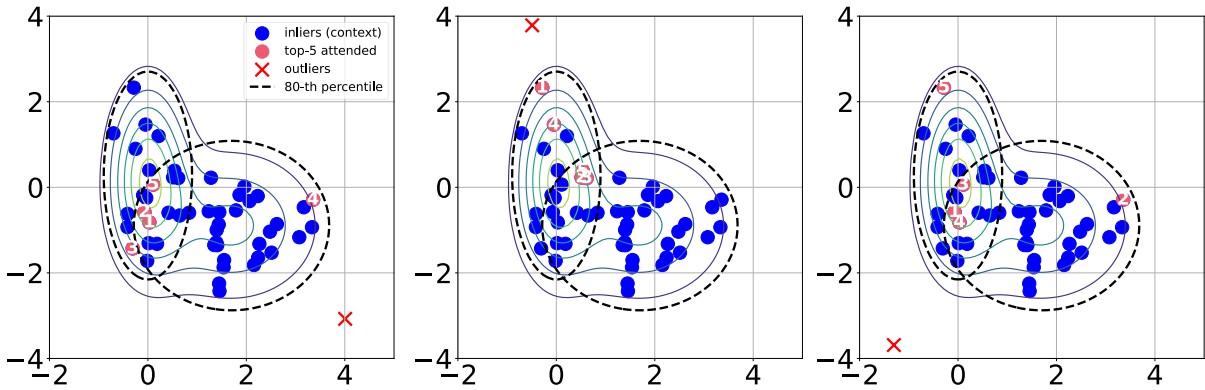

Figure 9: Top-5 attended inliers of 3 outliers at different positions of the GMM

## F    Ablation Analyses

In this section, we perform various ablations to study the effect of different design choices in FoMo-0D; namely, **F.1** maximum pretraining data dimensionality $D$, the number of routers $R$ on **F.2** cost and **F.3** performance, **F.4** context size (both for training and inference), **F.5** number of unique datasets used for pretraining (i.e., reuse periodicity $P$), data transformation $T$ during synthesis on **F.6** performance and **F.7** speed up, **F.8** data diversity and prolonged training, **F.9** quantile transforming the benchmark datasets preceding inference, and finally, **F.10** how different model sizes affect performance.

Unless stated otherwise, most ablation results are performed using FoMo-0D with $D = 20$, as it is faster to pretrain under these many varying settings.

### F.1    Effect of pretraining dimensionality $D$

***How does** FoMo-0D**'s generalization performance change by increasing dimensionality of the pretraining data***?

We start by comparing FoMo-0D pretrained on datasets with up to $D = 20$ versus $D = 100$ dimensions. Note that learning on higher dimensional datasets is harder, as evident from the relatively larger pretraining loss as shown in Appendix Figure 5. While the statement is accurate in general, it is also partly because subspace outliers "hide" better in higher dimensions.

Comparing Table 1 ($D = 100$) with Table 2 ($D = 20$) w.r.t. $p$-values over All datasets, we find that FoMo-0D at larger scale does better, where **all** $p$-values are larger for $D = 100$ than $D = 20$. We find that FoMo-0D with $D = 20$ performs well on datasets with $d \leq 20$ (i.e., "on its own game"), however beyond its pretraining setting, e.g. on datasets with $d \leq 50$, $D = 100$ is superior to $D = 20$ as shown in Appendix Table 13.

### F.2    Effect of routers on cost

***What is the running time and memory cost of** FoMo-0D **with & w/out router-based attention***?

Figure 10(left) shows the average inference time per test sample, comparing FoMo-0D using a router-based attention mechanism with $R = 500$ routers (in green) versus FoMo-0D using typical attention without any routers (in blue). As inference context size increases, running time for traditional attention grows quadratically while router mechanism scales linearly.[9]

Similarly, memory cost with routers is considerably lower when using routers, especially for larger context sizes, as shown in Figure 10(middle).

---

[9]Note that the inference time is reported on CPUs to show scalability. On GPUs, w/ 5K context size, see Appendix Figure 6, where typical attention takes advantage of parallelism (6.5ms), while router-based attention is slightly slower (7.7 ms w/ 500 routers) due to its **two** sequential self-attentions; see Eq.s (4) and (5).

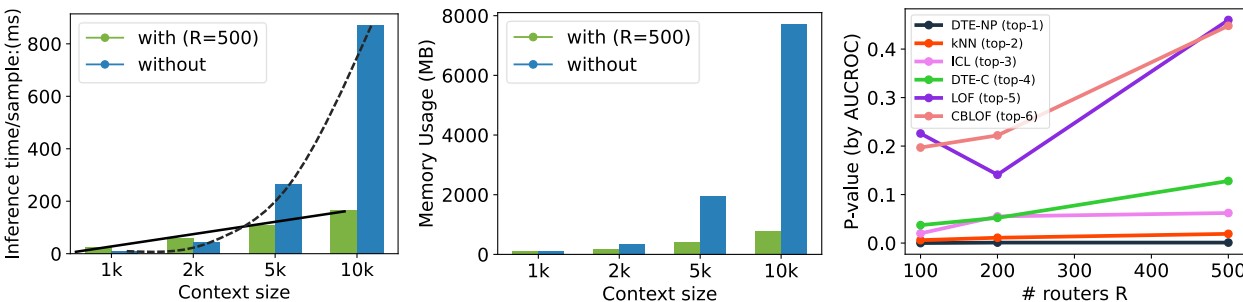

Figure 10: `FoMo-0D` w/ *router mechanism saves time and memory* while *more #routers perform better*, offering a cost-performance trade-off: (left) inference-time (ms) per sample and (middle) memory cost (MB) with & w/out routers by varying context size; (right) performance (based on *p*-value against top baselines, higher is better) vs. number of routers. (setting: $D = 20$, $P = 1$)

### F.3 Effect of routers on performance

***What is the impact of the number $R$ of routers (or representatives) on performance?***

Router-based mechanism allows to trade-off running time with expressiveness of the attention and hence performance. Figure 10(right) shows the *p*-values of the Wilcoxon signed rank test as the number of routers $R$ is increased from 100 to 200 and 500, comparing `FoMo-0D` to each of the top-6 baselines. We notice that `FoMo-0D` performance tends to increase monotonically with more routers.

### F.4 Effect of context size

***What is the impact of context size, both during model pretraining as well as during inference?***

To study how performance changes by context size, we train `FoMo-0D` with varying context size in {1K,2K,5K} and employ each pretrained model for inference with varying context size in {1K,2K,5K,10K}. Table 5 shows the results, where performance is depicted by the average rank of `FoMo-0D` (the lower, the better).

Table 5: Average rank (based on comparison to 30 baselines w.r.t. AUROC) of `FoMo-0D` across datasets under *different context sizes* for training and inference. Smaller ranks imply better performance. (setting: $D = 20$, $R = 500$, $P = 1$)

|          | Infer:1K | Infer:2K | Infer:5K | Infer:10K |
|----------|----------|----------|----------|-----------|
| Train:1K | 13.816   | 14.623   | 15.193   | 15.439    |
| Train:2K | 13.079   | 13.219   | 13.439   | 13.561    |
| Train:5K | 13.088   | 13.211   | 13.307   | 13.430    |

We find that training with a larger context improves performance at any inference context size. On the other hand, perhaps counter-intuitively, `FoMo-0D` with smaller inference context size does better. We conjecture that is because the #routers-to-context size ratio increases with a larger context size at inference, limiting the expressive power of the "bottleneck" attention mechanism. The pairwise statistical tests among the $3 \times 4 = 12$ models support these observations, as shown in Figure 11. Interestingly, when the training context size is large enough at 5K, inference with 10K samples generalizes beyond training with no statistical evidence for performance difference (at 0.05) from other inference context sizes.

### F.5 Effect of number of unique datasets

***How do `FoMo-0D` performances compare when pretrained on unique vs. reused datasets, via varying periodicity $P$?***

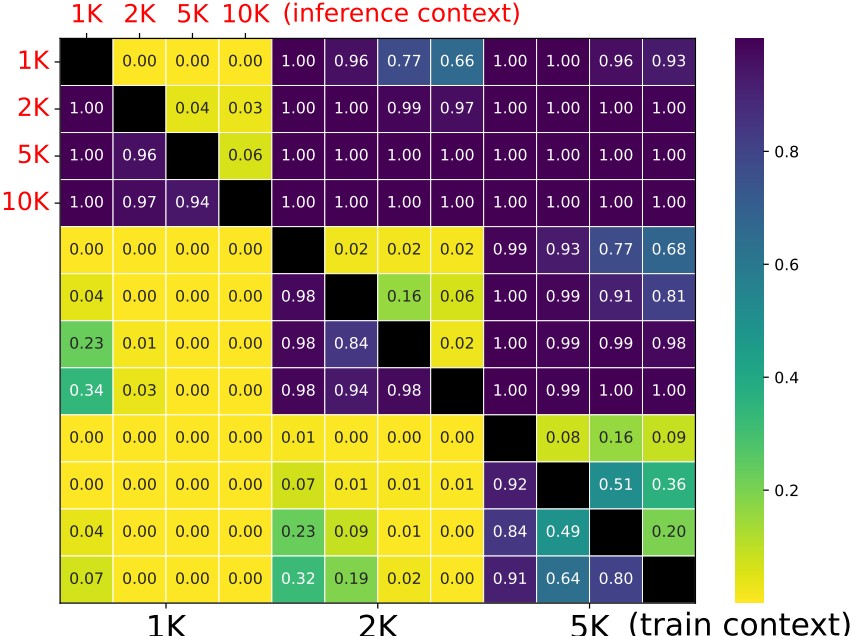

Figure 11: $p$-values of the pairwise Wilcoxon signed rank test between models (larger $p$ implies col-method is better than row-method) w/ different context sizes for **training** (1K/2K/5K, $1st/2nd/3rd$ four grids, in **black**) and inference (1K/2K/5K/10K, every $1st/2nd/3rd/4th$ grid, in red): Larger training context improves overall performance, while smaller inference context is preferable.

Table 6: Ablation results on dataset reuse across epochs with varying $P\in\{1,50,100\}$ show stable $p$-values against the top-5 baselines, where there is no statistical evidence to suggest performance difference between FoMo-0D with $D = 20$ and the top $3rd$ baseline at 0.05 w.r.t. pairwise Wilcoxon signed rank test comparisons, while it continues to significantly outperform the top $5th$ baseline (LOF) when $d \leq 50$. (setting: $D=20$, $R=500$, context size=5K, w/out transformation $T$)

| | $P = 1$ (#unique datasets: 8K) | | | | | $P = 50$ (#unique datasets: 8×50 = 400K) | | | | | $P = 100$ (#unique datasets: 8×100=800K) | | | | |
|---|---|---|---|---|---|---|---|---|---|---|---|---|---|---|---|
| top-5 | DTE-NP | kNN | ICL | DTE-C | LOF | DTE-NP | kNN | ICL | DTE-C | LOF | DTE-NP | kNN | ICL | DTE-C | LOF |
| All | 0.001 | 0.019 | 0.062 | 0.128 | 0.460 | 0.001 | 0.019 | 0.089 | 0.159 | 0.394 | 0.001 | 0.015 | 0.072 | 0.121 | 0.290 |
| $d \leq 20$ | 0.583 | 0.755 | 0.943 | 0.736 | **0.998** | 0.572 | 0.789 | **0.968** | 0.616 | **0.993** | 0.439 | 0.678 | **0.953** | 0.550 | **0.972** |
| $d \leq 50$ | 0.415 | 0.750 | 0.869 | **0.962** | **0.999** | 0.347 | 0.794 | 0.893 | 0.946 | **0.997** | 0.293 | 0.697 | 0.890 | 0.924 | **0.994** |

Next we study the effect of dataset *reuse at epoch level* (w/out transformation) on performance as presented in Section 3.3. We vary reuse periodicity $P$ in $\{1, 50, 100\}$, and accordingly, increase the number of unique datasets used for pretraining across epochs. As shown in Table 6, FoMo-0D (w/ $D = 20$) performs similarly with varying dataset reuse. In fact, it is competitive even with $P = 1$, remaining no different from the $3rd$ best baseline (ICL) across All (57) datasets, while significantly outperforming the top $5th$ (LOF) across (24) datasets with $d \leq 20$ as well as (38) with $d \leq 50$.

## F.6   Effect of transformation $T$ for synthesis

***How do* FoMo-0D *performances compare when pretrained on datasets with vs. w/out linear transformation*?**

Setting $P = 1$, we next study the impact of linear transformation $T$. Table 7 presents the results, where we compare reuse of the *same* 8K unique datasets across epochs (w/out $T$), versus *transforming* these datasets with $T$ at every epoch with different parameters (w/ $T$). FoMo-0D performance remains stable; no statistical evidence for performance difference from the top $3rd$ model on All datasets, while significantly outperforming the top $5th$ across those with $d \leq 20$ and $d \leq 50$. This suggests that $T$ can be employed without sacrificing performance to save time during pretraining.

Table 7: Ablation results on performance w/ & w/out linear transformation $T$ show stable $p$-values against the top-5 baselines, with no statistical evidence for performance difference between FoMo-0D with $D = 20$ and the top $3rd$ baseline at 0.05 w.r.t. pairwise Wilcoxon signed rank test comparisons. (setting: $D = 20$, $R = 500$, context size=5K, $P = 1$)

| top-5 | w/out transformation $T$ | | | | | w/ transformation $T$ | | | | |
|---|---|---|---|---|---|---|---|---|---|---|
| | DTE-NP | kNN | ICL | DTE-C | LOF | DTE-NP | kNN | ICL | DTE-C | LOF |
| All | 0.001 | 0.019 | 0.062 | 0.128 | 0.460 | 0.002 | 0.015 | 0.226 | 0.210 | 0.280 |
| $d \leq 20$ | 0.583 | 0.755 | 0.943 | 0.736 | **0.998** | 0.648 | 0.708 | **0.988** | 0.718 | **0.955** |
| $d \leq 50$ | 0.415 | 0.750 | 0.869 | **0.962** | **0.999** | 0.264 | 0.382 | **0.971** | 0.900 | **0.963** |

### F.7  Speed up by $T$

***What is the time saving on data synthesis with linear transformation?***

Figure 12 shows the distribution of pretraining running-time per epoch with and w/out data transformation. Specifically, we compare (left) generating 8K unique datasets/epoch on-the-fly and (right) first generating 500 unique datasets on-the-fly and then transforming each one 15 times using $T$ with different parameters to reach 8K datasets at each epoch.

Notice that pretraining with $T$ takes about 450 sec./epoch on average, while without $T$ it requires 1200 sec./epoch to generate 8K unique datasets and gradient descent across 1000 steps. Different from other ablation results, which are based on the $D = 20$ model, here we report the running times for our $D = 100$ model. Overall, our final FoMo-0D took ≈**25 hours** for pre-training (450 sec. ×200 epochs). Importantly, this is a one-time cost that amortizes across many downstream tasks with as low as **7.7 ms inference time** per test sample (see Table 3 and Appendix Figure 6).

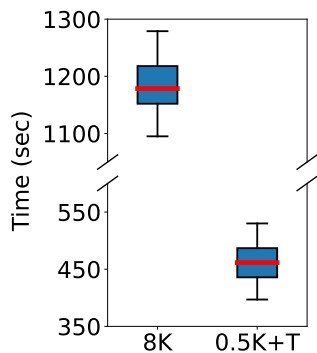

Figure 12: Runtime/epoch dist.n over 100 epochs for FoMo-0D ($D$=100) with (left) $P$=100, i.e. 8K unique datasets/epoch vs. (right) 0.5K unique+7.5K transformed datasets/epoch.

### F.8  Effect of data diversity and prolonged training

***How does* FoMo-0D*'s performance change by increasing pretraining data diversity and number of training epochs?***

Originally we have trained FoMo-0D w/ $D = 100$ using 0.5K unique + 7.5K transformed datasets over 200 epochs. As mentioned earlier, learning in higher dimensions tends to incur a larger loss in general but also specifically here, as subspace outliers are harder to detect in high dimensions.

Toward reducing the loss further, we resume the pretraining for another 100 epochs. Further, to simplify the tasks and thereby increase data diversity, we also decrease the inlier/outlier labeling percentile threshold from 90% to 80% during on-the-fly data generation in the last 100 epochs. In Figure 13, we present the training loss of FoMo-0D ($D = 100$) trained with 0.5K unique + 7.5K transformed datasets/epoch over 200 epochs ($90th$ percentile as labeling threshold) and then 100 additional epochs ($80th$ percentile as the threshold) to show how data diversity and amount affect model performance.

Figure 14 compares FoMo-0D's performance (w/ $D = 100$) to top-5 baselines w.r.t. $p$-values of the paired Wilcoxon signed rank test on datasets with $d \leq 100$, after the first 200 epochs versus after 300 epochs. The increase in all the $p$-values showcases the benefit of additional training.

### F.9  Effect of applying quantile transform on benchmark datasets

***What is the impact of quantile data transform preceding inference on performance?***

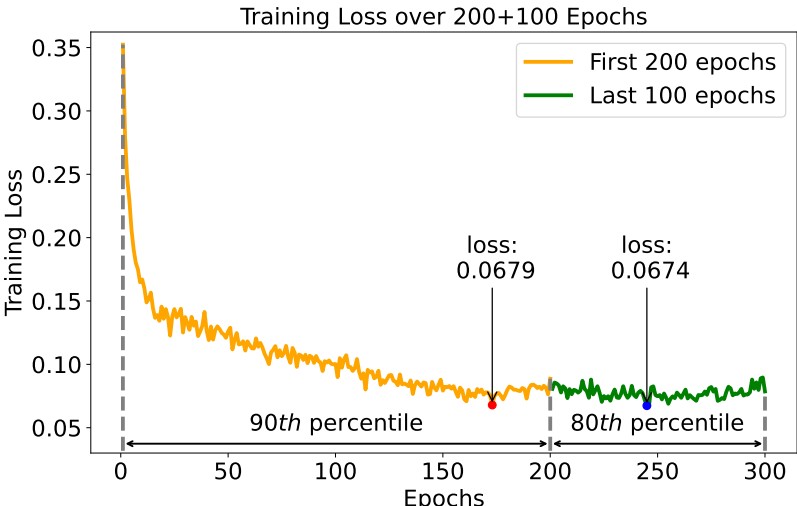

Figure 13: (best in color) Training loss of FoMo-0D ($D = 100$) with 0.5K unique + 7.5K transformed datasets/epoch for 200 epochs (in orange), followed with additional 100 epochs of training (in green). For the first 200 epochs we train with $90th$ percentile as the inlier/outlier threshold, which we reduce to $80th$ in the subsequent 100 epochs.

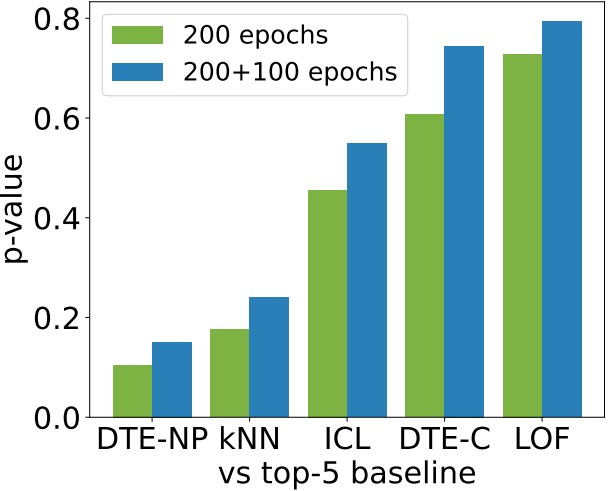

Figure 14: $p$-values increase with additional 100 epochs of pretraining, i.e. FoMo-0D w/ $D = 100$ performs better against top-5 baselines on datasets w/ $d \leq 100$.

We pretrain FoMo-0D on synthetic datasets from a simple data prior based on GMMs. The real-world benchmark datasets, on the other hand, may exhibit features with distributions different from Gaussians. To close the gap, we apply a quantile transform (denoted QT) on the benchmark datasets prior to feeding them to FoMo-0D for inference, which transforms the features to exhibit a more Gaussian-like probability distribution.

Figure 15 compares the performance of three FoMo-0D w/ $D = 100$ variants with and w/out QT against the top-5 baselines w.r.t. the $p$-values of the paired Wilcoxon signed rank test. FoMo-0D tends to perform better as suggested by larger $p$-values when QT is applied.

### F.10 Effect of Model Size

To understand how the performance of FoMo-0D scales with model sizes, we vary the number of transformer layers $L = 1, 2, 3$, where the default is $L = 4$ (see in **Model architecture**). We present the p-values of

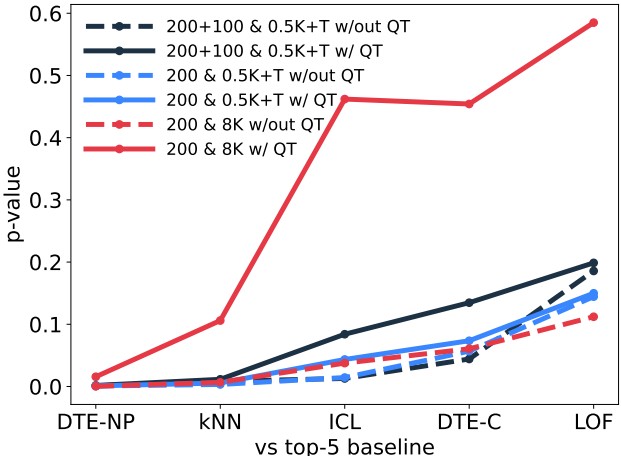

Figure 15: $p$-values increase, i.e. FoMo-0D performance improves, against top-5 baselines with quantile transform (QT) preceding inference, for 3 different settings of FoMo-0D w/ $D = 100$.

Table 8: $p$-values of the one-sided Wilcoxon signed rank test, comparing FoMo-0D (with $D = 100$) to **top 10 baselines** with default hyperparameters (HPs), and **top $4^{\mathrm{avg}}$** baselines[6] with **avg.** performance over varying HPs (denoted w/ $^{\mathrm{avg}}$) over all (57) datasets. FoMo-0D improves (i.e. higher $p$-values) as number of transformer layers $L$ increases. At $L = 1$, in-context learning ability appears to be quite limited. (setting: $D = 100$, $R = 500$, train/inference context size=5K, w/ quantile transform, $\alpha = 0.05$)

| FoMo-0D | #parameters | DTE-NP | $k$NN | ICL | DTE-C | LOF | CBLOF | Feat.Bag. | SLAD | DDPM | OCSVM | DTE-NP$^{\mathrm{avg}}$ | $k$NN$^{\mathrm{avg}}$ | ICL$^{\mathrm{avg}}$ | DTE-C$^{\mathrm{avg}}$ |
|---|---|---|---|---|---|---|---|---|---|---|---|---|---|---|---|
| $L=1$ | 1.34M | $1.66 \times 10^{-9}$ | $4.30 \times 10^{-9}$ | $1.25 \times 10^{-6}$ | $1.34 \times 10^{-7}$ | $5.08 \times 10^{-6}$ | $5.44 \times 10^{-8}$ | $1.31 \times 10^{-4}$ | $1.07 \times 10^{-6}$ | $1.30 \times 10^{-6}$ | $1.46 \times 10^{-7}$ | $2.68 \times 10^{-9}$ | $1.53 \times 10^{-8}$ | $2.13 \times 10^{-7}$ | 0.395 |
| $L=2$ | 2.52M | 0.006 | 0.036 | 0.157 | 0.259 | 0.442 | 0.557 | 0.703 | 0.431 | 0.759 | 0.805 | 0.021 | 0.134 | 0.333 | 1.000 |
| $L=3$ | 3.70M | 0.016 | 0.098 | 0.372 | 0.579 | 0.572 | 0.871 | 0.808 | 0.652 | 0.921 | 0.961 | 0.085 | 0.335 | 0.652 | 1.000 |
| $L=4$ | 4.89M | 0.016 | 0.106 | 0.462 | 0.454 | 0.585 | 0.750 | 0.823 | 0.759 | 0.901 | 0.895 | 0.112 | 0.315 | 0.670 | 1.000 |

different model sizes in Table 8, where a p-value $> 0.05$ means no statistical evidence to suggest performance difference between FoMo-0D and the compared baseline. We can observe that FoMo-0D is not comparable to the top-10 baselines with one layer, and shows improved performance (i.e., p-value increases and becomes larger than 0.05) as the number of layers increases, which suggests that scaling up the model size might help improve FoMo-0D's performance. On the other hand, we can see that the detection performance didn't increase a lot from $L = 3$ to 4, which suggests there exist diminishing returns in scaling the model size, and to further improve performance, one has to consider other methods (e.g., increase prior complexity, more pre-training epochs) besides scaling up only the model size.

## G    Generalization Analyses

### G.1    Generalization to Out-of-Distribution Synthetic Datasets

We conduct analyses to understand FoMo's ability to generalize on out-of-distribution synthetic GMM datasets. Besides the in-distribution setting for pre-training (i.e., $\boldsymbol{\mu} \in [-5, 5], \boldsymbol{\Sigma} \in (0, 5], m \leq 5, d \leq 100$), we consider the following out-of-distribution settings: **(a)** mean and covariance significantly out of range, with $\boldsymbol{\mu} \in [-50, -5] \cup [5, 50], \boldsymbol{\Sigma} \in [5, 50]$, denoted as "$|\boldsymbol{\mu}|, |\boldsymbol{\Sigma}| \in [5, 50]$"; **(b)** number of clusters significantly out of range, denoted as "$m \in [5, 50]$"; **(c)** number of dimensions significantly out of range, denoted as "$d \in [100, 500]$"; **(d)** binary outliers with values either 0 or 1 in one dimension from the sub-dimensions, denoted as "binary"; **(e)** "all", which combines all the variants above. For each setting, we generate 1000 datasets with random seeds from 0 to 999, where on each dataset, we simulate 1000 test points with an outlier rate of 5% and evaluate FoMo-0D with a context length of 5000. We present the results with averaged performance over 1000 datasets for each setting in Table 9.

Table 9: Average metric score $\pm$ standard dev. over 1000 seeds for different out-of-distribution (OOD) synthetic GMMs. FoMo-0D remains robust against OOD test datasets as in (a)–(d), maintaining similar performance to in-distribution performance (top). Performance is affected more when datasets are OOD w.r.t. multiple factors combined as in (e).

| Dataset | AUROC | AUCPR | F1 |
|---|---|---|---|
| ID: in-distribution | $98.55 \pm 2.73$ | $91.17 \pm 13.07$ | $86.74 \pm 15.43$ |
| **(a)** OOD w.r.t. $|\boldsymbol{\mu}|, |\boldsymbol{\Sigma}| \in [5, 50]$ | $94.79 \pm 7.53$ | $80.62 \pm 21.19$ | $76.32 \pm 19.85$ |
| **(b)** OOD w.r.t. $m \in [5, 50]$ | $97.69 \pm 3.59$ | $86.72 \pm 15.57$ | $81.20 \pm 16.23$ |
| **(c)** OOD w.r.t. $d \in [100, 500]$ | $96.22 \pm 9.01$ | $86.37 \pm 23.27$ | $83.23 \pm 22.08$ |
| **(d)** OOD w.r.t. binary variable | $100.00 \pm 0.00$ | $100.00 \pm 0.06$ | $99.97 \pm 0.34$ |
| **(e)** OOD w.r.t. all combined | $85.44 \pm 16.96$ | $64.17 \pm 35.07$ | $63.99 \pm 33.53$ |

We can observe different extents of performance degradation when applying out-of-distribution variations. Compared to other single variations, FoMo-0D seems to suffer more from inflating the mean and covariances, as due to the significant deviation in the parameters of the GMMs, inliers generated under such a setting are seemingly "outliers" w/o any reference points. Surprisingly, although FoMo-0D is only trained on continuous data, it can almost perfectly classify binary outliers hidden in one of the sub-dimensions, suggesting FoMo-0D could potentially generalize to discrete data at test time.

However, with all variations added, FoMo-0D becomes less capable compared to one single out-of-distribution variation, although there might exist some signals (e.g., binary labels) in favor of its decision-making process, for which training a powerful model with more comprehensive priors could possibly alleviate the issue.

### G.2 Generalization to Out-of-GMM-Distribution Real-World Datasets

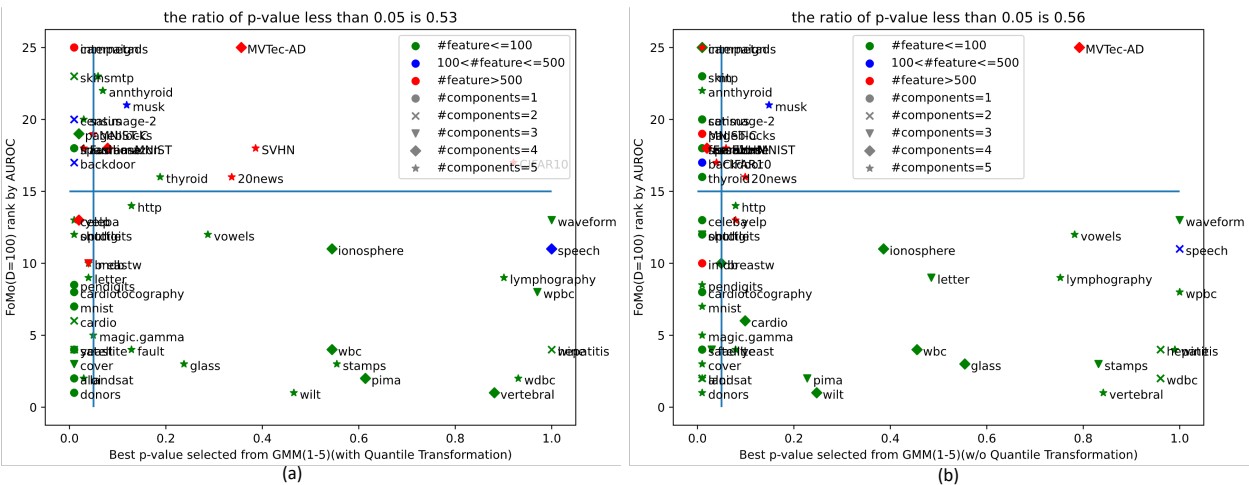

Figure 16: Goodness of fit test results for 57 datasets in ADBench. The x-axis is the $p$-value of a dataset, where a small $p$-value indicates GMMs are not a good fit for the dataset, and the y-axis is the rank of FoMo-0D ($D$=100) on that dataset (the smaller the better). We plot $p$-value = 0.05 (vertical) and rank = 15 (horizontal) as references. The left figure (a) is with quantile transformation while figure (b) is without quantile transformation. We use different colors to represent datasets at different dimensions, and use different markers to represent different numbers of clusters.

To understand how FoMo-0D generalizes from the pre-training GMM data priors to complex real-world datasets when performing zero-shot OD, we conduct the goodness of fit test Huber-Carol et al. (2012) for datasets in ADBench. We fit GMMs to each real-world dataset $D_{\mathrm{real}}$ with up to 5 components (as with our pre-training datasets), then sample $D_{\mathrm{syn}}$ from the best-parameter-fitted GMM and perform a two-sample

test[10] on $D_{\text{real}}$ and $D_{\text{syn}}$, with the null hypothesis that they come from the same distribution. A smaller $p$-value ($\leq 0.05$) of such a test provides evidence toward rejecting the null, which suggests GMM is not a good fit for the dataset (i.e., the pre-training data distribution is different from the test data).

We present the results in Figure 16, depicting the $p$-value (of the goodness-of-GMM-fit test) vs. FoMo-0D 's performance rank (the lower the better) among 30 baselines. We report the result both with quantile transformation in Figure 16(a) and without quantile transformation in 16(b). Since the two figures are highly similar, our next analysis will primarily focus on the figure with quantile transformation to align with our model's implementation. We plot the vertical and horizontal lines as $p$-value $= 0.05$ and rank $= 15$. For $p$-value $\geq 0.05$ and rank $< 15$, we observe that performance is good on datasets with relatively large $p$-value where we cannot reject the null (i.e. GMM is a relatively good fit). This is where arguably FoMo-0D recalls its data prior distribution and generalizes to datasets similar to those seen during pretraining. We also see, for $p$-value $< 0.05$ and rank $\geq 15$, datasets with relatively poor performance where we can reject the null (i.e. GMM is not a good fit). These can be attributed to falling short in generalization to OOD datasets.

On the other hand, we observe many datasets concentrate on $p$-value $< 0.05$ and rank $< 15$, where $p$-value is small (GMM not a good fit) yet the performance is competitive — those are the datasets on which FoMo-0D is likely to have achieved out-of-distribution generalization. It remains an open (theoretical) question to understand what (algorithm, if any) FoMo-0D might have learned that generalizes to out-of-distribution datasets. It is also an open (empirical) quest to explore whether a more complex data prior, beyond GMMs, could further push the performance up and by how much.

### G.3 Generalization to Out of Distribution (OOD) Detection Tasks

We further evaluate FoMo-0D on more complex datasets (e.g., ImageNet-level). Specifically, we employ OpenOOD Zhang et al. (2023), an out-of-distribution (OOD) detection benchmark, where models are trained on labeled in-distribution datasets, with $K$ known classes, and then evaluated on out-of-distribution datasets, aiming to detect $K+1, K+2, ...$ novel classes. Although OOD detection is inherently different from OD, we can construct an OD dataset from OOD datasets, treating all $K$ class samples as inliers and the $K+1, K+2, ...$ OOD samples as outliers. For the in-distribution datasets, we choose ImageNet1K, which contains 1000 categories of images, and ImageNet200, a subset of ImageNet1K containing 200 categories. We further choose SSB-hard, NINCO, iNaturalist, Textures, and OpenImage-O as the out-of-distribution datasets, which gives us a total number of $2 \times 5 = 10$ datasets that are ImageNet-level complex.

Following Han et al. (2022), we create 10 new OD datasets from OpenOOD containing 10,000 samples with 5% outliers, and use the embedding from the last average pooling layer of ResNet18 He et al. (2016) as the feature (512) for each sample. Comparing FoMo-0D with the top-4 (on our original testbed) baselines in the order of: DTE-NP, $k$NN, ICL, DTE-C, we follow Livernoche et al. (2024) and report mean (standard dev.) over 5 runs (seed=0/1/2/3/4) on each dataset. We present the results with in-distribution datasets being ImageNet200 and ImageNet1K in Table 10 and 11, respectively.

Table 10: Average AUROC score $\pm$ standard dev. over five seeds for in-distribution dataset being **ImageNet200**. We use blue and green respectively to mark the top-1 and the top-2 method.

| dataset | DTE-NP | kNN | ICL | DTE-C | FoMo-0D |
|---|---|---|---|---|---|
| ssb-hard | $58.03 \pm 0.00$ | $58.14 \pm 0.00$ | $60.52 \pm 0.25$ | $60.74 \pm 1.88$ | $58.34 \pm 1.55$ |
| ninco | $53.28 \pm 0.00$ | $54.14 \pm 0.00$ | $59.56 \pm 0.63$ | $58.83 \pm 1.54$ | $55.16 \pm 2.19$ |
| inaturalist | $29.38 \pm 0.00$ | $29.51 \pm 0.00$ | $35.96 \pm 1.10$ | $41.77 \pm 2.84$ | $38.85 \pm 3.29$ |
| textures | $59.28 \pm 0.00$ | $59.91 \pm 0.00$ | $66.40 \pm 0.69$ | $70.33 \pm 3.18$ | $59.89 \pm 2.07$ |
| openimageo | $52.82 \pm 0.00$ | $53.79 \pm 0.00$ | $55.20 \pm 0.69$ | $59.09 \pm 1.50$ | $54.77 \pm 1.19$ |
| average | $50.56$ | $51.10$ | $55.53$ | $58.15$ | $53.40$ |

We further report the $p$-value of the Wilcoxon signed rank test between the baselines and FoMo-0D on the 10 datasets from OpenOOD, as well as on the expanded benchmark combining those 10 with our original

---

[10]We use e-test from https://www.rdocumentation.org/packages/energy/versions/1.7-11/topics/eqdist.etest

Table 11: Average AUROC score $\pm$ standard dev. over five seeds for in-distribution dataset being **ImageNet1K**. We use blue and green respectively to mark the top-1 and the top-2 method.

| dataset | DTE-NP | kNN | ICL | DTE-C | FoMo-0D |
|---|---|---|---|---|---|
| ssb-hard | $55.63 \pm 0.00$ | $55.94 \pm 0.00$ | $58.79 \pm 1.20$ | $59.17 \pm 1.82$ | $56.73 \pm 2.65$ |
| ninco | $48.23 \pm 0.00$ | $49.10 \pm 0.00$ | $55.25 \pm 0.87$ | $57.60 \pm 3.93$ | $52.70 \pm 2.70$ |
| inaturalist | $30.24 \pm 0.00$ | $30.28 \pm 0.00$ | $35.03 \pm 1.42$ | $41.96 \pm 3.13$ | $38.94 \pm 4.59$ |
| textures | $54.38 \pm 0.00$ | $55.43 \pm 0.00$ | $61.30 \pm 0.95$ | $63.10 \pm 3.72$ | $55.18 \pm 2.92$ |
| openimageo | $54.31 \pm 0.00$ | $54.91 \pm 0.00$ | $54.02 \pm 0.43$ | $58.71 \pm 2.08$ | $56.95 \pm 3.89$ |
| average | 48.56 | 49.13 | 52.88 | 56.11 | 52.10 |

ADBench (10+57) in Table 12. In terms of metric values, FoMo-0D performs 2nd or 3rd best across OOD datasets. $p$-values show that it significantly outperforms DTE-NP and kNN (p>0.95, such that the p-value < 0.05 for rejecting the null hypothesis and accepting the alternative hypothesis that the "baseline-minus-FoMo-0D" gap is smaller than zero) and is no different from ICL (2nd best after DTE-C). These results demonstrate that FoMo-0D generalizes beyond OD datasets and maintains strong zero-shot OD performance on complex, ImageNet-level OOD benchmarks.

Table 12: $p$-value of the Wilcoxon signed rank test (alternative: "greater") between baselines and FoMo-0D on OpenOOD and combined benchmark on AUROC. A small $p$-value ($\leq 0.05$) means that there is statistical evidence for the alternative hypothesis such that baselines achieve higher metric performance than FoMo-0D.

| method | DTE-NP | kNN | ICL | DTE-C |
|---|---|---|---|---|
| OpenOOD | 1 | 0.9951 | 0.1875 | 0.0009 |
| OpenOOD+ADBench | 0.1271 | 0.3308 | 0.3153 | 0.1265 |

Interestingly, we observe that ICL and DTE-C outperform DTE-NP and $k$NN on the OpenOOD datasets, whereas on ADBench, DTE-NP and $k$NN are the top-2 methods outperforming ICL and DTE-C. We hypothesize this is because it is harder for non-parametric methods like DTE-NP and kNN to estimate meaningful decision boundaries in high dimensions (e.g., 512). In contrast, the performance of FoMo-0D is consistently competitive, where the $p$-values on the combined testbed (OpenOOD+ADBench) show that FoMo-0D is as competitive as all the top baselines across 67 diverse datasets, while maintaining zero-shot detection ability.

## H    Performance Profile Plots

To enable a comprehensive comparison of different methods, we plot rank (w.r.t. AUROC, lower is better) distribution in Figure 17, and adopt $\tau$ performance profile plots as described in Dolan & Moré (2002). These plots display the cumulative distribution of the $\tau$ metric—which quantifies suboptimality relative to the best-performing method. By computing sorted $\tau$ values along with their cumulative probabilities, we then use the area under each CDF curve as a global performance indicator, where a larger area signifies superior performance.

Figure 18, Figure 19, and Figure 20 illustrate performance profile plots of FoMo-0D and other baselines across all datasets. The results show that FoMo-0D (D=100) ranks at **top-5 (w.r.t. AUROC), top-3 (w.r.t. AUPR) and top-1 (w.r.t. F1)**, respectively, outperforming many baselines.

Moreover, the performance of FoMo-0D (D=100) is even better (i.e., ranked within top-2) when tested on datasets with dimensions less than 100. As shown in Figure 21, Figure 22, and Figure 23, the area under the curve of FoMo-0D (D=100) ranks at **top-1 (w.r.t. AUROC), top-2 (w.r.t. AUPR) and top-2 (w.r.t. F1)**, respectively, under this setting.

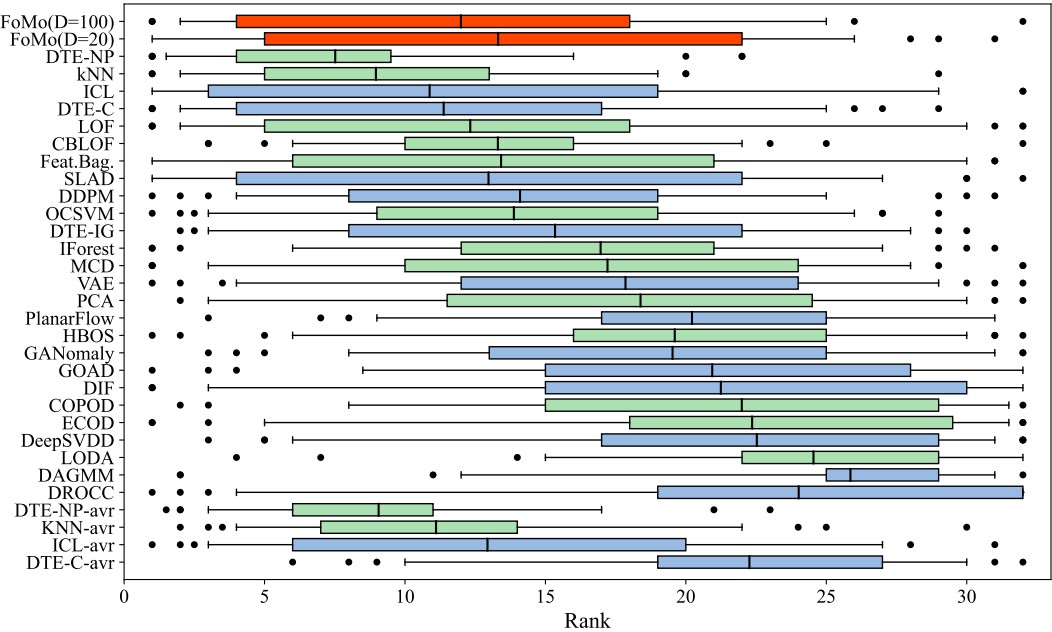

Figure 17: (best in color) Rank (w.r.t. AUROC, lower is better) distribution across all **57** real-world datasets shown via boxplots for (from top to bottom) `FoMo-0D` in red, all **26** baselines ordered by mean $p$-value[6] (shallow and deep baselines in green and blue), and **top 4** baselines' $^{\mathrm{avg}}$ variants. The vertical line depicts the mean, the box shows the 25-75%, bars range 5-95%, and circles show the datasets at the tails.

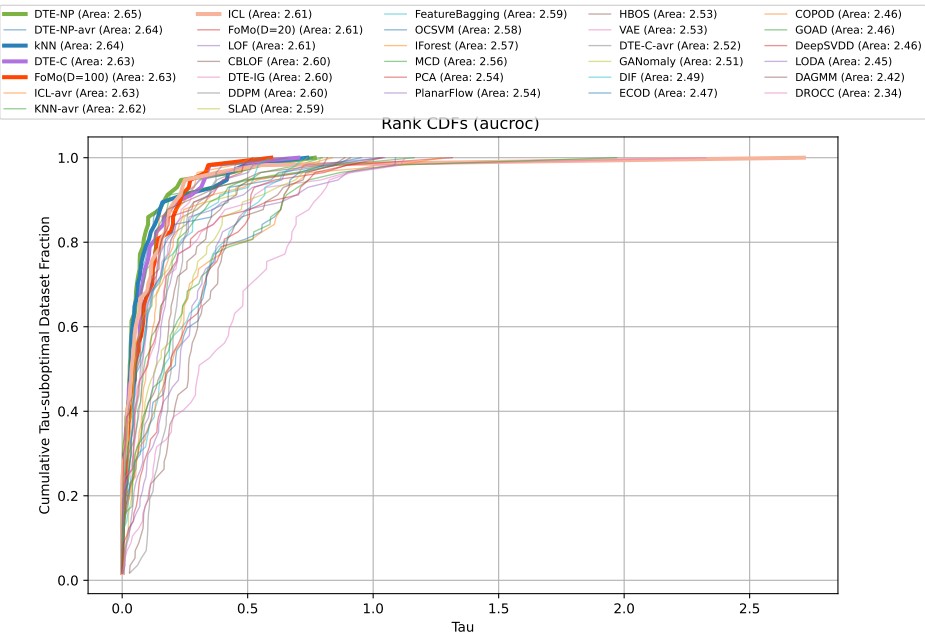

Figure 18: `FoMo-0D` ranks in **top-5** based on the performance profile plots of all detectors w.r.t. **AUROC** across **all datasets**. In the plot, x-axis represents the $\tau$ values—performance ratios that compare each method's metric to the best performance achieved, while y-axis displays the cumulative fraction of test datasets for which a method's performance is within the $\tau$ value. We use the area under each CDF curve as a global performance indicator, where a larger area signifies superior performance.

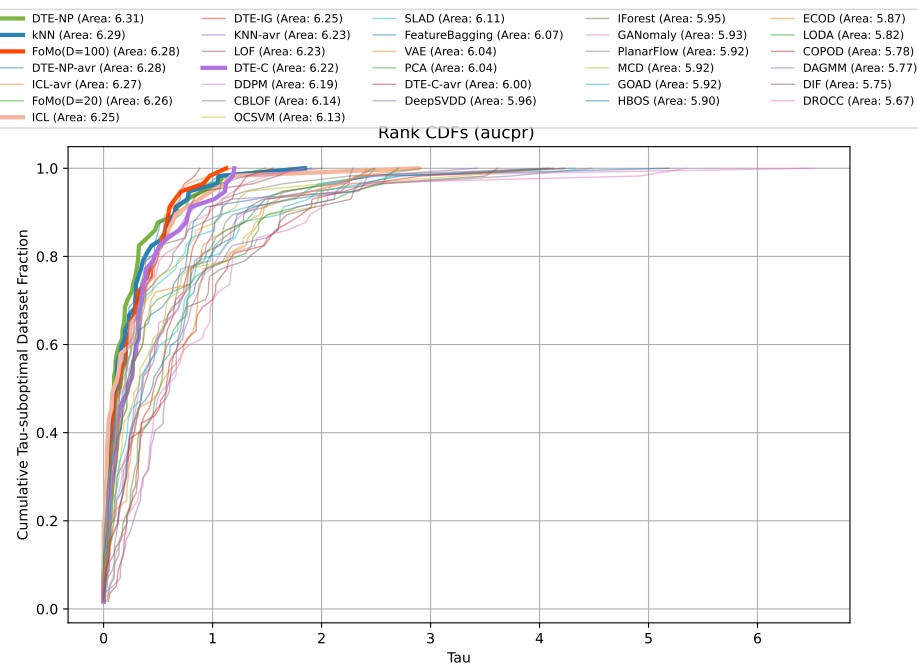

Figure 19: `FoMo-0D` ranks at **top-3** based on the performance profile plots of all detectors w.r.t. **AUPR** across **all datasets**. In the plot, x-axis represents the $\tau$ values—performance ratios that compare each method's metric to the best performance achieved, while y-axis displays the cumulative fraction of test datasets for which a method's performance is within the $\tau$ value. We use the area under each CDF curve as a global performance indicator, where a larger area signifies superior performance.

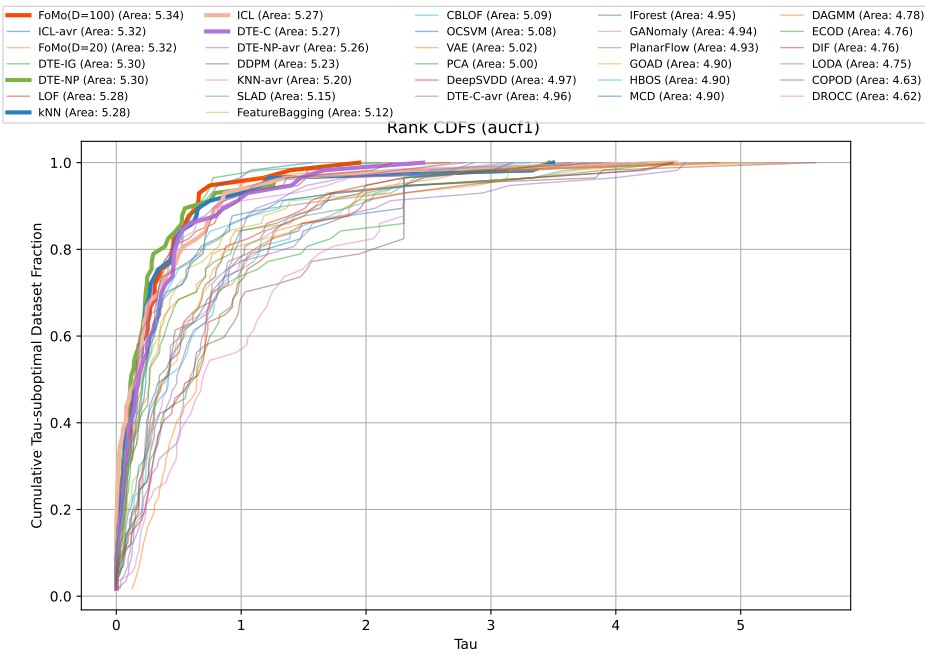

Figure 20: `FoMo-0D` ranks at **top-1** based on the performance profile plots of all detectors w.r.t. F1 across all datasets. In the plot, x-axis represents the $\tau$ values—performance ratios that compare each method's metric to the best performance achieved, while y-axis displays the cumulative fraction of test datasets for which a method's performance is within the $\tau$ value. We use the area under each CDF curve as a global performance indicator, where a larger area signifies superior performance.

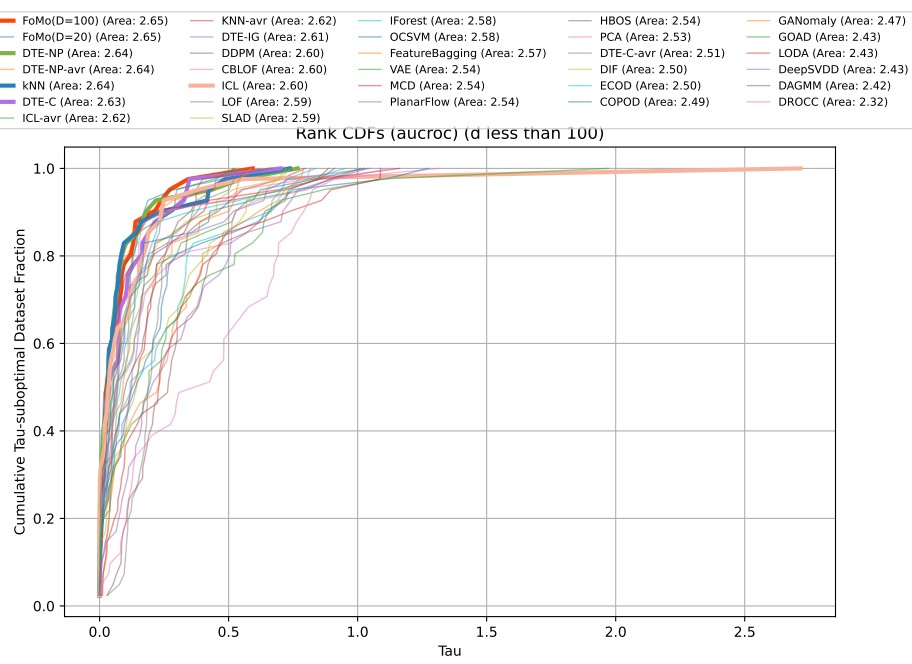

Figure 21: `FoMo-0D` (D=100) ranks at **top-1** based on the performance profile plots of all detectors w.r.t. **AU-ROC** in *datasets with dimensions less than $d \leq 100$*. In the plot, x-axis represents the $\tau$ values—performance ratios that compare each method's metric to the best performance achieved, while y-axis displays the cumulative fraction of test datasets for which a method's performance is within the $\tau$ value. We use the area under each CDF curve as a global performance indicator, where a larger area signifies superior performance.

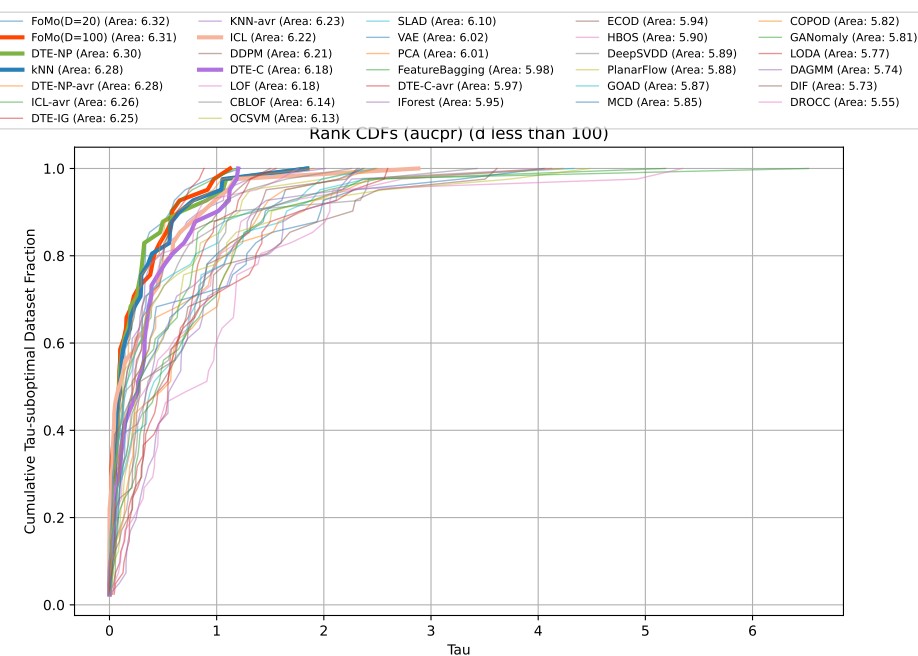

Figure 22: `FoMo-0D` (D=100) ranks at **top-2** based on the performance profile plots of all detectors w.r.t. **AUPR** in *datasets with dimensions less than $d \leq 100$*. In the plot, x-axis represents the $\tau$ values—performance ratios that compare each method's metric to the best performance achieved, while y-axis displays the cumulative fraction of test datasets for which a method's performance is within the $\tau$ value. We use the area under each CDF curve as a global performance indicator, where a larger area signifies superior performance.

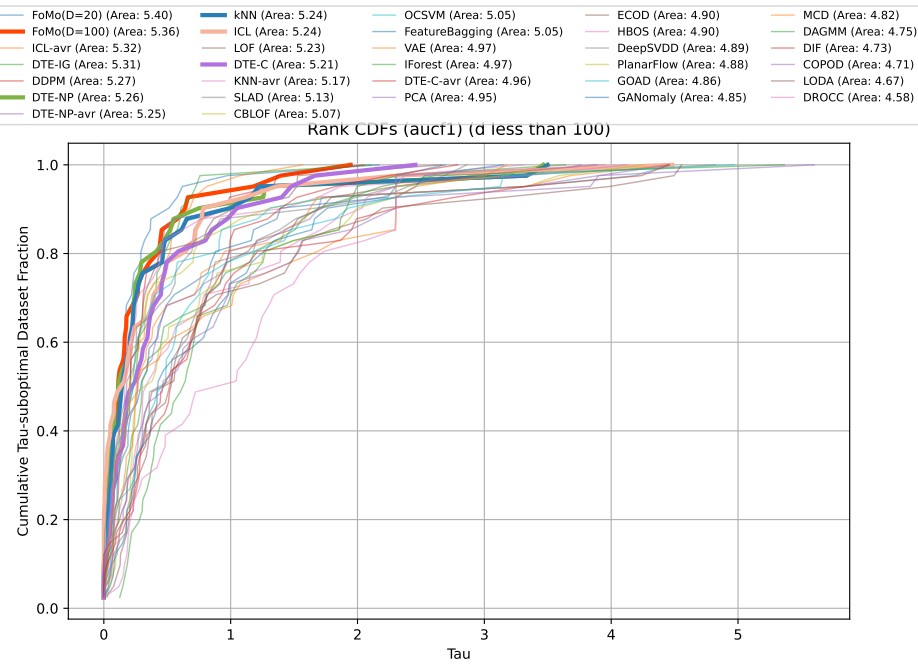

Figure 23: FoMo-0D (D=100) ranks at **top-2** based on the performance profile plots of all detectors w.r.t. **F1** in *datasets with dimensions less than $d \leq 100$*. In the plot, x-axis represents the $\tau$ values—performance ratios that compare each method's metric to the best performance achieved, while y-axis displays the cumulative fraction of test datasets for which a method's performance is within the $\tau$ value. We use the area under each CDF curve as a global performance indicator, where a larger area signifies superior performance.

## I Full Results

Tables 14.1 & 14.2, 15.1 & 15.2, and 16.1 & 16.2 respectively show the AUROC, AUPR, and F1 scores of the top-4 baselines, DTE-NP, $k$NN, ICL, and DTE-C as well as their corresponding $^{\text{avg}}$ model with the average performance across HPs, as listed in Table 4.

Tables 17.1&17.2, 18.1&18.2, and 19.1&19.2 respectively show the AUROC, AUPR, and F1 scores of all methods across all benchmark datasets. In all these tables, the last four rows show the avg_rank of methods across datasets, and $p$-values of the Wilcoxon signed rank test comparing FoMo-0D w/ $D = 100$ with other baselines. The preceding four rows are the same for FoMo-0D w/ $D = 20$, when ranking 31 models (26 baselines + 4 $^{\text{avg}}$ variants of top-4 baselines + FoMo-0D w/ $D = 20$).

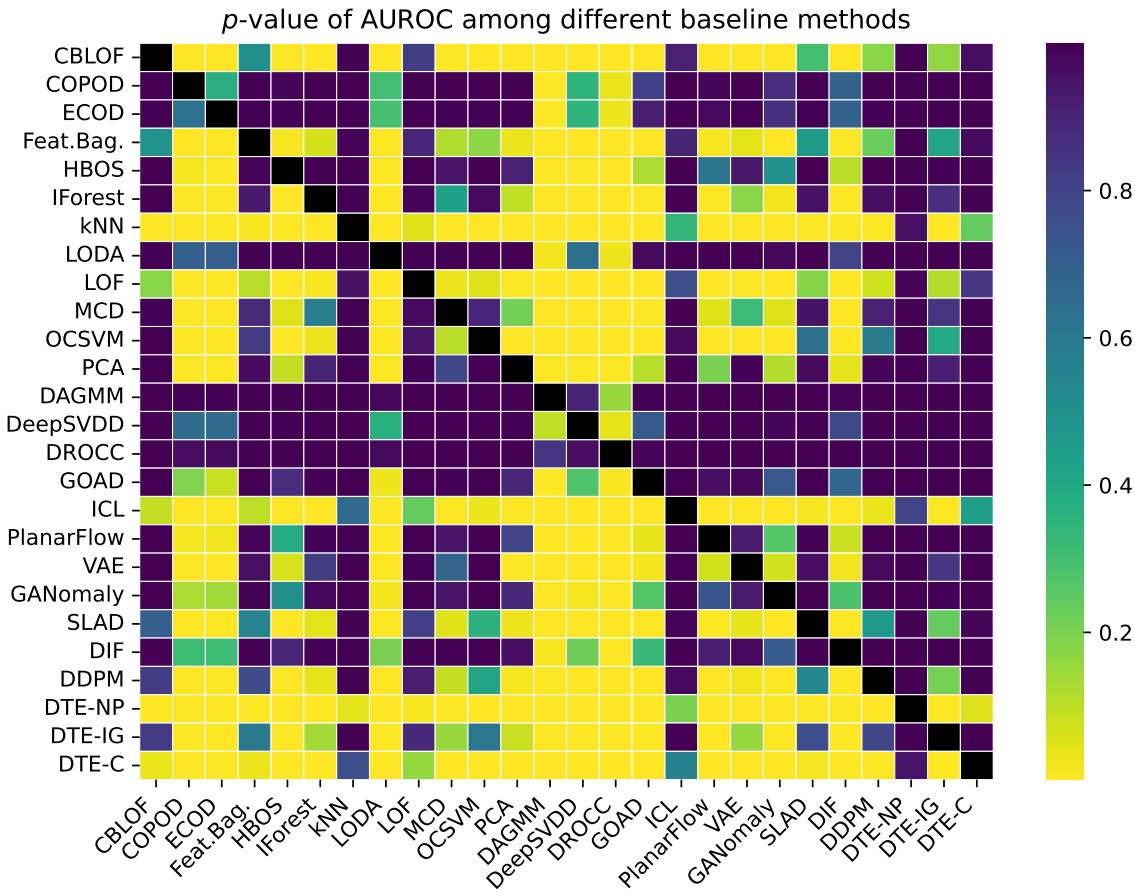

Figure 24: Pairwise *p*-values among baseline methods based on the Wilcoxon signed rank test w.r.t. AUROC performances across datasets.

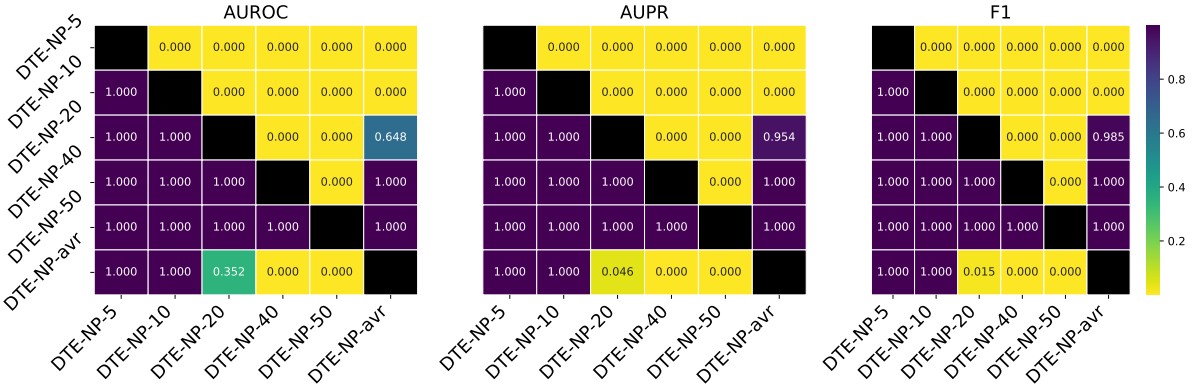

Figure 25: *p*-values w.r.t. AUROC/AUPR/F1 among different HP configurations of **DTE-NP** (i.e., $k \in \{5, 10, 20, 40, 50\}$), along with the $^{\text{avg}}$ model with the average performance across HPs.

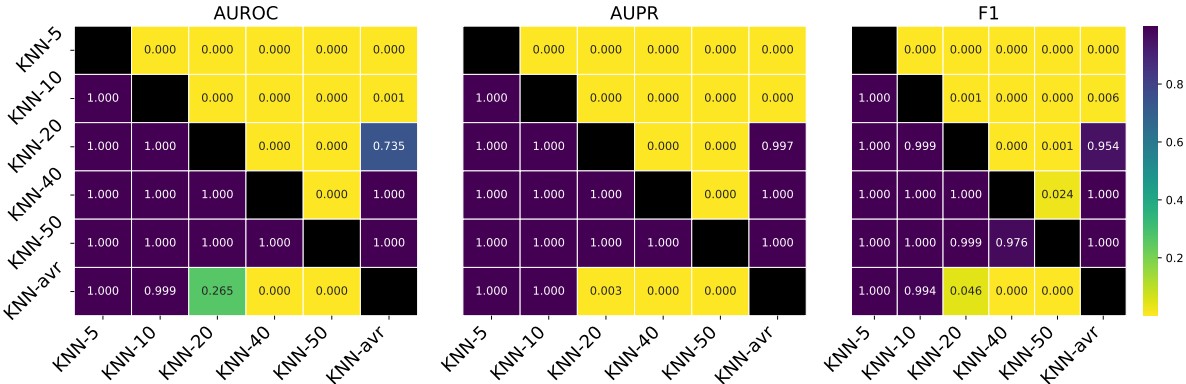

Figure 26: *p*-values w.r.t. AUROC/AUPR/F1 among different HP configurations of *k***NN** (i.e., $k \in \{5, 10, 20, 40, 50\}$), along with the $^{\text{avg}}$ model with the average performance across HPs.

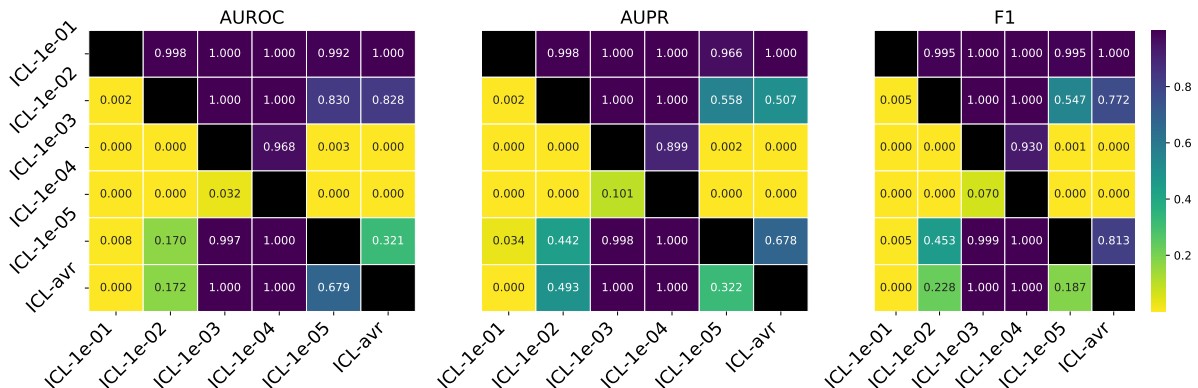

Figure 27: *p*-values w.r.t. AUROC/AUPR/F1 among different HP configurations of **ICL** (i.e., `learning_rate` $\in \{10^{-1}, 10^{-2}, 10^{-3}, 10^{-4}, 10^{-5}\}$), along with the $^{\text{avg}}$ model with the average performance across HPs.

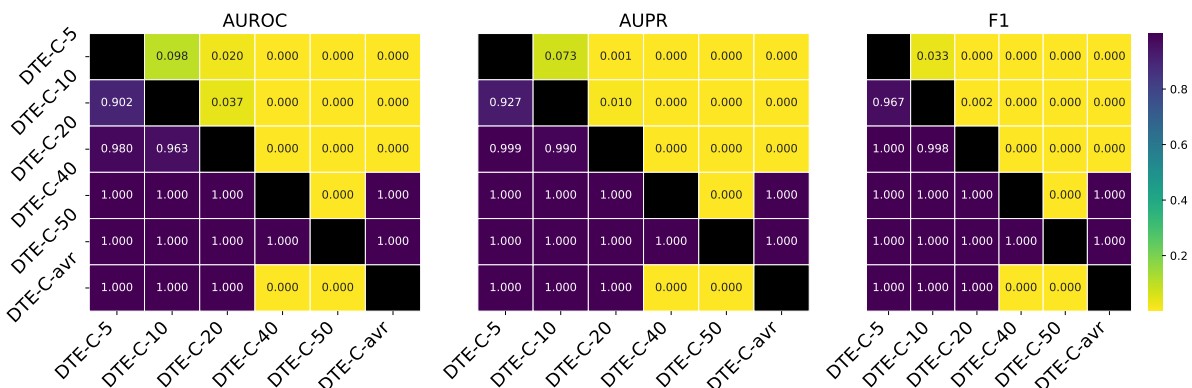

Figure 28: *p*-values w.r.t. AUROC/AUPR/F1 among different HP configurations of **DTE-C** (i.e., $k \in \{5, 10, 20, 40, 50\}$), along with the $^{\text{avg}}$ model with the average performance across HPs.

Table 13: Comparison of methods across datasets. (top row) Rank w.r.t. AUROC performance avg.'ed over 57 datasets is presented for FoMo-0D (with $D = 100$), **top-10 baselines** with default HPs, and **top-4**[6] baselines with performance **avg.**'ed over varying HPs (denoted w/ $^{\text{avg}}$); followed by $p$-values of the pairwise Wilcoxon signed rank test, comparing FoMo-0D to each baseline (from top to bottom) over All (57) datasets, those (24) w/ $d \leq 20$, (38) w/ $d \leq 50$, (42) w/ $d \leq 100$ and (46) datasets w/ $d \leq 500$ dimensions. FoMo-0D performs as well as **(i.e., statistically no different from) the 2$nd$ best model** ($k$NN, w/ $p = 0.106$) across All datasets, while it is **comparable to ($p > 0.05$) or better than ($p > 0.95$) all baselines** over datasets w/ $d \leq 100$ (aligned w/ pretraining where $D = 100$) *and $d \leq 500$ (generalizing beyond pretraining)*.

| | FoMo-0D | DTE-NP | $k$NN | ICL | DTE-C | LOF | CBLOF | Feat.Bag. | SLAD | DDPM | OCSVM | DTE-NP$^{\text{avg}}$ | $k$NN$^{\text{avg}}$ | ICL$^{\text{avg}}$ | DTE-C$^{\text{avg}}$ |
|---|---|---|---|---|---|---|---|---|---|---|---|---|---|---|---|
| Rank(avg) | 11.886 | 7.553 | 9.018 | 10.851 | 11.36 | 12.316 | 13.342 | 13.386 | 12.982 | 14.061 | 13.851 | 9.079 | 11.105 | 12.991 | 22.263 |
| All | - | 0.016 | 0.106 | 0.462 | 0.454 | 0.585 | 0.750 | 0.823 | 0.759 | 0.901 | 0.895 | 0.112 | 0.315 | 0.670 | **1.000** |
| $d \leq 20$ | - | 0.428 | 0.665 | **0.987** | 0.727 | 0.911 | 0.940 | **0.987** | 0.868 | 0.758 | **0.968** | 0.781 | 0.868 | **0.990** | **1.000** |
| $d \leq 50$ | - | 0.734 | 0.923 | **0.992** | 0.973 | 0.989 | 0.987 | **0.999** | 0.948 | **0.985** | 0.986 | 0.948 | **0.967** | **0.989** | **1.000** |
| $d \leq 100$ | - | 0.415 | 0.700 | 0.949 | **0.953** | 0.970 | 0.971 | **0.996** | 0.876 | **0.980** | 0.978 | 0.752 | 0.860 | **0.958** | **1.000** |
| $d \leq 200$ | - | 0.315 | 0.605 | 0.923 | 0.919 | 0.944 | **0.977** | **0.990** | 0.904 | **0.970** | 0.983 | 0.663 | 0.789 | 0.937 | **1.000** |
| $d \leq 500$ | - | 0.220 | 0.569 | 0.827 | 0.894 | **0.960** | 0.968 | **0.994** | 0.910 | **0.960** | 0.979 | 0.607 | 0.756 | 0.846 | **1.000** |

Table 14.1: Average AUROC $\pm$ standard dev. over five seeds for the semi-supervised setting of DTE-NP, $k$NN with varying hyperparameter (HP) values; $k \in \{5, 10, 20, 40, 50\}$. Also reported is the $^{\text{avg}}$ model. We use **bold** and underline respectively to mark the **best** and the worst performance of each model to showcase the variability of performance across different HP settings.

| dataset | DTE-NP-5 | DTE-NP-10 | DTE-NP-20 | DTE-NP-40 | DTE-NP-50 | DTE-NP-avr | KNN-5 | KNN-10 | KNN-20 | KNN-40 | KNN-50 | KNN-avr |
|---|---|---|---|---|---|---|---|---|---|---|---|---|
| aloi | 50.69±0.00 | 51.02±0.00 | 51.26±0.00 | 51.58±0.00 | 51.69±0.00 | 51.25±0.00 | 51.04±0.00 | 51.33±0.00 | 51.63±0.00 | 51.97±0.00 | 52.08±0.00 | 51.61±0.00 |
| amazon | 60.76±0.00 | 60.69±0.00 | 60.53±0.00 | 60.17±0.00 | 60.22±0.00 | 60.47±0.00 | 60.58±0.00 | 60.52±0.00 | 60.23±0.00 | 60.02±0.00 | 59.91±0.00 | 60.25±0.00 |
| annthyroid | 93.01±0.00 | 92.89±0.00 | 92.66±0.00 | 92.38±0.00 | 92.26±0.00 | 92.64±0.00 | 92.81±0.00 | 92.60±0.00 | 92.34±0.00 | 91.98±0.00 | 91.73±0.00 | 92.30±0.00 |
| backdoor | 94.48±0.42 | 93.72±0.46 | 92.67±0.46 | 91.20±0.45 | 90.68±0.45 | 92.55±0.44 | 93.71±0.46 | 92.58±0.46 | 91.14±0.46 | 89.18±0.47 | 88.39±0.51 | 91.00±0.47 |
| breastw | 99.10±0.28 | 98.91±0.35 | 98.59±0.34 | 98.40±0.35 | 98.36±0.28 | 98.67±0.28 | 99.09±0.24 | 99.11±0.27 | 99.16±0.22 | 99.21±0.18 | 99.21±0.17 | 99.16±0.21 |
| campaign | 78.34±0.00 | 78.71±0.00 | 78.91±0.00 | 78.93±0.00 | 78.90±0.00 | 78.76±0.00 | 78.48±0.00 | 78.74±0.00 | 78.84±0.00 | 78.65±0.00 | 78.60±0.00 | 78.66±0.00 |
| cardio | 91.53±0.00 | 92.03±0.00 | 92.46±0.00 | 93.06±0.00 | 93.28±0.00 | 92.47±0.00 | 92.00±0.00 | 92.44±0.00 | 92.99±0.00 | 93.85±0.00 | 94.08±0.00 | 93.07±0.00 |
| cardiotocography | 60.40±0.00 | 61.63±0.00 | 63.14±0.00 | 65.05±0.00 | 65.81±0.00 | 63.21±0.00 | 62.11±0.00 | 63.39±0.00 | 65.12±0.00 | 67.85±0.00 | 68.91±0.00 | 65.48±0.00 |
| celeba | 70.39±0.33 | 72.58±0.26 | 74.81±0.34 | 76.87±0.38 | 77.47±0.37 | 74.42±0.28 | 72.91±0.29 | 75.24±0.40 | 77.50±0.47 | 79.14±0.38 | 79.68±0.37 | 76.90±0.35 |
| census | 72.18±0.34 | 72.34±0.17 | 72.28±0.10 | 71.93±0.17 | 71.80±0.17 | 72.11±0.16 | 72.23±0.29 | 72.36±0.12 | 71.94±0.19 | 71.37±0.21 | 71.28±0.16 | 71.84±0.15 |
| cover | 97.90±0.17 | 97.72±0.14 | 97.40±0.18 | 96.99±0.23 | 96.84±0.24 | 97.37±0.19 | 97.51±0.15 | 97.19±0.15 | 96.75±0.22 | 96.21±0.28 | 96.00±0.31 | 96.73±0.22 |
| donors | 99.72±0.03 | 99.61±0.03 | 99.43±0.06 | 99.14±0.09 | 99.02±0.10 | 99.38±0.06 | 99.51±0.06 | 99.24±0.08 | 98.85±0.10 | 98.20±0.13 | 97.90±0.14 | 98.74±0.09 |
| fault | 58.34±0.00 | 58.37±0.00 | 58.70±0.00 | 60.00±0.00 | 60.43±0.00 | 59.17±0.00 | 58.73±0.00 | 58.76±0.00 | 60.12±0.00 | 61.71±0.00 | 61.79±0.00 | 60.22±0.00 |
| fraud | 95.70±0.93 | 95.67±0.93 | 95.64±0.93 | 95.60±0.92 | 95.60±0.92 | 95.64±0.92 | 95.59±0.97 | 95.58±0.89 | 95.55±0.89 | 95.54±0.89 | 95.62±0.88 | 95.57±0.93 |
| glass | 96.08±0.39 | 93.04±1.06 | 89.82±1.12 | 87.89±1.10 | 87.31±1.40 | 90.83±0.91 | 92.13±0.94 | 88.67±0.98 | 87.24±1.18 | 84.93±2.92 | 83.55±2.61 | 87.30±1.59 |
| hepatitis | 99.84±0.20 | 99.27±0.51 | 96.89±0.96 | 93.15±1.69 | 91.97±1.76 | 96.22±0.88 | 96.77±1.47 | 86.88±2.21 | 85.50±2.34 | 85.46±1.92 | 84.88±2.09 | 87.90±1.75 |
| http | 99.99±0.00 | 99.98±0.01 | 99.95±0.00 | 99.93±0.01 | 99.91±0.02 | 99.95±0.01 | 100.00±0.00 | 99.99±0.02 | 99.95±0.01 | 99.95±0.01 | 99.95±0.01 | 99.96±0.01 |
| imdb | 50.48±0.00 | 50.38±0.00 | 50.32±0.00 | 50.28±0.00 | 50.27±0.00 | 50.35±0.00 | 50.08±0.00 | 50.04±0.00 | 50.29±0.00 | 50.23±0.00 | 50.23±0.00 | 50.18±0.00 |
| internetads | 70.96±0.00 | 68.65±0.00 | 66.86±0.00 | 65.97±0.00 | 65.82±0.00 | 67.65±0.00 | 68.08±0.00 | 65.48±0.00 | 65.02±0.00 | 65.04±0.00 | 65.04±0.00 | 65.73±0.00 |
| ionosphere | 98.48±0.60 | 98.13±0.74 | 97.84±0.64 | 96.83±0.71 | 96.21±0.79 | 97.50±0.63 | 97.32±0.85 | 97.62±0.81 | 96.33±0.76 | 92.80±1.64 | 91.53±1.65 | 95.12±0.92 |
| landsat | 68.99±0.00 | 68.02±0.00 | 66.46±0.00 | 64.73±0.00 | 64.16±0.00 | 66.47±0.00 | 68.25±0.00 | 66.48±0.00 | 64.36±0.00 | 62.49±0.00 | 61.93±0.00 | 64.70±0.00 |
| letter | 36.12±0.00 | 35.66±0.00 | 34.78±0.00 | 33.72±0.00 | 33.40±0.00 | 34.74±0.00 | 35.43±0.00 | 34.54±0.00 | 33.17±0.00 | 32.11±0.00 | 31.69±0.00 | 33.39±0.00 |
| lymphography | 99.88±0.25 | 99.79±0.32 | 99.79±0.32 | 99.76±0.31 | 99.76±0.31 | 99.80±0.30 | 99.87±0.10 | 99.85±0.05 | 99.85±0.05 | 99.88±0.08 | 99.88±0.08 | 99.87±0.06 |
| magic.gamma | 83.91±0.00 | 83.49±0.00 | 82.87±0.00 | 82.05±0.00 | 81.73±0.00 | 82.81±0.00 | 83.27±0.00 | 82.64±0.00 | 81.85±0.00 | 80.76±0.00 | 80.30±0.00 | 81.76±0.00 |
| mammography | 87.65±0.00 | 87.73±0.00 | 87.68±0.00 | 87.42±0.00 | 87.29±0.00 | 87.55±0.00 | 87.58±0.00 | 87.75±0.00 | 87.38±0.00 | 86.97±0.00 | 86.78±0.00 | 87.29±0.00 |
| mnist | 94.22±0.00 | 93.93±0.00 | 93.57±0.00 | 93.20±0.00 | 93.08±0.00 | 93.60±0.00 | 93.85±0.00 | 93.45±0.00 | 93.00±0.00 | 92.55±0.00 | 92.36±0.00 | 93.04±0.00 |
| musk | 100.00±0.00 | 100.00±0.00 | 100.00±0.00 | 100.00±0.00 | 100.00±0.00 | 100.00±0.00 | 100.00±0.00 | 100.00±0.00 | 100.00±0.00 | 100.00±0.00 | 100.00±0.00 | 100.00±0.00 |
| optdigits | 95.00±0.00 | 93.97±0.00 | 92.52±0.00 | 90.87±0.00 | 90.28±0.00 | 92.53±0.00 | 93.72±0.00 | 92.09±0.00 | 90.20±0.00 | 87.66±0.00 | 86.67±0.00 | 90.07±0.00 |
| pageblocks | 89.04±0.00 | 89.40±0.00 | 89.56±0.00 | 89.37±0.00 | 89.23±0.00 | 89.32±0.00 | 89.65±0.00 | 89.86±0.00 | 89.88±0.00 | 89.30±0.00 | 89.18±0.00 | 89.57±0.00 |
| pendigits | 99.90±0.00 | 99.88±0.00 | 99.83±0.00 | 99.51±0.00 | 99.38±0.00 | 99.70±0.00 | 99.87±0.00 | 99.79±0.00 | 99.21±0.00 | 98.58±0.00 | 98.39±0.00 | 99.17±0.00 |
| pima | 82.21±1.82 | 79.74±1.61 | 77.98±1.38 | 77.28±1.35 | 77.14±1.33 | 78.87±1.43 | 77.44±2.07 | 76.14±1.36 | 76.39±1.21 | 76.55±1.25 | 76.38±1.29 | 76.58±1.39 |
| satellite | 82.40±0.00 | 82.09±0.00 | 81.55±0.00 | 80.71±0.00 | 80.38±0.00 | 81.43±0.00 | 82.24±0.00 | 81.56±0.00 | 80.56±0.00 | 79.22±0.00 | 78.76±0.00 | 80.47±0.00 |
| satimage-2 | 99.68±0.00 | 99.73±0.00 | 99.79±0.00 | 99.82±0.00 | 99.82±0.00 | 99.77±0.00 | 99.71±0.00 | 99.80±0.00 | 99.79±0.00 | 99.78±0.00 | 99.77±0.00 | 99.77±0.00 |
| shuttle | 99.94±0.00 | 99.92±0.00 | 99.91±0.00 | 99.91±0.00 | 99.91±0.00 | 99.92±0.00 | 99.91±0.00 | 99.90±0.00 | 99.89±0.00 | 99.89±0.00 | 99.89±0.00 | 99.89±0.00 |
| skin | 99.66±0.05 | 99.52±0.04 | 99.28±0.02 | 98.77±0.09 | 98.54±0.09 | 99.15±0.05 | 99.51±0.06 | 99.17±0.05 | 98.70±0.12 | 97.43±0.05 | 96.90±0.09 | 98.34±0.04 |
| smtp | 92.94±2.55 | 92.84±2.56 | 93.14±2.20 | 93.14±2.15 | 93.14±2.20 | 93.04±2.32 | 92.90±2.48 | 92.97±2.52 | 93.25±2.06 | 93.11±2.28 | 93.24±2.25 | 93.10±2.30 |
| spambase | 84.06±0.00 | 83.49±0.00 | 82.97±0.00 | 82.50±0.00 | 82.36±0.00 | 83.07±0.00 | 83.36±0.00 | 82.68±0.00 | 82.22±0.00 | 81.83±0.00 | 81.72±0.00 | 82.36±0.00 |
| speech | 41.12±0.00 | 39.00±0.00 | 37.59±0.00 | 37.37±0.00 | 37.01±0.00 | 38.42±0.00 | 36.36±0.00 | 36.15±0.00 | 36.37±0.00 | 36.40±0.00 | 36.25±0.00 | 36.31±0.00 |
| stamps | 97.88±0.33 | 97.04±0.33 | 95.60±0.78 | 94.59±1.07 | 94.29±1.08 | 95.88±0.66 | 96.19±1.24 | 94.35±1.35 | 93.67±1.09 | 93.44±1.21 | 93.33±1.25 | 94.20±1.17 |
| thyroid | 98.58±0.00 | 98.63±0.00 | 98.64±0.00 | 98.63±0.00 | 98.59±0.00 | 98.61±0.00 | 98.68±0.00 | 98.67±0.00 | 98.70±0.00 | 98.69±0.00 | 98.69±0.00 | 98.68±0.00 |
| vertebral | 79.59±2.23 | 67.05±2.09 | 56.99±2.07 | 49.07±1.60 | 47.08±1.43 | 59.96±1.83 | 57.33±3.80 | 49.40±1.87 | 45.06±1.20 | 41.08±1.52 | 40.08±1.23 | 46.59±1.67 |
| vowels | 82.11±0.00 | 81.62±0.00 | 80.46±0.00 | 78.11±0.00 | 76.87±0.00 | 79.83±0.00 | 82.21±0.00 | 80.20±0.00 | 77.82±0.00 | 77.22±0.00 | 76.30±0.00 | 78.30±0.00 |
| waveform | 74.42±0.00 | 75.40±0.00 | 76.12±0.00 | 76.79±0.00 | 76.90±0.00 | 75.93±0.00 | 75.21±0.00 | 76.04±0.00 | 76.78±0.00 | 77.47±0.00 | 77.60±0.00 | 76.62±0.00 |
| wbc | 99.62±0.27 | 99.36±0.33 | 99.17±0.41 | 99.12±0.37 | 99.09±0.38 | 99.27±0.34 | 98.88±0.50 | 98.82±0.25 | 98.86±0.38 | 99.08±0.34 | 99.10±0.34 | 98.95±0.35 |
| wdbc | 99.62±0.26 | 99.47±0.29 | 99.26±0.25 | 99.18±0.22 | 99.15±0.20 | 99.34±0.23 | 99.23±0.27 | 99.08±0.21 | 99.08±0.22 | 99.08±0.22 | 99.08±0.17 | 99.11±0.22 |
| wilt | 66.98±0.00 | 63.87±0.00 | 60.18±0.00 | 56.26±0.00 | 55.02±0.00 | 60.46±0.00 | 63.66±0.00 | 59.33±0.00 | 55.21±0.00 | 51.13±0.00 | 49.77±0.00 | 55.82±0.00 |
| wine | 99.90±0.14 | 99.73±0.09 | 99.33±0.18 | 98.88±0.19 | 98.57±0.30 | 99.28±0.12 | 99.17±0.18 | 97.96±0.77 | 98.15±0.47 | 97.65±0.43 | 97.68±0.41 | 98.12±0.40 |
| wpbc | 89.29±1.07 | 79.39±1.04 | 69.61±1.14 | 63.21±1.50 | 61.80±1.64 | 72.67±0.81 | 64.27±2.37 | 58.20±1.67 | 57.03±2.09 | 55.89±1.96 | 55.32±1.96 | 58.14±1.82 |
| yeast | 44.73±0.00 | 44.61±0.00 | 44.33±0.00 | 43.88±0.00 | 43.68±0.00 | 44.25±0.00 | 44.74±0.00 | 44.64±0.00 | 44.23±0.00 | 43.31±0.00 | 42.85±0.00 | 43.95±0.00 |
| yelp | 68.67±0.00 | 68.22±0.00 | 67.75±0.00 | 67.22±0.00 | 66.98±0.00 | 67.47±0.00 | 68.07±0.00 | 67.74±0.00 | 67.12±0.00 | 66.53±0.00 | 66.36±0.00 | 67.16±0.00 |
| MNIST-C | 84.73±0.00 | 84.21±0.00 | 83.58±0.00 | 82.87±0.00 | 82.64±0.00 | 83.60±0.00 | 84.11±0.00 | 83.39±0.00 | 82.64±0.00 | 81.81±0.00 | 81.53±0.00 | 82.70±0.00 |
| FashionMNIST | 90.13±0.00 | 89.91±0.00 | 89.65±0.00 | 89.37±0.00 | 89.27±0.00 | 89.67±0.00 | 89.87±0.00 | 89.55±0.00 | 89.27±0.00 | 88.96±0.00 | 88.86±0.00 | 89.30±0.00 |
| CIFAR10 | 67.81±0.00 | 67.63±0.00 | 67.47±0.00 | 67.32±0.00 | 67.20±0.00 | 67.50±0.00 | 67.53±0.00 | 67.39±0.00 | 67.21±0.00 | 67.09±0.00 | 67.06±0.00 | 67.26±0.00 |
| SVHN | 62.12±0.00 | 61.80±0.00 | 61.48±0.00 | 61.19±0.00 | 61.11±0.00 | 61.54±0.00 | 61.69±0.00 | 61.37±0.00 | 61.06±0.00 | 60.82±0.00 | 60.74±0.00 | 61.14±0.00 |
| MVTec-AD | 89.65±0.57 | 85.94±0.49 | 82.74±0.49 | 80.49±0.49 | 79.94±0.46 | 83.75±0.49 | 81.33±0.47 | 79.41±0.45 | 78.35±0.51 | 77.55±0.42 | 77.39±0.43 | 78.81±0.45 |
| 20news | 60.08±0.67 | 58.74±0.78 | 57.43±0.82 | 56.90±0.88 | 56.97±0.87 | 58.02±0.80 | 57.53±0.84 | 56.74±0.86 | 56.28±0.90 | 56.07±0.92 | 55.62±0.88 | 56.45±0.88 |
| agnews | 67.87±0.00 | 67.26±0.00 | 66.42±0.00 | 65.52±0.00 | 65.21±0.00 | 66.46±0.00 | 67.05±0.00 | 66.15±0.00 | 65.24±0.00 | 64.32±0.00 | 64.00±0.00 | 65.35±0.00 |

Table 14.2: Average AUROC ± standard dev. over five seeds for the semi-supervised setting of ICL and DTE-C baselines with varying hyperparameter (HP) values; For ICL, the learning rate ∈ {0.1, 0.02, 0.001, 0.0001, 1e − 05}, for DTE-C, $k \in$ {5, 10, 20, 40, 50}. Also reported is the [avg] model. We use **bold** and underline respectively to mark the **best** and the worst performance of each model to showcase the variability of performance across different HP settings.

| dataset | ICL-0.1 | ICL-0.02 | ICL-0.001 | ICL-0.0001 | ICL-1e-05 | ICL-avr | DTE-C-5 | DTE-C-10 | DTE-C-20 | DTE-C-40 | DTE-C-50 | DTE-C-avr |
|---|---|---|---|---|---|---|---|---|---|---|---|---|
| aloi | 47.74±0.28 | 47.12±0.51 | 46.81±0.47 | 48.42±0.24 | 48.06±0.23 | 47.63±0.15 | 50.20±0.21 | **50.84±0.10** | 50.16±0.34 | 50.26±0.33 | 50.00±0.00 | 50.29±0.09 |
| amazon | 53.07±0.19 | **53.44±0.19** | 52.75±0.68 | 53.31±0.21 | 53.18±0.15 | 53.15±0.19 | **56.20±1.62** | 55.07±4.20 | 56.12±2.73 | 50.00±0.00 | 50.00±0.00 | 53.48±1.33 |
| annthyroid | 84.02±9.46 | 72.68±3.79 | 87.29±2.69 | **88.84±2.35** | 88.52±1.40 | 84.27±2.31 | 97.47±0.10 | 97.65±0.11 | **97.73±0.15** | 97.40±0.22 | 50.00±0.00 | 88.05±0.05 |
| backdoor | 93.03±0.66 | 92.91±0.70 | 93.30±0.44 | **93.93±0.58** | 93.32±0.71 | 93.30±0.48 | 88.65±1.08 | 92.06±1.04 | **92.64±0.84** | 50.00±0.00 | 50.00±0.00 | 74.67±0.36 |
| breastw | 98.96±0.47 | 98.87±0.20 | **99.19±0.34** | 99.11±0.18 | 97.61±0.65 | 98.75±0.18 | 93.45±1.34 | 94.34±1.25 | 96.71±0.91 | 98.85±0.45 | **99.26±0.07** | 96.52±0.53 |
| campaign | 76.07±0.20 | 74.61±1.97 | 78.88±0.42 | 79.79±0.91 | **82.30±0.38** | 78.33±0.52 | **79.18±1.09** | 78.24±2.09 | 78.49±1.13 | 50.00±0.00 | 50.00±0.00 | 67.18±0.74 |
| cardio | 73.99±7.79 | **84.98±4.75** | 82.30±3.36 | 78.68±2.18 | 68.59±0.64 | 77.71±1.76 | **88.07±0.51** | 87.54±0.69 | 87.66±0.63 | 50.00±0.54 | 50.00±0.00 | 72.66±0.21 |
| cardiotocography | 48.99±2.62 | 54.18±3.25 | 53.24±3.25 | 50.67±2.55 | 47.20±1.33 | 50.85±1.24 | 60.09±2.19 | 59.05±1.34 | 50.16±1.26 | 50.00±0.00 | 50.00±0.00 | 55.90±0.23 |
| celeba | 79.38±2.17 | 79.15±2.47 | 76.13±1.88 | 78.39±1.78 | **79.43±1.44** | 78.49±0.93 | **82.95±1.26** | 81.59±0.79 | 80.16±1.26 | 50.00±0.00 | 50.00±0.00 | 68.94±0.44 |
| census | 70.41±2.07 | 67.52±8.79 | 73.93±0.78 | **75.03±0.46** | 74.37±0.53 | 72.25±1.63 | **70.95±0.91** | 68.04±0.85 | 68.05±2.58 | 68.05±2.58 | 50.00±0.00 | 61.41±0.71 |
| cover | 93.59±3.09 | 82.32±9.14 | 91.92±4.58 | **94.97±2.91** | 94.27±3.40 | 91.42±1.74 | 97.57±0.86 | **97.81±0.75** | 96.61±0.63 | 96.01±0.63 | 50.00±0.00 | 78.40±0.14 |
| donors | 86.20±10.29 | 97.76±1.57 | 99.37±0.30 | 99.37±0.30 | **99.51±0.14** | 96.30±1.95 | **98.68±0.15** | 97.56±0.53 | 95.64±0.27 | 58.94±17.89 | 50.00±0.00 | 80.17±3.55 |
| fault | 63.37±3.29 | 63.04±1.93 | 61.44±0.87 | 62.83±1.01 | 62.83±1.01 | 62.47±1.14 | **59.16±1.69** | 58.41±1.41 | 59.04±1.53 | 50.21±0.42 | 50.21±0.42 | 55.41±0.63 |
| fraud | 93.24±1.45 | 93.21±1.25 | 94.97±2.91 | **95.84±0.87** | 93.90±1.05 | 94.23±0.87 | **94.49±1.56** | 93.08±2.20 | 33.50±2.01 | 50.00±0.00 | 50.00±0.00 | 76.21±1.08 |
| glass | 84.02±8.38 | 94.66±3.83 | 99.25±0.37 | **99.86±0.29** | 99.12±0.55 | 95.27±1.45 | 93.46±0.91 | 84.89±2.89 | 84.89±2.89 | 66.42±8.60 | 50.00±0.00 | 76.95±2.58 |
| hepatitis | 99.95±0.11 | 98.76±1.84 | 99.93±0.14 | **100.00±0.00** | 99.97±0.06 | 99.98±0.02 | **99.54±0.58** | 98.90±1.01 | 98.90±1.08 | 74.74±7.86 | 50.00±0.00 | 83.01±5.81 |
| http | 99.96±0.07 | 99.97±0.02 | 99.88±0.01 | 100.00±0.00 | 100.00±0.00 | 99.69±0.48 | 99.54±0.08 | **99.64±0.28** | 99.39±0.06 | 99.39±0.07 | 50.00±0.00 | 49.27±1.12 |
| imdb | 52.16±0.31 | 51.88±0.40 | 52.28±0.15 | 52.35±0.12 | **53.06±0.15** | 52.35±0.12 | 47.68±3.79 | 47.68±2.08 | 47.81±2.08 | **50.00±0.00** | 50.00±0.00 | 66.79±0.73 |
| internetads | 72.61±1.19 | 73.97±0.44 | **74.09±1.75** | 73.47±0.45 | 68.51±0.74 | 72.53±0.65 | **79.02±1.49** | 77.71±0.82 | 77.23±2.48 | 50.00±0.00 | 50.00±0.00 | 84.42±1.20 |
| ionosphere | 96.81±2.22 | 96.13±3.45 | 98.90±0.41 | 98.91±0.31 | 98.14±0.79 | 97.78±1.23 | 94.67±1.52 | 94.49±2.41 | **95.23±1.37** | 85.77±2.62 | 44.90±23.57 | 89.59±0.09 |
| landsat | 65.71±2.05 | 60.63±1.79 | 65.95±0.76 | **67.85±0.68** | 61.82±1.06 | 64.39±0.59 | **51.80±0.94** | 50.31±2.61 | 48.67±2.62 | 50.22±0.45 | 50.00±0.00 | 50.20±0.98 |
| letter | 48.14±3.53 | 42.74±3.41 | 40.70±2.13 | 40.32±1.11 | 47.20±2.50 | 43.82±1.97 | 36.75±1.19 | 37.72±1.31 | 37.40±0.89 | **50.00±0.00** | 50.00±0.00 | 42.38±0.25 |
| lymphography | 100.00±0.00 | 100.00±0.00 | 100.00±0.00 | 100.00±0.00 | 100.00±0.00 | 100.00±0.00 | 98.97±0.31 | **99.40±0.22** | 99.40±0.22 | 93.82±8.49 | 50.00±0.00 | 88.19±1.78 |
| magic.gamma | 69.73±3.02 | 76.94±3.25 | 77.61±0.70 | **77.81±0.88** | 78.36±1.34 | 76.09±1.17 | 86.42±0.59 | **87.42±0.35** | 87.21±0.73 | 64.88±18.23 | 50.00±0.00 | 75.19±3.50 |
| mammography | 79.90±5.19 | **85.08±3.05** | 79.26±2.14 | 80.75±1.40 | 77.35±0.77 | 80.47±0.62 | 83.00±3.62 | **86.02±1.23** | 84.85±2.49 | 85.40±1.03 | 50.00±0.00 | 77.85±0.91 |
| mnist | 75.53±1.43 | 76.05±3.81 | 79.13±1.20 | 87.05±1.12 | **89.10±0.77** | 81.37±0.93 | **90.21±0.70** | 86.98±1.70 | 85.58±2.12 | 50.00±0.00 | 50.00±0.00 | 72.55±0.64 |
| musk | 100.00±0.00 | 100.00±0.00 | 100.00±0.00 | 98.47±0.03 | 85.92±1.84 | 97.18±0.37 | 100.00±0.00 | 100.00±0.00 | 100.00±0.00 | 50.00±0.00 | 80.00±0.00 | 80.00±0.00 |
| optdigits | 91.22±1.88 | 94.15±0.97 | 95.17±0.09 | 98.47±0.03 | 95.08±1.06 | 94.82±0.49 | 86.29±1.29 | 74.81±6.84 | 77.57±4.38 | 88.80±0.13 | 50.00±0.00 | 67.73±1.71 |
| pageblocks | 89.21±1.10 | **90.77±1.73** | 88.70±1.01 | 88.58±0.56 | 88.53±0.27 | 89.16±0.29 | 89.86±0.39 | **90.29±0.81** | 89.23±0.26 | 88.80±0.13 | 50.00±0.00 | 81.66±0.20 |
| pendigits | 87.01±9.78 | 96.45±2.40 | 96.17±2.68 | 96.78±1.19 | 95.10±0.89 | 94.60±2.36 | **98.24±0.47** | 97.42±0.83 | 96.89±0.68 | 50.00±0.00 | 50.00±0.00 | 78.51±0.15 |
| pima | 74.66±3.27 | 73.04±1.77 | **79.77±1.92** | 78.06±0.74 | 75.09±2.65 | 76.12±1.04 | 70.12±1.89 | 67.22±3.15 | 66.78±2.78 | 56.11±12.22 | 69.39±5.15 | 68.95±0.96 |
| satellite | 74.62±9.67 | 83.23±2.89 | 85.43±0.52 | **89.24±0.57** | 85.35±0.48 | 83.57±2.48 | **80.21±1.18** | 79.01±0.76 | 78.54±0.62 | 78.54±0.62 | 50.00±0.00 | 68.77±2.26 |
| satimage-2 | 94.41±2.46 | **93.04±12.69** | 99.80±0.06 | 99.55±0.09 | 98.15±0.57 | 86.99±2.56 | **99.61±0.13** | 99.17±0.15 | 98.38±0.56 | 69.34±23.69 | 50.00±0.00 | 83.30±4.71 |
| shuttle | 99.43±0.50 | 99.89±0.05 | **99.99±0.00** | 99.99±0.00 | 99.97±0.01 | 99.85±0.11 | **99.76±0.00** | 99.74±0.01 | 99.70±0.01 | 99.28±0.22 | 50.00±0.00 | 89.70±0.05 |
| skin | 53.71±32.40 | 86.42±5.89 | 86.94±5.22 | 71.25±17.43 | 70.72±13.69 | 73.81±10.45 | 91.69±0.28 | **92.12±0.27** | 91.95±0.22 | 91.63±0.28 | 50.00±0.00 | 83.48±0.19 |
| smtp | 81.23±10.57 | **87.97±4.57** | 90.53±5.25 | 89.48±4.52 | 92.63±4.06 | 88.37±5.41 | 95.06±1.20 | 95.61±1.24 | **95.84±1.23** | 95.84±1.22 | 50.00±0.00 | 86.47±0.96 |
| spambase | 79.20±2.89 | 79.36±5.07 | 83.16±0.34 | **83.39±0.40** | 78.42±0.49 | 80.71±0.86 | 82.98±0.37 | 83.29±0.49 | **83.74±0.38** | 50.00±0.00 | 50.00±0.00 | 70.00±0.04 |
| speech | 50.06±3.03 | 50.96±2.48 | **53.72±1.46** | 51.29±2.33 | 46.64±1.97 | 50.53±1.10 | 38.02±1.49 | 38.55±1.07 | 38.72±1.79 | **50.00±0.00** | 50.10±26.50 | 43.06±0.55 |
| stamps | 77.62±9.15 | 84.33±6.20 | **97.29±0.48** | 96.70±0.85 | 96.70±0.85 | 90.11±2.02 | 93.01±1.38 | 91.27±2.53 | 86.48±2.57 | 90.17±4.76 | 50.10±26.50 | 82.21±5.22 |
| thyroid | 94.79±1.60 | 95.24±1.04 | 96.20±0.74 | **96.78±1.19** | 93.73±0.80 | 95.35±0.60 | 98.75±0.07 | **98.92±0.02** | 98.92±0.04 | **98.94±0.05** | 50.00±0.00 | 89.11±0.02 |
| vertebral | 53.80±4.97 | **58.76±7.63** | 75.60±6.18 | 54.07±0.26 | 81.92±1.33 | 70.61±1.11 | 67.07±2.73 | 65.09±3.50 | 62.93±4.76 | 56.35±4.78 | 48.40±5.99 | 59.97±2.61 |
| vowels | 73.59±6.94 | 79.11±2.66 | 84.59±3.49 | 99.55±0.09 | 83.99±1.93 | 81.22±1.57 | 87.25±1.51 | 63.97±2.75 | 66.24±1.45 | 50.00±0.00 | 50.00±0.00 | 72.28±0.42 |
| waveform | 70.94±1.78 | **73.85±6.07** | 70.23±2.82 | 61.96±1.35 | 64.69±1.66 | 68.33±2.15 | 65.16±1.34 | 85.96±4.40 | 99.28±0.22 | 50.00±0.00 | 50.00±0.00 | 59.07±0.68 |
| wbc | 98.13±1.33 | 99.08±0.62 | 99.89±0.19 | **99.90±0.16** | 99.97±0.01 | 99.34±0.42 | 86.07±4.41 | 98.76±0.26 | 86.45±7.40 | **97.03±1.02** | 50.00±0.00 | 81.10±1.77 |
| wdbc | 96.66±2.87 | 99.05±0.72 | 99.51±0.25 | **99.67±0.15** | 99.29±0.25 | 98.84±0.43 | 98.96±0.46 | 87.06±0.069 | **99.01±0.41** | 50.22±16.16 | 45.17±0.66 | 78.42±3.73 |
| wilt | 52.75±1.46 | 61.09±5.94 | 83.81±3.04 | 84.63±1.21 | 79.81±0.93 | 72.42±2.57 | 84.17±0.27 | 70.67±1.94 | 81.09±1.72 | 84.90±1.24 | 50.00±0.00 | 77.44±0.57 |
| wine | 99.64±0.73 | 99.73±0.29 | **99.92±0.12** | 99.81±0.39 | 99.84±0.26 | 99.79±0.33 | **99.91±0.11** | 59.96±3.66 | 99.81±0.29 | 66.04±29.91 | 49.98±0.04 | 83.09±5.95 |
| wpbc | 91.76±2.53 | 76.46±9.54 | 95.40±1.42 | **95.51±1.23** | 92.46±1.77 | 90.32±2.50 | 68.39±4.15 | 48.44±1.44 | 68.60±3.44 | 51.85±2.56 | 49.96±5.03 | 61.89±2.06 |
| yeast | 53.40±0.49 | 47.02±1.77 | 54.08±0.78 | 47.38±1.07 | 52.73±0.21 | 46.62±1.16 | 46.82±1.26 | 63.02±0.36 | 49.01±2.16 | 44.44±2.06 | **50.00±0.00** | 47.74±0.50 |
| yelp | 80.78±0.30 | 54.26±0.22 | **85.02±0.14** | 54.07±0.26 | 83.37±0.12 | 83.71±1.22 | **62.00±1.65** | 89.56±0.58 | 60.99±2.26 | 50.00±0.00 | 50.00±0.00 | 56.47±0.84 |
| MNIST-C | 87.80±0.17 | 83.24±0.51 | 90.91±0.06 | 94.91±0.11 | 88.57±0.10 | 83.46±0.11 | 85.01±0.43 | 86.12±0.43 | 85.25±0.30 | **97.03±1.02** | 50.00±0.00 | 71.28±0.16 |
| FashionMNIST | 59.53±0.31 | **90.44±0.09** | 90.91±0.06 | 90.95±0.03 | 89.25±0.13 | 89.74±0.07 | 90.13±0.19 | **90.30±0.13** | 90.27±0.15 | 50.00±0.00 | 50.00±0.00 | 74.14±0.06 |
| CIFAR10 | 45.12±3.69 | 64.12±0.37 | **65.72±0.09** | 65.89±0.27 | 61.54±0.06 | 92.90±0.15 | 68.78±0.24 | 68.59±0.36 | **68.87±0.31** | 50.00±0.00 | 50.00±0.00 | 61.25±0.12 |
| SVHN | 59.44±0.27 | 61.25±0.11 | **61.73±0.07** | 61.54±0.06 | 60.61±0.08 | 60.91±0.03 | 63.02±0.30 | 63.02±0.36 | 62.97±0.38 | 50.00±0.00 | 50.00±0.00 | 57.80±0.05 |
| MVTec-AD | 93.31±0.36 | 93.73±0.30 | **94.26±0.35** | 94.27±0.32 | 89.82±0.44 | 93.08±0.33 | 86.78±0.49 | **89.56±0.58** | 89.38±0.93 | 50.75±1.11 | 50.13±0.45 | 73.32±0.42 |
| 20news | 57.95±0.57 | 59.21±0.72 | **59.52±0.72** | 60.25±0.48 | 55.58±0.17 | 58.50±0.44 | 64.14±1.95 | 64.01±0.79 | 61.86±0.96 | 50.00±0.28 | 50.00±0.28 | 58.00±0.60 |
| agnews | 56.84±0.12 | 57.43±0.39 | 57.59±0.12 | 57.89±0.09 | 56.85±0.16 | 57.32±0.11 | **68.75±0.81** | 65.60±1.74 | 65.95±1.66 | 50.00±0.00 | 50.00±0.00 | 60.06±0.42 |

Table 15.1: Average AUPR ± standard dev. over five seeds for the semi-supervised setting of DTE-NP, *k*NN baselines with varying hyperparameter (HP) values; $k \in \{5, 10, 20, 40, 50\}$. Also reported is the avg model. We use **bold** and underline respectively to mark the **best** and the worst performance of each model to showcase the variability of performance across different HP settings.

| dataset | DTE-NP-5 | DTE-NP-10 | DTE-NP-20 | DTE-NP-40 | DTE-NP-50 | DTE-NP-avr | KNN-5 | KNN-10 | KNN-20 | KNN-40 | KNN-50 | KNN-avr |
|---|---|---|---|---|---|---|---|---|---|---|---|---|
| aloi | 5.95±0.00 | 5.99±0.00 | 6.02±0.00 | 6.06±0.00 | 6.07±0.00 | 6.02±0.00 | 6.02±0.00 | 6.07±0.00 | 6.09±0.00 | 6.13±0.00 | 6.15±0.00 | 6.09±0.00 |
| amazon | 11.68±0.00 | 11.68±0.00 | 11.68±0.00 | 11.61±0.00 | 11.62±0.00 | 11.65±0.00 | 11.69±0.00 | 11.70±0.00 | 11.65±0.00 | 11.60±0.00 | 11.59±0.00 | 11.65±0.00 |
| annthyroid | 67.49±0.00 | 66.73±0.00 | 66.04±0.00 | 65.39±0.00 | 64.87±0.00 | 66.11±0.00 | 68.07±0.00 | 67.90±0.00 | 67.26±0.00 | 66.27±0.00 | 65.73±0.00 | 67.06±0.00 |
| backdoor | 55.90±0.99 | 47.16±1.45 | 38.31±1.02 | 31.44±1.47 | 29.58±0.37 | 40.48±0.81 | 46.70±1.22 | 37.36±1.35 | 29.58±0.58 | 24.34±0.41 | 22.34±0.53 | 32.06±0.76 |
| breastw | 98.51±0.56 | 98.19±0.58 | 97.56±0.51 | 97.13±0.62 | 97.05±0.45 | 97.69±0.40 | 98.97±0.28 | 99.01±0.31 | 99.08±0.23 | 99.15±0.17 | 99.16±0.16 | 99.08±0.22 |
| campaign | 48.48±0.00 | 49.05±0.00 | 49.77±0.00 | 49.77±0.00 | 49.51±0.00 | 49.31±0.00 | 49.04±0.00 | 49.89±0.00 | 50.45±0.00 | 49.47±0.00 | 49.33±0.00 | 49.64±0.00 |
| cardio | 76.90±0.00 | 77.73±0.00 | 78.30±0.00 | 79.19±0.00 | 79.53±0.00 | 78.33±0.00 | 77.22±0.00 | 78.33±0.00 | 78.33±0.00 | 80.67±0.00 | 81.15±0.00 | 79.30±0.00 |
| cardiotocography | 56.55±0.00 | 57.18±0.00 | 58.19±0.00 | 59.42±0.00 | 59.95±0.00 | 58.26±0.00 | 57.43±0.00 | 58.37±0.00 | 59.44±0.00 | 61.41±0.00 | 62.19±0.00 | 59.77±0.00 |
| celeba | 10.56±0.44 | 11.63±0.49 | 12.74±0.52 | 13.92±0.58 | 14.30±0.59 | 12.63±0.51 | 11.99±0.57 | 13.26±0.61 | 14.50±0.58 | 15.70±0.65 | 16.10±0.68 | 14.31±0.60 |
| census | 21.14±0.39 | 21.38±0.54 | 21.16±0.43 | 20.67±0.41 | 20.52±0.42 | 20.97±0.43 | 21.36±0.76 | 21.22±0.39 | 20.59±0.33 | 20.00±0.42 | 19.94±0.44 | 20.62±0.44 |
| cover | 63.67±3.21 | 57.85±3.52 | 51.55±3.10 | 44.58±2.49 | 42.11±2.29 | 51.95±2.90 | 55.15±3.45 | 48.67±2.84 | 41.44±2.04 | 33.72±1.51 | 31.69±1.35 | 42.14±2.20 |
| donors | 93.23±0.80 | 91.25±0.77 | 88.17±0.95 | 83.92±1.29 | 82.34±1.32 | 87.78±0.99 | 89.44±0.96 | 85.33±1.15 | 80.15±1.33 | 73.68±1.32 | 71.00±1.30 | 79.92±1.13 |
| fault | 62.03±0.00 | 61.58±0.00 | 61.29±0.00 | 61.98±0.00 | 62.31±0.00 | 61.84±0.00 | 61.98±0.00 | 61.16±0.00 | 61.92±0.00 | 63.67±0.00 | 64.06±0.00 | 62.56±0.00 |
| fraud | 40.60±6.67 | 43.77±5.38 | 43.03±4.92 | 39.91±4.75 | 38.80±4.94 | 41.22±5.02 | 42.35±5.61 | 44.96±3.90 | 41.19±3.73 | 37.33±4.15 | 36.42±4.12 | 40.45±4.07 |
| glass | 60.15±6.89 | 47.75±5.62 | 37.27±4.92 | 31.23±3.12 | 30.48±2.85 | 41.38±4.46 | 44.05±6.38 | 32.96±3.74 | 29.87±3.75 | 26.62±2.65 | 26.04±3.61 | 31.91±3.73 |
| hepatitis | 99.47±0.73 | 97.98±1.43 | 91.71±2.26 | 81.65±3.68 | 78.95±3.45 | 89.95±1.80 | 91.10±4.41 | 69.28±4.16 | 64.12±5.59 | 64.29±5.08 | 64.33±6.16 | 70.62±4.48 |
| http | 98.52±0.37 | 95.38±2.26 | 88.66±1.01 | 84.43±2.65 | 80.40±4.55 | 89.48±1.90 | 100.00±0.00 | 98.01±3.98 | 91.24±1.42 | 91.44±1.25 | 91.28±1.36 | 94.39±1.31 |
| imdb | 9.11±0.00 | 9.09±0.00 | 9.06±0.00 | 9.07±0.00 | 9.06±0.00 | 9.08±0.00 | 8.92±0.00 | 8.94±0.00 | 8.99±0.00 | 8.98±0.00 | 8.99±0.00 | 8.96±0.00 |
| internetads | 52.20±0.00 | 49.76±0.00 | 48.19±0.00 | 47.56±0.00 | 47.45±0.00 | 49.03±0.00 | 49.22±0.00 | 47.29±0.00 | 46.93±0.00 | 46.95±0.00 | 46.94±0.00 | 47.47±0.00 |
| ionosphere | 98.72±0.48 | 98.46±0.54 | 98.27±0.42 | 97.44±0.50 | 96.93±0.61 | 97.96±0.46 | 97.86±0.60 | 98.11±0.52 | 97.04±0.60 | 94.12±1.45 | 92.90±1.65 | 96.01±0.78 |
| landsat | 56.14±0.00 | 54.25±0.00 | 50.75±0.00 | 46.43±0.00 | 45.17±0.00 | 50.55±0.00 | 54.85±0.00 | 50.62±0.00 | 45.18±0.00 | 41.32±0.00 | 40.50±0.00 | 46.49±0.00 |
| letter | 8.86±0.00 | 8.78±0.00 | 8.67±0.00 | 8.54±0.00 | 8.50±0.00 | 8.67±0.00 | 8.70±0.00 | 8.58±0.00 | 8.41±0.00 | 8.27±0.00 | 8.22±0.00 | 8.44±0.00 |
| lymphography | 97.27±5.45 | 96.07±6.79 | 96.00±6.79 | 95.68±6.60 | 95.68±6.60 | 96.16±6.43 | 98.61±1.02 | 98.43±0.52 | 98.43±0.52 | 98.70±0.83 | 98.70±0.83 | 98.57±0.65 |
| magic.gamma | 86.30±0.00 | 85.80±0.00 | 85.28±0.00 | 84.56±0.00 | 84.29±0.00 | 85.26±0.00 | 85.86±0.00 | 85.25±0.00 | 84.51±0.00 | 83.61±0.00 | 83.25±0.00 | 84.50±0.00 |
| mammography | 42.14±0.00 | 41.51±0.00 | 40.67±0.00 | 40.87±0.00 | 40.50±0.00 | 41.04±0.00 | 41.27±0.00 | 40.55±0.00 | 40.24±0.00 | 38.97±0.00 | 38.10±0.00 | 39.83±0.00 |
| mnist | 74.43±0.00 | 73.09±0.00 | 71.84±0.00 | 70.69±0.00 | 70.36±0.00 | 72.08±0.00 | 72.72±0.00 | 71.40±0.00 | 70.09±0.00 | 69.02±0.00 | 68.60±0.00 | 70.36±0.00 |
| musk | 100.00±0.00 | 100.00±0.00 | 100.00±0.00 | 100.00±0.00 | 100.00±0.00 | 100.00±0.00 | 100.00±0.00 | 100.00±0.00 | 100.00±0.00 | 100.00±0.00 | 100.00±0.00 | 100.00±0.00 |
| optdigits | 34.44±0.00 | 30.53±0.00 | 26.28±0.00 | 22.67±0.00 | 21.61±0.00 | 27.11±0.00 | 29.11±0.00 | 24.76±0.00 | 21.10±0.00 | 17.68±0.00 | 16.62±0.00 | 21.85±0.00 |
| pageblocks | 62.78±0.00 | 62.52±0.00 | 62.20±0.00 | 61.02±0.00 | 60.30±0.00 | 61.76±0.00 | 67.60±0.00 | 67.74±0.00 | 67.87±0.00 | 66.41±0.00 | 66.13±0.00 | 67.15±0.00 |
| pendigits | 97.68±0.00 | 97.31±0.00 | 96.28±0.00 | 90.01±0.00 | 86.69±0.00 | 93.59±0.00 | 96.99±0.00 | 95.65±0.00 | 81.40±0.00 | 70.28±0.00 | 67.39±0.00 | 82.34±0.00 |
| pima | 80.27±1.65 | 78.05±2.13 | 75.87±2.40 | 74.73±2.71 | 74.49±2.71 | 76.68±2.22 | 75.66±2.91 | 73.62±2.59 | 73.42±2.98 | 73.71±2.79 | 73.63±2.86 | 74.01±2.75 |
| satellite | 85.98±0.00 | 85.74±0.00 | 85.17±0.00 | 84.15±0.00 | 83.72±0.00 | 84.95±0.00 | 86.01±0.00 | 85.31±0.00 | 84.02±0.00 | 82.19±0.00 | 81.56±0.00 | 83.82±0.00 |
| satimage-2 | 96.10±0.00 | 96.64±0.00 | 97.02±0.00 | 97.39±0.00 | 97.42±0.00 | 97.21±0.00 | 96.69±0.00 | 97.21±0.00 | 97.39±0.00 | 97.42±0.00 | 97.42±0.00 | 97.22±0.00 |
| shuttle | 99.16±0.00 | 98.76±0.00 | 98.72±0.00 | 98.78±0.00 | 98.77±0.00 | 98.84±0.00 | 97.86±0.00 | 97.34±0.00 | 97.28±0.00 | 97.22±0.00 | 97.20±0.00 | 97.38±0.00 |
| skin | 98.92±0.23 | 98.31±0.20 | 96.81±0.31 | 94.52±0.43 | 93.30±0.50 | 96.37±0.25 | 98.31±0.34 | 96.30±0.30 | 94.09±0.51 | 88.39±0.08 | 86.43±0.46 | 92.71±0.16 |
| smtp | 56.70±7.16 | 54.77±7.80 | 54.75±7.81 | 54.76±7.81 | 48.74±10.23 | 53.94±7.83 | 50.26±5.73 | 50.20±5.74 | 50.18±5.75 | 50.33±5.85 | 50.41±5.72 | 50.27±5.76 |
| spambase | 83.93±0.00 | 83.42±0.00 | 83.03±0.00 | 82.73±0.00 | 82.63±0.00 | 83.15±0.00 | 83.32±0.00 | 82.70±0.00 | 82.41±0.00 | 82.17±0.00 | 82.11±0.00 | 82.54±0.00 |
| speech | 3.02±0.00 | 2.89±0.00 | 2.70±0.00 | 2.76±0.00 | 2.70±0.00 | 2.82±0.00 | 2.80±0.00 | 2.73±0.00 | 2.74±0.00 | 2.76±0.00 | 2.74±0.00 | 2.75±0.00 |
| stamps | 82.50±3.71 | 77.11±4.30 | 69.99±5.29 | 65.85±6.16 | 64.64±6.16 | 72.02±4.82 | 73.26±7.70 | 65.58±7.71 | 63.12±6.69 | 62.09±7.20 | 61.57±7.18 | 65.12±7.10 |
| thyroid | 77.22±0.00 | 77.53±0.00 | 77.26±0.00 | 76.43±0.00 | 74.75±0.00 | 76.64±0.00 | 80.94±0.00 | 81.09±0.00 | 81.50±0.00 | 81.90±0.00 | 81.93±0.00 | 81.47±0.00 |
| vertebral | 43.77±5.50 | 31.72±3.49 | 24.99±2.97 | 21.10±2.37 | 20.29±2.40 | 28.38±3.31 | 25.07±3.19 | 21.55±2.53 | 19.65±2.31 | 18.09±1.91 | 17.76±2.02 | 20.42±2.34 |
| vowels | 31.68±0.00 | 30.30±0.00 | 29.54±0.00 | 27.85±0.00 | 27.32±0.00 | 29.34±0.00 | 30.21±0.00 | 28.75±0.00 | 27.41±0.00 | 24.27±0.00 | 22.44±0.00 | 26.62±0.00 |
| waveform | 26.96±0.00 | 26.71±0.00 | 25.68±0.00 | 24.67±0.00 | 24.49±0.00 | 25.70±0.00 | 27.00±0.00 | 25.82±0.00 | 23.77±0.00 | 24.13±0.00 | 23.87±0.00 | 24.92±0.00 |
| wbc | 96.59±2.18 | 93.20±4.25 | 90.32±6.04 | 90.14±5.60 | 88.96±6.03 | 91.84±4.57 | 89.48±5.58 | 89.07±3.41 | 88.67±4.37 | 91.70±3.59 | 91.82±3.52 | 90.15±3.97 |
| wdbc | 92.08±6.52 | 89.03±6.77 | 86.03±5.81 | 83.79±5.59 | 83.41±5.19 | 86.87±5.84 | 85.35±5.46 | 83.72±5.56 | 82.05±5.59 | 82.37±4.91 | 82.33±4.34 | 83.17±5.03 |
| wilt | 13.43±0.00 | 12.36±0.00 | 11.30±0.00 | 10.40±0.00 | 10.12±0.00 | 11.52±0.00 | 12.25±0.00 | 11.04±0.00 | 10.11±0.00 | 9.33±0.00 | 9.09±0.00 | 10.36±0.00 |
| wine | 99.42±0.77 | 98.32±0.52 | 96.09±1.31 | 93.12±1.51 | 91.59±2.00 | 95.71±0.92 | 95.18±1.66 | 88.85±3.38 | 88.36±2.97 | 88.39±0.08 | 85.79±1.56 | 88.72±2.32 |
| wpbc | 75.30±1.88 | 61.19±1.49 | 51.53±2.17 | 46.62±2.23 | 45.63±2.17 | 56.05±1.44 | 47.07±2.57 | 43.16±2.41 | 42.43±2.53 | 42.28±2.60 | 42.11±2.47 | 43.41±2.44 |
| yeast | 48.37±0.00 | 47.91±0.00 | 47.52±0.00 | 47.26±0.00 | 47.20±0.00 | 47.65±0.00 | 48.26±0.00 | 47.48±0.00 | 47.24±0.00 | 46.74±0.00 | 46.48±0.00 | 47.24±0.00 |
| yelp | 16.05±0.00 | 15.78±0.00 | 15.40±0.00 | 15.01±0.00 | 14.89±0.00 | 15.42±0.00 | 16.03±0.00 | 15.63±0.00 | 15.17±0.00 | 14.77±0.00 | 14.66±0.00 | 15.25±0.00 |
| MNIST-C | 47.21±0.00 | 46.26±0.00 | 45.35±0.00 | 44.50±0.00 | 44.24±0.00 | 45.51±0.00 | 46.20±0.00 | 45.18±0.00 | 44.32±0.00 | 43.50±0.00 | 43.23±0.00 | 44.49±0.00 |
| FashionMNIST | 59.52±0.00 | 59.05±0.00 | 58.57±0.00 | 58.09±0.00 | 57.96±0.00 | 58.64±0.00 | 59.15±0.00 | 58.64±0.00 | 58.19±0.00 | 57.74±0.00 | 57.60±0.00 | 58.26±0.00 |
| CIFAR10 | 19.77±0.00 | 19.59±0.00 | 19.43±0.00 | 19.31±0.00 | 19.28±0.00 | 19.48±0.00 | 19.62±0.00 | 19.50±0.00 | 19.37±0.00 | 19.27±0.00 | 19.24±0.00 | 19.40±0.00 |
| SVHN | 15.44±0.00 | 15.30±0.00 | 15.18±0.00 | 15.08±0.00 | 15.05±0.00 | 15.21±0.00 | 15.34±0.00 | 15.22±0.00 | 15.11±0.00 | 15.02±0.00 | 14.99±0.00 | 15.13±0.00 |
| MVTec-AD | 82.66±1.11 | 79.02±0.97 | 76.24±0.90 | 74.41±0.89 | 73.90±0.86 | 77.26±0.94 | 75.38±0.86 | 73.88±0.83 | 73.13±0.87 | 72.56±0.79 | 72.46±0.78 | 73.48±0.82 |
| 20news | 15.45±1.12 | 14.32±1.09 | 13.25±0.77 | 12.71±0.65 | 12.59±0.66 | 13.66±0.85 | 13.76±0.93 | 12.83±0.58 | 12.50±0.63 | 12.15±0.62 | 11.95±0.62 | 12.64±0.66 |
| agnews | 17.03±0.00 | 16.50±0.00 | 15.92±0.00 | 15.29±0.00 | 15.07±0.00 | 15.96±0.00 | 16.68±0.00 | 15.98±0.00 | 15.35±0.00 | 14.71±0.00 | 14.50±0.00 | 15.45±0.00 |

Table 15.2: Average AUPR ± standard dev. over five seeds for the semi-supervised setting of ICL and DTE-C baselines with varying hyperparameter (HP) values; For ICL, the learning rate ∈ {0.1, 0.02, 0.001, 0.0001, 1e − 05}, for DTE-C, $k \in \{5, 10, 20, 40, 50\}$. Also reported is the ^avg model. We use **bold** and underline respectively to mark the **best** and the worst performance of each model to showcase the variability of performance across different HP settings.

| dataset | ICL-0.1 | ICL-0.01 | ICL-0.001 | ICL-0.0001 | ICL-1e-05 | ICL-avr | DTE-C-5 | DTE-C-10 | DTE-C-20 | DTE-C-40 | DTE-C-50 | DTE-C-avr |
|---|---|---|---|---|---|---|---|---|---|---|---|---|
| aloi | 5.50±0.09 | 5.39±0.07 | 5.50±0.05 | **5.59±0.04** | 5.46±0.01 | 5.49±0.02 | 5.76±0.03 | 5.82±0.02 | 5.72±0.05 | 5.73±0.05 | **5.91±0.00** | 5.79±0.01 |
| amazon | 10.06±0.10 | **10.08±0.03** | 9.91±0.13 | 10.01±0.05 | 9.99±0.02 | 10.01±0.05 | 11.01±0.43 | 10.99±1.11 | **11.05±0.93** | 9.52±0.00 | 9.52±0.00 | 10.42±0.40 |
| annthyroid | **58.53±13.33** | 39.69±5.23 | 53.66±3.08 | 53.85±5.44 | 55.94±2.70 | 52.34±3.72 | 82.46±0.50 | 83.25±0.34 | **83.27±0.63** | 81.52±1.18 | 13.81±0.00 | 68.86±0.20 |
| backdoor | 85.81±2.45 | 88.14±1.38 | **88.99±1.12** | 88.96±1.20 | 86.24±0.75 | 87.63±1.03 | 43.90±3.90 | 61.21±2.58 | **63.64±1.61** | 4.83±0.09 | 4.83±0.09 | 35.68±0.79 |
| breastw | 98.65±0.81 | 98.46±0.51 | **98.98±0.57** | 98.50±0.42 | 95.32±1.11 | 97.98±0.31 | 88.69±2.54 | 89.49±2.00 | 94.04±1.49 | 98.56±0.68 | **99.22±0.11** | 94.00±0.69 |
| campaign | 47.46±0.41 | 44.03±2.05 | 47.94±0.25 | 49.22±0.91 | **51.41±0.51** | 48.01±0.47 | **49.90±2.01** | 46.77±1.41 | 48.40±1.60 | 20.25±0.00 | 20.25±0.00 | 37.11±0.64 |
| cardio | 48.81±14.30 | **65.70±11.26** | 63.55±5.07 | 61.75±2.55 | 39.67±1.62 | 53.90±3.02 | 69.32±0.43 | 70.19±0.63 | **70.25±0.47** | 17.78±0.59 | 17.55±0.00 | 49.02±0.29 |
| cardiotocography | 43.21±5.03 | **52.56±1.14** | 50.69±2.12 | 49.01±1.29 | 39.29±1.23 | 47.05±1.34 | **53.83±0.94** | 53.56±0.73 | 49.12±4.17 | 36.12±0.00 | 36.12±0.00 | 45.75±0.91 |
| celeba | 12.89±1.17 | **13.90±1.49** | 13.09±1.73 | 13.50±1.15 | 12.97±0.55 | 13.27±0.47 | **15.16±1.05** | 13.87±0.59 | 12.60±1.22 | 4.31±0.09 | 4.31±0.09 | 10.05±0.49 |
| census | 20.29±1.27 | 20.38±2.24 | 23.32±0.50 | **23.68±0.70** | 22.37±0.59 | 22.01±0.42 | **18.39±0.83** | 17.28±0.52 | 17.45±0.64 | 11.66±0.20 | 11.66±0.20 | 15.29±0.31 |
| cover | 22.58±13.63 | 10.32±4.93 | 35.90±22.12 | **47.50±19.76** | 40.81±22.18 | 31.42±3.97 | **71.28±4.54** | 61.74±4.63 | 38.26±3.29 | 18.86±15.32 | 11.20±0.14 | 35.03±0.82 |
| donors | 39.48±10.00 | 77.81±8.28 | 82.59±7.03 | 91.60±4.15 | **92.24±2.17** | 76.75±3.27 | **76.07±1.84** | 66.66±3.73 | 53.74±2.07 | 51.69±0.25 | 51.74±0.32 | 45.30±2.85 |
| fault | **65.37±3.40** | 64.58±1.81 | 63.83±0.74 | 64.32±0.96 | 64.41±1.09 | 64.41±1.09 | 63.92±1.40 | 62.97±0.60 | 63.62±1.06 | 56.20±2.31 | 1.94±0.07 | 58.79±0.41 |
| fraud | 51.45±12.34 | 49.97±9.27 | 56.24±9.47 | 62.41±10.30 | **72.24±6.51** | 58.46±7.58 | **68.85±10.35** | 57.17±4.68 | 24.97±21.19 | 20.53±3.60 | 0.34±0.03 | 30.33±4.25 |
| glass | 49.70±15.55 | 69.98±13.15 | 87.25±7.44 | 87.79±8.75 | 79.79±8.75 | 76.58±1.38 | 46.09±6.97 | 36.80±5.13 | 36.80±3.60 | 20.53±3.60 | 7.83±1.03 | 27.78±2.62 |
| hepatitis | 99.85±0.30 | 97.89±3.11 | 98.94±2.13 | 98.91±0.48 | 99.93±0.14 | 99.28±1.20 | 98.64±1.76 | 95.49±4.81 | 96.17±4.38 | 54.16±11.62 | 27.45±1.71 | 74.38±1.79 |
| http | 94.85±9.03 | 95.76±3.25 | 97.86±1.72 | 99.77±0.25 | 99.55±0.61 | 97.56±2.20 | 56.93±7.39 | 69.18±22.25 | 50.00±7.08 | 48.53±7.22 | 0.74±0.04 | 45.08±7.07 |
| imdb | 10.09±0.09 | 10.02±0.07 | 10.16±0.04 | 10.18±0.05 | 10.41±0.04 | 10.17±0.03 | 8.71±0.63 | 9.49±0.47 | 8.79±0.30 | 9.52±0.00 | 9.52±0.00 | 9.21±0.20 |
| internetads | 57.32±2.19 | 60.94±1.44 | 62.86±1.88 | 63.83±0.88 | 47.64±2.20 | 58.52±1.04 | 60.45±4.51 | 56.24±2.74 | 57.78±6.93 | 31.53±0.00 | 31.53±0.00 | 47.51±1.92 |
| ionosphere | 96.94±2.27 | 97.20±2.57 | 99.00±0.36 | 98.91±0.48 | 97.54±1.39 | 97.92±1.11 | 96.52±0.77 | 96.49±1.17 | 86.50±4.57 | 86.50±4.57 | 56.47±15.94 | 86.53±4.28 |
| landsat | 58.66±1.95 | 56.49±1.07 | 55.72±1.20 | 55.69±0.77 | 52.73±0.98 | 55.86±0.60 | 36.22±0.76 | 36.06±2.33 | 34.01±2.31 | 34.27±0.09 | 34.32±0.09 | 34.98±0.81 |
| letter | 11.22±1.58 | 10.84±0.96 | 9.41±0.34 | 9.71±0.55 | 15.87±1.71 | 11.41±0.43 | 8.92±0.10 | 9.01±0.23 | 8.96±0.09 | 11.76±0.00 | 11.76±0.00 | 10.09±0.04 |
| lymphography | 100.00±0.00 | 100.00±0.00 | 100.00±0.00 | 100.00±0.00 | 100.00±0.00 | 100.00±0.00 | 87.03±7.20 | 93.30±2.48 | 83.45±12.01 | 72.50±16.37 | 52.03±0.00 | 68.82±5.96 |
| magic-gamma | 73.92±3.66 | 81.23±2.83 | 82.64±0.71 | 82.47±0.98 | 83.45±0.84 | 80.74±1.03 | 89.01±0.45 | 89.46±0.29 | 89.15±0.44 | 66.79±18.08 | 52.03±0.00 | 77.29±3.52 |
| mammography | 33.51±8.47 | 37.72±7.54 | 27.06±2.69 | 26.94±1.92 | 28.51±2.57 | 28.51±2.57 | 35.02±6.35 | 37.47±2.68 | 32.22±4.57 | 35.30±2.06 | 4.54±0.00 | 28.91±1.92 |
| mnist | 49.27±1.39 | 50.40±3.02 | 54.46±1.34 | 63.49±1.45 | 65.20±1.41 | 56.56±0.65 | 60.64±1.13 | 56.20±2.31 | 55.17±3.05 | 16.86±0.00 | 16.86±0.00 | 41.15±1.11 |
| musk | 100.00±0.00 | 86.99±1.74 | 100.00±0.00 | 100.00±0.00 | 100.00±0.00 | 89.48±1.94 | 100.00±0.00 | 100.00±0.00 | 100.00±0.00 | 48.08±0.00 | 48.08±0.00 | 62.46±0.00 |
| optdigits | 76.76±8.77 | 36.36±3.03 | 88.29±3.65 | 61.20±0.68 | 42.39±5.22 | 41.08±1.34 | 18.29±1.20 | 11.55±3.63 | 12.82±3.07 | 5.59±0.00 | 6.14±0.00 | 10.77±1.02 |
| pageblocks | 27.13±3.47 | 66.26±4.89 | 63.86±2.44 | 61.28±3.91 | 68.37±1.14 | 64.91±1.13 | 64.41±1.43 | 68.08±1.72 | 63.78±4.40 | 59.09±4.25 | 5.59±0.00 | 54.53±1.01 |
| pendigits | 64.80±3.36 | 66.20±12.32 | 63.38±8.59 | 73.08±7.23 | 48.40±11.06 | 58.53±6.17 | 53.63±4.87 | 45.92±5.63 | 40.53±4.47 | 4.44±0.00 | 17.28±0.00 | 29.79±0.96 |
| pima | 39.64±20.69 | 71.44±2.85 | 78.53±2.86 | 76.72±1.68 | 75.58±3.60 | 75.31±2.15 | 67.82±2.03 | 65.53±3.71 | 66.04±3.31 | 70.01±2.92 | 34.42±0.19 | 67.61±2.81 |
| satellite | 74.29±3.85 | 81.18±2.88 | 88.43±0.25 | 90.38±0.44 | 86.65±0.60 | 86.65±2.16 | 85.28±0.60 | 84.72±0.41 | 84.72±0.41 | 55.55±14.95 | 68.66±8.30 | 71.75±2.86 |
| satimage-2 | 76.76±8.77 | 77.53±35.36 | 96.77±0.59 | 95.65±0.53 | 50.77±13.44 | 71.56±7.98 | 83.45±5.57 | 46.25±6.02 | 46.25±6.02 | 24.56±27.32 | 2.42±0.00 | 43.90±6.54 |
| shuttle | 97.77±1.43 | 99.19±0.23 | 99.04±1.73 | 99.91±0.05 | 99.72±0.12 | 99.28±0.28 | 94.26±0.06 | 94.06±0.16 | 93.19±0.08 | 88.77±1.78 | 17.28±0.00 | 76.73±0.36 |
| skin | 44.77±20.35 | 73.00±9.87 | 89.36±6.27 | 37.40±6.86 | 50.36±5.65 | 56.71±7.09 | 68.88±0.57 | 70.08±0.63 | 69.80±0.53 | 69.53±0.66 | 34.42±0.19 | 62.54±0.43 |
| smtp | 38.06±20.67 | 42.88±7.84 | 38.77±14.99 | 82.51±0.72 | 36.22±4.31 | 38.67±3.38 | 83.44±0.47 | 52.14±4.82 | 41.15±7.61 | 15.36±9.94 | 0.07±0.01 | 31.76±4.31 |
| spambase | 80.80±1.90 | 81.18±2.88 | 85.16±0.70 | 85.42±0.58 | 82.51±0.72 | 83.00±0.50 | 50.08±5.85 | 85.11±0.52 | 84.12±0.39 | 57.05±0.00 | 73.05±0.13 | 73.05±0.13 |
| speech | 3.74±1.02 | 3.79±0.38 | 3.92±0.39 | 3.46±0.20 | 3.41±0.23 | 3.66±0.23 | 2.78±0.12 | 3.32±0.58 | 2.85±0.16 | 3.26±0.00 | 3.26±0.00 | 3.09±0.15 |
| stamps | 45.19±8.30 | 52.05±12.46 | 80.41±3.57 | 80.49±4.25 | 74.72±8.62 | 66.57±4.58 | 61.33±7.30 | 57.64±7.65 | 53.34±6.09 | 56.82±10.61 | 24.85±19.23 | 50.80±7.62 |
| thyroid | 72.79±3.97 | 56.56±5.89 | 56.38±5.72 | 60.13±6.20 | 30.60±3.56 | 55.29±1.31 | 80.34±1.28 | 81.93±2.54 | 82.86±1.27 | 83.14±1.08 | 4.81±0.00 | 66.61±0.60 |
| vertebral | 25.76±4.65 | 31.37±8.07 | 50.83±13.40 | 59.35±3.61 | 54.90±6.08 | 44.44±4.60 | 34.31±5.07 | 31.45±6.44 | 28.95±5.65 | 25.00±4.75 | 21.17±1.01 | 28.18±4.88 |
| vowels | 24.00±15.67 | 21.57±7.57 | 28.35±6.64 | 28.35±6.64 | 22.16±3.78 | 24.91±3.58 | 39.58±4.05 | 41.45±3.27 | 44.91±1.07 | 6.64±0.00 | 6.64±0.00 | 27.84±1.66 |
| waveform | 51.44±2.31 | 34.95±18.32 | 28.41±9.48 | 9.51±0.97 | 20.22±2.30 | 28.91±5.68 | 9.88±0.96 | 9.59±0.83 | 10.41±0.55 | 67.56±7.06 | 5.65±0.00 | 8.24±0.18 |
| wbc | 82.98±11.29 | 91.66±5.01 | 99.16±1.35 | 99.16±1.35 | 96.58±2.18 | 93.88±3.76 | 36.71±3.33 | 33.53±4.39 | 40.09±7.65 | 15.20±19.59 | 8.72±1.13 | 37.32±1.43 |
| wdbc | 66.53±18.38 | 85.94±10.45 | 89.36±6.27 | 92.02±6.00 | 89.28±3.59 | 84.63±3.40 | 76.31±10.20 | 73.63±7.98 | 76.35±9.33 | 15.20±19.59 | 5.21±0.94 | 49.34±5.19 |
| wilt | 11.08±3.73 | 99.19±0.23 | 32.19±7.32 | 32.00±2.89 | 31.22±2.08 | 23.88±1.71 | 25.16±0.41 | 27.94±1.32 | 21.04±1.52 | 25.25±1.80 | 10.13±0.00 | 21.90±0.77 |
| wine | 98.06±3.88 | 98.33±1.88 | 99.48±0.75 | 97.94±4.12 | 98.11±3.43 | 98.38±2.75 | 99.47±0.66 | 98.39±1.77 | 98.70±1.77 | 45.36±23.65 | 14.95±1.08 | 71.37±4.46 |
| wpbc | 82.28±2.77 | 66.72±11.51 | 85.45±3.99 | 85.99±2.53 | 83.48±3.70 | 80.78±4.52 | 60.51±5.10 | 61.17±3.87 | 59.53±3.80 | 40.41±3.04 | 38.36±1.82 | 52.00±2.61 |
| yeast | 48.32±2.44 | 49.17±1.53 | 49.01±0.39 | 49.32±0.70 | 47.21±0.41 | 48.61±0.81 | 49.65±0.87 | 50.38±1.14 | 49.94±1.24 | 47.04±1.19 | 50.90±0.00 | 49.58±0.27 |
| yelp | 9.79±0.11 | 9.91±0.05 | 9.82±0.17 | 9.84±0.06 | 9.59±0.03 | 9.79±0.05 | 14.09±1.54 | 12.96±1.75 | 13.22±1.41 | 9.52±0.00 | 9.52±0.00 | 11.86±0.62 |
| MNIST-C | 44.47±0.54 | 48.19±0.57 | 50.18±0.26 | 50.34±0.31 | 45.98±0.37 | 47.83±0.12 | 46.68±0.62 | 56.00±0.23 | 46.12±0.32 | 9.52±0.00 | 9.52±0.00 | 31.85±0.20 |
| FashionMNIST | 58.47±0.45 | 64.11±0.18 | 64.96±0.32 | 65.22±0.32 | 55.84±0.54 | 61.72±0.18 | 54.92±0.36 | 19.61±0.15 | 56.31±0.95 | 9.50±0.00 | 9.50±0.00 | 37.25±0.17 |
| CIFAR10 | 14.30±0.27 | 17.49±0.41 | 19.10±0.21 | 18.89±0.21 | 13.77±0.14 | 16.71±0.19 | 19.95±0.09 | 19.79±0.18 | 19.79±0.18 | 9.52±0.00 | 9.52±0.00 | 15.68±0.07 |
| SVHN | 13.92±0.24 | 15.32±0.05 | 15.82±0.01 | 15.74±0.09 | 15.01±0.13 | 15.16±0.06 | 15.61±0.09 | 15.55±0.11 | 15.55±0.11 | 9.52±0.00 | 9.52±0.00 | 13.16±0.04 |
| MVTec-AD | 86.94±0.92 | 88.30±0.67 | 88.91±0.75 | 88.97±0.75 | 83.64±0.91 | 87.35±0.76 | 83.04±1.01 | 84.66±1.01 | 84.41±1.22 | 88.89±1.52 | 38.21±1.62 | 65.84±0.58 |
| 20news | 12.37±0.20 | 13.20±0.34 | 13.52±0.34 | 13.98±0.15 | 11.83±0.29 | 12.98±0.23 | 17.28±1.40 | 15.75±0.40 | 14.55±0.94 | 9.44±0.39 | 9.44±0.39 | 13.29±0.47 |
| agnews | 12.45±0.12 | 12.84±0.25 | 13.04±0.08 | 13.05±0.04 | 12.55±0.05 | 12.79±0.06 | 18.40±0.64 | 16.51±0.99 | 16.18±1.53 | 9.52±0.00 | 9.52±0.00 | 14.03±0.43 |

Table 16.1: Average F1 score ± standard dev. over five seeds for the semi-supervised setting of DTE-NP, kNN baselines with varying hyperparameter (HP) values; $k \in \{5, 10, 20, 40, 50\}$. Also reported is the avg model. We use **bold** and underline respectively to mark the **best** and the worst performance of each model to showcase the variability of performance across different HP settings.

| dataset | DTE-NP-5 | DTE-NP-10 | DTE-NP-20 | DTE-NP-40 | DTE-NP-50 | DTE-NP-avr | KNN-5 | KNN-10 | KNN-20 | KNN-40 | KNN-50 | KNN-avr |
|---|---|---|---|---|---|---|---|---|---|---|---|---|
| aloi | 5.90±0.00 | 5.84±0.00 | 5.70±0.00 | 5.90±0.00 | 5.97±0.00 | 5.86±0.00 | 5.90±0.00 | 5.64±0.00 | 5.97±0.00 | 6.17±0.00 | 6.37±0.00 | 6.01±0.00 |
| amazon | 10.80±0.00 | 10.80±0.00 | 10.20±0.00 | 10.20±0.00 | 11.00±0.00 | 10.60±0.00 | 11.40±0.00 | 10.20±0.00 | 10.60±0.00 | 11.20±0.00 | 11.20±0.00 | 10.92±0.00 |
| annthyroid | 62.55±0.00 | 61.80±0.00 | 60.67±0.00 | 58.99±0.00 | 58.80±0.00 | 60.56±0.00 | 61.99±0.00 | 60.49±0.00 | 58.43±0.00 | 58.24±0.00 | 56.74±0.00 | 59.18±0.00 |
| backdoor | 64.15±1.04 | 52.30±1.87 | 40.62±1.46 | 30.25±1.34 | 26.96±1.20 | 42.86±1.32 | 52.53±1.63 | 60.05±0.33 | 58.71±1.50 | 20.21±0.78 | 17.52±0.83 | 31.87±1.22 |
| breastw | 96.72±0.64 | 96.23±0.39 | 96.17±0.47 | 95.99±0.39 | 95.99±0.39 | 96.22±0.43 | 96.00±0.44 | 96.05±0.33 | 95.87±0.28 | 95.99±0.39 | 95.93±0.32 | 95.97±0.31 |
| campaign | 49.94±0.00 | 50.62±0.00 | 51.14±0.00 | 51.38±0.00 | 51.57±0.00 | 50.93±0.00 | 50.37±0.00 | 51.27±0.00 | 51.27±0.00 | 51.70±0.00 | 51.29±0.00 | 51.10±0.00 |
| cardio | 63.64±0.00 | 61.36±0.00 | 61.93±0.00 | 63.64±0.00 | 64.20±0.00 | 62.95±0.00 | 61.93±0.00 | 61.93±0.00 | 61.93±0.00 | 67.61±0.00 | 69.32±0.00 | 65.00±0.00 |
| cardiotocography | 44.64±0.00 | 45.71±0.00 | 47.00±0.00 | 48.50±0.00 | 49.14±0.00 | 47.00±0.00 | 46.35±0.00 | 46.78±0.00 | 47.85±0.00 | 50.86±0.00 | 51.93±0.00 | 48.76±0.00 |
| celeba | 15.83±0.69 | 17.05±0.43 | 18.17±0.61 | 19.02±0.69 | 19.30±0.60 | 17.87±0.57 | 17.08±0.58 | 18.41±0.65 | 19.30±0.81 | 20.27±0.68 | 20.48±0.68 | 19.11±0.61 |
| census | 22.22±0.54 | 21.93±0.52 | 21.46±0.25 | 21.38±0.48 | 21.12±0.29 | 21.62±0.14 | 22.23±0.42 | 21.48±0.40 | 21.47±0.57 | 21.33±0.65 | 21.26±0.50 | 21.55±0.24 |
| cover | 69.15±2.12 | 66.87±2.35 | 63.15±2.07 | 55.99±2.08 | 53.06±2.23 | 61.65±2.14 | 65.04±1.92 | 60.56±2.04 | 52.76±1.83 | 42.69±1.94 | 39.92±1.99 | 52.19±1.92 |
| donors | 97.27±0.36 | 96.20±0.45 | 94.49±0.55 | 91.70±0.90 | 90.52±0.86 | 94.05±0.60 | 94.98±0.62 | 92.36±0.59 | 88.71±0.99 | 80.36±1.90 | 76.62±1.50 | 86.60±0.98 |
| fault | 56.02±0.00 | 55.72±0.00 | 55.57±0.00 | 56.91±0.00 | 57.36±0.00 | 56.32±0.00 | 55.57±0.00 | 55.87±0.00 | 57.06±0.00 | 57.50±0.00 | 58.25±0.00 | 56.85±0.00 |
| fraud | 48.18±4.56 | 49.60±3.28 | 49.00±3.95 | 46.66±3.52 | 45.46±3.60 | 47.78±3.53 | 47.78±3.53 | 49.39±5.24 | 45.76±0.00 | 42.58±3.16 | 41.64±3.36 | 45.61±3.48 |
| glass | 47.81±5.78 | 35.14±2.75 | 27.98±4.21 | 18.37±2.40 | 17.81±2.98 | 29.42±2.61 | 29.87±9.62 | 22.55±6.87 | 18.58±4.19 | 17.23±3.73 | 17.23±3.73 | 21.09±5.16 |
| hepatitis | 98.94±1.41 | 94.16±2.13 | 81.68±4.18 | 75.95±4.78 | 71.99±4.14 | 84.54±2.23 | 81.98±4.50 | 66.04±4.53 | 62.31±6.09 | 60.39±6.21 | 60.04±7.30 | 66.15±4.71 |
| http | 98.50±0.38 | 95.10±2.50 | 88.26±1.38 | 82.85±3.80 | 78.96±6.40 | 88.73±2.54 | 100.00±0.00 | 98.57±2.86 | 92.67±0.91 | 92.67±0.91 | 92.67±0.91 | 95.32±0.75 |
| imdb | 5.20±0.00 | 5.40±0.00 | 5.40±0.00 | 5.20±0.00 | 5.20±0.00 | 5.28±0.00 | 5.40±0.00 | 5.40±0.00 | 5.40±0.00 | 5.00±0.00 | 5.00±0.00 | 5.24±0.00 |
| internetads | 55.16±0.00 | 51.63±0.00 | 48.37±0.00 | 46.47±0.00 | 46.20±0.00 | 49.57±0.00 | 51.90±0.00 | 46.20±0.00 | 45.11±0.00 | 45.11±0.00 | 45.11±0.00 | 46.68±0.00 |
| ionosphere | 92.33±1.17 | 92.05±1.63 | 91.63±1.09 | 90.41±1.35 | 89.19±1.72 | 91.12±1.24 | 90.23±1.86 | 91.81±1.87 | 89.45±1.91 | 85.06±3.37 | 82.75±3.12 | 87.86±1.86 |
| landsat | 52.29±0.00 | 51.24±0.00 | 49.06±0.00 | 45.99±0.00 | 45.39±0.00 | 48.79±0.00 | 51.46±0.00 | 49.29±0.00 | 45.76±0.00 | 42.69±0.00 | 41.34±0.00 | 46.11±0.00 |
| letter | 1.00±0.00 | 1.00±0.00 | 1.00±0.00 | 1.00±0.00 | 1.00±0.00 | 1.00±0.00 | 1.00±0.00 | 1.00±0.00 | 1.00±0.00 | 1.00±0.00 | 1.00±0.00 | 1.00±0.00 |
| lymphography | 97.89±4.21 | 93.61±7.97 | 94.67±6.53 | 92.71±6.09 | 91.66±7.34 | 94.11±5.93 | 91.57±5.96 | 90.69±3.65 | 90.69±3.65 | 92.96±4.97 | 92.96±4.97 | 91.77±3.69 |
| magic.gamma | 76.79±0.00 | 76.20±0.00 | 75.49±0.00 | 74.84±0.00 | 74.75±0.00 | 75.61±0.00 | 76.17±0.00 | 76.17±0.00 | 74.60±0.00 | 73.49±0.00 | 73.00±0.00 | 74.48±0.00 |
| mammography | 41.92±0.00 | 41.15±0.00 | 44.23±0.00 | 44.23±0.00 | 44.23±0.00 | 43.15±0.00 | 40.38±0.00 | 43.46±0.00 | 43.85±0.00 | 43.85±0.00 | 43.46±0.00 | 43.00±0.00 |
| mnist | 72.71±0.00 | 72.29±0.00 | 71.57±0.00 | 70.43±0.00 | 69.86±0.00 | 71.37±0.00 | 71.86±0.00 | 71.29±0.00 | 69.86±0.00 | 69.71±0.00 | 69.71±0.00 | 70.49±0.00 |
| musk | 100.00±0.00 | 100.00±0.00 | 100.00±0.00 | 100.00±0.00 | 100.00±0.00 | 100.00±0.00 | 100.00±0.00 | 100.00±0.00 | 100.00±0.00 | 100.00±0.00 | 100.00±0.00 | 100.00±0.00 |
| optdigits | 30.00±0.00 | 24.00±0.00 | 12.00±0.00 | 7.33±0.00 | 6.67±0.00 | 16.00±0.00 | 21.33±0.00 | 12.00±0.00 | 6.67±0.00 | 2.67±0.00 | 2.00±0.00 | 8.93±0.00 |
| pageblocks | 59.41±0.00 | 59.22±0.00 | 59.61±0.00 | 58.43±0.00 | 58.43±0.00 | 59.02±0.00 | 59.02±0.00 | 59.22±0.00 | 60.20±0.00 | 56.08±0.00 | 56.27±0.00 | 58.16±0.00 |
| pendigits | 94.23±0.00 | 92.31±0.00 | 91.03±0.00 | 80.13±0.00 | 78.21±0.00 | 87.18±0.00 | 90.38±0.00 | 90.38±0.00 | 73.72±0.00 | 64.74±0.00 | 62.82±0.00 | 76.41±0.00 |
| pima | 74.73±2.13 | 72.94±2.46 | 71.48±1.96 | 71.44±2.36 | 71.93±1.99 | 72.50±1.97 | 71.03±2.53 | 70.18±2.12 | 71.58±2.16 | 71.67±2.10 | 72.03±2.02 | 71.30±2.09 |
| satellite | 72.20±0.00 | 71.76±0.00 | 70.68±0.00 | 69.79±0.00 | 69.20±0.00 | 70.73±0.00 | 71.81±0.00 | 70.73±0.00 | 69.84±0.00 | 67.93±0.00 | 67.29±0.00 | 69.52±0.00 |
| satimage-2 | 90.14±0.00 | 90.14±0.00 | 90.14±0.00 | 92.96±0.00 | 92.96±0.00 | 91.27±0.00 | 90.14±0.00 | 91.55±0.00 | 92.96±0.00 | 92.96±0.00 | 92.96±0.00 | 92.11±0.00 |
| shuttle | 98.35±0.00 | 98.23±0.00 | 98.15±0.00 | 98.12±0.00 | 98.12±0.00 | 98.19±0.00 | 98.23±0.00 | 98.06±0.00 | 98.06±0.00 | 98.09±0.00 | 98.09±0.00 | 98.11±0.00 |
| skin | 97.12±0.46 | 96.83±0.38 | 96.14±0.17 | 94.73±0.23 | 94.30±0.27 | 95.82±0.14 | 96.71±0.41 | 95.85±0.17 | 94.56±0.29 | 91.73±0.16 | 90.60±0.28 | 93.89±0.13 |
| smtp | 68.05±5.12 | 83.36±7.16 | 81.00±0.00 | 81.00±4.08 | 63.34±8.31 | 67.73±4.99 | 80.92±4.47 | 78.72±5.61 | 79.49±5.02 | 77.95±6.52 | 76.84±4.81 | 78.79±5.12 |
| spambase | 80.88±0.00 | 80.29±0.00 | 80.23±0.00 | 79.81±0.00 | 79.45±0.00 | 80.13±0.00 | 80.52±0.00 | 79.93±0.00 | 79.15±0.00 | 78.98±0.00 | 78.80±0.00 | 79.48±0.00 |
| speech | 3.28±0.00 | 3.28±0.00 | 1.64±0.00 | 1.64±0.00 | 3.28±0.00 | 2.62±0.00 | 3.28±0.00 | 3.28±0.00 | 3.28±0.00 | 3.28±0.00 | 3.28±0.00 | 3.28±0.00 |
| stamps | 85.93±2.66 | 80.63±2.92 | 74.04±4.45 | 68.12±7.14 | 66.98±6.79 | 75.14±4.45 | 78.57±8.41 | 68.89±8.23 | 62.67±6.19 | 60.52±7.08 | 61.78±6.58 | 66.49±7.15 |
| thyroid | 75.27±0.00 | 75.27±0.00 | 74.19±0.00 | 74.19±0.00 | 74.62±0.00 | 74.62±0.00 | 75.27±0.00 | 73.12±0.00 | 72.04±0.00 | 73.12±0.00 | 74.19±0.00 | 73.55±0.00 |
| vertebral | 49.03±4.93 | 32.73±5.11 | 24.06±3.90 | 15.07±1.64 | 12.51±2.44 | 26.08±3.24 | 23.02±5.17 | 17.18±2.01 | 12.19±3.52 | 9.07±3.30 | 8.51±2.65 | 13.99±2.95 |
| vowels | 28.00±0.00 | 28.00±0.00 | 28.00±0.00 | 30.00±0.00 | 30.00±0.00 | 28.80±0.00 | 26.00±0.00 | 26.00±0.00 | 30.00±0.00 | 26.00±0.00 | 26.00±0.00 | 26.80±0.00 |
| waveform | 26.00±0.00 | 26.00±0.00 | 27.00±0.00 | 28.00±0.00 | 28.00±0.00 | 27.00±0.00 | 27.00±0.00 | 27.00±0.00 | 27.00±0.00 | 27.00±0.00 | 27.00±0.00 | 27.00±0.00 |
| wbc | 89.35±3.08 | 88.05±3.35 | 88.05±3.35 | 89.16±2.38 | 89.16±2.38 | 88.75±2.48 | 88.65±4.53 | 84.82±3.86 | 83.72±5.17 | 89.16±2.38 | 89.16±2.38 | 86.10±2.43 |
| wdbc | 85.05±6.11 | 83.36±7.16 | 81.00±4.08 | 81.00±4.08 | 81.00±4.08 | 82.28±4.91 | 80.92±4.47 | 78.72±5.61 | 78.72±5.61 | 76.84±4.81 | 76.84±4.81 | 78.79±5.12 |
| wilt | 3.50±0.00 | 2.33±0.00 | 1.56±0.00 | 0.78±0.00 | 0.78±0.00 | 1.79±0.00 | 2.33±0.00 | 1.56±0.00 | 0.78±0.00 | 0.78±0.00 | 0.78±0.00 | 1.25±0.00 |
| wine | 97.73±3.52 | 93.65±2.20 | 86.39±3.51 | 83.56±1.93 | 80.50±3.68 | 88.37±1.61 | 85.91±3.49 | 80.47±6.23 | 80.89±5.12 | 78.83±4.40 | 78.83±4.40 | 80.99±2.84 |
| wpbc | 75.11±2.68 | 63.70±1.39 | 53.31±2.33 | 45.02±2.91 | 43.40±2.66 | 56.11±1.23 | 50.17±2.76 | 42.79±2.99 | 47.14±0.00 | 37.87±3.42 | 39.00±3.24 | 41.99±2.22 |
| yeast | 45.76±0.00 | 46.35±0.00 | 46.35±0.00 | 46.75±0.00 | 46.75±0.00 | 46.39±0.00 | 46.75±0.00 | 45.96±0.00 | 47.14±0.00 | 46.75±0.00 | 46.94±0.00 | 46.71±0.00 |
| yelp | 19.40±0.00 | 18.80±0.00 | 17.40±0.00 | 16.40±0.00 | 16.60±0.00 | 17.72±0.00 | 18.80±0.00 | 17.60±0.00 | 17.40±0.00 | 16.20±0.00 | 16.80±0.00 | 17.36±0.00 |
| MNIST-C | 47.51±0.00 | 46.59±0.00 | 45.57±0.00 | 44.74±0.00 | 44.61±0.00 | 45.84±0.00 | 46.30±0.00 | 45.24±0.00 | 44.46±0.00 | 43.87±0.00 | 43.70±0.00 | 44.72±0.00 |
| FashionMNIST | 59.75±0.00 | 59.17±0.00 | 58.76±0.00 | 58.16±0.00 | 57.94±0.00 | 58.76±0.00 | 59.05±0.00 | 58.48±0.00 | 57.87±0.00 | 57.27±0.00 | 57.21±0.00 | 57.97±0.00 |
| CIFAR10 | 23.23±0.00 | 23.00±0.00 | 22.89±0.00 | 22.70±0.00 | 22.59±0.00 | 22.88±0.00 | 22.85±0.00 | 22.89±0.00 | 22.78±0.00 | 22.47±0.00 | 22.47±0.00 | 22.69±0.00 |
| SVHN | 19.19±0.00 | 18.99±0.00 | 18.90±0.00 | 18.80±0.00 | 18.58±0.00 | 18.89±0.00 | 18.95±0.00 | 18.79±0.00 | 18.70±0.00 | 18.42±0.00 | 18.38±0.00 | 18.65±0.00 |
| MVTec-AD | 75.68±0.99 | 71.31±0.96 | 68.50±0.99 | 66.57±0.72 | 66.05±0.61 | 69.62±0.83 | 67.06±1.03 | 65.80±0.80 | 65.00±0.73 | 64.20±0.60 | 64.09±0.54 | 65.23±0.73 |
| 20news | 18.08±1.41 | 16.56±1.63 | 14.56±1.60 | 12.95±1.47 | 12.86±1.43 | 15.00±1.49 | 15.11±1.90 | 13.63±1.52 | 12.70±1.43 | 11.85±1.43 | 10.81±1.54 | 12.82±1.49 |
| agnews | 20.80±0.00 | 20.30±0.00 | 19.60±0.00 | 18.35±0.00 | 17.95±0.00 | 19.40±0.00 | 20.10±0.00 | 19.25±0.00 | 18.10±0.00 | 16.85±0.00 | 16.40±0.00 | 18.14±0.00 |

Table 16.2: Average F1 score ± standard dev. over five seeds for the semi-supervised setting of ICL and DTE-C baselines with varying hyperparameter (HP) values; For ICL, the learning rate $\in \{0.1, 0.02, 0.001, 0.0001, 1e-05\}$, for DTE-C, $k \in \{5, 10, 20, 40, 50\}$. Also reported is the $^{avg}$ model. We use **bold** and underline respectively to mark the **best** and the underline worst performance of each model to showcase the variability of performance across different HP settings.

| dataset | ICL-0.1 | ICL-0.01 | ICL-0.001 | ICL-0.0001 | ICL-1e-05 | ICL-avr | DTE-C-5 | DTE-C-10 | DTE-C-20 | DTE-C-40 | DTE-C-50 | DTE-C-avr |
|---|---|---|---|---|---|---|---|---|---|---|---|---|
| aloi | 4.51±0.69 | 4.34±0.42 | **5.28**±0.47 | 4.68±0.30 | 4.16±0.38 | 4.59±0.07 | **4.75**±0.27 | 4.27±0.19 | 4.28±0.10 | 4.51±0.17 | 0.00±0.00 | 3.56±0.03 |
| amazon | **10.44**±0.46 | 9.76±0.34 | 9.92±0.84 | 10.08±0.35 | 9.52±0.43 | 9.94±0.32 | 11.48±0.97 | **11.96**±1.68 | 11.60±2.07 | 0.00±0.00 | 0.00±0.00 | 7.01±0.72 |
| annthyroid | 54.87±13.24 | 42.25±3.55 | 53.45±4.13 | 54.72±5.45 | **85.37**±1.01 | 52.56±3.29 | 77.23±0.25 | 77.94±0.28 | 77.53±0.85 | 75.43±0.93 | 0.00±0.00 | 61.63±0.20 |
| backdoor | 87.17±0.98 | **87.32**±0.99 | 87.11±1.09 | 86.85±0.95 | 86.76±1.00 | 86.76±1.00 | 46.19±8.39 | 83.03±2.14 | **84.50**±0.60 | 75.43±0.93 | 0.00±0.00 | 42.75±1.54 |
| breastw | 95.98±0.34 | 96.07±0.94 | 96.80±0.40 | 97.44±0.55 | 96.11±0.75 | 96.48±0.28 | 88.80±1.59 | 90.10±1.35 | 92.46±1.78 | 95.31±0.70 | **96.11**±1.44 | 92.50±0.79 |
| campaign | 48.12±0.36 | 46.81±1.72 | 50.68±0.66 | 51.37±0.85 | 53.40±0.51 | 50.07±0.49 | 51.98±0.70 | 52.45±1.07 | 52.33±1.00 | 0.00±0.00 | 0.00±0.00 | 31.35±0.44 |
| cardio | 49.09±11.28 | **61.93**±5.57 | 58.86±1.59 | 57.95±2.30 | 40.57±4.28 | 53.68±2.83 | 58.30±0.58 | 57.84±0.43 | 58.07±0.23 | 0.34±0.68 | **62.02**±0.00 | 34.91±0.09 |
| cardiotocography | 36.1±1.28 | **41.07**±1.73 | 39.18±4.49 | 35.36±2.14 | 32.66±1.59 | 36.88±1.27 | 39.91±1.05 | 39.48±1.54 | 37.73±1.73 | 62.02±0.00 | 62.02±0.00 | 48.23±0.39 |
| celeba | 15.42±2.29 | **17.97**±2.55 | 17.20±1.92 | 17.46±1.17 | 16.17±0.65 | 16.84±0.88 | 19.18±2.74 | 17.12±1.45 | 14.31±2.28 | 0.00±0.00 | 0.00±0.00 | 10.12±1.04 |
| census | 22.72±1.73 | 24.06±2.05 | 25.86±1.34 | 27.15±1.12 | 24.06±1.31 | 24.76±0.50 | 17.58±1.42 | 17.54±1.79 | 16.44±1.41 | 0.00±0.00 | 0.00±0.00 | 10.31±0.52 |
| cover | 26.77±15.24 | 25.80±1.34 | 53.70±16.88 | 53.70±16.88 | 44.34±20.24 | 36.61±2.59 | 76.51±2.37 | 68.92±4.22 | 46.54±3.93 | 0.00±0.00 | 0.00±0.00 | 38.39±0.62 |
| donors | 43.71±11.52 | 42.68±17.00 | 74.95±1.27 | 89.28±2.66 | 92.77±1.39 | 78.24±2.61 | 87.99±1.87 | 75.05±7.18 | 63.08±1.70 | 11.80±23.60 | 0.00±0.00 | 47.58±4.61 |
| fault | 60.33±3.36 | 83.59±4.47 | 58.90±6.77 | 66.88±4.88 | 58.87±1.51 | 62.30±4.91 | 55.57±1.57 | 55.33±1.66 | 55.16±1.66 | 97.03±0.00 | 96.91±0.24 | 72.00±0.47 |
| fraud | 57.54±10.13 | 57.05±16.03 | 84.05±6.11 | 87.24±5.04 | 79.18±3.21 | 70.87±3.12 | 75.61±1.76 | 54.25±5.90 | 22.55±24.73 | 0.00±0.00 | 0.00±0.00 | 30.48±4.66 |
| glass | 43.53±20.21 | 94.69±7.81 | 93.86±1.63 | 82.50±5.41 | 82.50±5.41 | 98.65±2.11 | 35.43±5.07 | 34.48±4.93 | 31.36±0.69 | 19.45±5.66 | 19.45±5.66 | 24.15±2.73 |
| hepatitis | 99.64±0.71 | 96.07±3.07 | 99.64±0.71 | 99.64±0.71 | 99.64±0.71 | 99.14±0.43 | 96.40±3.01 | 94.63±4.90 | 92.51±3.12 | 51.68±9.78 | 51.68±9.78 | 70.84±3.40 |
| http | 93.91±10.35 | 97.69±1.51 | 99.36±0.19 | 99.36±0.19 | 99.14±0.43 | 99.23±2.45 | 88.08±11.97 | 50.80±8.02 | 18.33±10.68 | 16.75±12.06 | 18.97±10.60 | 24.79±12.88 |
| imdb | 10.52±0.65 | 10.44±0.54 | 9.76±0.15 | 9.76±0.15 | 10.40±0.33 | 10.19±0.23 | 6.64±0.89 | 8.40±1.23 | 7.32±1.04 | 0.00±0.00 | 0.00±0.00 | 4.47±0.43 |
| internetads | 55.92±2.66 | 57.45±0.61 | 57.77±1.10 | 58.26±0.50 | 49.02±1.45 | 55.68±0.94 | 67.99±1.98 | 65.87±0.95 | 64.95±2.24 | 41.85±0.00 | 41.85±0.00 | 56.50±0.87 |
| ionosphere | 92.64±4.66 | 91.41±4.67 | 93.86±1.63 | 94.48±0.71 | 94.49±1.56 | 93.38±2.21 | 89.67±1.44 | 89.41±1.53 | 89.52±1.13 | 78.12±2.07 | 49.44±16.82 | 79.23±4.14 |
| landsat | 49.50±1.24 | 47.91±1.88 | 54.51±0.52 | 54.25±0.74 | 47.97±1.09 | 50.83±0.68 | 35.45±0.79 | 35.47±3.76 | 31.67±3.98 | 50.29±8.34 | 54.46±0.00 | 41.47±2.39 |
| letter | 6.80±4.21 | 4.00±1.55 | 3.60±1.36 | 3.20±0.75 | 11.60±2.06 | 5.84±1.11 | 2.40±1.50 | 3.00±1.55 | 3.20±1.17 | 0.00±0.00 | 0.00±0.00 | 1.72±0.37 |
| lymphography | 100.00±0.00 | 100.00±0.00 | 100.00±0.00 | 100.00±0.00 | 100.00±0.00 | 100.00±0.00 | 77.11±6.79 | 79.84±1.53 | 74.75±5.79 | 60.02±26.09 | 0.00±0.00 | 58.34±6.75 |
| magic.gamma | 64.88±2.50 | 69.99±2.98 | 69.99±0.87 | 70.21±0.50 | 71.64±0.93 | 69.34±1.08 | 79.82±0.48 | 80.94±0.33 | 80.19±0.73 | 89.73±7.81 | 96.10±0.00 | 85.36±1.69 |
| mammography | 36.62±8.76 | 38.15±8.41 | 27.69±1.80 | 29.08±2.63 | 18.15±1.23 | 29.94±2.59 | 32.69±5.73 | 34.62±2.19 | 35.08±2.68 | 39.69±2.19 | 0.00±0.00 | 28.42±1.40 |
| mnist | 45.37±1.84 | 46.20±3.18 | 50.54±1.26 | 59.20±1.30 | 62.66±2.09 | 52.79±0.77 | 63.86±1.77 | 56.74±2.71 | 53.74±4.35 | 0.00±0.00 | 0.00±0.00 | 34.87±1.48 |
| musk | 100.00±0.00 | 100.00±0.00 | 100.00±0.00 | 78.70±0.90 | 89.36±1.65 | 89.36±1.65 | 100.00±0.00 | 100.00±0.00 | 100.00±0.00 | 0.00±0.00 | 0.00±0.00 | 60.00±0.00 |
| optdigits | 29.87±5.68 | 41.20±3.49 | 44.67±5.56 | 71.73±0.53 | 46.00±5.11 | 46.69±2.00 | 14.80±1.81 | 6.40±3.88 | 9.33±5.72 | 0.00±0.00 | 0.00±0.00 | 6.11±1.52 |
| pageblocks | 62.16±2.84 | **64.08**±2.96 | 62.31±2.65 | 63.88±1.08 | 62.20±1.05 | 62.93±0.29 | 62.24±0.98 | 62.71±1.24 | 61.25±0.32 | 60.47±0.69 | 0.00±0.00 | 49.33±0.37 |
| pendigits | 46.03±17.81 | 60.51±10.58 | 59.23±5.15 | 66.03±5.18 | 51.03±3.57 | 56.56±5.52 | 63.97±6.36 | 54.36±7.40 | 43.46±7.50 | 0.00±0.00 | 0.00±0.00 | 32.36±1.43 |
| pima | 68.77±4.96 | 68.29±2.03 | 74.95±1.27 | 71.40±1.67 | 68.83±2.08 | 70.45±1.78 | 66.14±2.35 | 63.89±4.28 | 64.71±3.39 | 67.99±2.34 | 65.96±4.44 | 65.74±2.10 |
| satellite | 65.94±10.44 | 72.95±2.96 | 76.27±0.81 | 74.22±0.57 | 74.22±0.57 | 73.62±2.69 | 72.06±0.60 | 72.97±0.77 | 72.63±0.32 | 91.51±9.02 | 96.02±0.00 | 81.04±1.96 |
| satimage-2 | 38.03±8.59 | 72.68±32.26 | 91.83±0.56 | 89.58±1.13 | 56.34±12.38 | 69.69±7.51 | 78.31±3.03 | 60.85±5.87 | 46.20±5.07 | 26.20±32.11 | 0.00±0.00 | 42.31±7.14 |
| shuttle | 97.17±1.11 | 98.27±0.10 | 98.83±0.14 | 98.91±0.12 | 88.38±0.21 | 89.31±0.24 | 98.00±0.01 | 97.98±0.00 | 97.73±0.10 | 92.83±2.84 | 54.71±0.80 | 77.31±0.58 |
| skin | 38.03±28.70 | 72.09±8.03 | 67.99±9.92 | 54.29±12.30 | 49.74±13.81 | 56.43±4.55 | 82.15±0.39 | 82.23±0.37 | 81.66±0.57 | 80.48±0.35 | 0.00±0.00 | 76.25±0.32 |
| smtp | 39.28±18.81 | 59.49±7.84 | 54.93±13.50 | 48.81±9.06 | 50.03±10.03 | 50.51±3.54 | 69.59±3.95 | 69.59±3.95 | 49.07±5.05 | 27.32±14.61 | 87.61±0.00 | 43.11±3.67 |
| spambase | 76.97±2.12 | 76.88±3.76 | 76.27±0.81 | 80.23±0.62 | 74.93±0.44 | 77.87±0.46 | 80.19±0.18 | 80.27±0.41 | 80.56±0.29 | 87.61±0.00 | 87.61±0.00 | 83.25±0.07 |
| speech | 2.95±1.23 | 3.61±1.23 | 3.28±1.04 | 2.95±2.41 | 4.59±1.61 | 3.48±0.92 | 3.93±0.80 | 2.95±1.61 | 3.28±1.80 | 0.00±0.00 | 14.60±29.20 | 2.03±0.25 |
| stamps | 34.84±12.18 | 42.45±13.97 | 83.03±3.34 | 76.24±6.92 | 73.47±7.50 | 62.00±6.23 | 63.62±10.79 | 58.37±9.83 | 57.63±9.24 | 56.93±11.83 | 0.00±0.00 | 50.23±10.89 |
| thyroid | **68.39**±4.17 | 61.29±1.80 | 61.08±4.63 | 63.87±3.57 | 33.70±5.95 | 57.68±1.74 | 73.76±2.21 | 76.13±1.72 | 75.27±0.00 | 78.28±0.80 | 0.00±0.00 | 60.69±0.57 |
| vertebral | 23.17±6.96 | 29.15±9.05 | 52.53±11.60 | 64.81±3.19 | 54.89±6.76 | 44.91±2.82 | 43.88±5.22 | 36.20±11.73 | 32.59±0.94 | 24.68±8.71 | 3.48±6.96 | 28.18±7.79 |
| vowels | 22.40±17.59 | 18.80±6.52 | 30.80±7.44 | 30.80±12.43 | 24.00±7.04 | 32.28±5.79 | 40.00±5.51 | 40.80±3.25 | 45.20±0.98 | 0.00±0.00 | 0.00±0.00 | 25.20±1.36 |
| waveform | 47.80±3.06 | 38.20±17.57 | 35.40±7.47 | 11.69±1.62 | 28.40±4.22 | 25.36±2.94 | 11.20±1.72 | 11.80±2.14 | 12.20±1.60 | 67.85±5.79 | 0.00±0.00 | 7.04±0.23 |
| wbc | 78.86±7.89 | 84.03±7.23 | 96.89±4.35 | 95.71±4.90 | 90.32±5.34 | 90.65±0.56 | 40.59±7.11 | 70.31±4.12 | 77.59±7.50 | 40.96±12.93 | 0.00±0.00 | 36.84±4.92 |
| wdbc | 64.32±18.21 | 80.42±5.68 | 84.10±6.72 | 87.13±4.26 | 78.33±5.25 | 89.16±3.80 | 78.62±5.69 | 34.81±6.65 | 10.96±12.93 | 10.43±20.87 | 0.00±0.00 | 47.39±3.91 |
| wilt | 6.15±7.00 | 11.36±4.93 | 37.04±8.60 | 39.61±4.50 | 89.16±3.80 | 26.35±2.35 | 19.53±2.50 | 19.92±4.34 | 5.37±2.97 | 16.96±5.74 | 54.71±0.80 | 12.36±2.38 |
| wine | 97.95±4.09 | 96.72±3.06 | 98.25±2.23 | 98.18±3.64 | 97.67±3.23 | 97.76±2.03 | 98.86±1.44 | 95.05±5.74 | 97.27±3.64 | 41.73±22.73 | 0.00±0.00 | 66.58±4.38 |
| wpbc | 81.91±5.11 | 63.78±10.33 | 88.98±3.08 | 88.64±3.69 | 83.46±2.71 | 81.35±3.33 | 58.05±5.72 | 59.07±3.42 | 36.98±5.14 | 36.48±3.67 | 98.96±14.97 | 53.91±5.76 |
| yeast | 46.31±2.91 | 48.95±1.78 | 49.35±0.54 | 49.31±0.62 | 48.13±0.93 | 48.41±0.87 | 49.23±1.21 | 48.78±0.62 | 47.93±0.48 | 16.82±1.91 | 98.22±0.00 | 58.93±0.44 |
| yelp | 7.64±0.70 | 8.20±0.38 | 7.52±0.65 | 7.56±0.29 | 7.56±0.37 | 7.70±0.20 | 16.12±2.67 | 14.76±3.88 | 14.68±3.19 | 0.00±0.00 | 0.00±0.00 | 9.11±1.45 |
| MNIST-C | 45.87±0.49 | 50.05±0.46 | 51.49±0.27 | 51.46±0.39 | 47.26±0.49 | 49.22±0.12 | 50.07±0.51 | 47.93±0.48 | 58.65±0.27 | 0.00±0.00 | 0.00±0.00 | 29.35±0.11 |
| FashionMNIST | 56.83±0.32 | 63.23±0.30 | 64.48±0.20 | 64.41±0.16 | 58.13±0.29 | 61.42±0.13 | 59.78±0.30 | 59.24±0.30 | 23.92±0.30 | 0.00±0.00 | 0.00±0.00 | 35.54±0.10 |
| CIFAR10 | 16.33±0.58 | 20.51±0.66 | 22.84±0.00 | 22.87±0.57 | 15.56±0.45 | 19.62±0.19 | 24.23±0.19 | 23.86±0.44 | 19.47±0.17 | 0.00±0.00 | 0.00±0.00 | 14.40±0.12 |
| SVHN | 16.44±0.90 | 19.22±0.23 | 20.00±0.14 | 20.07±0.30 | 18.85±0.22 | 18.92±0.20 | 19.80±0.22 | 19.47±0.17 | 19.61±0.14 | 0.00±0.00 | 0.00±0.00 | 11.78±0.08 |
| MVTec-AD | 79.81±0.59 | 81.35±0.82 | 82.39±0.82 | 82.20±0.46 | 77.51±0.66 | 80.65±0.56 | 76.79±0.49 | 78.35±0.88 | 78.60±1.12 | 63.03±1.80 | 63.09±1.51 | 71.97±0.92 |
| 20news | 12.82±0.61 | 14.12±1.10 | 15.05±1.36 | 16.54±1.42 | 12.79±0.95 | 14.26±0.73 | 19.22±2.15 | 16.99±0.93 | 15.62±1.07 | 0.18±0.35 | 0.18±0.35 | 10.44±0.45 |
| agnews | 14.17±0.19 | 14.40±0.24 | 14.47±0.12 | 14.53±0.20 | 13.69±0.12 | 14.25±0.08 | 22.14±0.86 | 20.77±1.57 | 19.76±2.41 | 0.00±0.00 | 0.00±0.00 | 12.53±0.63 |

Table 17.1: Average AUROC ± standard dev. over five seeds for the semi-supervised setting on ADBench. Rank of each model among 32 models (26 baselines + 4 ^avg variants of top-4 baselines + 2 FoMo-0D variants w/ $D = 100$ and $D = 20$) per dataset is provided (in parentheses) (the lower, the better). We use blue and green respectively to mark the top-1 and the top-2 method. Last four rows show avg_rank of methods across datasets, and $p$-values of the Wilcoxon signed rank test comparing FoMo-0D ($D = 100$) with other baselines. The previous four rows are the same for FoMo-0D ($D = 20$), when ranking 31 models (26 baselines + 4 ^avg variants of top-4 baselines + FoMo-0D w/ $D = 20$).

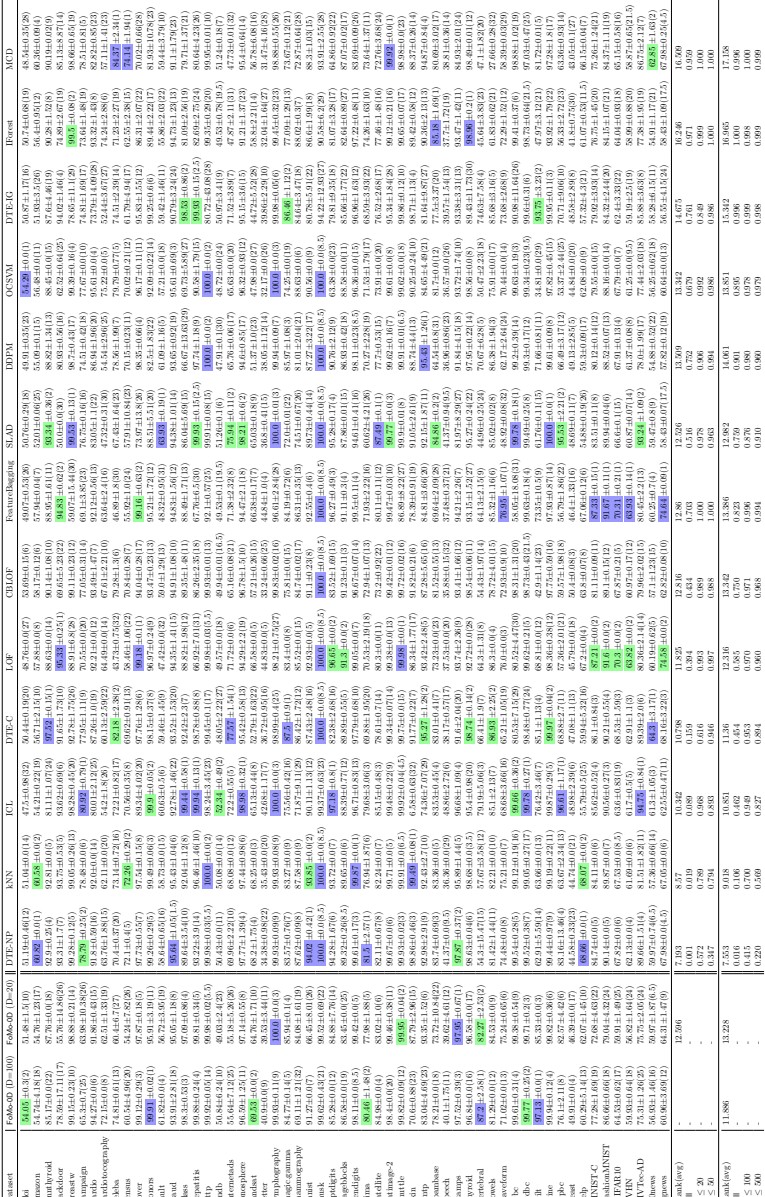

Table 17.2: Average AUROC ± standard dev. over five seeds for the semi-supervised setting on ADBench. Rank of each model per dataset is provided (in parentheses) (the lower, the better). We use blue and green respectively to mark the top-1 and the top-2 method.

| Dataset | VAE | PCA | PlanarFlow | HBOS | GANomaly | GOAD | DIF | COPOD | ECOD | DeepSVDD | LODA | DAGMM | DROCC | DTE-NP$^{avg}$ | KNN$^{avg}$ | ICL$^{avg}$ | DTE-C$^{avg}$ |
|---|---|---|---|---|---|---|---|---|---|---|---|---|---|---|---|---|---|
| aloi | 54.04±0.0(3.5) | 54.04±0.0(3.5) | 48.52±2.34(29) | 52.23±0.0(7) | 53.94±1.65(5) | 48.01±0.02(30) | 51.1±0.0(13) | 49.51±0.0(24) | 51.73±0.0(8) | 50.89±2.05(15) | 49.24±2.75(25) | 50.84±2.97(17) | 50.0±0.0(22) | 51.25±0.0(11) | 51.61±0.0(9) | 47.63±0.15(31) | 50.29±0.09(21) |
| amazon | 5.94±0.0(32) | 54.9±0.0(16) | 49.94±2.12(31) | 56.32±0.0(13) | 53.42±0.94(22) | 56.07±0.0(14) | 51.43±0.34(27) | 56.78±0.0(9) | 53.79±0.0(20) | 51.2±4.42(28) | 52.23±2.84(24) | 50.47±2.01(29) | 50.0±0.0(30) | 60.47±0.0(3) | 60.25±0.0(5) | 53.15±0.19(23) | 53.48±1.33(21) |
| annthyroid | 85.45±0.0(20) | 85.19±0.0(21) | 93.19±2.14(3) | 66.02±0.0(31) | 67.62±5.54(30) | 83.01±5.16(25) | 88.36±0.0(16) | 76.77±0.0(28) | 78.45±0.0(26) | 55.01±3.62(32) | 77.35±5.75(27) | 72.23±15.13(29) | 88.5±2.31(12) | 92.64±0.0(6) | 92.3±0.0(7) | 84.27±2.33(23) | 88.65±0.05(17) |
| backdoor | 64.67±0.74(23) | 64.57±0.65(24) | 76.03±11.56(18) | 70.81±0.89(21) | 87.5±1.43(13) | 92.9±14.48(28) | 53.73±0.76(15) | 50.0±0.0(30) | 50.0±0.0(30) | 91.14±2.62(11) | 47.62±22.63(32) | 54.36±19.91(27) | 94.25±0.73(3) | 92.55±0.44(9) | 91.0±0.47(12) | 93.3±0.48(8) | 74.67±0.36(20) |
| breastw | 99.22±0.18(7) | 99.21±0.17(8) | 97.93±0.82(22) | 99.23±0.17(6) | 94.75±2.73(25) | 62.31±0.0(31) | 96.34±1.19(31) | 99.46±0.09(3) | 99.14±0.22(11) | 96.96±0.92(23) | 98.13±0.35(21) | 69.52±10.2(27) | 97.32±3.19(32) | 98.67±0.28(18) | 99.16±0.21(9) | 98.75±0.18(16) | 99.52±0.53(24) |
| campaign | 77.47±0.0(11.5) | 77.47±0.0(11.5) | 69.75±3.75(21) | 77.06±0.0(13) | 69.21±3.28(22) | 47.89±12.32(32) | 57.87±0.0(30) | 68.25±0.0(30) | 76.84±0.0(15) | 62.21±12.88(27) | 58.88±4.48(29) | 61.47±2.73(28) | 50.0±0.0(31) | 78.76±0.0(3) | 78.66±0.0(4) | 78.35±0.52(7) | 67.18±0.74(24) |
| cardio | 96.55±0.0(1) | 96.54±0.0(1(2) | 88.9±0.93(8) | 80.7±0.0(24) | 86.06±4.33(15) | 96.01±0.27(3) | 68.25±0.0(30) | 93.16±0.0(9) | 79.3±0.0(1) | 65.43±4.37(31) | 91.34±2.93(17) | 77.92±8.85(26) | 62.14±23.76(32) | 92.47±0.0(11) | 93.07±0.0(10) | 77.71±3.76(27) | 72.66±0.21(29) |
| cardiotocography | 78.9±0.0(2) | 78.89±0.0(3) | 69.88±5.65(9) | 61.24±0.0(21) | 62.77±8.62(18) | 70.06±1.4(4) | 41.79±0.0(32) | 66.35±0.0(12) | 76.42±0.0(10) | 47.75±8.50(29) | 72.79±7.6(7) | 67.11±9.03(11) | 45.98±16.75(31) | 63.21±0.0(17) | 65.48±0.0(13) | 50.85±1.24(28) | 55.9±0.23(24) |
| celeba | 80.32±0.48(4) | 80.53±0.7(3) | 71.64±7.91(18) | 76.68±0.66(10) | 52.25±11.79(29) | 43.8±10.0(31) | 76.33±0.63(11) | 75.72±0.59(12) | 56.17±22.46(28) | 62.46±13.17(26) | 62.46±13.17(26) | 63.81±4.28(25) | 74.42±4.28(15) | 74.42±4.28(15) | 76.9±0.35(9) | 78.49±0.93(8) | 68.94±0.44(21) |
| census | 70.32±0.22(9) | 70.51±0.21(10) | 59.33±2.89(21) | 62.5±0.47(16) | 68.07±3.66(14) | 35.24±4.19(32) | 61.45±2.04(18) | 50.0±0.0(30.5) | 50.0±0.0(30.5) | 54.16±4.33(27) | 51.12±11.19(29) | 52.24±1.19(28) | 55.36±3.82(25) | 72.11±0.16(4) | 71.84±0.15(6) | 72.25±1.63(3) | 61.41±0.71(19) |
| cover | 94.35±0.15(16) | 94.41±0.14(15) | 47.52±8.02(31) | 71.11±0.82(27) | 76.35±10.11(24) | 13.83±13.34(32) | 61.45±2.04(18) | 88.2±0.27(21) | 91.86±0.21(18) | 54.16±4.33(27) | 94.93±5.07(14) | 75.94±14.06(25) | 55.36±3.82(25) | 97.35±0.19(9) | 96.73±0.12(10) | 91.42±1.74(19) | 78.4±0.14(23) |
| donors | 88.6±0.23(19) | 88.12±0.57(21) | 91.64±3.32(15) | 81.19±0.61(25) | 75.34±11.69(27) | 33.57±16.0(32) | 90.04±1.79(16) | 81.5±0.21(24) | 88.74±0.38(18) | 72.95±17.81(28) | 63.52±27.27(30) | 62.15±16.19(31) | 74.18±22.09(28) | 98.74±0.09(7) | 98.73±0.0(8) | 96.3±1.95(10) | 80.17±3.55(26) |
| fault | 55.87±0.0(20.5) | 55.87±0.0(20.5) | 53.63±0.13(24) | 53.06±0.0(26) | 59.5±5.15(17) | 58.89±6.01(14) | 62.31±0.0(5) | 49.14±0.0(28) | 50.37±0.0(28) | 54.31±1.62(25) | 50.27±1.88(29) | 52.85±7.19(27) | 59.75±3.53(23) | 60.22±0.0(7) | 60.0±0.0(6) | 62.47±1.14(2) | 55.41±0.63(24) |
| fraud | 95.4±0.0(75.5) | 95.38±0.08(7) | 90.72±2.26(25) | 95.02±0.67(9) | 93.25±2.85(21) | 69.75±3.21(31) | 92.58±2.27(29) | 94.3±1.41(16) | 94.89±1.27(11) | 83.13±6.0(28) | 89.05±8.12(26) | 85.33±6.76(27) | 50.0±0.0(32) | 95.57±0.93(4) | 94.23±0.87(17) | 95.27±3.45(6) | 76.21±1.08(30) |
| glass | 73.55±1.66(25) | 73.44±2.22(34) | 85.33±6.2(16) | 82.59±3.23(18) | 79.77±8.72(20) | 59.03±12.64(32) | 96.45±1.39(5) | 76.0±1.94(33) | 71.14±3.54(26) | 83.67±16.2(17) | 67.34±5.30(30) | 65.33±15.73(30) | 64.89±23.64(31) | 90.83±0.91(9) | 95.27±3.45(6) | 95.27±3.45(6) | 76.95±2.58(22) |
| hepatitis | 84.84±2.27(19.5) | 84.48±2.29(22) | 95.8±1.67(13) | 84.84±6.78(19.5) | 50.14±4.84.85(30) | 84.5±3.25(21) | 86.07±2.32(12) | 80.9±1.22(25) | 73.84±1.99(27) | 99.57±0.24(7) | 68.98±3.97(29) | 70.22±6.66(28) | 51.8±17.93(32) | 96.22±0.88(11) | 99.09±0.48(6) | 99.98±0.02(0.5) | 84.42±1.2(23) |
| http | 99.94±0.01(12) | 99.95±0.01(10) | 70.87±0.84(19) | 98.58±1.04(22) | 69.86±0.23(31) | 99.68±0.31(16) | 99.36±0.07(19) | 65.94±0.0(16) | 66.01±0.0(15) | 49.97±5.71(15) | 47.72±45.49(32) | 49.47±5.05(28) | 99.95±0.01(10) | 99.95±0.01(10) | 65.73±0.0(18) | 99.98±0.02(0.5) | 89.59±0.09(27) |
| imdb | 47.97±0.0(28.5) | 47.97±0.0(28.5) | 49.23±2.75(22) | 49.94±6.0(16.5) | 53.89±2.03(20) | 51.58±0.71(3) | 93.59±2.96(21) | 51.05±0.0(8) | 46.88±0.0(32) | 97.2±1.26(7) | 47.23±2.24(31) | 48.6±0.38(35) | 61.14±28.54(32) | 50.35±0.0(12) | 95.12±0.92(16) | 52.95±0.12(1) | 49.27±1.12(21) |
| internetads | 65.12±0.0(22.5) | 65.12±0.0(22.5) | 70.87±0.84(9) | 49.18±0.0(30) | 69.86±0.23(31) | 65.65±0.21(19) | 93.59±2.06(21) | 65.94±0.0(16) | 66.01±0.0(15) | 72.96±5.24(3) | 58.73±3.84(24) | 49.47±5.05(28) | 73.95±5.98(29) | 67.65±0.03(13) | 65.73±0.0(18) | 72.53±0.45(6) | 66.79±0.73(14) |
| ionosphere | 89.76±1.2(34) | 89.11±1.31(25) | 96.86±1.21(9) | 70.68±2.85(31) | 93.89±2.03(20) | 91.54±3.35(22) | 93.59±2.06(21) | 49.29±0.0(26) | 71.77±1.43(30) | 97.2±1.26(7) | 85.56±3.59(26) | 73.95±5.98(29) | 73.95±5.98(29) | 97.5±0.63(5) | 95.12±0.92(16) | 97.78±1.23(3) | 83.01±5.81(27) |
| landsat | 30.23±0.0(31) | 43.9±0.0(30) | 50.85±2.13(24) | 50.85±2.13(24) | 55.34±10.03(19) | 40.52±2.32(32) | 54.07±0.0(1) | 42.01±0.0(3) | 45.37±0.0(3) | 36.4±3.05(18) | 30.2±0.94(32) | 56.27±3.46(18) | 53.86±2.57(21) | 66.47±0.0(6) | 64.7±0.0(11) | 64.39±0.59(12) | 50.2±0.98(25) |
| letter | 30.23±0.0(31) | 30.33±0.0(30) | 38.73±3.53(13) | 35.91±0.0(19) | 34.02±0.74(23) | 31.08±0.55(29) | 74.07±0.0(1) | 36.53±0.0(17) | 45.37±0.0(3) | 36.4±3.05(18) | 30.2±0.94(32) | 38.97±8.38(12) | 34.74±0.0(21) | 34.74±0.0(21) | 33.39±0.0(24) | 43.82±1.97(6) | 42.38±0.25(8) |
| lymphography | 99.86±0.05(14) | 99.86±0.05(14) | 74.12±2.75(20) | 99.69±0.17(19) | 99.09±0.86(24) | 99.89±0.08(11) | 99.84±0.25(17) | 99.53±0.2(21) | 99.52±0.15(22) | 99.73±0.3(18) | 67.04±13.87(31) | 94.94±3.85(29) | 55.26±11.04(2) | 99.87±0.06(13) | 99.87±0.0(3) | 100.0±0.0(3) | 88.19±1.78(30) |
| magic.gamma | 70.64±0.0(23.5) | 70.64±0.0(23.5) | 74.12±2.75(20) | 74.53±0.0(18) | 59.18±1.6(32) | 69.46±2.39(26) | 63.86±0.0(28) | 68.0±0.0(27) | 63.58±0.0(29) | 62.97±1.07(30) | 70.53±1.36(25) | 59.23±4.32(31) | 78.83±0.66(12) | 82.81±0.0(10) | 81.76±0.0(11) | 76.09±1.17(14) | 75.19±3.5(17) |
| mammography | 89.58±0.17(5) | 89.93±0.0(3) | 87.93±5.71(23) | 85.01±0.0(16) | 85.54±7.43(14) | 69.94±8.59(31) | 73.87±4.2(27) | 91.51±5.7(4)(30) | 71.5±7.4(30) | 71.5±7.4(30) | 89.62±0.87(4) | 76.03±14.65(25) | 87.55±0.0(10) | 87.55±0.0(10) | 87.29±0.0(11) | 80.47±0.02(22) | 77.85±0.91(24) |
| mnist | 90.21±0.0(10.5) | 90.21±0.0(10.5) | 81.9±2.67(30) | 62.34±0.0(29) | 77.96±6.44(23) | 90.07±0.35(13) | 50.21±0.0(30) | 50.0±0.0(31.5) | 50.0±0.0(31.5) | 66.37±11.03(27) | 64.74±1.74(28) | 83.13±1.64(19) | 83.13±1.64(19) | 93.6±0.0(3) | 93.04±0.0(4) | 81.37±0.93(21) | 72.55±0.64(24) |
| msk | 100.0±0.0(8.5) | 100.0±0.0(8.5) | 76.65±18.72(31) | 100.0±0.0(8.5) | 100.0±0.0(8.5) | 100.0±0.0(8.5) | 74.74±12.58(19) | 99.71±0.0(19) | 50.0±0.0(26.5) | 39.45±18.65(30) | 99.67±0.35(20) | 95.01±4.27(26) | 32.99±33.18(32) | 92.53±0.0(8) | 94.82±0.49(5) | 97.18±0.37(25) | 67.73±1.71(20) |
| optdigits | 58.17±0.0(24.5) | 58.17±0.0(24.5) | 34.12±8.29(31) | 89.92±0.0(11) | 72.83±9.94(31) | 67.46±4.94(21) | 48.64±0.0(28) | 80.85±0.0(29) | 50.0±0.0(26.5) | 99.99±0.01(17) | 32.77±8.62(32) | 40.04±20.45(29) | 89.32±0.0(8.5) | 89.32±0.0(8.5) | 89.16±0.29(10) | 89.16±0.29(10) | 80.0±0.0(30) |
| pageblocks | 86.16±0.0(20) | 86.12±0.0(21) | 84.85±1.19(23) | 65.62±0.0(32) | 72.83±9.94(31) | 88.05±1.19(13) | 87.89±0.0(16) | 80.85±0.0(29) | 87.95±0.0(14) | 87.39±1.53(30) | 83.62±2.6(24) | 82.8±10.44(26) | 75.91±13.38(29) | 89.77±0.0(6) | 89.7±0.0(6) | 89.16±0.29(10) | 81.66±0.0(28) |
| pendigits | 94.5±0.0(18) | 94.37±0.0(19) | 83.45±6.13(27) | 93.55±0.0(20) | 67.93±21.6(30) | 89.97±2.03(24) | 89.71±0.0(25) | 90.74±0.0(23) | 92.93±0.0(21) | 46.29±11.35(32) | 92.13±1.02(22) | 56.49±21.83(31) | 57.13±15.93(31) | 99.17±0.0(6) | 99.17±0.0(6) | 76.12±1.04(8) | 78.51±0.15(28) |
| pima | 73.18±1.93(12) | 72.28±1.99(14) | 72.17±2.71(15) | 74.76±1.6(9) | 60.45±3.01(28) | 62.31±13.7(25) | 58.71±0.0(25) | 66.59±1.32(23) | 60.56±1.57(27) | 57.99±2.85(29) | 62.68±7.55(24) | 54.54±6.6(31) | 47.53±16.52(32) | 78.87±1.43(4) | 76.58±1.39(7) | 76.12±1.04(8) | 68.95±0.96(21) |
| satellite | 74.14±0.25(19) | 66.63±0.0(31) | 72.3±1.73(25) | 84.5±0.0(2) | 79.87±0.56(13) | 68.76±0.92(28) | 66.91±0.0(30) | 68.34±0.0(29) | 62.22±0.0(32) | 76.19±2.66(18) | 69.73±1.15(36) | 72.79±2.01(23) | 73.38±4.44(21) | 81.43±0.0(9) | 80.47±0.0(10) | 83.57±2.48(5) | 68.77±2.26(27) |
| satimage-2 | 98.98±0.04(18) | 98.17±0.0(21) | 96.67±0.57(27) | 97.05±0.0(22) | 97.57±1.19(24) | 98.99±0.06(17) | 86.9±0.0(31) | 97.92±0.0(25) | 97.09±0.0(25) | 92.94±2.84(29) | 82.67±4.43(30) | 99.22±0.42(15) | 99.15±0.05(2) | 99.77±0.0(3) | 99.77±0.0(3) | 96.99±2.56(26) | 83.3±4.71(32) |
| shuttle | 99.35±0.0(21) | 99.36±0.0(20) | 86.5±3.58(28) | 98.64±0.0(24) | 97.53±0.74(25) | 70.44±16.26(31) | 66.91±0.0(30) | 99.47±0.0(19) | 99.33±0.0(22) | 99.79±0.07(13) | 71.68±33.88(30) | 84.58±18.68(29) | 50.0±0.0(32) | 99.92±0.0(4.5) | 99.89±0.0(9) | 99.85±0.11(11) | 89.7±0.05(26) |
| skin | 66.05±0.0(19) | 59.73±0.31(28) | 84.22±6.88(22) | 76.92±0.35(20) | 48.48±2.87(30) | 64.95±2.25(26) | 87.55±0.76(16) | 47.21±0.18(31) | 49.14±0.18(29) | 85.24±6.6(19) | 75.51±5.43(21) | 67.91±30.02(24) | 57.13±15.93(31) | 98.34±0.04(5) | 93.1±2.3(7) | 73.81±10.45(22) | 83.48±0.19(18) |
| smtp | 81.93±5.67(25) | 81.81±7.32(26) | 84.22±6.88(22) | 82.75±5.31(24) | 54.55±5.98(32) | 78.78±13.12(28) | 95.11±1.36(3) | 91.15±1.58(12) | 87.62±0.91(22) | 89.79±0.07(13) | 40.08±33.97(32) | 87.08±5.31(17) | 57.13±15.93(31) | 93.4±2.3(7) | 98.8±1.14(9) | 88.37±5.41(14) | 78.42±3.73(29) |
| spambase | 81.4±0.0(14.5) | 81.4±0.0(14.5) | 82.26±3.35(10) | 77.88±0.0(19) | 82.57±1.34(8) | 81.78±0.36(11) | 41.31±0.0(32) | 72.09±0.0(25) | 99.31±0.0(11) | 70.24±5.02(26) | 72.39±6.9(24) | 69.41±4.42(29) | 75.37±4.45(21) | 83.07±0.0(6) | 82.36±0.0(9) | 80.71±0.86(16) | 70.0±0.04(27) |
| speech | 36.38±0.0(27.5) | 36.38±0.0(27.5) | 48.56±4.54(7) | 36.66±0.0(24) | 38.7±5.43(15) | 36.63±1.14(25) | 37.03±0.0(22) | 35.96±0.0(31) | 35.96±0.0(31) | 48.88±2.88(5) | 38.02±2.67(18) | 50.66±3.9(2) | 48.96±2.21(4) | 38.42±0.0(16) | 36.31±0.0(30) | 50.53±1.1(3) | 43.06±0.55(8) |
| stamps | 93.28±1.28(14) | 92.7±1.68(16) | 87.3±10.69(23) | 91.8±1.0(19) | 68.27±7.84(31) | 88.05±1.19(13) | 81.46±15.36(28) | 93.14±0.4(15) | 87.62±0.91(22) | 71.09±3.68(30) | 91.03±3.55(17) | 80.11±11.56(29) | 90.15±22.93(32) | 95.88±0.66(6) | 94.2±1.17(8) | 90.11±0.22(21) | 82.21±5.22(26) |
| thyroid | 98.55±0.0(9.5) | 93.79±1.58(23) | 84.42±0.51(13) | 98.65±0.0(5) | 94.78±3.22(25) | 95.15±0.85(23) | 91.09±2.25(25) | 93.81±0.0(26) | 97.55±0.0(15) | 88.77±3.69(32) | 96.06±1.65(19) | 91.08±6.68(29) | 94.96±1.55(24) | 98.61±0.0(7) | 98.68±0.0(3.5) | 95.35±0.6(21) | 89.11±0.02(31) |
| vertebral | 42.63±2.75(27) | 42.08±3.49(28) | 49.78±7.82(19) | 40.09±3.97(30) | 74.82±28.64(29) | 67.53±0.72(8.5) | 26.34±2.49(32) | 86.37±4.37(27) | 73.86±5.42(30) | 92.16±4.34(24) | 31.66±5.28(31) | 50.6±11.93(31) | 43.8±3.23(32) | 67.5±0.0(10) | 46.59±1.67(22) | 99.79±0.33(7) | 59.97±2.61(10) |
| vowels | 52.12±0.02(30) | 52.99±0.0(29) | 54.59±10.55(26) | 53.31±0.0(27) | 63.11±9.82(20) | 48.48±0.0(31) | 82.51±3.26(7) | 52.83±2.93(27) | 41.47±0.0(22) | 55.73±4.43(23) | 55.52±8.1(24) | 42.55±11.56(31) | 79.83±0.0(14) | 72.67±0.81(9) | 81.22±1.57(13) | 72.28±0.42(18) |
| waveform | 64.84±0.0(21) | 64.68±0.0(23) | 64.8±2.42(22) | 69.28±0.0(15) | 97.56±7.58(4) | 64.99±3.1(29) | 72.36±0.0(11) | 72.36±0.0(11) | 44.64±0.0(22) | 59.04±2.58(20) | 60.96±5.08(25) | 51.89±7.39(30) | 67.7±5.0(18) | 44.25±0.0(24) | 76.62±0.0(2) | 68.33±2.15(17) | 59.07±0.68(28) |
| wbc | 99.25±0.0(27)(13) | 99.35±0.0(3)(10) | 95.97±0.81(23) | 99.01±0.41(17) | 95.91±3.0(24) | 90.14±0.2(15) | 97.46±1.82(22) | 99.4±0.25(7) | 99.39±0.25(8) | 91.44±3.64(25) | 87.86±1.50(21) | 86.76±15.33(27) | 99.27±0.34(12) | 98.05±0.35(18) | 99.34±0.42(11) | 99.34±0.42(11) | 81.1±1.77(28) |
| wdbc | 99.14±0.36(14.5) | 99.11±0.22(14.5) | 98.86±4.76(19) | 98.55±0.28(23) | 96.22±3.1(28) | 98.96±0.25(18) | 68.03±7.24(31) | 99.18±0.21(13) | 96.72±0.71(27) | 74.24±6.0(27) | 96.98±2.04(36) | 73.78±25.27(30) | 40.08±33.97(32) | 99.11±0.22(16) | 99.11±0.22(16) | 98.84±0.43(20) | 78.42±3.73(29) |
| wilt | 35.41±0.0(28) | 26.07±0.0(32) | 74.62±4.18(8) | 39.1±0.0(26) | 44.01±5.48(22) | 55.02±0.0(18) | 55.02±0.0(18) | 32.09±0.0(31) | 37.48±0.0(27) | 34.41±1.7(30) | 41.1±7.0(25) | 41.81±7.29(24) | 49.49±12.62(20) | 55.92±0.0(17) | 67.16±0.0(5) | 72.42±2.57(10) | 77.44±0.57(6) |
| wine | 94.25±1.47(20) | 93.79±1.58(23) | 95.38±2.4(19) | 95.63±2.63(18) | 94.78±3.22(25) | 94.11±1.86(21) | 86.37±5.42(30) | 50.0±0.0(31.5) | 56.88±0.0(27) | 90.94±4.75(26) | 75.45±2.22(25) | 49.63±3.46(32) | 43.78±4.34(32) | 67.5±0.0(10) | 83.46±0.11(9) | 99.79±0.33(7) | 62.9±0.15(20) |
| wpbc | 54.42±2.76(24) | 52.54±2.3(26) | 57.46±2.93(20) | 60.01±2.67(16) | 60.11±5.48(17) | 82.51±3.26(7) | 82.51±3.26(7) | 49.5±2.47(30) | 49.5±2.47(30) | 82.67±5.47(5) | 51.32±3.74(29) | 46.99±2.56(31) | 72.67±0.81(9) | 58.14±1.82(19) | 60.91±0.03(13) | 72.42±2.57(10) | 61.89±2.06(15) |
| yeast | 42.89±0.01(29) | 43.24±0.0(26) | 54.07±3.33(19) | 42.88±0.0(28) | 47.64±6.7(11) | 64.99±3.1(20) | 28.44±0.0(32) | 60.21±0.0(14) | 57.39±0.0(20) | 49.9±3.3(30) | 54.5±4.02(26) | 53.36±2.12(28) | 44.25±0.0(24) | 44.25±0.0(24) | 43.95±0.0(25) | 46.62±1.16(14) | 47.74±0.5(10) |
| yelp | 59.14±0.04(19) | 59.16±0.0(18) | 53.6±2.03(28) | 59.95±0.0(15) | 56.11±1.6(24) | 49.14±0.2(15) | 97.46±1.82(22) | 64.24±0.27(7) | 60.21±0.0(14) | 49.9±3.3(30) | 56.26±3.78(23) | 49.87±1.25(31) | 50.67±1.15(29) | 67.16±0.0(5) | 53.71±0.22(27) | 53.71±0.22(27) | 56.47±0.84(22) |
| MNIST-C | 78.35±0.0(17.5) | 78.35±0.0(17.5) | 71.21±1.5(24) | 70.43±0.0(25) | 80.08±1.16(13) | 79.34±0.0(16) | 54.05±2.27(30) | 99.18±0.21(13) | 50.0±0.0(31.5) | 64.72±4.6(27) | 69.39±5.31(26) | 63.71±6.37(28) | 51.55±14.44(30) | 82.7±0.0(8) | 82.7±0.0(8) | 83.46±0.11(9) | 71.28±0.16(23) |
| FashionMNIST | 87.6±0.0(16.5) | 87.6±0.0(16.5) | 82.19±0.96(22) | 75.42±0.0(26) | 89.43±0.52(10) | 88.03±0.34(15) | 65.53±1.77(29) | 99.18±0.21(13) | 50.0±0.0(31.5) | 75.45±2.22(25) | 79.28±4.1(23) | 70.8±4.89(28) | 49.99±2.47(32) | 89.07±0.0(9) | 89.7±0.07(8) | 89.71±0.07(8) | 74.14±0.06(27) |
| CIFAR10 | 67.42±0.0(11.5) | 67.42±0.0(11.5) | 62.75±1.09(21) | 57.89±0.0(26) | 67.53±0.72(8.5) | 67.53±0.72(8.5) | 67.53±0.72(8.5) | 54.98±0.0(29) | 56.88±0.0(27) | 61.62±4.35(23) | 54.5±4.02(26) | 53.97±2.96(30) | 49.63±3.46(32) | 67.5±0.0(10) | 67.26±0.0(13) | 62.9±0.15(20) | 61.25±0.12(24) |
| SVHN | 60.79±0.0(11.5) | 60.79±0.0(11.5) | 58.87±0.87(21.5) | 54.65±0.0(25) | 61.25±0.77(9.5) | 60.75±0.4(9.17) | 92.98±0.67(29) | 50.0±0.0(30.5) | 50.0±0.0(30.5) | 53.92±3.05(27) | 54.5±4.02(26) | 53.36±2.12(28) | 49.99±2.47(32) | 61.14±0.0(11) | 60.91±0.03(13) | 60.91±0.03(13) | 57.8±0.05(23) |
| MVTec-AD | 76.37±1.91(21) | 76.37±1.91(21) | 72.8±2.39(27) | 75.97±1.84(23) | 81.1±1.8(12) | 77.06±2.1(30) | 81.99±2.78(10) | 50.0±0.0(31.5) | 50.0±0.0(31.5) | 89.55±2.19(5) | 72.31±3.37(28) | 64.69±5.63(29) | 60.52±11.42(30) | 83.75±0.49(9) | 78.81±0.45(16) | 93.08±0.33(3) | 73.32±0.42(26) |
| 20news | 54.59±0.74(23) | 54.39±0.41(24) | 51.59±3.0(31) | 53.57±0.35(26) | 59.68±1.87(8) | 54.92±1.04(20) | 54.92±1.04(20) | 55.05±0.0(27) | 54.11±0.0(24)(25) | 55.64±4.17(19) | 53.45±4.0(27) | 51.48±4.08(32) | 52.77±4.09(30) | 58.02±0.8(12) | 58.5±0.44(10) | 58.5±0.44(10) | 58.0±0.6(13) |
| agnews | 56.9±0.0(22.5) | 56.9±0.0(22.5) | 50.15±1.19(30) | 55.69±0.0(25) | 58.63±1.14(16) | 59.87±0.86(15) | 52.02±0.89(28) | 55.05±0.0(27) | 55.11±0.0(26) | 49.83±5.53(31) | 56.95±3.54(21) | 51.03±3.6(29) | 49.65±0.76(32) | 66.66±0.0(7) | 65.35±0.0(8) | 57.32±0.11(20) | 60.06±0.42(14) |
| Rank(avg) |  |  |  |  |  |  |  |  |  |  |  |  |  |  |  |  |  |
| All | 17.202 | 17.719 | 19.43 | 18.868 | 18.772 | 20.237 | 20.491 | 21.219 | 21.605 | 21.737 | 23.667 | 24.965 | 23.158 | 8.605 | 10.632 | 12.447 | 21.439 |
| d ≤ 20 | 0.962 | 0.971 | 0.997 | 1.000 | 0.999 | 1.000 | 1.000 | 1.000 | 1.000 | 1.000 | 1.000 | 1.000 | 1.000 | 0.007 | 0.062 | 0.437 | 1.000 |
| d ≤ 50 | 1.000 | 1.000 | 1.000 | 1.000 | 1.000 | 1.000 | 1.000 | 1.000 | 1.000 | 1.000 | 1.000 | 1.000 | 1.000 | 0.813 | 0.924 | 0.999 | 1.000 |
| Rank(avg) |  |  |  |  |  |  |  |  |  |  |  |  |  |  |  |  |  |
| All | 17.851 | 18.368 | 20.254 | 19.623 | 19.509 | 20.956 | 21.246 | 22.026 | 22.395 | 24.561 | 24.561 | 25.895 | 24.0 | 9.079 | 11.105 | 12.991 | 22.263 |
| d ≤ 100 | 0.998 | 0.999 | 1.000 | 1.000 | 1.000 | 1.000 | 1.000 | 1.000 | 1.000 | 1.000 | 1.000 | 1.000 | 1.000 | 0.112 | 0.315 | 0.670 | 1.000 |
| d ≤ 500 | 1.000 | 1.000 | 1.000 | 1.000 | 1.000 | 1.000 | 1.000 | 1.000 | 1.000 | 1.000 | 1.000 | 1.000 | 1.000 | 0.607 | 0.752 | 0.958 | 1.000 |
|  |  |  |  |  |  |  |  |  |  |  |  |  |  |  | 0.756 | 0.846 |  |

Table 18.1: Average AUPR ± standard dev. over five seeds for the semi-supervised setting on ADBench. Rank of each model per dataset is provided (in parentheses) (the lower, the better). We use blue and green respectively to mark the top-1 and the top-2 method.

| Dataset | FoMo-0D (D=100) | FoMo-0D (D=20) | DTE-NP | kNN | ICL | DTE-C | LOF | CBLOF | FeatureBagging | SLAD | DDPM | OCSVM | DTE-IG | IForest | MCD |
|---|---|---|---|---|---|---|---|---|---|---|---|---|---|---|---|
| aloi | 4.93±0.05(2) | 6.23±0.22(10.5) | 6.05±0.06(15) | 6.02±0.01(16.5) | 5.5±0.1(30) | 5.76±0.04(26) | 6.54±0.0(5) | 6.4±0.02(9) | 6.8±0.07(3) | 5.99±0.04(18) | 5.07±0.06(20) | 6.52±0.0(7) | 5.98±0.01(14)(19) | 5.82±0.01(23) | 5.55±0.07(29) |
| amazon | 11.1±1.46(9.5) | 10.68±0.14(19) | 11.72±0.0(1) | 11.09±0.0(3) | 10.19±0.08(23) | 11.15±0.65(7.5) | 11.04±0.0(14) | 11.46±0.08(6) | 11.06±0.02(12.5) | 9.74±0.02(29) | 10.76±0.02(16) | 11.06±0.0(12.5) | 10.16±1.08(35) | 11.07±0.19(11) | 11.7±0.03(2) |
| annthyroid | 48.83±0.0(22) | 49.52±0.0(22) | 68.15±0.38(4) | 46.54±1.41(9) | 45.83±2.16(27) | 82.88±0.59(1) | 53.53±0.0(19) | 63.62±4.1(10) | 48.89±8.07(25) | 70.58±0.92(2) | 62.88±2.59(11) | 60.11±0.0(13) | 49.87±9.05(21) | 59.02±5.39(15) | 59.7±0.0(14) |
| backdoor | 43.86±1.4.97(11) | 14.76±10.63(19) | 45.7±12.5(10) | 89.18±1.04(1) | 89.16±1.04(1) | 62.44±2.39(6) | 53.47±2.6(7) | 9.07±1.37(22) | 49.52±8.43(8) | 4.83±0.1(32) | 14.2±0.63(20) | 7.66±0.06(26) | 81.99±4.24(5) | 9.37±1.48(21) | 22.16±13.73(17) |
| breastw | 99.03±0.34(12) | 98.15±0.20(17) | 99.19±0.14(5.5) | 98.92±0.32(13) | 96.79±1.61(21) | 88.25±1.06(27) | 80.01±0.10.99(29) | 99.06±0.24(11) | 52.39±10.9(31) | 99.47±0.21(2) | 98.6±0.51(15) | 99.35±0.21(4) | 81.41±7.86(28) | 99.39±0.10(3) | 98.27±1.23(16) |
| campaign | 34.21±0.47(26) | 37.78±8.11(23) | 99.95±0.67(2) | 98.54±1.41(9) | 48.9±0.98(9) | 46.9±0.71(17) | 40.24±0.0(21) | 48.56±0.59(13) | 33.31±6.99(27) | 48.11±0.12(14) | 48.87±0.29(10) | 49.43±0.0(6) | 46.17±2.19(18) | 45.73±1.85(19) | 47.91±1.49(16) |
| cardio | 78.92±0.0(7) | 76.73±2.29(13) | 77.41±0.85(11) | 77.22±0.0(12) | 47.91±1.1.47(29) | 69.29±1.27(19) | 70.15±0.0(17) | 80.94±2.89(5) | 71.55±3.14(16) | 69.19±0.91(20) | 69.28±0.91(20) | 83.59±0.0(4) | 78.63±2.69(8) | 67.07±0.62(23) | 99.09±0.0(1) |
| cardiotocography | 63.55±0.0(6) | 55.67±1.79(19) | 58.68±1.14(13) | 57.43±0.0(15) | 48.66±3.84(26) | 53.34±1.26(21) | 57.32±0.0(16) | 61.71±1.92(8) | 57.04±2.15(17) | 49.37±0.19(25) | 51.31±1.98(23) | 66.19±0.0(5) | 39.59±5.95(31) | 62.85±3.42(7) | 52.83±0.7(22) |
| celeba | 8.39±0.38(23) | 11.9±1.9(28) | 10.65±0.49(18) | 11.92±0.5(16) | 11.92±0.5(16) | 14.19±2.31(11) | 70.5±0.43(5) | 18.5±4.43(5) | 3.87±0.26(30) | 2.9±0.92(32) | 18.03±2.6(6) | 20.27±0.9(3) | 11.7±1.35(17) | 19.02±3.4(4) | — |
| census | 16.01±2.97(16) | 14.2±1.57(23) | 21.05±0.67(5) | 21.68±0.63(3) | 21.2±0.6(14) | 17.94±1.09(13) | 13.71±0.42(26) | 20.34±0.62(8) | 12.02±0.38(29) | 14.96±4.41(19) | 19.67±0.6(12) | 20.32±0.66(9) | 16.3±1.77(15) | 14.19±0.73(24) | 25.94±1.47(1) |
| cover | 72.37±8.38(5) | 68.27±2.5(6) | 59.97±10.57(8) | 55.79±3.74(9) | 34.48±16.39(13) | 71.33±3.89(10) | 52.97±2.19(1) | 15.96±0.8(22) | 78.1±18.95(3) | 6.96±5.18(26) | 73.27±3.36(4) | 22.28±1.01(18) | 80.43±4.6(2) | 8.66±1.53(25) | 3.14±0.18(28) |
| donors | 98.53±0.48(1) | 75.72±5.93(9) | 83.55±4.56(6) | 89.09±0.94(4) | 98.35±0.87(2) | 71.33±3.89(10) | 63.39±1.89(12) | 46.46±1.12(14) | 65.25±10.23(11) | 46.61(9) | 26.66±2.91(28) | 42.71±0.86(18) | 95.77±2.6(3) | 40.51±3.39(20) | 31.24±13.62(26) |
| fault | 63.08±0.0(8) | 60.86±3.58(17) | 62.17±0.14(11) | 61.98±0.0(13) | 63.1±6.0(7) | 63.73±0.0(24) | 59.4±0.0(32) | 61.29±1.82(15) | 60.84±0.82(31) | 61.0(9) | 61.12±0.0(16) | 61.12±0.0(16) | 63.83±1.23(5) | 59.19±2.02(22) | 63.37±5.99(6) |
| fraud | 55.13±13.77(8) | 34.75±7.84(20) | 42.1±7.63(14) | 38.68±7.19(17) | 53.88±8.78(10) | 62.14±10.02(4) | 55.09±8.19(9) | 27.77±2.14(27) | 63.11±6.38(2) | 44.97±5.43(13) | 69.21±3.46(1) | 29.64±5.04(24) | 51.14±8.64(11) | 18.22±3.66(29) | 60.06±3.89(6) |
| glass | 83.6±4.63(2) | 70.36±9.67(5) | 37.38±15.51(13) | 42.32±8.47(8) | 92.25±8.32(1) | 41.51±5.91(9) | 38.12±9.91(12) | 31.7±2.74(16) | 36.09±6.99(14) | 41.15±3.98(11) | 31.21±1.3(17) | 26.76±7.68(21) | 80.57±12.83(3) | 21.37±3.58(25) | 20.29±3.2(27) |
| hepatitis | 99.1±1.14(5) | 99.65±0.59(14.5) | 82.32±10.26(14) | 82.32±0.0(14) | 99.83±0.38(1) | 55.82±3.85(8) | 43.67±10.79(31) | 63.36±6.81(23) | 44.6±9.55(30) | 99.79±0.47(3) | 95.14±1.34(9) | 77.63±3.49(15) | 99.3±0.45(2) | 55.36±6.09(26) | 56.8±8.0(24) |
| http | 90.58±5.01(11) | 96.39±2.57(7) | 97.1±4.04(6) | 10.0±0.0(1) | 70.82±2.96(17) | 55.45±5.61(20) | 97.12±3.92(5) | 90.31±1.45(13) | 8.21±1.15(30) | 88.09±7.46(15) | 99.99±0.69(2) | 99.88±0.26(3) | 78.8±42.55(16) | 53.48±12.08(21) | 92.16±1.88(9) |
| imdb | 10.53±2.36(2) | 9.08±0.51(15.5) | 9.08±0.0(24) | 8.92±0.0(24) | 10.24±0.13(4) | 9.98±0.0(5) | 9.08±0.0(23) | 8.95±0.0(23) | 9.79±0.03(8) | 8.7±0.0(31) | 8.85±0.0(26) | 8.85±0.0(26) | 10.06±1.49(6) | 8.97±0.17(20.5) | 9.45±0.04(11) |
| internetads | 38.29±5.22(27) | 39.5±7.8(25) | 51.3±2.02(10) | 49.22±0.0(13) | 60.03±1.39(4) | 55.22±3.65(7) | 50.43±0.01(1) | 47.04±0.08(21) | 60.52±1.05(3) | 60.52±1.05(3) | 47.7±0.16(16) | 48.15±0.0(15) | 58.68±6.22(5) | 29.2±1.74(32) | 34.36±0.0(28) |
| ionosphere | 97.75±0.76(8) | 97.77±0.44(7) | 95.75±0.69(5) | 54.85±0.0(6) | 99.0±0.32(1) | 96.83±0.38(3.4) | 50.4±4.0(32) | 96.4±1.4(19) | 97.26±1.65(11) | 95.38±0.51(17.5) | 96.43±0.5(15) | 97.45±0.53(10) | 96.91±2.1(12) | 91.7±1.89(23) | 66.66±0.39(14) |
| landsat | 58.23±0.0(4) | 54.75±2.34(7) | 54.52±4.05(8) | 54.85±0.0(6) | 53.12±1.57(9) | 36.75±1.23(23) | 61.37±0.0(2) | 36.89±0.24(22) | 61.49±0.23(1) | 45.1±0.17(14) | 34.83±0.91(26) | 37.01±0.0(21) | 10.23±1.12(9) | 47.31±3.5(12) | 39.68±4.51(17) |
| letter | 9.33±0.0(12) | 9.13±0.39(14) | 8.57±0.13(22) | 8.7±0.0(20) | 12.8±1.17(3) | 8.95±0.13(15) | 8.33±0.0(25) | 8.33±0.0(25) | 11.66±0.51(4) | 8.93±0.05(16.5) | 9.53±0.51(11) | 8.26±0.0(28) | 8.22±0.0.21(27) | 8.22±0.0(27) | 8.1±0.4(29) |
| lymphography | 99.15±1.26(10) | 10.00±0.0(2.5) | 99.34±0.93(6.5) | 99.17±0.94(9) | 10.00±0.02(2.5) | 86.77±9.24(25) | 84.16±1.27(27) | 98.26±0.34(15) | 72.73±15.29(29) | 99.34±0.12(6.5) | 99.3±1.02(8) | 10.00±0.0(2.5) | 99.76±0.55(5) | 94.38±3.28(21.5) | 86.76±6.31(26) |
| magic.gamma | 87.5±0.18(5) | 87.97±0.12(3.5) | 86.15±0.74(8) | 85.86±0.0(9) | 81.33±0.57(13) | 89.98±0.1(1) | 86.36±0.0(16) | 80.24±0.0(16) | 86.89±0.56(6) | 77.34±0.0(19) | 87.97±0.84(3.5) | 79.16±0.0(17) | 80.27±1.07(15) | 80.73±1.24(11) | 77.21±6.09(21) |
| mammography | 36.28±1.26(16) | 45.99±0.99(3) | 42.00±0.86(5) | 41.27±0.0(8) | 17.11±3.75(30) | 39.8±1.26(11) | 34.07±0.0(17) | 41.06±0.1(10) | 29.3±1.15(19) | 18.99±1.65(28) | 19.93±3.92(27) | 40.52±0.0(11) | 33.16±3.91(13) | 37.94±3.21(14) | 7.96±0.2(32) |
| mnist | 57.97±0.0(16) | 32.15±3.35(28) | 73.08±1.11(11) | 72.72±0.0(12) | 68.45±1.45(17) | 56.26±1.26(18) | 70.97±0.0(14) | 64.09±3.0(9) | 69.29±1.07(6) | 63.82±1.39(14) | 63.42±1.39(14) | 66.2±0.0(10) | 56.1±8.07(19) | 54.1±5.6.22(22) | 55.75±6.4(20) |
| musk | 97.0±2.27(19) | 94.34±6.15(21) | 100.0±0.0(8.5) | 72.72±0.0(12) | 100.0±0.0(8.5) | 100.0±0.0(8.5) | 100.0±0.0(8.5) | 100.0±0.0(8.5) | 100.0±0.0(8.5) | 100.0±0.0(8.5) | 100.0±0.0(8.5) | 100.0±0.0(8.5) | 88.87±21.88(25) | 40.39±26.1(30) | 66.32±12.08(28) |
| optdigits | 31.88±0.0(7) | 20.6±5.31(14) | 31.75±4.86(8) | 29.11±0.0(9) | 5.00±1.8.5.9(1) | 15.34±2.17(17) | 41.63±0.0(2) | 13.97±1.21(18) | 41.23±2.99(4) | 36.3±0.94(6) | 25.56±4.25(11) | 6.92±0.9(23) | 22.06±15.09(12) | 15.41±3.21(16) | 7.1±0.17(22) |
| pageblocks | 62.67±0.0(15) | 60.24±0.0(19) | 67.45±0.1(7) | 67.6±0.0(6) | 68.11±2.3(5) | 66.22±1.23(9) | 71.07±0.0(2) | 70.6±0.11(3) | 70.16±1.6(4) | 64.7±0.79(11) | 62.1±0.95(16) | 64.25±0.0(12) | 57.46±2.99(25) | 43.42±2.0(30) | 63.17±0.04(14) |
| pendigits | 66.33±0.0(9) | 86.15±0.0(4) | 91.89±3.3(3) | 96.99±0.0(1) | 66.41±7.58(8) | 48.44±5.92(16) | 78.55±0.0(7) | 51.24±0.47(15) | 85.67±2.53(5) | 35.35±0.15(22) | 61.14±4.73(10) | 51.78±0.0(14) | 59.2±10.53(11) | 58.79±5.21(12) | 13.2±0.07(30) |
| pima | 79.27±1.74(2) | 76.92±2.16(4) | 70.71±2.44(1) | 75.37±2.99(7) | 78.63±1.94(3) | 67.98±2.86(22) | 68.4±3.79(21) | 62.19±8.71(19) | 69.54±3.79(18) | 50.01±13.26(24) | 71.18±2.06(15.5) | 64.91±5.79(11) | 69.64±4.28(17) | 73.65±2.07(10) | 41.7±6.16(30) |
| satellite | 88.05±0.0(2) | 86.85±0.53(4) | 85.83±0.72(9) | 86.00±0.0(6) | 87.62±0.2(3) | 84.79±0.35(13) | 85.86±0.0(7) | 77.28±0.33(27) | 85.82±0.05(10) | 88.64±0.07(1) | 85.09±0.18(11) | 80.9±0.0(20) | 81.71±1.91(17) | 82.35±0.88(15) | 79.93±2.95(21) |
| satimage-2 | 92.75±0.0(14) | 92.32±1.27(15) | 96.16±0.0(7) | 96.69±0.0(6) | 94.7±1.19(10) | 68.21±3.15(28) | 88.46±0.0(18) | 88.17±1.47(18) | 90.65±0.96(17) | 95.44±0.20(9) | 88.05±5.1(19) | 96.92±0.0(3.5) | 83.13±4.52(22) | 79.66±5.62(11) | 98.31±0.0(1) |
| shuttle | 99.36±0.17(4) | 99.59±0.15(3) | 98.14±0.48(9) | 97.86±0.0(14) | 99.72±0.0(1) | 99.03±0.11(23) | 99.75±0.0(1) | 96.77±0.12(18) | 46.35±25.97(31) | 98.04±0.01(11) | 97.91±0.26(13) | 97.67±0.0(15) | 99.35±0.09(5) | 98.61±0.34(8) | 90.9±0.0(25) |
| smtp | 38.18±9.0(17) | 67.06±3.85(11) | 94.78±2.33(4) | 50.53±5.92(6) | 3.81±3.83(25) | 50.37±6.13(7) | 61.68±1.85(18) | 49.7±6.04(10) | 49.21±1.13(24) | 78.73±7.58(6) | 76.37±5.79(7) | 66.31±0.51(12) | 96.85±2.54(2) | 64.58±1.09(14) | 62.39±0.42(17) |
| spambase | 80.99±0.0(18.5) | 78.91±6.94(22) | 83.65±0.5(26) | 85.32±0.0(8) | 86.78±0.59(2) | 83.8±0.52(5) | 72.71±0.0(29) | 82.03±0.41(14) | 68.4±2.58(31) | 85.64±0.1(3) | 72.89±0.42(28) | 82.19±0.0(12) | 80.99±2.36(18.5) | 88.26±1.32(1) | 8.178±2.94(17) |
| speech | 2.9±0.31(17) | 2.82±0.39(22.5) | 3.17±0.0(10) | 2.8±0.0(25) | 3.38±0.5(5.5) | 2.85±0.5(12.20) | 3.15±0.0(11) | 2.7±0.02(32) | 2.98±0.1(15) | 3.1±0.06(12) | 3.0±0.29(14) | 2.78±0.0(27) | 2.58±0.48(18) | 3.23±1.0(8) | 2.83±0.07(21) |
| stamps | 86.75±5.73(12) | 92.75±5.82(5) | 82.47±3.48(3) | 71.68±8.33(17) | 79.54±5.4(4) | 57.65±8.4(18) | 64.84±8.17(12) | 62.19±8.71(14) | 65.59±8.71(9) | 56.01±13.26(24) | 64.74±12.87(13) | 64.91±5.79(11) | 72.8±10.02(13) | 58.84±6.84(16) | 41.7±6.16(30) |
| thyroid | 67.0±0.0(19) | 59.43±0.0(26) | 81.03±0.31(17) | 80.94±0.0(18) | 51.51±12.73(29) | 91.07±0.97(2) | 60.57±0.0(25) | 77.55±0.25(11) | 81.51±0.0(6.31) | 36.49±17.53(31) | 74.01±0.84(17) | 78.92±0.0(12) | 45.67±16.28(30) | 79.66±5.62(11) | 80.08±0.33(10) |
| vertebral | 69.39±3.97(1) | 55.17±0.73(14) | 25.21±4.9(15) | 26.11±2.49(13) | 58.75±7.5(2) | 35.1±5.39(7) | 33.87±4.3(8) | 25.2±4±3.76(14) | 32.89±4.98(9) | 19.87±4.37(27) | 35.84±9.27(6) | 22.23±2.0(12) | 51.5±10.55(4) | 20.78±3.28(23) | 20.96±2.2(23) |
| vowels | 24.1±0.0(16) | 23.31±0.0(18) | 31.59±1.59(8) | 94.78±2.33(4) | 27.39±5.75(13) | 38.1±4.89(4) | 21.68±1.5(18) | 23.85±4.55(17) | 40.97±2.44(22) | 39.23±1.65(3) | 35.84±9.27(6) | 27.43±0.0(12) | 33.63±6.25(5) | 11.97±0.50(24) | 4.36±0.01(32) |
| waveform | 9.95±0.0(19) | 19.5±1.38(12) | 27.87±0.0(4) | 50.53±5.92(6) | 15.63±0.08(13) | 9.99±1.13(18) | 30.66±0.0(1) | 22.89±1.47(9) | 28.73±3.9(10) | 5.31±0.0(32) | 9.31±1.05(21) | 19.61±1.09(11) | 19.61±0.76(17) | 10.53±0.76(17) | 7.83±0.02(27) |
| wbc | 96.46±2.49(3) | 94.29±4.42(7) | 94.17±0.0(4) | 92.03±5.36(14) | 92.01±5.36(14) | 92.0±0.13(13) | 24.89±3.49(30) | 86.83±6.29(20) | 12.68±6.7.89(32) | 98.1±1.35(1) | 93.84±2.45(10) | 97.15±0.77(2) | 94.24±3.35(8) | 94.24±3.35(8) | 90.16±7.96(17) |
| wdbc | 93.82±8.3(2) | 92.75±5.82(5) | 90.47±7.91(7) | 82.03±3.28(18) | 9.56±6.12(1) | 68.88±2.12.42(24) | 93.64±3.06(4) | 75.67±6.52(22) | 93.67±2.48(3) | 89.00±6.97(8) | 84.3±4.44(13) | 97.41±5.59(9) | 92.07±6.84(6) | 71.98±8.57(23) | 55.26±4.5(27) |
| wilt | 17.46±0.0(1) | 12.2±1.6(14) | 12.25±0.0(13) | 95.11±1.81(11) | 98.26±3.89(7) | 29.94±3.31(3) | 25.41±1.3(6(5) | 15.74±0.0(12) | 8.09±0.16(24) | 12.17±2.0(29) | 17.24±0.46(10) | 7.12±0.0(29) | 52.1±7.69(2) | 8.81±0.51(22) | 21.49±0.0(18) |
| wine | 99.62±0.77(4) | 98.48±3.04(5) | 69.09±13.91(17) | 69.02±13.91(17) | 98.26±3.89(7) | 99.85±0.23(2) | 89.95±2.64(12) | 86.77±2.55(16) | 88.68±3.76(14.5) | 87.45±6.21(3) | 97.65±0.72(8) | 88.68±2.29(14.5) | 99.71±0.65(3) | 67.12±7.62(25) | 83.13±9.43(17) |
| wpbc | 67.36±4.34(8) | 75.17±5.73(4) | 46.11±2.74(15) | 46.11±2.74(15) | 59.31±5.37(1) | 60.35±4.48(10) | 41.2±2.62(21) | 44.8±1.47(18) | 49.97±2.44(22) | 56.56±8.08(17) | 54.61±3.75(12) | 40.88±3.04(23) | 65.78±8.55(9) | 40.73±3.15(24) | 45.16±1.42(17) |
| yeast | 51.0±0.0(4) | 50.24±0.0(8) | 48.26±0.49(22) | 48.26±0.0(21) | 49.55±1.36(14) | 49.74±0.72(12) | 48.94±0.0(19) | 50.74±0.02(6) | 49.89±0.68(9) | 50.56±0.08(7) | 51.05±1.65(3) | 47.95±0.0(23) | 82.85±3.49(7) | 46.78±0.37(26.5) | 45.67±0.0(31) |
| yelp | 13.9±3.0(8) | 14.35±0.69(7) | 16.35±0.0(1) | 16.03±0.0(4) | 10.4±0.09(26) | 16.41±0.36(6) | 16.14±0.0(2) | 13.72±0.03(10) | 16.08±0.07(3) | 9.96±0.05(29) | 12.8±0.02(17) | 13.42±0.0(11) | 12.25±1.58(20) | 13.15±0.29(13) | 13.81±0.02(9) |
| MNIST-C | 38.4±3.34(19) | 33.32±6.13(21) | 47.42±0.0(5) | 46.2±0.0(8) | 51.47±1.1(3) | 47.16±1.36(6) | 51.89±0.0(2) | 42.54±0.12(13) | 52.15±0.36(1) | 46.89±0.14(7) | 47.78±0.12(14) | 41.57±0.0(15) | 44.08±1.87(11) | 32.8±2.43(22) | 25.81±4.48(27) |
| FashionMNIST | 52.19±2.78(19) | 40.62±3.93(24) | 59.79±0.0(5) | 59.15±0.0(8) | 17.39±0.59(17) | 47.16±1.36(6) | 63.61±0.0(3) | 57.75±0.25(18) | 61.94±0.43(11) | 59.6±0.1(7) | 57.14±0.12(12) | 56.53±0.0(14) | 53.73±2.28(18) | 44.73±1.88(23) | 37.41±6.46(25) |
| CIFAR10 | 14.67±0.33(18) | 15.34±0.69(25) | 19.91±0.0(5) | 19.02±0.0(8) | 17.39±0.59(17) | 19.68±0.86(7) | 22.17±0.0(2) | 19.73±0.18(6) | 22.2±0.34(1) | 19.08±0.07(4) | 15.08±0.04(12) | 19.42±0.01(14) | 16.69±1.63(20) | 16.46±0.6(21) | 15.92±1.04(22) |
| SVHN | 73.25±1.79(19) | 71.96±2.43(21) | 15.4±0.0(4) | 15.4±0.0(8) | 15.47±0.47(5) | 15.47±1.05(14) | 15.97±0.0(2) | 15.66±0.11(11) | 16.08±0.12(11) | 15.4±0.26(6) | 72.50±2.43(12) | 73.03±2.82(18) | 14.21±0.95(20) | 18.85±0.49(21) | 12.82±0.93(24) |
| MVTec-AD | 13.51±0.84(13) | 15.56±1.39(3) | 13.64±1.35(2) | 13.17±0.0(13) | 10.63±1.68(12) | 17.31±1.96(1) | 15.04±0.67(5) | 14.86±2.84(14) | 15.77±5.0(1) | 13.59±0.34(11) | 13.05±0.74(1) | 11.83±0.52(20) | 82.85±3.49(7) | 70.0±3.0(7.23) | 50.51±2.92(8) |
| 20news | 15.51±0.0(14) | 16.68±0.0(6) | 17.35±0.0(5) | 16.68±0.0(6) | 15.42±0.38(9) | 14.62±0.88(17) | 25.96±0.0(2) | 13.78±0.01(13) | 25.91±0.0(1) | 12.3±0.0(31) | 12.3±0.0(27) | 12.82±0.0(14) | 11.56±0.42(21) | 11.94±0.32(21) | 15.45±1.06(14) |
| agnews | — | — | — | — | — | — | — | — | — | — | — | — | — | — | 14.62±0.41(11) |
| Rank(avg) | 7.561 | 12.132 | 7.561 | 9.263 | 9.877 | 12.088 | 12.0 | 13.605 | 13.509 | 12.439 | 12.763 | 13.377 | 12.754 | 17.991 | 18.193 |
| All | 0.006 | — | 0.006 | 0.050 | 0.118 | 0.709 | 0.708 | 0.997 | 0.959 | 0.905 | 0.755 | 0.848 | 0.746 | 1.000 | 0.999 |
| d ≤ 20 | 0.005 | — | 0.005 | 0.708 | 0.914 | 0.987 | 0.998 | 0.997 | 1.000 | 0.963 | 0.921 | 0.983 | 0.755 | 1.000 | 1.000 |
| d ≤ 50 | 0.371 | — | 0.371 | 0.651 | 0.729 | 0.997 | 0.995 | 0.997 | 1.000 | 0.934 | 0.969 | 0.986 | 0.990 | 1.000 | 1.000 |
| Rank(avg) | 7.965 | 12.825 | 7.965 | 9.754 | 10.368 | 12.614 | 12.526 | 14.202 | 14.123 | 12.947 | 13.325 | 13.921 | 13.377 | 18.798 | 18.93 |
| All | 0.082 | — | 0.082 | 0.347 | 0.568 | 0.883 | 0.720 | 0.953 | 0.978 | 0.805 | 0.926 | 0.873 | 0.984 | 1.000 | 1.000 |
| d ≤ 100 | 0.485 | — | 0.485 | 0.728 | 0.898 | 0.993 | 0.989 | 0.989 | 1.000 | 0.871 | 0.960 | 0.944 | 0.997 | 1.000 | 1.000 |
| d ≤ 500 | 0.340 | — | 0.340 | 0.627 | 0.816 | 0.980 | 0.946 | 0.988 | 0.999 | 0.894 | 0.957 | 0.941 | 0.993 | 1.000 | 1.000 |

Table 18.2: Average AUPR ± standard dev. over five seeds for the semi-supervised setting on ADBench. Rank of each model per dataset is provided (in parentheses) (the lower, the better). We use blue and green respectively to mark the top-1 and the top-2 method.

| Dataset | VAE | PCA | PlanarFlow | HBOS | GANomaly | GOAD | DIF | COPOD | ECOD | DeepSVDD | LODA | DAGMM | DROCC | DTE-NP^avg | KNN^avg | ICL^avg | DTE-C^avg |
|---|---|---|---|---|---|---|---|---|---|---|---|---|---|---|---|---|---|
| aloi | 6.54±0.0(5) | 6.54±0.0(5) | 5.48±0.3(32) | 6.42±0.0(8) | 8.00±1.27(1) | 5.7±0.24(28) | 5.8±0.0(24) | 5.72±0.0(27) | 6.06±0.0(14) | 6.23±0.35(10.5) | 5.93±0.0(21) | 6.07±0.35(13) | 5.91±0.0(22) | 6.02±0.0(16.5) | 6.09±0.0(12) | 5.49±0.02(31) | 5.79±0.01(25) |
| amazon | 10.72±0.0(17.5) | 10.72±0.0(17.5) | 9.56±0.59(30) | 11.1±0.0(9.5) | 9.93±0.19(27) | 10.94±0.21(15) | 9.89±0.41(28) | 11.15±0.0(7.5) | 10.4±0.0(21) | 10.24±1.27(22) | 10.18±0.78(24) | 9.53±0.46(31) | 9.52±0.0(32) | 11.65±0.0(4.5) | 11.65±0.0(4.5) | 10.01±0.05(26) | 10.42±0.0(20) |
| annthyroid | 56.74±0.0(17) | 56.57±0.0(18) | 65.15±8.63(8) | 39.03±0.0(29) | 34.33±0.62(30) | 58.74±5.0(16) | 17.81±1.03(18) | 29.61±0.0(31) | 40.02±0.0(28) | 27.83±5.93(32) | 49.01±6.73(23) | 48.03±17.56(26) | 63.72±3.08(9) | 66.11±0.0(7) | 67.06±0.0(6) | 52.34±3.72(20) | 68.86±0.2(3) |
| backdoor | 7.97±0.24(24) | 7.9±0.13(25) | 32.15±23.78(14) | 8.56±0.26(23) | 27.87±6.63(16) | 6.31±1.94(28) | 17.81±1.03(18) | 4.84±0.1(30.5) | 4.84±0.1(30.5) | 84.77±2.79(3) | 5.96±3.84(29) | 7.5±3.45(27) | 84.59±1.93(4) | 40.48±0.81(12) | 32.06±0.76(15) | 87.63±1.03(2) | 35.68±0.79(13) |
| breastw | 99.17±0.17(7) | 99.19±0.15(5.5) | 97.47±1.08(20) | 99.08±0.27(9.5) | 93.78±2.46(25) | 98.77±0.37(14) | 52.05±5.33(32) | 99.44±0.12(3) | 99.16±0.2(8) | 96.01±1.28(23) | 96.76±0.62(22) | 90.95±8.37(26) | 63.19±22.35(30) | 97.69±0.4(19) | 99.08±0.22(9.5) | 97.98±0.31(18) | 94.0±0.69(24) |
| campaign | 48.84±0.0(11.5) | 48.84±0.0(11.5) | 42.77±2.92(20) | 49.69±0.0(3) | 39.17±4.25(22) | 23.09±7.18(31) | 24.99±0.0(30) | 51.05±0.0(1) | 49.51±0.0(5) | 36.95±12.7(25) | 29.75±5.8(29) | 32.35±4.69(28) | 20.25±0.0(32) | 49.31±0.0(7) | 49.64±0.0(4) | 48.01±0.47(15) | 37.11±0.64(24) |
| cardio | 86.25±0.0(1) | 86.17±0.0(2) | 68.92±1.77(21) | 50.7±0.0(24) | 67.71±4.44(22) | 67.52±0.82(4) | 29.26±0.0(32) | 74.88±0.0(4) | 78.55±0.0(9) | 38.89±5.52(31) | 72.47±5.87(15) | 55.86±7.98(25) | 51.15±24.55(27) | 78.33±0.0(10) | 79.3±0.0(6) | 53.9±3.02(26) | 49.02±0.29(28) |
| cardiotocography | 69.09±0.0(1) | 69.08±0.0(2) | 59.27±4.03(12) | 16.77±0.78(8) | 54.94±3.83(20) | 4.01±1.21(29) | 33.51±0.0(32) | 56.07±0.0(18) | 68.98±0.0(3) | 7.09±4.32(27) | 60.56±7.01(9) | 59.7±7.63(11) | 43.91±12.35(30) | 58.26±0.0(14) | 59.77±0.0(10) | 47.05±1.34(27) | 45.75±0.91(29) |
| celeba | 20.05±1.11(1.5) | 20.05±1.11(1.5) | 12.85±4.44(14) | 14.01±0.32(25) | 7.47±5.61(26) | 8.69±0.99(32) | 14.66±1.25(21) | 16.48±0.82(9) | 16.9±0.79(7) | 9.04±4.27(32) | 9.46±6.75(21) | 7.65±0.16(25) | 7.65±0.16(25) | 12.63±0.51(15) | 14.31±0.6(10) | 13.27±0.47(13) | 10.05±0.49(19) |
| census | 19.82±0.55(11) | 20.03±0.59(10) | 14.68±1.56(20) | 5.42±0.6(27) | 17.49±2.11(14) | 1.09±0.18(32) | 14.66±1.25(21) | 11.73±0.29(30.5) | 11.73±0.29(30.5) | 15.35±1.07(17) | 13.42±3.92(27) | 13.17±0.9(28) | 14.25±1.1(22) | 20.97±0.43(6) | 20.62±0.44(7) | 22.01±0.42(2) | 15.29±0.31(18) |
| cover | 16.05±0.88(21) | 16.17±0.86(20) | 1.98±0.6(31) | 5.42±0.6(27) | 1.09±0.18(32) | 9.0±1.93(32) | 2.23±0.32(30) | 12.26±0.85(23) | 19.22±1.54(19) | 2.69±1.53(29) | 22.56±9.16(17) | 31.33±5.85(15) | 31.33±5.85(15) | 51.96±2.9(10) | 42.14±2.2(11) | 31.42±3.97(14) | 35.03±0.82(12) |
| donors | 36.01±0.72(23) | 35.22±1.21(24) | 49.31±14.82(13) | 36.33±1.85(22) | 23.9±11.25(30) | 62.14±0.82(12) | 37.26±4.14(21) | 11.73±0.29(30.5) | 41.27±0.97(19) | 42.75±27.48(17) | 25.39±21.33(20) | 19.54±11.02(31) | 30.2±17.77(27) | 87.78±0.99(5) | 79.92±1.13(7) | 76.75±3.37(8) | 45.3±2.85(16) |
| fault | 60.35±0.0(20) | 60.35±0.0(20) | 62.81±9.37(3) | 53.89±0.0(28) | 62.14±4.52(12) | 60.8±0.0(18) | 60.8±0.0(18) | 53.19±0.0(29) | 51.71±0.0(30) | 55.46±1.48(20) | 54.49±2.63(27) | 57.81±4.16(24) | 57.81±4.16(24) | 61.78±0.99(5) | 62.56±0.0(9) | 64.4±1.09(3) | 58.79±0.41(23) |
| fraud | 28.74±5.54(26) | 26.93±1.91(28) | 30.93±6.56(26) | 32.25±5.42(22) | 60.24±1.15(5) | 29.44±44.24(66)(25) | 2.08±0.82(31) | 38.43±3.99(18) | 33.2±4.48(21) | 48.33±17.13(12) | 36.59±15.17(19) | 15.57±20.11(30) | 0.33±0.03(32) | 41.22±5.02(15) | 40.45±4.07(16) | 58.46±7.58(7) | 30.33±4.25(23) |
| glass | 18.51±3.73(30) | 20.96±5.76(28) | 30.93±6.56(26) | 27.61±6.5(20) | 26.04±10.3(22) | 18.33±7.23(31) | 18.33±7.23(31) | 20.09±4.04(28) | 25.02±6.94(23) | 52.35±22.04(7) | 15.55±2.58(32) | 18.62±12.43(29) | 23.14±13.98(24) | 41.38±4.46(10) | 31.91±3.73(15) | 76.58±1.38(4) | 27.78±2.62(19) |
| hepatitis | 64.48±5.1(21) | 64.85±5.06(20) | 89.63±3.08(12) | 63.49±5.99(22) | 73.25±1.71(15) | 65.77±5.43(19) | 89.1±6.91(13) | 56.08±3.5(25) | 45.84±3.47(29) | 98.73±1.05(7) | 50.15±7.4(28) | 54.37±8.05(27) | 34.91±13.15(32) | 89.95±1.8(11) | 70.62±4.48(18) | 99.28±1.2(6) | 74.38±1.79(16) |
| http | 90.42±1.67(12) | 91.69±1.53(10) | 52.23±4.21(22) | 38.95±4.21(28) | 19.29±40.33(11) | 68.38±8.01(18) | 68.38±8.01(18) | 46.31±2.11(24) | 25.18±0.82(28) | 36.09±31.85(27) | 7.46±9.79(31) | 57.53±32.61(19) | 57.53±32.61(19) | 89.48±1.9(14) | 94.39±1.31(8) | 97.56±2.2(4) | 45.08±7.07(25) |
| imdb | 8.71±0.0(29.5) | 8.71±0.0(29.5) | 9.5±0.67(10) | 9.01±0.0(19) | 52.89±1.61(1) | 8.8±0.1(27) | 10.3±0.46(3) | 9.3±0.0(12) | 8.48±0.0(32) | 9.68±1.4(9) | 8.73±0.39(28) | 9.22±0.3(13) | 9.89±0.52(7) | 9.08±0.0(15.5) | 8.96±0.0(22) | 10.17±0.03(5) | 9.21±0.2(14) |
| internetads | 46.97±0.0(22.5) | 46.97±0.0(22.5) | 47.57±1.08(17) | 30.79±0.0(30) | 95.38±1.33(17.5) | 47.43±0.86(20) | 30.56±0.37(31) | 61.74±0.0(1) | 75.64±1.71(30) | 96.82±3.69(17) | 39.32±2.28(26) | 31.78±3.63(29) | 43.08±5.72(24) | 49.03±0.0(14) | 47.47±0.0(19) | 58.52±0.04(6) | 47.51±1.92(18) |
| ionosphere | 91.42±1.37(24) | 90.94±1.25(25) | 97.64±0.86(9) | 64.63±3.76(32) | 95.38±1.33(17.5) | 93.17±2.64(22) | 94.44±1.96(21) | 78.49±3.06(28) | 89.09±0.81(3) | 69.54±0.73(28) | 85.15±3.63(27) | 77.5±4.94(29) | 71.72±21.88(31) | 97.96±0.46(4) | 96.01±0.78(16) | 97.92±1.11(6) | 86.53±4.28(26) |
| landsat | 40.29±7.81(15) | 32.72±0.0(29) | 34.19±0.94(27) | 60.12±0.0(3) | 37.14±8.33(30) | 31.21±0.81(3) | 37.37±0.0(19) | 33.82±0.0(28) | 31.09±0.0(32) | 27.54±11.4(23) | 35.7±5.77(24) | 40.28±2.2(16) | 37.55±1.93(18) | 50.55±0.0(10) | 46.49±0.03(13) | 55.86±0.6(5) | 34.98±0.81(25) |
| letter | 8.0±0.0(32) | 8.01±0.0(31) | 9.19±0.81(13) | 8.73±0.0(19) | 8.45±0.09(23) | 8.13±0.05(28) | 24.7±0.0(1) | 8.85±0.0(18) | 10.65±0.0(7) | 8.93±0.52(16.5) | 8.03±0.25(30) | 10.37±1.67(8) | 15.74±6.69(2) | 8.67±0.0(21) | 8.44±0.0(24) | 11.41±0.43(5) | 10.09±0.04(10) |
| lymphography | 98.59±1.01(12) | 98.40±0.55(14) | 78.51±2.91(18) | 96.55±2.11(18) | 65.83±2.36(30) | 98.76±0.88(11) | 98.1±2.64(16) | 93.9±2.85(23) | 94.38±1.43(21.5) | 96.82±3.69(17) | 24.13±13.29(32) | 73.47±15.88(28) | 30.87±35.63(31) | 96.16±6.43(20) | 98.57±0.65(13) | 100.0±0.0(2.5) | 68.82±5.96(30) |
| magic.gamma | 75.27±0.0(25) | 75.2±0.0(26) | 78.51±2.91(18) | 77.15±0.0(22) | 76.13±2.42(23) | 76.13±2.42(23) | 65.76±0.0(31) | 72.22±0.0(27) | 67.92±0.0(29) | 69.54±0.73(28) | 75.78±1.08(24) | 64.5±4.62(32) | 83.19±0.66(12) | 85.26±0.0(10) | 84.5±0.0(11) | 80.74±13.0(14) | 77.29±3.52(20) |
| mammography | 41.76±0.0(25) | 41.65±0.0(7) | 18.52±9.52(29) | 21.32±0.0(26) | 37.08±23.89(15) | 27.82±3.84(22) | 11.17±0.0(31) | 54.68±0.0(2) | 55.2±0.0(1) | 27.54±11.4(23) | 43.21±2.04(4) | 22.0±17.15(25) | 27.24±2.23(24) | 41.0±0.0(10) | 39.83±0.0(12) | 28.51±2.57(21) | 28.91±1.92(20) |
| mnist | 64.99±0.0(12.5) | 64.99±0.0(12.5) | 55.22±3.33(21) | 22.21±0.0(29) | 47.97±4.73(23) | 65.09±0.57(11) | 20.19±0.0(30) | 16.86±0.0(31.5) | 16.86±0.0(31.5) | 46.0±0.55(5) | 34.07±7.67(27) | 46.06±7.88(24) | 59.72±1.98(15) | 72.0±0.0(3) | 71.36±0.0(5) | 56.56±0.65(17) | 41.15±1.11(26) |
| musk | 100.0±0.0(8.5) | 100.0±0.0(8.5) | 32.68±33.48(31) | 100.0±0.0(8.5) | 100.0±0.0(8.5) | 7.78±1.15(21) | 72.21±0.0(26) | 96.13±0.0(20) | 99.91±0.17(17) | 99.0±1.0(4) | 90.8±10.84(23) | 70.61±23.94(27) | 15.65±19.61(32) | 100.0±0.0(8.5) | 100.0±0.0(8.5) | 89.48±1.94(24) | 62.46±0.0(29) |
| optdigits | 6.01±0.0(25) | 6.02±0.0(24) | 3.93±0.47(31.5) | 42.38±0.0(3) | 11.57±3.94(19) | 63.5±1.17(13) | 5.14±4.0(28) | 5.59±0.0(26.5) | 5.59±0.0(26.5) | 52.05±3.89(27) | 3.93±6.45(31.5) | 4.95±2.39(29) | 19.15±3.94(15) | 27.11±0.0(10) | 21.85±0.0(13) | 41.08±5.34(5) | 10.77±1.02(20) |
| pageblocks | 59.39±0.0(20) | 59.35±0.0(21) | 58.26±5.01(24) | 22.48±0.0(32) | 46.08±16.77(29) | 59.1±0.0(22) | 59.1±0.0(22) | 41.51±0.0(31) | 58.54±0.0(23) | 52.05±3.89(27) | 48.57±3.38(28) | 60.26±12.84(18) | 60.26±12.84(18) | 61.76±0.0(17) | 67.15±0.0(8) | 64.91±1.13(10) | 54.53±1.01(26) |
| pendigits | 39.14±0.0(19) | 38.63±0.0(20) | 14.47±4.88(29) | 43.33±0.0(17) | 14.65±15.39(27) | 33.35±2.85(23) | 22.36±0.0(26) | 30.86±0.0(24) | 41.45±0.0(18) | 9.34±7.78(32) | 37.23±7.94(21) | 11.71±9.8(31) | 14.57±3.43(28) | 93.59±0.0(2) | 82.34±0.0(6) | 58.53±6.17(13) | 29.79±0.96(25) |
| pima | 71.49±3.69(13) | 71.18±3.39(15.5) | 71.23±2.96(14) | 75.88±2.36(6) | 61.66±4.66(27) | 65.15±8.8(24) | 56.78±3.69(30) | 69.07±2.47(19) | 64.77±2.3(25) | 59.75±1.75(28) | 59.36±7.6(29) | 56.48±5.33(31) | 53.42±13.53(32) | 76.68±2.22(5) | 74.01±2.75(9) | 75.51±2.15(8) | 67.61±2.81(23) |
| satellite | 81.04±0.11(19) | 77.79±0.0(25) | 77.86±2.47(24) | 86.49±0.0(5) | 81.83±0.75(16) | 78.96±0.46(23) | 63.25±0.0(32) | 73.33±0.0(29) | 69.57±0.0(31) | 81.1±1.97(18) | 79.77±0.03(22) | 75.98±3.34(28) | 77.46±6.34(26) | 84.95±0.0(12) | 83.82±0.04(14) | 85.84±2.16(8) | 71.75±2.86(30) |
| satimage-2 | 92.94±0.28(13) | 91.92±0.0(16) | 62.47±5.18(29) | 87.68±0.0(20) | 80.25±15.95(23) | 95.89±0.11(8) | 94.87±0.01(22) | 85.27±0.0(21) | 95.2±0.0(24) | 76.28±8.21(26) | 93.72±0.69(12) | 47.48±30.14(30) | 79.32±13.48(25) | 96.92±0.3(3.5) | 97.38±0.0(17) | 99.28±0.28(6) | 43.9±6.54(31) |
| shuttle | 96.27±0.0(19.5) | 96.27±0.0(19.5) | 51.66±12.93(30) | 53.37±5.59(20) | 93.94±4.7(24) | 60.16±26.92(28) | 63.01±1.72(15) | 98.05±0.0(18) | 30.49±0.2(31) | 98.03±0.13(12) | 55.74±0.66(20) | 65.98±23.68(27) | 13.35±0.0(32) | 98.84±0.0(7) | 92.71±0.16(5) | 56.71±7.09(19) | 76.73±0.36(26) |
| skin | 40.14±0.33(27) | 36.30±0.33(28) | 74.74±17.4(8) | 7.88±0.0(29) | 31.88±2.2(30) | 42.18±1.84(20) | 49.46±6.64(12) | 73.58±0.0(26) | 71.26±0.0(40) | 53.04±7.11(21) | 53.04±7.11(21) | 60.37±21.79(23) | 62.62±1.8(13) | 96.37±0.25(3) | 92.71±0.16(5) | 83.01±0.5(10) | 62.54±0.43(16) |
| smtp | 49.38±6.44(13) | 49.5±6.1(11) | 0.77±0.4(30) | 9.0±0.0(22) | 0.1±0.02(32) | 32.4±8.48(19) | 50.5±0.0(32) | 0.99±0.05(29) | 08.01±5.66(1) | 30.73±22.82(21) | 8.10±5.47(24) | 20.92±26.93(22) | 8.69±19.27(23) | 53.94±7.83(4) | 50.27±5.76(8) | 38.67±3.38(16) | 31.76±4.31(20) |
| spambase | 81.84±0.0(15.5) | 81.84±0.0(15.5) | 85.36±2.63(4) | 78.42±0.0(23) | 83.63±1.4(7) | 82.09±0.21(13) | 32.4±8.48(19) | 73.58±0.0(26) | 71.26±0.0(40) | 75.26±2.44(24) | 80.16±4.45(20) | 74.22±2.55(25) | 79.07±3.1(21) | 83.15±0.0(9) | 82.54±0.0(11) | 83.01±0.5(10) | 73.0±0.13(27) |
| speech | 2.77±0.0(28.5) | 2.77±0.0(28.5) | 3.26±0.57(7) | 3.21±0.0(9) | 2.76±0.21(30) | 2.81±0.31(24) | 3.9±0.35(2) | 2.79±0.0(26) | 2.87±0.0(19) | 3.38±0.38(5.5) | 2.97±0.96(16) | 3.95±0.75(1) | 3.57±0.73(4) | 2.82±0.0(22.5) | 2.75±0.0(31) | 3.66±0.2(3) | 3.09±0.15(13) |
| stamps | 59.92±7.99(15) | 58.81±7.82(17) | 52.41±12.56(21) | 52.28±4.56(22) | 33.47±10.36(31) | 49.57±17.72(25) | 49.13±8.19(26) | 56.43±3.1(20) | 49.0±3.86(27) | 42.62±9.94(29) | 57.17±11.21(19) | 46.54±22.11(28) | 28.48±21.94(32) | 72.02±4.82(6) | 65.12±7.1(10) | 66.57±5.4.58(8) | 50.8±7.62(23) |
| thyroid | 81.33±0.0(6) | 81.34±0.0(5) | 75.79±6.78(15) | 76.95±0.0(13) | 53.61±24.51(28) | 80.09±0.89(9) | 60.91±0.0(24) | 30.19±0.0(32) | 64.03±0.0(22) | 78.56±9.59(19) | 64.26±6.24(21) | 63.08±15.77(23) | 74.35±3.86(16) | 76.64±0.0(14) | 81.47±0.0(4) | 55.29±1.31(27) | 66.61±0.6(20) |
| vertebral | 17.85±1.85(30) | 19.26±1.41(28) | 22.97±3.46(20) | 18.86±2.45(29) | 23.36±3.83(18) | 21.39±4.98(22) | 28.64±2.59(10) | 15.48±1.84(32) | 19.93±0.83(26) | 23.42±3.17(17) | 16.72±1.76(31) | 35.99±4.15(31) | 23.35±10.4(19) | 28.38±3.31(11) | 20.42±2.34(25) | 98.38±2.75(6) | 71.37±4.46(21) |
| vowels | 10.1±0.0(27) | 10.51±0.0(25) | 9.67±2.52(28) | 7.88±0.0(29) | 21.63±1.94(19) | 20.94±2.43(20) | 43.28±0.0(1) | 7.06±0.0(31) | 17.72±0.0(21) | 16.88±1.93(22) | 10.43±2.54(26) | 7.32±3.21(30) | 13.19±9.94(23) | 29.34±0.0(10) | 26.62±0.0(14) | 24.91±3.58(15) | 52.0±2.61(13) |
| waveform | 8.4±0.0(25) | 8.41±0.0(24) | 25.08±5.76(7) | 9.0±0.0(22) | 13.33±5.54(14) | 8.96±0.64(23) | 5.7±0.0(31) | 9.88±0.0(29) | 7.35±0.0(29) | 11.52±3.39(15) | 7.8±1.04(28) | 6.07±0.89(30) | 20.07±6.96(10) | 25.7±0.0(6) | 24.92±0.0(8) | 28.91±5.68(2) | 8.24±0.18(26) |
| wbc | 93.22±2.73(11) | 94.3±1.87(6) | 70.99±10.49(25) | 87.73±5.11(19) | 73.87±14.24(23) | 91.95±3.26(15) | 79.24±1.88(31) | 93.16±2.9(12) | 93.11±2.88(13) | 56.51±11.12(27) | 75.74±17.2(22) | 56.8±29.54(26) | 23.95±26.86(31) | 91.84±4.57(16) | 90.15±3.97(18) | 93.88±3.76(9) | 37.32±1.43(28) |
| wdbc | 83.26±6.58(15) | 82.05±4.51(17) | 77.46±13.28(21) | 77.84±2.61(20) | 58.98±15.46(26) | 78.77±4.26(19) | 9.88±2.9(32) | 83.78±3.4(14) | 61.04±3.15(25) | 84.32±8.89(12) | 54.84±16.75(28) | 30.92±26.02(30) | 12.23±18.04(31) | 86.87±5.84(10) | 83.17±5.03(16) | 84.63±3.4(11) | 49.34±5.19(29) |
| wilt | 7.25±0.0(28) | 6.41±0.0(32) | 17.07±2.4(11) | 7.87±0.0(26) | 8.85±1.26(21) | 10.86±1.37(18) | 11.08±0.0(17) | 6.87±0.0(31) | 7.68±0.0(27) | 7.08±0.17(30) | 7.96±0.92(25) | 8.43±1.12(23) | 9.61±2.4(20) | 11.52±0.0(16) | 10.36±0.0(19) | 23.88±1.71(6) | 21.9±0.7(7) |
| wine | 69.47±6.95(23) | 69.22±6.18(24) | 78.85±9.76(18) | 77.71±9.93(20) | 47.63±29.53(30) | 70.1±6.29(22) | 53.48±10.47(27) | 52.34±5.29(28) | 32.56±4.25(31) | 78.56±9.59(19) | 57.85±15.76(26) | 50.92±36.74(29) | 18.5±14.36(32) | 50.71±0.92(10) | 88.72±2.32(13) | 98.38±2.75(6) | 71.37±4.46(21) |
| wpbc | 40.26±2.88(25) | 40.03±2.79(26) | 45.45±2.24(16) | 42.61±2.22(20) | 38.88±3.82(27) | 70.57±6.66(6) | 38.17±2.2(29) | 46.82±0.0(27) | 49.43±0.0(15) | 74.88±5.58(5) | 38.1±3.27(28) | 37.19±3.38(30) | 35.99±4.15(31) | 56.05±1.44(11) | 43.41±2.44(19) | 80.78±4.52(3) | 52.0±2.61(13) |
| yeast | 46.46±0.0(30) | 46.78±0.0(28.5) | 47.04±1.92(26) | 49.78±4.0(10) | 50.77±2.18(5) | 50.77±2.18(5) | 43.96±0.0(32) | 46.82±0.0(27) | 49.43±0.0(15) | 49.21±3.88(16) | 48.95±3.57(18) | 9.31±0.44(31) | 49.76±4.94(11) | 47.65±0.0(24) | 47.24±0.0(25) | 48.61±0.81(20) | 49.58±0.27(13) |
| yelp | 12.76±0.0(19) | 12.77±0.0(18) | 10.69±0.61(25) | 13.04±4.0(15) | 11.38±0.24(24) | 13.13±0.39(14) | 9.05±0.14(32) | 13.25±0.0(12) | 11.89±0.0(21) | 10.02±1.01(28) | 11.77±1.34(23) | 23.41±9.88(26) | 10.1±0.59(27) | 15.43±0.0(5) | 15.25±0.0(6) | 9.79±0.05(30) | 11.86±0.62(22) |
| MNIST-C | 40.34±0.0(17) | 40.33±0.0(18) | 34.1±1.28(20) | 21.6±0.0(29) | 41.23±3.04(25) | 41.23±3.04(25) | 43.44±1.57(7) | 9.52±0.0(31.5) | 9.52±0.0(31.5) | 31.44±3.04(25) | 32.64±5.25(23) | 29.66±7.59(28) | 29.63±12.73(29) | 45.51±0.0(9) | 44.49±0.0(10) | 47.83±0.12(4) | 31.85±0.0(24) |
| FashionMNIST | 56.16±0.0(15.5) | 56.16±0.0(15.5) | 46.78±1.12(21) | 34.86±0.0(27) | 56.59±0.4(13) | 56.59±0.4(13) | 16.15±1.44(30) | 9.52±0.0(31.5) | 9.5±0.0(31.5) | 14.03±1.08(26) | 46.91±3.85(20) | 29.66±7.59(28) | 12.36±1.69(29) | 58.64±0.0(9) | 58.26±0.0(10) | 61.72±0.18(4) | 37.25±0.17(26) |
| CIFAR10 | 19.23±0.0(14.5) | 19.23±0.0(14.5) | 14.24±0.4(19) | 13.97±0.0(27) | 19.4±0.4(12.5) | 19.4±0.4(12.5) | 10.44±0.55(32) | 12.1±0.0(30) | 12.63±0.0(28) | 14.03±1.08(26) | 16.88±1.96(18) | 12.04±1.54(31) | 11.77±1.04(29) | 19.44±0.0(10) | 19.4±0.0(12.5) | 16.71±0.19(19) | 15.68±0.07(24) |
| SVHN | 14.86±0.0(16.5) | 14.86±0.0(16.5) | 14.24±0.4(19) | 11.96±0.0(27) | 14.93±0.18(15) | 14.93±0.18(15) | 10.75±2.28(30) | 9.52±0.0(31.5) | 9.52±0.0(31.5) | 12.4±0.95(28) | 12.71±1.55(25) | 1.43±0.99(28) | 11.43±0.99(28) | 15.21±0.0(9) | 15.13±0.0(11) | 15.16±0.06(10) | 13.16±0.04(23) |
| MVTec-AD | 71.61±2.31(22) | 72.05±2.65(20) | 67.85±3.08(25) | 67.62±2.71(26) | 75.56±2.55(13) | 72.62±2.74(19) | 37.58±1.63(31.5) | 37.58±1.63(31.5) | 37.58±1.63(31.5) | 83.78±3.32(5) | 65.71±3.94(28) | 58.13±6.28(30) | 59.28±10.9(29) | 77.26±0.94(9) | 73.48±0.82(16) | 87.35±0.76(3) | 13.29±0.47(14) |
| 20news | 11.49±0.65(22.5) | 11.34±0.4(25) | 10.57±1.52(30) | 11.14±0.27(28) | 11.49±0.51(22.5) | 11.49±0.51(22.5) | 10.44±1.03(31) | 11.09±0.32(29) | 11.27±0.12(26) | 12.9±1.9(6) | 11.23±1.35(27) | 10.17±1.17(32) | 12.0±1.2.09(19) | 13.66±0.85(10) | 12.64±0.66(17) | 12.99±0.23(15) | 13.29±0.47(14) |
| agnews | 11.62±0.0(23.5) | 11.62±0.0(23.5) | 9.72±0.37(32) | 11.16±0.0(25) | 12.42±0.31(18) | 12.42±0.31(18) | 10.44±1.03(31) | 11.07±0.0(26) | 10.93±0.02(27) | 10.22±1.85(28) | 12.08±0.09(20) | 10.17±1.33(29.5) | 9.74±0.34(31) | 15.45±0.0(8) | 15.45±0.0(8) | 12.79±0.06(16) | 14.03±0.43(13) |
| Rank(avg) | 17.482 | 17.904 | 19.132 | 19.316 | 18.561 | 19.061 | 22.325 | 22.044 | 21.500 | 19.114 | 22.939 | 24.009 | 22.158 | 10.035 | 11.719 | 11.184 | 19.965 |
| All | 0.995 | 0.996 | 1.000 | 1.000 | 0.999 | 0.999 | 1.000 | 1.000 | 1.000 | 1.000 | 1.000 | 1.000 | 1.000 | 0.106 | 0.380 | 0.403 | 1.000 |
| d≤20 | 1.000 | 1.000 | 1.000 | 1.000 | 1.000 | 1.000 | 1.000 | 1.000 | 1.000 | 1.000 | 1.000 | 1.000 | 1.000 | 0.868 | 0.963 | 0.990 | 1.000 |
| d≤50 | 1.000 | 1.000 | 1.000 | 1.000 | 1.000 | 1.000 | 1.000 | 1.000 | 1.000 | 1.000 | 1.000 | 1.000 | 1.000 | 0.806 | 0.932 | 0.952 | 1.000 |
| Rank(avg) | 18.149 | 18.57 | 19.956 | 20.149 | 19.281 | 19.816 | 23.202 | 22.939 | 22.342 | 19.939 | 23.816 | 24.939 | 22.982 | 10.579 | 12.228 | 11.746 | 20.825 |
| All | 0.998 | 0.998 | 1.000 | 1.000 | 1.000 | 1.000 | 1.000 | 1.000 | 1.000 | 1.000 | 1.000 | 1.000 | 1.000 | 0.623 | 0.759 | 0.830 | 1.000 |
| d≤100 | 0.999 | 1.000 | 1.000 | 1.000 | 1.000 | 1.000 | 1.000 | 1.000 | 1.000 | 1.000 | 1.000 | 1.000 | 1.000 | 0.876 | 0.902 | 0.967 | 1.000 |
| d≤500 | 0.999 | 0.999 | 1.000 | 1.000 | 1.000 | 1.000 | 1.000 | 1.000 | 1.000 | 1.000 | 1.000 | 1.000 | 1.000 | 0.849 | 0.892 | 0.982 | 1.000 |

Table 19.1: Average F1 score ± standard dev. over five seeds for the semi-supervised setting on ADBench. Rank of each model per dataset is provided (in parentheses) (the lower, the better). We use blue and green respectively to mark the top-1 and the top-2 method.

| Dataset | FoMo-0D (D=100) | FoMo-0D (D=20) | DTE-NP | kNN | ICL | DTE-C | LOF | CBLOF | FeatureBagging | SLAD | DDPM | OCSVM | DTE-IG | IForest | MCD |
|---|---|---|---|---|---|---|---|---|---|---|---|---|---|---|---|
| abil | 7.82±0.41(4) | 6.49±0.77(12) | 5.82±0.07(17) | 5.9±0.0(15) | 4.91±0.56(22) | 4.2±0.2(26.5) | 8.16±0.0(3) | 6.74±0.08(10) | 8.93±0.57(2) | 5.32±0.11(19) | 6.76±0.19(9) | 7.29±0.0(8) | 5.12±0.68(21) | 4.2±0.26(26.5) | 3.41±0.14(31) |
| amazon | 11.72±3.24(4) | 10.72±1.21(6) | 10.8±0.0(15) | 11.4±0.0(7.5) | 9.4±0.32(28) | 11.8±1.33(2) | 10.0±0.0(23) | 11.52±0.5(6) | 10.0±0.0(14.23) | 10.16±0.22(21) | 11.08±0.11(11) | 12.0±0.0(1) | 10.48±2.37(20) | 11.28±0.64(10) | 11.32±0.23(9) |
| annthyroid | 49.25±0.9(24) | 51.5±0.0(17) | 61.8±4.15(94) | 0.09±0.0(3) | 49.4±0.88(23) | 7.7±2.8(0.5+1) | 49.63±0.0(22) | 56.7±3.3(12) | 50.67±5.76(18) | 65.99±0.62(21) | 57.23±2.96(11) | 53.94±0.0(13) | 48.8±7.04(23) | 53.02±4.22(14) | 50.37±0.0(19) |
| backdoor | 45.12±1.91(11) | 12.82±15.17(19) | 51.8±8.17(20) | 1.0±0.0(3) | 95.4±0.32(18) | 8.7±1.1(14) | 72.4±2.2(7) | 7.7±1.1(24) | 58.5±5.7(6.8) | 58.3±0.3(10) | 9.6±0.62(20) | 7.94±8.8(23) | 8.4±5.2(16.4) | 4.07±2.4(29) | 19.49±27.36(18) |
| breastw | 95.92±0.79(12) | 96.81±0.49(4) | 96.66±0.67(15.5) | 95.77±0.19(17) | 95.91±0.68(13) | 88.18±2.86(26) | 85.4±5.50(27) | 95.78±0.27(15.5) | 60.99±1.4.7(30) | 96.87±0.65(3) | 95.04±0.75(20) | 96.66±1.1(5.5) | 74.03±10.32(29) | 96.9(1) | 95.84±6.67(14) |
| campaign | 39.72±6.9(24) | 40.61±9.3(23) | 50.98±0.6(114) | 50.37±0.9(7) | 51.03±0.73(3) | 52.12±0.62(11) | 42.24±0.0(20) | 49.29±0.2(11) | 37.15±6.73(26) | 49.83±0.0(9) | 50.4±0.68(6) | 49.59±0.0(119) | 47.85±2.0(18) | 43.7±0.91(19) | 48.33±1.62(16) |
| cardio | 72.16±0.0(15) | 68.86±2.67(9) | 63.07±0.0(13) | 61.93±0.0(17) | 52.16±5.49(27) | 58.3±6.76(23) | 62.5±0.0(16) | 70.0±5.04(8) | 62.95±3.04(14.5) | 60.8±0.0(19) | 61.7±1.73(18) | 70.45±0.0(6.5) | 36.82±1.29(30) | 67.5±3.32(10) | 59.09±0.0(21) |
| cardiotocography | 56.22±0.0(6) | 48.41±1.52(13.5) | 46.78±1.44(19) | 46.35±0.0(20.5) | 38.93±2.89(23) | 38.37±2.23(25) | 48.28±0.0(15.5) | 51.42±3.49(10) | 48.41±1.57(13.5) | 33.82±0.47(30) | 38.84±2.75(24) | 57.94±0.0(5) | 31.67±2.63(31) | 56.1±2.73(7) | 36.48±1.78(28) |
| celeba | 8.41±0.88(28) | 7.38±1.96(29) | 15.81±0.69(18) | 15.81±0.69(16) | 17.19±0.83(16) | 17.35±3.48(14) | 1.91±0.47(32) | 25.32±7.1(4.5) | 2.92±0.89(31) | 13.69±1.66(20) | 26.02±2.89(4) | 27.37(1) | 19.11±5.61(10.5) | 17.33±2.29(15) | 25.12±4.44(6) |
| census | 17.4±5.36(17) | 7.93±5.08(28) | 22.21±0.6(5) | 22.52±0.54(4) | 23.96±0.54(3) | 17.43±2.28(16) | 13.09±0.42(23) | 21.46±0.28(8) | 3.47±1.3(30) | 8.65±1.85(27) | 20.27±0.48(12) | 20.67±0.38(11) | 17.47±2.81(15) | 10.54±1.4(25) | 29.4±0.37(1) |
| cover | 69.92±7.24(6) | 69.59±1.32(7) | 6.84±7.4(8) | 6.51±2.15(9) | 94.91±0.67(3) | 71.04±0.46(5) | 82.4(2+1) | 13.99±1.06(23) | 56.11±4.4(12) | 9.14±8.13(27) | 76.86±1.33(4) | 24.15±1.58(17) | 77.79±3.9(13) | 11.61±1.24(25) | 3.44±0.31(28) |
| donors | 37.83(8+0.79(1) | 77.32±3.52(10) | 92.9±2.83(6) | 94.91±0.67(3) | 97.22±1.04(2) | 82.17±2.54(8) | 74.47±2.04(11) | 48.48±1.33(14) | 56.11±1.4.4(12) | 55.86±8.74(13) | 25.01±7.13(27) | 39.52±1.55(21) | 93.11±3.02(5) | 43.46±3.54(18) | 33.32±1.42(25) |
| fault | 57.5±0.0(8) | 55.39±3.22(20) | 56.2±0.4(16) | 56.5±2.15(9) | 55.57±0.0(19) | 56.23±1.73(15) | 50.67±0.0(31) | 56.1±4.9.87(32) | 56.1±0.5(32) | 60.06±0.24(3) | 58.4±5±1.15(5) | 55.13±0.0(23) | 55.81±0.56(18) | 53.64±1.34(25.5) | 56.37±1.62(13) |
| fraud | 61.34±1.1.8(6) | 44.63±8.94(20) | 48.36±5.41(13) | 43.22±4.87(18) | 97.43±5.97(10) | 68.23±1.31.1(12) | 59.47±4.5(8) | 34.39±0.60(26) | 67.65±4.19(3) | 47.39±1.57(15) | 73.16±2.29(1) | 41.52±5.56(21) | 55.61±10.96(12) | 29.03±4.09(29) | 56.19±3.88(11) |
| glass | 77.6±0.0(7.5) | 66.91±1.91(14) | 24.58±16.83(15) | 25.87±13.76(13) | 57.63±0.9(11) | 37.46±6.36(8) | 37.46±6.20(8) | 23.73±14.21(17) | 22.4±6.7.76(16) | 32.81±1.36(10) | 33.03±2.67(9) | 1.497±7.54(30) | 73.8±1.2(2) | 16.18±7.01(26) | 16.25±0.99(25) |
| hepatitis | 90.6(1±0.67(2.5) | 97.4±3.7(6) | 79.03±1.95(14) | 81.29±5.04(11) | 99.64±0.79(2.5) | 92.6±3.32(8) | 41.95±10.62(29) | 66.93±8.55(16) | 41.27±12.6(30) | 99.6(1±0.79(2.5) | 86.69±1.26(9) | 66.01±4.93(17) | 99.6(1±0.79(2.5) | 5.0±5.54(24) | 49.49±1.04(28) |
| http | 89.24±5.79(13) | 96.0±1.94(7) | 97.43±3.53(4) | 100.0±0.0(1) | 60.7±5.56(17) | 24.59±15.3(23) | 96.78±3.13(6) | 91.29±1.52(12) | 0.0±0.0(31.5) | 88.49±9.31(15) | 99.68±0.3(3) | 99.75±0.3(2) | 78.82±42.94(16) | 93.05±1.5(19) | 93.05±1.5(19) |
| imdb | 10.12±4.42(6) | 7.6±1.82(10) | 5.2±0.0(29) | 5.4±0.0(26) | 10.56±0.54(2) | 7.24±1.62(12) | 6.4±0.0(18.5) | 6.96±0.09(14.5) | 6.96±0.09(25) | 10.24±0.22(4) | 5.56±0.09(25) | 5.8±0.0(21) | 10.36±5.13(3) | 6.2±0.49(20) | 7.44±0.22(11) |
| internetads | 40.82±7.48(25) | 33.8±3.72(27) | 53.21±3.77(11) | 51.9±0.0(13) | 55.87±0.91(4) | 64.78±2.48(1) | 54.62±0.0(8) | 45.76±0.24(21) | 45.87±0.35(20) | 57.88±0.3(2) | 45.87±0.35(20) | 46.2±0.0(18) | 54.95±5.79(7) | 26.41±4.44(32) | 33.42±0.0(28) |
| ionosphere | 92.31±1.33(6) | 89.99±1.04(12) | 91.51±2.0(9.8) | 90.47±2.17(11) | 94.15±1.61(21) | 89.58±0.9(14) | 87.53±3.2±6(18) | 91.96±2.4(17) | 87.65±2.26(18) | 92.65±1.24(4) | 78.88±0.95(4) | 92.62±1.51(5) | 89.68±4.23(13) | 83.4±3.5(22) | 88.64±1.58(15.5) |
| landsat | 52.66±0.0(4) | 49.93±1.75(9) | 51.22±2.55(7) | 51.46±0.0(6) | 53.82±0.03(2) | 38.29±2.94(23) | 53.64±0.0(3) | 38.33±0.19(22) | 53.97(±0.17(1) | 46.93±0.04(12) | 40.23±1.02(19) | 38.5±0.0(21) | 30.26±4.37(32) | 43.27±1.34(14) | 47.7±9.54(11) |
| letter | 4.0±0.0(11.5) | 0.2±0.0(32) | 95.78±5.8(18) | 94.5±6.47(10) | 7.2±0.04(7) | 2.4±1.6(17.5) | 10.0±0.0(27.5) | 1.0±0.0(27.5) | 8.6±1.34(5.5) | 1.6±0.5(30) | 3.6±0.89(15) | 1.0±0.0(27.5) | 3.4±0.55(16) | 3.8±1.1(13.5) | 2.4±0.89(17.5) |
| lymphography | 96.39±5.14(7) | 100.0±0.0(2.5) | 78.78±0.07(15) | 76.17±0.0(9) | 100.0±0.0(2.5) | 82.01±3.83(20) | 74.87±7.44(27) | 89.28±1.85(19) | 65.34±17.16(29) | 99.47±4.18(5) | 95.19±6.67(9) | 100.0±0.0(2.5) | 97.89±4.71(6) | 85.05±4.72(23) | 83.73±4.92(24) |
| magic.gamma | 77.89±0.22(6) | 89.04±2.57(5) | 42.28±1.03(10) | 40.38±0.0(12) | 69.55±0.47(15) | 8.67±0.8(2) | 76.08±0.0(10) | 69.24±0.01(17) | 76.85±0.7(7) | 39.38±0.97(13) | 24.62±4.04(26) | 41.92±0.0(11) | 35.31±7.05(20) | 69.64±1.25(14) | 67.89±0.21(19) |
| mammography | 37.0±0.62(17) | 45.31±0.82(5) | 7.2±1.2(1) | 71.96(±0.0(2) | 17.38±3.62(29) | 58.46±2.88(16) | 71.4±0.0(3) | 49.23±0.0(3) | 68.89±1.44(6) | 60.37±5.89(15) | 24.62±4.04(26) | 64.29±0.0(10) | 50.43±8.41(22) | 39.23±2.48(14) | 26.26±0.63(32) |
| mnist | 62.0±0.0(14) | 31.71±4.35(28) | 80.67±0.48(3) | 71.56(1+1.2(1) | 64.89±1.93(9) | 58.46±2.88(16) | 71.4±0.0(3) | 65.94±0.4(8) | 68.89±1.44(6) | 67.0±0.39(7) | 63.57±5.89(15) | 50.43±8.41(22) | 50.43±8.41(22) | 52.6±4.64(20) | 55.97±2.01(18) |
| musk | 95.2±3.23(18) | 92.16±9.17(20) | 27.2±0.73(9) | 100.0±0.0(8.5) | 83.1±5.97(25) | 100.0±0.0(8.5) | 100.0±0.0(8.5) | 1.6±0.37(30) | 100.0±0.0(8.5) | 100.0±0.0(8.5) | 5.8±2.09(13) | 100.0±0.0(8.5) | 88.66±25.36(13) | 35.88±24.78(30) | 53.6±1.4.5(29) |
| optdigits | 37.33±0.0(7) | 22.4±7.5(11) | 27.2±0.73(9) | 21.33±0.0(12) | 57.73±5.8(4) | 62.12±0.4(8) | 53.33±0.02(2) | 65.89±0.28(3) | 47.33±4.45(3) | 39.87±6.73(6) | 28.4±7.0(6.8) | 0.67±0.02(3.5) | 27.07±18.37(10) | 12.8±6.26(15) | 33.42±0.0(24) |
| pageblocks | 61.18±0.0(10) | 64.71±0.0(15) | 59.29±0.26(12) | 59.02±0.0(13.5) | 64.9±1.29(4) | 62.12±0.4(7.8) | 65.88±0.0(2) | 65.29±0.28(3) | 63.45±1.67(6) | 60.2±0.0(11) | 50.31±0.73(22) | 55.69±0.0(18) | 54.59±2.54(20) | 42.63±2.29(29) | 57.57±0.0(11.7) |
| pendigits | 69.23±0.0(8) | 80.13±0.0(5) | 83.46±0.0(3) | 90.28±0.0(1) | 61.15±5.82(10) | 56.03±8.28(14) | 76.28±0.0(7) | 49.23±0.7(16) | 83.33±3.51(4) | 41.36±1.05(17) | 64.62±2.62(9) | 53.21±0.0(15) | 59.74±9.95(11) | 57.95±4.75(12) | 14.36±0.35(29) |
| pima | 74.65(1±0.84(2) | 72.13±1.63(5) | 71.91±0.99(12) | 70.56±2.49(8) | 73.54±2.63(3) | 65.35±3.35(21) | 66.75±3.0(18) | 68.78±2.76(14) | 68.43±3.04(16) | 58.87±2.87(27) | 73.74±0.27(7) | 68.59±2.02(15) | 63.17±3.7(22) | 67.12±0.64(21) | 63.15±5.1(29) |
| satellite | 77.75±0.0(3) | 72.08±0.6(6) | 95.78±5.8(18) | 71.81±0.0(13) | 74.99±0.46(5) | 72.33±0.63(11) | 72.64±0.0(9) | 64.02±0.07(36) | 72.6±0.08(10) | 73.74±0.27(7) | 90.31±3.17(9) | 91.55±0.0(4) | 70.57±3.05(15) | 96.71±0.53(16) | 95.77(±0.0(1) |
| satimage-2 | 97.72±0.2(15) | 98.14±0.64(6) | 98.3±0.0(17) | 90.14±0.0(7.5) | 88.45±1.54(14) | 66.48±2.71(28) | 81.69±0.0(19) | 92.96(±0.0(2) | 84.51±1.0(17) | 88.73±0.0(11.5) | 50.71±4.98(13) | 80.02±0.42(8) | 78.31±5.23(23) | 89.58±1.61(9) | 84.62±0.0(25) |
| shuttle | 49.67±1.9(25) | 73.54±2.57(15) | 98.3±0.0(17) | 96.35(1±0.62(1) | 1.09±0.97(32) | 82.23±0.4(16) | 70.8±2.09(18) | 81.39±0.38(7) | 59.02±1.62(19) | 73.42±5.35(16) | 58.3±0.08(7) | 80.02±0.42(8) | 93.36±2.92(5) | 78.06±0.72(11) | 76.76±0.35(12) |
| skin | 36.65±2.8(20) | 68.64±4.98(11) | 69.50(±4.42(2.5) | 69.5±4.3(17.5) | 6.96±12.38(25) | 69.5±4.3(17.5) | 65.82±5.88(14) | 69.5±4.3(17.5) | 0.0±0.0(29) | 69.50(±4.42(2.5) | 56.19±1.77(15) | 69.5±4.43(17.5) | 37.91±24.53(19) | 0.0±0.0(29) | 0.0±0.0(29) |
| smtp | 74.27±0.0(21) | 72.76±8.49(24) | 80.67±0.48(3) | 3.28±0.0(13) | 2.62±1.87(24) | 80.02±0.23(7) | 73.957±0.0(22) | 78.78±0.49(12) | 71.52±1.85(26) | 63.55±0.95(31) | 78.56±0.0(13) | 78.56±0.0(13) | 75.18±2.65(19) | 80.09±1.34(5) | 77.7±2.09(17) |
| spambase | 2.95±1.23(20) | 3.61±2.1(17) | 4.192(1±0.2.5) | 3.28±0.0(13) | 71.1(7±7.83(4) | 3.93±1.4(15.5) | 3.28±0.0(13) | 1.6±4.0(30.5) | 2.90±0.73(20) | 31.5±0.13(2) | 2.95±1.37(20) | 3.28±0.0(13) | 1.97±1.8(28.5) | 3.93±2.49(5.5) | 2.62±1.47(24) |
| speech | 58.7(2±5.7(2) | 8.5±1.1.95(3) | 58.35±5.1.4.5(5) | 86.41±3.23(13) | 71.1(7±7.83(4) | 57.97±11.6(19) | 63.52±13.2(13) | 64.39±11.86(11) | 64.7±12.55(10) | 50.99±12.77(24) | 62.98±12.09(15) | 63.44±9.99(14) | 70.23±11.35(7) | 63.02±8.8(17) | 30.97±8.32(31) |
| stamps | 59.1(1±0.92(3.5) | 63.44±0.0(21) | 71.84±0.0(967) | 73.5±2.59(6.5) | 56.18±5.72(10) | 75.48±0.73(6.5) | 52.09±0.0(28) | 71.49±1.08(10.5) | 64.7±12.55(10) | 71.18±1.18(15) | 75.44±2.073(15) | 73.27±0.60(5.5) | 47.74±10.66(30) | 18.9±1.18(21) | 73.12±0.0(14) |
| thyroid | 43.14(1±0.84(1) | 57.9±0.46(3) | 21.56±1.78(15) | 22.93±5.0(14) | 63.9(9±5.17(2) | 42.13±1.49(9) | 33.08±6.21(10) | 25.713±3.79(13) | 33.33±8.67(10) | 14.2±0.47(25) | 35.84±1.38(3) | 20.37±5.46(18) | 46.58±102(24) | 44.46±0.86(27) | 46.27±0.18(25) |
| vertebral | 28.0±0.0(10.5) | 28.4±2.58(4) | 26.0±0.09(8) | 26.0±0.0(13.5) | 24.4±8.29(17) | 37.2±4.15(4) | 34.0±0.0(7) | 19.6±2.6(22) | 35.2±4.15(6) | 38.84±4.38(8) | 28.0±0.0(10.5) | 28.0±0.0(10.5) | 36.0±2.45(5) | 15.2±3.63(23) | 12.52±0.3(23) |
| vowels | 8.0±0.0(25) | 28.4±2.58(4) | 80.67±0.48(3) | 27.0±0.0(7) | 26.8±5.81(9) | 12.2±2.77(16) | 28.0±0.0(5) | 26.1±1.52(11) | 35.2±4.15(6) | 2.2±1.1(32) | 12.0±1.221(7.5) | 28.0±0.0(10.5) | 24.8±4.27(13) | 10.2±2.17(19) | 9.0±0.0(22) |
| waveform | 91.15±6.4(13) | 87.43±9.19(10) | 89.35±3.4.5(5) | 86.41±3.23(13) | 92.98±4.1.57(1) | 32.49±10.28(29) | 20.27±9.64(31) | 80.68±9.87(18) | 6.29±14.06(32) | 85.23±6.41(7) | 86.03±5.16(15) | 89.84±2.99(4) | 62.64±10.31(24) | 88.25±2.4(19) | 79.13±1.06(20) |
| wbc | 92.6(1±5.52(1) | 87.81±6.07(4) | 85.05±6.3(8) | 78.7±2.32(17) | 92.8±4.1.57(1) | 35.1.8±2.59(3) | 85.63±6.59(6) | 80.68±9.87(18) | 87.11±5.67(5) | 85.23±6.41(7) | 79.28±7.36(13) | 80.34±2.47(11) | 89.47±7.99(3) | 70.91±11.05(21) | 58.73±7.26(25) |
| wilt | 77.04(1±0.0(1) | 28.79±0.0(4) | 24.21±1.49(19) | 2.33±0.0(20) | 35.1.8±2.59(3) | 19.23±0.0(14) | 16.73±0.0(9) | 1.09±0.43(29) | 19.14±13.28(7) | 7.0±0.0(14) | 20.23±0.91(6) | 1.17±0.0(28) | 64.75(1±4.54(2) | 2.02±0.33(21) | 7.78±0.0(13) |
| wine | 99.32(1±1.36(3) | 98.18±4.64(6) | 92.13±1.85(8) | 57.16±5.61(11) | 59.32±1.52(3) | 98.29±2.42(5) | 80.84±3.22(14) | 75.73±7.02(17) | 82.7±5.26(12) | 100.0(±0.0(1) | 90.31±3.17(9) | 78.28±3.76(15) | 99.32±1.52(3) | 71.05±4.25(19) | 74.67±15.13(18) |
| wpbc | 58.02±1.22(9) | 71.46±4.08(4) | 68.16±14.02(6) | 49.09±2.1.14 | 90.53(1±0.97(1) | 57.76±6.4.1(10) | 41.26±4.76(21) | 44.45±2.96(17) | 38.86±5.39(22) | 87.0(1±3.64(2) | 50.71±4.98(13) | 35.79±1.33(27) | 59.51±9.36(8) | 36.63±1.93(24) | 41.28±3.93(20) |
| yeast | 50.49±0.0(6) | 49.7±0.0(8) | 46.15±0.04(26) | 46.75±0.0(20) | 50.36±1.38(7) | 49.23±1.12(13) | 47.73±0.0(18) | 51.36±0.11(4) | 47.5±1.51(19) | 49.27±0.22(12) | 50.85±2.02(5) | 46.55±0.022(22) | 49.66±2.28(9) | 44.46±0.86(27) | 46.27±0.18(25) |
| yelp | 16.12±5.95(11) | 17.4±1.44(6) | 17.4±1.43(6) | 18.8±0.0(4) | 8.88±0.73(27) | 14.72±1.51(17) | 20.6(1±0.0(2) | 13.4±0.2(20.5) | 20.72±0.23(1) | 7.12±0.11(31) | 16.08±0.18(12) | 15.2±0.0(16) | 14.6±2.17(18) | 13.8±0.62(14) | 12.52±0.3(23) |
| MNIST-C | 39.11±2.44(19) | 35.11±6.03(20) | 47.51±0.0(7) | 46.3±0.0(8) | 52.07±1.25(3) | 48.68±1.3(5) | 52.96(1±0.0(2) | 42.94±0.16(13) | 53.2(1±0.6(1) | 47.66±0.22(6) | 42.44±0.25(14) | 42.32±0.05(15) | 46.21±5.11(9) | 34.17±2.59(21) | 25.69±4.92(27) |
| FashionMNIST | 53.44±2.28(19) | 51.6±4.09(7.24) | 39.11±6.07(24) | 58.05±0.0(9) | 62.9±1.07(3) | 59.22±0.99(8) | 63.8(1±0.0(2) | 57.14±0.32(12) | 63.78±0.56(11) | 59.77±0.26(5) | 56.67±0.27(13) | 56.44±0.01(14) | 54.66±2.68(18) | 45.33±2.4(23) | 37.35±5.77(25) |
| CIFAR10 | 20.03±1.05(18) | 17.4±5.1.08(24) | 19.1(9±0.0(7) | 22.85±0.0(11) | 23.79±1.09(3) | 23.56±0.99(2) | 27.0(7±0.0(2) | 53.26±0.0(32) | 27.22±0.71(11) | 24.67±0.0(8) | 22.96±0.41(8) | 22.9±0.0(12) | 22.81±0.0(12) | 18.19±1.18(21) | 17.91±2.05(22) |
| SVHN | 17.72±0.63(18) | 15.9±0.89(22) | 18.85±0.0(6) | 19.32±0.0(15) | 19.55±0.99(2) | 19.23±0.0(14) | 19.25±0.0(4) | 18.65±0.37(12.5) | 18.65±0.37(12.5) | 19.46±0.45(1) | 18.73±0.23(10) | 18.43±0.0(15) | 17.5±1.78(19) | 16.4±1.06(21) | 15.31±1.94(25) |
| MVTec-AD | 64.2±1.7(21) | 64.4±2.1(22) | 75.69±2.88(7) | 67.37±3.1(12) | 82.6(1±2.36(1) | 78.98±3.08(4) | 67.29±3.16(13) | 66.48±3.06(15) | 67.48±3.26(11) | 81.77(1±2.84(18) | 65.02±2.84(18) | 64.6±2.67(19.5) | 75.87±4.33(6) | 64.6±2.65(19.5) | 72.35±3.2(8) |
| 20news | 14.59±2.22(10) | 17.59±2.2(13) | 17.83(1±1.58(2) | 14.61±1.48(9) | 16.43±1.74(6) | 18.50(1±1.69(1) | 16.77±1.37(4) | 12.83±2.11(16) | 16.04±1.39(5) | 14.09±1.43(14) | 10.55±1.27(24) | 11.31±1.11(19) | 14.33±4.65(12) | 10.99±1.66(26) | 14.72±1.95(8) |
| agnews | 17.61±6.0(10) | 20.08±1.0(4) | 20.7±0.0(5) | 14.61±1.48(9) | 17.79±0.72(9) | 23.93±3.44(3) | 20.6(1±0.0(1) | 15.35±0.25(11) | 30.53(1±0.37(2) | 13.33±0.14(17) | 12.52±0.18(21) | 13.7±0.0(15) | 14.11±3.85(14) | 12.88±0.65(22) | 13.08±0.32(19) |
| Rank (avg) | | | 8.754 | 9.202 | 9.772 | 11.158 | 12.280 | 13.658 | 13.456 | 11.719 | 12.64 | 14.658 | 13.482 | 17.81 | 18.912 |
| All | | | 0.037 | 0.324 | 0.118 | 0.626 | 0.764 | 0.952 | 0.949 | 0.597 | 0.907 | 0.923 | 0.603 | 1.000 | 1.000 |
| d ≤ 20 | | | 0.767 | 0.905 | 0.910 | 0.958 | 1.000 | 0.999 | 1.000 | 0.979 | 0.990 | 0.996 | 0.868 | 1.000 | 1.000 |
| d ≤ 50 | | | 0.768 | 0.955 | 0.666 | 0.993 | 1.000 | 0.999 | 1.000 | 0.963 | 0.993 | 0.996 | 0.956 | 1.000 | 1.000 |
| Rank (avg) | 11.272 | 12.456 | 9.193 | 10.728 | 10.263 | 11.667 | 12.781 | 14.272 | 14.009 | 12.211 | 13.228 | 14.193 | 14.114 | 18.57 | 19.702 |
| All | - | - | 0.143 | 0.555 | 0.550 | 0.790 | 0.736 | 0.973 | 0.966 | 0.809 | 0.962 | 0.963 | 0.979 | 1.000 | 1.000 |
| d ≤ 100 | - | - | 0.640 | 0.889 | 0.868 | 0.986 | 0.986 | 0.991 | 0.999 | 0.912 | 0.967 | 0.982 | 0.996 | 1.000 | 1.000 |
| d ≤ 500 | - | - | 0.448 | 0.786 | 0.806 | 0.960 | 0.944 | 0.991 | 0.997 | 0.928 | 0.969 | 0.979 | 0.992 | 1.000 | 1.000 |

Table 19.2: Average F1 score ± standard dev. over five seeds for the semi-supervised setting on ADBench. Rank of each model per dataset is provided (in parentheses) (the lower, the better). We use blue and green respectively to mark the top-1 and the top-2 method.

| Dataset | VAE | PCA | PlanarFlow | HBOS | GANomaly | GOAD | DIF | COPOD | ECOD | DeepSVDD | LODA | DAGMM | DROCC | DTE-NP | KNN | ICL | DTE-C |
|---|---|---|---|---|---|---|---|---|---|---|---|---|---|---|---|---|---|
| aloi | 7.63±0.0(5.5) | 7.63±0.0(5.5) | 3.83±0.0.72(29) | 7.43±0.0(7) | 9.35±1.2(1) | 5.73±1.45(18) | 3.91±0.0(28) | 4.58±0.0(24) | 4.44±0.0(25) | 5.17±0.92(20) | 6.6±1.58(11) | 5.98±1.67(14) | 0.0±0.0(32) | 5.86±0.0(16) | 6.01±0.0(13) | 4.59±0.07(23) | 3.56±0.03(30) |
| amazon | 11.0±0.0(12.5) | 11.0±0.0(12.5) | 9.48±1.62(26.5) | 10.6±0.0(18.5) | 8.96±0.74(30) | 11.56±0.26(5) | 9.32±0.66(29) | 11.4±0.07.5) | 10.0±0.0(23) | 11.76±2.19(3) | 10.64±1.18(17) | 9.48±1.23(26.5) | 0.0±0.0(32) | 10.6±0.0(18.5) | 10.92±0.0(14) | 9.94±0.32(25) | 7.01±0.72(31) |
| annthyroid | 50.0±0.0(21) | 50.0±0.0(21) | 60.0±7.76(7) | 35.96±0.0(29) | 34.27±2.48(30) | 55.77±4.62(13) | 58.99±0.0(9) | 13.65±0.0(31) | 38.39±0.0(28) | 23.33±5.12(32) | 46.78±5.94(26) | 45.66±16.44(27) | 57.42±2.53(10) | 60.56±0.0(6) | 59.18±0.0(8) | 52.56±3.29(16) | 61.63±0.2(5) |
| backdoor | 8.5±1.27(21) | 8.3±1.0(22) | 36.73±22.17(14) | 6.93±0.61(25) | 21.92±0.41(16) | 4.79±3.47(27) | 20.28±2.02(17) | 0.0±0.0(31) | 0.0±0.0(31) | 82.96±3.28(5) | 4.64±4.15(28) | 5.25±3.94(26) | 85.44±1.14(3) | 42.86±1.32(12) | 31.87±1.22(15) | 86.76±1.0(2) | 42.75±1.54(13) |
| breastw | 96.12±0.47(10) | 95.78±0.45(15.5) | 94.09±1.69(22) | 96.93±0.34(2) | 90.05±3.37(25) | 95.66±0.34(19) | 56.81±4.58(31) | 96.41±0.34(8) | 94.63±0.53(21) | 91.85±0.77(24) | 95.67±0.47(18) | 83.52±11.1(28) | 48.27±26.55(32) | 96.22±0.43(9) | 95.97±0.31(11) | 96.48±0.0(7) | 92.5±0.79(23) |
| campaign | 48.4±0.0(14) | 48.8±0.0(13) | 42.11±2.88(21) | 47.91±0.0(17) | 40.92±4.2(22) | 22.62±9.05(31) | 27.11±0.0(30) | 49.27±0.0(12) | 48.38±0.0(15) | 37.89±12.9(25) | 30.74±5.47(29) | 34.13±3.43(27) | 0.0±0.0(32) | 50.93±0.0(5) | 51.1±0.0(2) | 50.07±0.49(8) | 31.35±0.44(28) |
| cardio | 76.14±0.0(1.5) | 76.14±0.0(1.5) | 59.77±1.94(30) | 56.25±0.0(24) | 58.75±3.47(22) | 74.89±0.93(3) | 27.27±0.0(32) | 70.45±0.06(15) | 73.86±0.0(4) | 70.11±0.59(5) | 43.41±3.82(12) | 53.07±6.55(28) | 46.93±23.41(28) | 62.95±0.0(14.5) | 65.0±0.0(11) | 53.68±2.83(25) | 34.91±0.09(31) |
| cardiotocography | 61.59±0.0(2.5) | 61.59±0.0(2.5) | 49.4±6.53(11) | 41.42±0.0(22) | 46.35±9.46(20.5) | 22.62±9.05(31) | 31.33±0.0(32) | 48.28±0.0(15.5) | 62.88±0.0(1) | 37.08±5.46(26) | 55.11±7.7(8) | 52.32±9.99(9) | 33.95±12.37(29) | 47.6±0.0(18) | 48.76±0.0(12) | 36.88±1.27(27) | 48.23±0.39(17) |
| celeba | 27.0±0.41(3) | 27.17±0.0(49)(2) | 17.91±7.49(12) | 22.68±0.93(9) | 11.0±7.59(23) | 3.98±2.98(30) | 10.79±1.42(24) | 22.81±0.78(7) | 22.78±0.0(14.5) | 38.41±4.19(29) | 13.26±8.39(21) | 14.19±5.61(19) | 8.62±0.84(26) | 17.87±0.57(13) | 19.11±0.61(10.5) | 16.84±0.88(17) | 10.12±1.04(25) |
| census | 20.76±0.0(27) | 20.82±0.33(9) | 13.86±2.4(22) | 10.77±0.62(24) | 18.27±3.91(14) | 4.96±2.13(29) | 14.44±1.82(20) | 0.0±0.0(31.5) | 0.0±0.0(31.5) | 19.28±1.44(13) | 14.15±8.41(21) | 14.46±2.49(19) | 15.55±1.44(18) | 21.62±0.14(6) | 21.55±0.24(7) | 24.76±0.5(2) | 10.31±0.52(26) |
| cover | 16.21±1.68(22) | 16.24±1.53(21) | 2.59±2.48(30) | 10.75±1.26(26) | 25.69±34.37(16) | 4.29±4.13(32) | 1.19±0.85(31) | 18.82±0.81(20) | 51.56±0.0(29) | 3.43±3.44(29) | 24.19±12.26(19) | 12.16±11.56(24) | 41.87±7.55(12) | 61.65±2.14(10) | 52.19±1.92(11) | 36.61±2.59(15) | 38.39±0.62(14) |
| donors | 37.75±0.93(22) | 37.3±1.8(24) | 47.84±14.58(15) | 24.36±3.68(28) | 18.98±16.47(31) | 4.29±4.13(32) | 37.46±5.83(23) | 0.0±0.0(31.5) | 44.6±1.04(17) | 41.44±30.4(19) | 21.04±27.81(30) | 21.94±14.54(29) | 29.37±27.54(26) | 94.05±0.6(4) | 86.6±0.98(7) | 78.24±2.61(9) | 47.58±4.61(16) |
| fault | 55.22±0.08(22) | 55.27±0.0(21) | 57.65±4.55(6) | 53.64±0.0(25.5) | 56.76±4.09(10) | 55.96±0.68(17) | 50.82±0.0(30) | 50.8±0.0(32) | 51.56±0.0(29) | 54.92±1.38(24) | 51.59±1.84(28) | 20.88±22.32(30) | 56.74±4.72(11) | 56.85±0.0(9) | 56.83±0.0(8) | 59.13±1.36(4) | 72.0±0.47(1) |
| fraud | 34.46±3.22(25) | 33.26±1.7(27) | 19.56±11.2(22) | 41.51±4.73(22) | 61.17±1.07(7) | 20.15±11.0(21) | 4.59±3.73(31) | 46.10±3.48(16) | 37.81±2.09(23) | 58.11±14.77(9) | 45.06±11.59(19) | 20.88±22.32(30) | 0.0±0.0(32) | 47.78±3.53(14) | 45.61±3.48(17) | 62.3±4.91(5) | 30.48±4.66(28) |
| glass | 18.03±5.11(24) | 15.77±8.65(27.5) | 19.56±11.2(22) | 27.68±10.39(12) | 90.05±3.37(25) | 24.83±10.99(14) | 60.41±4.55(5) | 53.78±0.81(25) | 15.77±8.65(27.5) | 45.39±22.67(11) | 14.6±4.71(31) | 47.51±7.54(27) | 15.48±13.14(29) | 29.42±62.61(11) | 21.00±5.16(19) | 70.87±3.12(4) | 70.84±3.4(15) |
| hepatitis | 60.51±6.9(21) | 60.56±7.59(20) | 79.75±3.37(13) | 58.08±1.24(22) | 65.51±0.66(19) | 57.86±7.77(23) | 81.14±4.43(12) | 2.16±1.15(28) | 37.6±3.88(31) | 93.82±1.29(7) | 46.76±9.9(28) | 29.29±17.28(32) | 84.54±2.23(10) | 66.15±4.71(18) | 98.65±2.11(5) | 97.23±2.45(3) | 24.79±2.88(22) |
| http | 91.94±1.25(11) | 92.71±1.43(10) | 14.43±12.31(25) | 3.64±3.65(27) | 99.51±4.78(10) | 56.39±17.61(18) | 14.18±12.07(26) | 70.15±1.29(29) | 25.0±2.46(21) | 25.0±2.46(21) | 1.05±0.96(30) | 48.95±34.37(19) | 0.0±0.0(31.5) | 88.73±2.54(14) | 95.32±0.75(8) | 97.23±2.45(3) | 24.79±2.88(22) |
| imdb | 5.6±0.0(23.5) | 5.6±0.0(23.5) | 9.24±1.45(8) | 6.4±0.0(18.5) | 6.96±0.35(14.5) | 5.64±0.65(22) | 11.44±0.0(77.1) | 6.6±0.0(16) | 5.0±0.0(30) | 9.96±2.71(7) | 7.04±1.08(13) | 8.92±1.37(9) | 4.4±6.03(32) | 5.28±0.0(27) | 5.24±0.0(28) | 10.19±0.23(5) | 4.47±0.43(31) |
| internetads | 45.65±0.0(26.5) | 45.65±0.0(26.5) | 55.82±1.78(5) | 27.17±0.0(31) | 52.72±0.54(14) | 46.14±0.52(19) | 31.14±0.52(30) | 50.0±0.0(14.5) | 50.0±0.0(14.5) | 54.29±5.92(10) | 41.36±2.56(24) | 31.85±5.57(29) | 38.42±6.06(26) | 49.57±0.0(16) | 46.68±0.0(17) | 55.68±0.94(6) | 56.5±0.87(3) |
| ionosphere | 79.82±2.6(24) | 78.99±2.57(26) | 90.83±1.6(10) | 69.49±1.68(29) | 86.17±2.5(20) | 83.42±5.88(23) | 85.86±1.47(21) | 60.53±2.51(28) | 64.5±2.11(31) | 93.07±1.23(3) | 77.34±4.54(27) | 69.33±4.23(30) | 60.19±21.57(32) | 91.12±1.24(9) | 87.86±1.86(17) | 93.38±2.21(2) | 79.23±4.14(25) |
| landsat | 38.7±4.50(30) | 33.9±8.0(28) | 35.77±1.67(25) | 52.14±0.0(5) | 35.12±1.16(26) | 32.96±1.19(30) | 34.43±0.0(27) | 33.8±0.0(29) | 30.76±0.0(31) | 36.89±4.89(24) | 36.97±4.47(24) | 40.8±3.0(18) | 40.8±3.0(18) | 48.79±0.0(10) | 46.11±0.0(13) | 50.83±0.68(8) | 41.47±2.39(16) |
| letter | 1.0±0.0(27.5) | 1.0±0.0(27.5) | 3.8±2.39(13.5) | 6.0±0.0(8) | 1.2±0.84(22) | 1.2±0.84(22) | 28.0±0.0(1) | 4.0±0.0(11.5) | 9.0±0.0(4) | 5.0±1.58(10) | 1.2±0.45(22) | 8.6±4.39(5.5) | 13.6±8.62(2) | 1.0±0.0(27.5) | 1.0±0.0(27.5) | 5.84±1.11(9) | 1.72±0.37(19) |
| lymphography | 92.96±5.55(14) | 90.86±4.08(17) | 91.07±0.32(16) | 88.49±3.88(20) | 83.31±1.43(25) | 93.13±5.46(13) | 94.47±7.77(11) | 86.67±6.42(22) | 86.47±4.27(21) | 89.82±9.49(18) | 24.05±20.79(32) | 26.15±35.82(31) | 26.15±35.82(31) | 94.11±5.93(12) | 91.77±3.69(15) | 100.0±0.0(2.5) | 58.34±6.75(30) |
| magic.gamma | 65.25±0.0(24) | 65.19±0.0(25) | 67.82±1.62(20) | 67.19±0.0(21) | 56.95±1.27(32) | 56.95±1.27(32) | 62.02±0.0(28) | 2.86±0.0(26) | 59.72±0.0(30) | 59.88±0.89(29) | 65.48±1.31(23) | 57.35±2.82(31) | 72.61±0.67(13) | 75.61±0.0(11) | 74.48±0.0(12) | 69.34±1.08(16) | 85.36±1.69(1) |
| mammography | 45.0±0.0(6) | 44.62±0.0(7) | 22.0±0.88(28) | 16.92±0.0(30) | 36.85±2.78(18) | 35.62±5.7(19) | 16.85±0.42(31) | 52.60±0.0(4.5) | 53.08±0.0(11) | 31.62±0.10.21(22) | 47.92±2.01(14) | 26.85±2.0(25) | 32.69±2.68(21) | 43.15±0.0(8) | 43.0±0.0(9) | 29.94±2.59(23) | 28.42±1.4(24) |
| mnist | 63.86±0.0(12.5) | 63.86±0.0(12.5) | 52.14±4.56(21) | 24.14±0.0(29) | 41.83±5.0(25) | 63.91±1.13(11) | 21.29±0.0(30) | 87.63±0.0(24) | 92.78±0.0(19) | 43.31±11.21(24) | 33.8±7.68(27) | 44.66±8.53(23) | 57.29±2.09(17) | 71.37±0.0(4) | 70.49±0.0(5) | 52.79±0.77(19) | 34.87±1.48(26) |
| musk | 100.0±0.0(8.5) | 100.0±0.0(8.5) | 35.05±28.17(31) | 100.0±0.0(8.5) | 4.93±4.07(19) | 100.0±0.0(8.5) | 0.67±0.0(23.5) | 0.0±0.0(30) | 92.78±0.0(19) | 0.0±0.0(30) | 90.72±5.41(21) | 70.72±21.53(26) | 12.16±17.44(32) | 16.0±0.0(14) | 8.93±0.0(17) | 46.69±2.0(4) | 60.0±0.0(28) |
| optdigits | 0.67±0.0(23.5) | 0.67±0.0(23.5) | 54.35±2.08(21) | 12.35±0.0(32) | 39.8±1.58(26) | 50.24±1.07(23) | 61.96±0.0(9) | 0.0±0.0(30) | 0.0±0.0(30) | 54.71±3.06(19) | 1.07±1.67(21) | 0.27±0.46(27) | 58.43±1.73(1) | 59.02±0.0(13.5) | 58.16±0.0(15) | 62.93±0.29(7) | 6.11±1.52(18) |
| pageblocks | 46.86±0.0(26.5) | 46.86±0.0(26.5) | 14.23±6.24(30) | 41.03±0.0(23) | 15.51±20.48(28) | 41.54±3.49(22) | 26.92±0.0(26) | 35.26±0.0(24) | 49.22±0.0(25) | 54.71±3.06(19) | 42.05±6.31(21) | 13.97±16.56(31) | 19.23±5.21(27) | 58.16±0.0(15) | 76.41±0.0(6) | 62.93±0.29(7) | 49.33±0.37(24) |
| pendigits | 44.23±0.0(18.5) | 44.23±0.0(18.5) | 67.92±2.87(17) | 69.6±2.76(11) | 58.93±5.42(28) | 59.19±11.77(25) | 54.47±4.52(30) | 63.16±2.25(23) | 58.54±1.89(28) | 55.95±2.36(29) | 61.14±5.31(24) | 54.05±5.38(31) | 50.01±13.57(32) | 72.5±1.97(4) | 71.3±2.0(6) | 70.45±1.78(9.5) | 65.74±2.1(20) |
| pima | 70.45±2.76(9.5) | 69.3±2.92(13) | 66.01±1.44(23) | 75.98±0.0(4) | 69.08±0.81(16) | 63.58±0.48(28) | 63.85±0.0(27) | 60.71±0.0(31) | 56.03±0.0(32) | 67.76±2.88(18) | 65.01±1.17(25) | 65.13±1.63(24) | 67.52±3.93(19) | 69.52±0.0(17) | 71.3±2.0(6) | 69.69±7.5(17) | 81.04±1.96(1) |
| satellite | 88.17±0.77(15) | 87.32±0.0(16) | 62.25±5.49(29) | 83.1±0.0(18) | 76.62±15.83(24) | 90.7±5.07(6) | 0.0±0.0(32) | 80.28±0.0(20) | 78.87±0.0(21) | 73.24±6.08(26) | 88.73±1.0(11.5) | 50.42±33.88(30) | 76.34±13.45(25) | 91.27±0.0(5) | 92.11±0.0(3) | 69.69±7.51(27) | 42.31±7.14(31) |
| satimage-2 | 95.78±0.0(20.5) | 95.78±0.0(20.5) | 46.15±11.43(30) | 95.07±0.0(22) | 91.15±8.04(24) | 56.26±30.2(28) | 97.86±0.0(14) | 96.07±0.0(19) | 91.8±0.0(23) | 98.11±0.09(11.5) | 53.11±44.55(29) | 67.9±24.09(27) | 98.19±0.0(10) | 98.19±0.0(10) | 98.11±0.0(11.5) | 98.31±0.0(24.5) | 77.31±0.58(26) |
| shuttle | 44.73±0.84(26) | 37.91±0.84(28) | 78.59±11.23(8) | 58.3±0.0(29) | 31.3±2.79(29) | 72.67±0.95(17) | 72.67±0.95(17) | 0.2±0.58(31) | 22.0±0.36(30) | 43.26±2.59(27) | 55.77±10.14(22) | 55.71±28.67(23) | 78.44±1.35(10) | 93.89±0.13(4) | 93.89±0.13(4) | 56.43±0.55(21) | 76.25±0.32(13) |
| skin | 69.59±4.42(2.5) | 69.5±4.43(7.5) | 0.0±0.0(29) | 0.0±0.0(29) | 0.0±0.0(29) | 48.56±12.43(17) | 68.05±5.72(12) | 0.0±0.0(29) | 69.51±4.43(7.5) | 34.0±0.23(28) | 8.76±5.05(24) | 26.32±34.3(22) | 13.75±80.75(23) | 67.73±4.99(13) | 69.50±3.35(2.5) | 50.51±3.54(16) | 43.11±3.67(18) |
| smtp | 74.49±0.03(15) | 78.5±0.0(14) | 77.62±3.62(18) | 74.93±0.0(20) | 79.33±1.0(9) | 78.81±0.63(11) | 50.98±0.0(23) | 0.0±0.0(29) | 60.51±0.0(29) | 69.55±3.79(28) | 71.03±5.1(27) | 68.45±3.55(30) | 73.94±3.0(23) | 80.13±0.0(6) | 79.48±0.0(8) | 77.87±0.46(16) | 83.25±0.0(71) |
| spambase | 3.28±0.0(13) | 3.28±0.0(14) | 1.64±1.16(30.5) | 1.61±1.09(9) | 1.97±1.37(28.5) | 2.95±1.37(20) | 4.59±1.8(4) | 3.28±0.0(13) | 3.28±0.0(13) | 1.31±1.37(32) | 2.3±1.87(26) | 3.28±1.16(13) | 2.95±2.14(20) | 80.13±0.0(6) | 3.28±0.0(13) | 3.48±0.92(8) | 2.03±0.25(27) |
| speech | 61.44±6.81(17) | 57.86±9.09(20) | 51.82±17.69(23) | 57.55±4.15(21) | 32.27±14.5(30) | 52.72±15.8(22) | 43.72±9.88(28) | 67.23±4.51(8) | 49.48±5.09(26) | 37.07±0.74(29) | 60.16±12.64(18) | 32.5±29.05(30) | 28.03±26.42(32) | 75.14±4.45(6) | 66.49±7.15(9) | 62.0±6.23(16) | 50.23±0.89(25) |
| stamps | 74.19±0.0(10.5) | 74.19±0.0(10.5) | 69.89±3.72(17) | 77.42±0.0(2) | 52.9±20.51(27) | 74.19±1.32(10.5) | 51.61±0.0(29) | 30.11±0.0(32) | 59.14±0.0(23.5) | 65.59±8.6(19) | 70.75±5.18(16) | 65.38±11.86(20) | 69.03±3.68(18) | 74.62±0.0(8) | 73.55±0.0(13) | 57.68±1.74(25) | 60.69±0.57(22) |
| thyroid | 14.07±1.0(26) | 13.93±1.31(28) | 68.31±14.38(21) | 9.53±5.46(30) | 19.76±4.38(19) | 18.25±10.64(20) | 56.13±3.76(26) | 28±0.63(32) | 12.61±2.46(29) | 16.71±5.9(33) | 8.42±4.46(31) | 21.19±13.5(17) | 16.99±21.41(22) | 88.37±1.61(10) | 13.99±2.95(27) | 44.91±2.82(5) | 28.18±7.79(11) |
| vertebral | 12.0±0.0(26.5) | 12.0±0.0(26.5) | 14.4±5.18(24) | 8.0±0.0(29) | 22.81±5.54(19) | 23.6±2.19(18) | 50.0±0.0(1) | 6.0±0.0(30) | 22.0±0.0(20) | 20.8±4.15(21) | 10.4±3.29(28) | 5.6±6.69(31) | 16.09±21.41(22) | 56.11±1.23(11) | 41.99±2.22(19) | 25.36±2.94(15) | 25.2±1.36(16) |
| waveform | 8.0±0.0(25) | 9.0±0.0(22) | 28.6±4.39(3) | 8.0±0.0(25) | 12.0±7.21(17.5) | 9.8±2.39(20) | 5.0±0.0(30) | 9.0±0.0(22) | 7.0±0.0(29) | 14.6±3.58(14) | 7.4±3.79(27) | 4.6±1.67(31) | 26.6±6.11(10) | 28.8±0.0(9) | 27.0±0.0(7) | 33.2±6.59(1) | 7.04±0.23(28) |
| ylc | 88.42±3.21(8) | 87.28±5.1(11) | 55.68±15.83(25) | 80.41±7.38(19) | 64.11±13.61(23) | 86.45±4.97(12) | 71.77±11.9(21) | 82.4±5.65(16.5) | 51.13±9.94(28) | 54.17±11.28(26) | 68.8±16.05(22) | 46.16±26.61(27) | 26.01±27.57(30) | 86.1±2.43(14) | 89.16±3.8(6) | 89.16±3.8(6) | 36.84±4.92(28) |
| wdbc | 78.69±6.98(18) | 78.78±1.28(16) | 75.08±11.73(20) | 67.95±6.63(24) | 75.89±10.72(20) | 75.82±5.69(19) | 3.35±5.65(32) | 79.55±3.91(12) | 51.13±9.94(28) | 83.34±7.3(9) | 52.65±20.70(27) | 32.5±29.05(30) | 8.7±19.44(31) | 88.75±2.8(7) | 78.79±5.12(15) | 78.86±2.34(14) | 47.39±3.91(29) |
| wilt | 1.95±0.0(22) | 1.56±0.0(24.5) | 3.27±4.59(18) | 0.0±0.0(32) | 6.15±5.11(15) | 12.45±2.0(10) | 10.51±0.0(12) | 1.56±0.0(24.5) | 4.28±0.0(17) | 69.81±9.29(20) | 0.86±0.58(30) | 5.68±3.33(16) | 1.48±1.39(26) | 1.25±0.0(27) | 80.99±2.84(13) | 26.35±2.35(5) | 12.36±2.38(11) |
| wine | 67.96±3.31(22) | 66.01±5.57(24) | 68.31±14.38(21) | 77.67±8.64(16) | 42.72±36.22(30) | 65.52±5.99(25) | 56.13±3.76(26) | 62.92±4.9(26.5) | 39.27±8.98(31) | 70.22±6.11(5) | 60.38±3.87(29.5) | 33.18±3.98(31) | 12.78±16.95(32) | 56.11±1.23(11) | 76.55±1.28(3) | 97.76±2.03(17) | 66.58±8.38(23) |
| wpbc | 36.48±3.41(25) | 33.62±3.01(29) | 42.84±4.39(18) | 44.59±3.59(16) | 45.61±5.66(15) | 34.22±3.42(28) | 66.34±4.21(16) | 42.6±0.0(32) | 36.21±1.91(26) | 49.47±5.66(11) | 37.28±3.79(23) | 33.18±3.98(31) | 31.9±5.22(32) | 41.99±2.22(19) | 81.35±3.33(3) | 48.41±0.87(15) | 53.91±5.76(12) |
| yeast | 44.26±0.18(29) | 43.39±0.0(31) | 47.77±2.46(17) | 44.38±0.0(28) | 13.86±1.31(22) | 53.21±2.43(2) | 43.79±0.0(30) | 16.0±0.0(33) | 46.35±0.0(24) | 10.4±1.77(25) | 13.8±2.17(19) | 8.2±2.18(28) | 6.88±6.41(32) | 46.71±0.0(23) | 17.26±0.0(7) | 7.7±0.2(30) | 58.40±0.44(1) |
| yelp | 16.2±0.0(19) | 16.2±0.0(19) | 11.12±1.38(24) | 16.2±0.0(19) | 43.79±0.0(30) | 15.36±0.38(15) | 8.16±0.33(29) | 16.0±0.0(13) | 13.4±0.0(20.5) | 10.4±1.77(25) | 13.8±2.17(19) | 8.2±2.18(28) | 6.88±6.41(32) | 17.72±0.0(5) | 17.26±0.0(7) | 49.22±0.12(4) | 9.11±1.45(26) |
| MNIST-C | 41.11±0.0(17.5) | 41.11±0.0(17.5) | 34.67±1.69(22) | 23.24±0.0(29) | 44.51±1.76(12) | 42.11±0.45(16) | 12.01±5.39(30) | 82.4±5.65(16.5) | 0.0±0.0(31.5) | 44.35±3.0(24) | 34.48±5.52(23) | 25.17±8.13(28) | 30.54±9.37(25) | 45.8±0.0(19) | 17.36±0.0(7) | 49.22±0.12(4) | 29.35±0.11(26) |
| FashionMNIST | 55.62±0.0(16.5) | 55.62±0.0(16.5) | 48.05±1.55(21) | 33.65±0.0(27) | 6.15±5.11(15) | 22.88±8.89(9.5) | 18.37±2.0(30) | 0.0±0.0(31.5) | 0.0±0.0(31.5) | 48.9±3.89(20) | 48.9±3.89(20) | 33.43±6.11(28) | 32.21±12.83(29) | 56.76±0.0(10) | 57.97±0.0(11) | 41.42±0.13(4) | 35.54±0.1(26) |
| CIFAR10 | 22.02±0.0(14.5) | 22.62±0.0(14.5) | 17.6±1.31(23) | 14.9±0.0(27) | 24.27±1.43(4) | 22.88±8.89(9.5) | 10.33±1.42(30) | 9.7±0.0(32) | 10.19±0.0(31) | 16.46±1.96(25) | 20.42±3.07(17) | 13.35±2.59(29) | 14.97±2.42(26) | 22.88±0.0(13) | 22.69±0.0(13) | 19.02±0.19(20) | 14.4±0.12(28) |
| SVHN | 18.27±0.0(16.5) | 18.27±0.0(16.5) | 17.53±0.97(20) | 13.1±0.0(27) | 19.19±0.89(6) | 18.49±0.42(14) | 11.37±0.0(78(30) | 62.92±4.9(26.5) | 15.2±1.66(24) | 15.2±1.66(24) | 15.33±2.34(23) | 12.99±2.1(28) | 13.18±1.68(26) | 18.89±0.0(10) | 18.65±0.0(12.5) | 80.65±0.56(3) | 11.78±0.08(29) |
| MVTec-AD | 10.31±1.2.64(25) | 63.41±2.78(24) | 60.38±33.19(29.5) | 62.65±2.52(28) | 67.28±2.76(14) | 66.34±4.21(16) | 66.34±4.21(16) | 62.92±4.9(26.5) | 76.78±3.79(5) | 63.64±3.63(15) | 60.38±3.87(29.5) | 9.61±2.52(30) | 51.15±10.03(32) | 69.62±0.83(10) | 65.23±0.73(17) | 14.26±0.73(13) | 71.97±0.92(9) |
| 20news | 10.79±1.83(21) | 10.5±1.24(25) | 9.78±1.2(31) | 9.38±0.78(31) | 14.54±2.93(11) | 10.7±1.03(23) | 10.12±2.67(28) | 10.89±0.77(20) | 10.76±1.05(22) | 13.64±3.63(15) | 9.95±1.97(29) | 9.61±2.52(30) | 12.55±3.73(18) | 12.82±1.49(17) | 14.26±0.73(13) | 14.26±0.73(13) | 10.44±0.45(27) |
| agnews | 12.25±0.0(23.5) | 12.25±0.0(23.5) | 9.78±1.2(31) | 11.45±0.0(26.5) | 14.44±0.8(12) | 13.15±0.69(18) | 10.71±1.01(29) | 11.55±0.0(25) | 11.45±0.0(26.5) | 10.66±3.57(30) | 13.52±1.56(16) | 10.77±5.43(28) | 3.72±4.16(32) | 19.4±0.0(7) | 18.14±0.0(8) | 14.25±0.08(13) | 12.53±0.63(20) |

| Rank(avg) | 16.816 | 17.825 | 19.316 | 19.783 | 18.693 | 18.439 | 21.395 | 21.807 | 22.404 | 18.991 | 21.798 | 23.649 | 22.947 | 10.509 | 12.105 | 10.947 | 19.298 |
| All | 0.997 | 0.998 | 1.000 | 1.000 | 1.000 | 1.000 | 1.000 | 1.000 | 1.000 | 1.000 | 1.000 | 1.000 | 1.000 | 0.371 | 0.731 | 0.403 | 1.000 |
| d ≤ 20 | 1.000 | 1.000 | 1.000 | 1.000 | 1.000 | 1.000 | 1.000 | 1.000 | 1.000 | 1.000 | 1.000 | 1.000 | 1.000 | 0.924 | 0.588 | 0.995 | 1.000 |
| d ≤ 50 | 1.000 | 1.000 | 1.000 | 1.000 | 1.000 | 1.000 | 1.000 | 1.000 | 1.000 | 1.000 | 1.000 | 1.000 | 1.000 | 0.949 | 0.993 | 0.961 | 1.000 |

| Rank(avg) | 17.539 | 18.456 | 20.14 | 20.579 | 19.465 | 19.132 | 22.167 | 22.675 | 23.237 | 19.763 | 22.684 | 24.544 | 23.798 | 11.053 | 12.649 | 11.474 | 20.088 |
| All | 0.999 | 0.999 | 1.000 | 1.000 | 1.000 | 1.000 | 1.000 | 1.000 | 1.000 | 1.000 | 1.000 | 1.000 | 1.000 | 0.670 | 0.897 | 0.757 | 1.000 |
| d ≤ 100 | 0.999 | 1.000 | 1.000 | 1.000 | 1.000 | 1.000 | 1.000 | 1.000 | 1.000 | 1.000 | 1.000 | 1.000 | 1.000 | 0.886 | 0.953 | 0.939 | 1.000 |
| d ≤ 500 | 0.999 | 1.000 | 1.000 | 1.000 | 1.000 | 1.000 | 1.000 | 1.000 | 1.000 | 1.000 | 1.000 | 1.000 | 1.000 | 0.849 | 0.947 | 0.880 | 1.000 |

# J Benchmark OD Datasets

Table 20: Description of all datasets in ADBench Livernoche et al. (2024). Datasets in blue are image and text datasets that are vectorized through pretrained encoders. We refer to the original paper for details.

| Dataset Name | # Samples | # Features | # Anomaly | % Anomaly | Category |
|---|---|---|---|---|---|
| ALOI | 49534 | 27 | 1508 | 3.04 | Image |
| annthyroid | 7200 | 6 | 534 | 7.42 | Healthcare |
| backdoor | 95329 | 196 | 2329 | 2.44 | Network |
| breastw | 683 | 9 | 239 | 34.99 | Healthcare |
| campaign | 41188 | 62 | 4640 | 11.27 | Finance |
| cardio | 1831 | 21 | 176 | 9.61 | Healthcare |
| Cardiotocography | 2114 | 21 | 466 | 22.04 | Healthcare |
| celeba | 202599 | 39 | 4547 | 2.24 | Image |
| census | 299285 | 500 | 18568 | 6.20 | Sociology |
| cover | 286048 | 10 | 2747 | 0.96 | Botany |
| donors | 619326 | 10 | 36710 | 5.93 | Sociology |
| fault | 1941 | 27 | 673 | 34.67 | Physical |
| fraud | 284807 | 29 | 492 | 0.17 | Finance |
| glass | 214 | 7 | 9 | 4.21 | Forensic |
| Hepatitis | 80 | 19 | 13 | 16.25 | Healthcare |
| http | 567498 | 3 | 2211 | 0.39 | Web |
| InternetAds | 1966 | 1555 | 368 | 18.72 | Image |
| Ionosphere | 351 | 32 | 126 | 35.90 | Oryctognosy |
| landsat | 6435 | 36 | 1333 | 20.71 | Astronautics |
| letter | 1600 | 32 | 100 | 6.25 | Image |
| Lymphography | 148 | 18 | 6 | 4.05 | Healthcare |
| magic.gamma | 19020 | 10 | 6688 | 35.16 | Physical |
| mammography | 11183 | 6 | 260 | 2.32 | Healthcare |
| mnist | 7603 | 100 | 700 | 9.21 | Image |
| musk | 3062 | 166 | 97 | 3.17 | Chemistry |
| optdigits | 5216 | 64 | 150 | 2.88 | Image |
| PageBlocks | 5393 | 10 | 510 | 9.46 | Document |
| pendigits | 6870 | 16 | 156 | 2.27 | Image |
| Pima | 768 | 8 | 268 | 34.90 | Healthcare |
| satellite | 6435 | 36 | 2036 | 31.64 | Astronautics |
| satimage-2 | 5803 | 36 | 71 | 1.22 | Astronautics |
| shuttle | 49097 | 9 | 3511 | 7.15 | Astronautics |
| skin | 245057 | 3 | 50859 | 20.75 | Image |
| smtp | 95156 | 3 | 30 | 0.03 | Web |
| SpamBase | 4207 | 57 | 1679 | 39.91 | Document |
| speech | 3686 | 400 | 61 | 1.65 | Linguistics |
| Stamps | 340 | 9 | 31 | 9.12 | Document |
| thyroid | 3772 | 6 | 93 | 2.47 | Healthcare |
| vertebral | 240 | 6 | 30 | 12.50 | Biology |
| vowels | 1456 | 12 | 50 | 3.43 | Linguistics |
| Waveform | 3443 | 21 | 100 | 2.90 | Physics |
| WBC | 223 | 9 | 10 | 4.48 | Healthcare |
| WDBC | 367 | 30 | 10 | 2.72 | Healthcare |
| Wilt | 4819 | 5 | 257 | 5.33 | Botany |
| wine | 129 | 13 | 10 | 7.75 | Chemistry |
| WPBC | 198 | 33 | 47 | 23.74 | Healthcare |
| yeast | 1484 | 8 | 507 | 34.16 | Biology |
| CIFAR10 | 5263 | 512 | 263 | 5.00 | Image |
| FashionMNIST | 6315 | 512 | 315 | 5.00 | Image |
| MNIST-C | 10000 | 512 | 500 | 5.00 | Image |
| MVTec-AD | 5354 | 512 | 1258 | 23.50 | Image |
| SVHN | 5208 | 512 | 260 | 5.00 | Image |
| Agnews | 10000 | 768 | 500 | 5.00 | NLP |
| Amazon | 10000 | 768 | 500 | 5.00 | NLP |
| Imdb | 10000 | 768 | 500 | 5.00 | NLP |
| Yelp | 10000 | 768 | 500 | 5.00 | NLP |
| 20newsgroups | 11905 | 768 | 591 | 4.96 | NLP |

## K  Differences to Prior Work on PFNs for Tabular Data

There exist applications of PFNs (originally developed by Müller et al. (2022)) that pre-date our proposed FoMo-0D, namely, TabPFN (Hollmann et al., 2023) for supervised classification, LC-PFN (Adriaensen et al., 2024) for learning curve extrapolation, PFN4BO (Müller et al., 2023) for Bayesian optimization, and ForecastPFN (Dooley et al., 2023) for time series forecasting.

Here we highlight the differences of our proposed FoMo-0D from these existing PFNs.

1. **First PFN4OD:** We employ prior-data fitted networks (PFNs) for outlier detection (OD) for the first time.

2. **First large-scale pretrained OD model:** FoMo-0D is the first model for zero-shot OD that is pretrained at large scale on a large collection of (synthetic) datasets, due to the minuscule nature of existing real-world OD benchmark datasets.

3. **New data prior:** Thanks to PFN's reliance on synthetically generated datasets, we establish a new data prior for OD, specifically for outlier synthesis.

4. **Data transformation for scale:** While drawing samples from a data prior may be relatively fast, pretraining a large foundation model requires many such draws for every step of each epoch. To speed up data synthesis on-the-fly, we are the first to leverage a linear transformation.

5. **Router-based attention for scale:** PFNs ingest the entire training dataset as context for in-context learning at inference time. To accommodate larger datasets at both training (for better generalization) and inference (for large-scale real-world datasets), we leveraged a "bottleneck" architecture for scalable self-attention, and in turn, larger context size.

## L  Discussion

**Summary:**  We introduced FoMo-0D, **the first foundation model for outlier detection** (OD) on tabular data. FoMo-0D is a prior-data fitted network (PFN), pretrained on a large number of *synthetic* datasets generated from a new data prior for OD, which can infer the posterior predictive distribution for test points in a new dataset in a **zero-shot** fashion where the training data is input as context, capitalizing on *in-context learning*.

Zero-shot OD implies **no additional OD model training or model selection**, given a new OD task. That is a revolution for OD (!), for which algorithm and hyperparameter selection are notoriously-hard *without any labeled data*, and also computationally taxing especially for today's modern deep OD models with numerous parameters *and* a long list of hyperparameters. What is more, FoMo-0D provides **extremely fast inference** thanks to a mere *single forward pass*, making it amenable for OD on data streams.

Building on the PFN paradigm (Müller et al., 2022), FoMo-0D breaks new ground not only conceptually by abolishing the burden of model training and selection, but also empirically: Against **26** different (both classical and modern) baselines on **57** public benchmark datasets from diverse domains, FoMo-0D performs on par with the top 2*nd* baseline, while significantly outperforming the majority of the baselines. Without the need to train any, let alone multiple models for HP tuning, FoMo-0D takes a mere **7.7 ms** per test sample for inference only.

**Limitations and Future Directions:**  FoMo-0D employs a simple straightforward data prior based on GMMs. While it is remarkable to see how far one can go with synthetic data from such a simple prior, future work can design more comprehensive data priors, inclusive of discrete features as well as other possible outlier types. We have also pretrained FoMo-0D solely on synthetic datasets, while future work can augment both synthetic and real-world datasets for pretraining.

Besides the lack of massive real-world datasets for tabular OD, a motivation for a data prior to pretrain purely on synthetic datasets comes from neural scaling laws (Kaplan et al., 2020; Zhai et al., 2022). Interestingly, the scaling laws for large Transformer models have shown that their generalization error tends to drop as a power law with the amount of training data (also, with number of parameters and amount of compute), but

the power law exponent is very small—suggesting that acquiring more colossal real-world datasets would be a slow, if not expensive approach to advancing ML/AI. Others have proposed ways to subset-select smaller, non-redundant "foundation datasets" (Sorscher et al., 2022; Paul et al., 2021), and emphasized the importance of task/dataset diversity in pretraining (Raventós et al., 2024). Arguably, synthetic data from a complex and diverse data prior is a potential gateway to obtaining non-redundant and diverse datasets for pretraining large foundation models like `FoMo-0D`. On the other hand, designing such a data prior requires a level of domain/prior knowledge.

Another improvement could be scaling up to even larger context (i.e. dataset) size and dimensionality. While `FoMo-0D` generalizes beyond pretrained context sizes and dimensionality, it is limited to and performs particularly well on downstream datasets of similar nature as our experiments showed. A promising direction for size generalization is using PFNs as extremely fast ensemble components at inference; since "*PFNs are quick enough to be used as ensemble members. The size constraints could therefore be overcome by boosting and bagging techniques*" (Nagler, 2023).

Further, our work focused on unsupervised OD with clean/inlier-only training data. Future work can study the unsupervised OD setting and pretraining with mixed/"contaminated" data in the transductive setting, where the unlabeled test data is the same as training data. In addition, we performed offline evaluation of `FoMo-0D` on static datasets, while its fast inference lends itself to streaming OD, which future work can explore. Technically, both extensions (unsupervised OD and streaming OD) are straightforward from the implementation perspective.

Our current work is limited to OD for tabular (or point-cloud) data. Our ideas can be extended to other data modalities, such as image, graph, and text outliers, to comprise other domains with critical OD applications such as video surveillance, fraud detection and LLM hallucination detection. To that end, the design of novel inlier/outlier priors would be an open direction. A promising approach here could be the use of pretrained generative models to draw synthesized image/text/etc. datasets for pretraining the PFN, in place of manually-designed data priors.

Finally, our quest here has been mainly experimental. Theoretically understanding why these models work as well as they do and investigating their failure cases are important yet open questions. As empirical future work, one could systematically stress-test `FoMo-0D` using synthetically generated test datasets that contain known outlier types and inlier distributions distinct from those used during pretraining, while varying the degree of out-of-distribution characteristics in a controlled manner.

As the first foundation model for OD, `FoMo-0D` inspires many promising directions for future research that could lead to fruition for additional practical applications.

## M  Reproducibility Statement

We expect that the disruptive nature of `FoMo-0D` will trigger future innovations in the OD literature, as well as a widespread adoption by practitioners thanks to its key desirable properties. To foster future research and accessibility in practice, we make all resources (our codebase used for prior data synthesis, data transformation, and pretraining as well as our pretrained model checkpoints) publicly available at `https://github.com/A-Chicharito-S/FoMo-0D`. Full implementation details are provided in Appendix C.

