# OpenReview forum: "FoMo-0D: A Foundation Model for Zero-shot Tabular Outlier Detection"
_TMLR — Accepted by TMLR_

### Review · Reviewer_oXoC · 2025-06-06

**Summary Of Contributions:**

The paper presents a model designed to perform outlier detection on tabular datasets without the need for dataset-specific training or hyperparameter tuning. The model, FoMo-0D, is based on the PFN framework, which enables it to compute a posterior predictive distribution conditioned on any new training dataset. To achieve generalization across tasks, the model is trained on millions of synthetically generated outlier detection problems, each constructed using GMMs to mimic diverse data characteristics.

The authors position FoMo-0D as a zero-shot outlier detection system, arguing that it can be directly applied to new datasets without model selection or retraining. They refer to it as a foundation model”for OD, citing its ability to generalize across a wide variety of tabular OD tasks. The model is lightweight (around 2 million parameters) and optimized for fast inference, making it practical for real-time or resource-constrained applications.

Evaluation is performed on 57 real-world datasets from the ODDS benchmark suite, where FoMo-0D is compared against 26 existing OD baselines, including classical methods such as Isolation Forest and newer neural approaches. The results suggest that FoMo-0D is competitive with, or outperforms, these methods on most tasks, despite having never seen real-world data during training.

**Audience:**

Yes

**Broader Impact Concerns:**

The paper does not currently include a Broader Impact Statement
While FoMo-0D offers promising generalization without dataset-specific tuning, this strength may also pose risks in high-stakes domains such as healthcare, finance, or security, where overly confident deployment without proper validation could lead to harmful false positives or negatives.
Lack of domain-specific calibration or uncertainty quantification may create a false sense of reliability. In real-world settings, outlier detection often supports decision-making in sensitive applications, and there are social and ethical costs of incorrect detections.
The model is trained entirely on synthetic data, biases or blind spots inherent in the synthetic task distribution may propagate into downstream use.

**Claims And Evidence:**

Yes

**Requested Changes:**

The authors should clarify and justify their use of the term foundation model. As commonly understood, foundation models support multiple downstream tasks and learn semantically rich, transferable representations. FoMo-0D, while general across tabular OD tasks, is limited to a single task trained on synthetic data. Either stronger evidence of broad generalization should be provided, or the terminology should be revised.

The explanation of zero-shot capabilities requires expansion. In related domains, zero-shot detection relies on semantic priors or auxiliary information. Here, the synthetic GMM training process is not clearly shown to generalize to real-world anomalies. The discussion in Appendix G should be brought forward and elaborated in the main paper.

The evaluation strategy across 57 datasets may obscure domain-specific needs. OD applications differ widely in their required sensitivity and specificity. The paper should discuss how FoMo-0D performs under these different constraints and whether it can be calibrated for critical use cases.

Additionally, the writing style could be more concise. The heavy use of em dashes and verbose phrasing hinders readability and gives the impression of (too much) LLM-assisted writing. A clearer presentation would improve the paper’s accessibility.

The limitations of relying entirely on synthetic data should be discussed more openly. Not all real-world anomaly types may be covered by GMM-based training, and this may constrain generalization.

**Strengths And Weaknesses:**

Strengths:
- Claimed Zero-Shot Capability.
- The model demonstrates rapid inference times (~7.7 ms per sample), making it suitable for real-time applications.

Weaknesses:

FoMo-0D is tailored exclusively for tabular datasets. While this domain is vast, many existing methods already achieve high performance due to the relatively low complexity of such data. For instance, traditional models like Isolation Forests [2] and Local Outlier Factor [3] have been effective in various tabular OD tasks. The paper does not sufficiently justify the need for a foundation model in this context, especially when simpler models perform adequately.

The term foundation model typically refers to models trained on broad data distributions that can generalize across multiple tasks and domains, often learning semantically rich embeddings, as demonstrated by GPT for text [5] and CLIP for vision-language tasks [4]. FoMo-0D, trained solely on synthetic GMM-based data, lacks evidence of such broad generalization or semantic understanding. The paper should clarify how FoMo-0D meets the criteria of a foundation model beyond its zero-shot capability. While it is fair to call the model general-purpose within the scope of tabular OD, calling it a foundation model implies stronger and more widely applicable representation learning, which is not clearly demonstrated.

More broadly, foundation models are typically capable of supporting a wide range of downstream tasks. In contrast, FoMo-0D is limited to outlier detection, albeit across multiple datasets. If the authors intend to argue for its status as a foundation model, they should either explore applications beyond OD or make this limitation explicit and discuss its implications.

The model’s zero-shot capability is an important claim, but the explanation for how it is achieved and validated remains underdeveloped. In other fields, zero-shot OD frequently involves the use of semantic priors or auxiliary information to generalize to unseen anomaly types [6, 7]. FoMo-0D’s synthetic-data-driven generalization approach is novel, but the paper does not convincingly show that the synthetic training sufficiently covers the variability and complexity of real-world anomalies. The authors could strengthen their position by discussing generalization challenges more openly, as touched upon briefly in Appendix G, and by presenting FoMo-0D’s performance on datasets with anomaly structures that differ significantly from the GMM-generated training tasks.

Furthermore, it remains unclear whether averaging results across 57 datasets is appropriate or informative. Different OD applications, such as fraud detection, manufacturing defect identification, or medical diagnostics, differ substantially in their tolerance for false positives and false negatives. Each domain has distinct requirements for sensitivity and specificity, and the model’s ability to accommodate such domain-specific trade-offs is not addressed. Without tuning or adaptation mechanisms, it’s uncertain whether FoMo-0D’s one-size-fits-all strategy is viable for high-stakes use cases in finance, security, or healthcare.

The writing style of the manuscript tends toward verbosity, and the frequent use of em dashes (—) makes the paper harder to follow and could be an indicator for extensive use of writing support by an LLM like ChatGPT. A more concise and direct presentation would significantly improve clarity and accessibility.

References:

1.	Shen, Y., Wen, H., & Akoglu, L. (2024). Zero-shot Outlier Detection via Prior-data Fitted Networks: Model Selection Bygone!. arXiv preprint arXiv:2409.05672.

2.	Liu, F. T., Ting, K. M., & Zhou, Z. H. (2008). Isolation forest. In 2008 Eighth IEEE International Conference on Data Mining (pp. 413-422). IEEE.

3.	Breunig, M. M., Kriegel, H. P., Ng, R. T., & Sander, J. (2000). LOF: identifying density-based local outliers. In Proceedings of the 2000 ACM SIGMOD international conference on Management of data (pp. 93-104).

4.	Radford, A., Kim, J. W., Hallacy, C., et al. (2021). Learning transferable visual models from natural language supervision. In Proceedings of the 38th International Conference on Machine Learning (pp. 8748-8763).

5.	Brown, T. B., Mann, B., Ryder, N., et al. (2020). Language models are few-shot learners. In Advances in Neural Information Processing Systems, 33, 1877-1901.

6.	Hendrycks, D., & Gimpel, K. (2017). A baseline for detecting misclassified and out-of-distribution examples in neural networks. arXiv preprint arXiv:1610.02136.

7.	Esmaeilpour, S., Liu, B., Robertson, E., & Shu, L. (2021). Zero-Shot Out-of-Distribution Detection Based on the Pre-trained Model CLIP. arXiv preprint arXiv:2109.02748. AAAI'22

---

> ### Author Response · Authors · 2025-06-29
> **Rebuttal to Weaknesses (1/2)**
>
> ### **1. The Need for Foundation Model**
>
> Simple models like iForest ad LOF are indeed heavily used in practice. iForest is very easy to train, less sensitive to hyperparameters (like number of trees, tree maximum depth) but limited to axis aligned cuts. LOF can adapt to varying densities on the manifold, but is sensitive to the nearest neighbor count k (its hyperparameter) and can be expensive at inference. (See [1] and the references therein.) The biggest shortcoming for those, as well as other shallow methods, remains: they do not do well on high dimensional datasets.
>
> Importantly, both models compare poorly to FoMo-0D (and other baselines) – see detailed results w.r.t. AUPRC in Tables 17.1 and 17.2: iForest avg rank is 18.798 and is outperformed by FoMo-0D significantly (all p-values are 1.0). LOF avg rank is 12.526 and is also significantly outperformed by FoMo-0D on datasets with dim<=500 (while p-value remains high at 0.72 across All datasets). Conclusions are similar w.r.t AUROC (Tables 16.1-16.2) and w.r.t. F1 (Tables 18.1-18.2).
>
> [1] The Need for Unsupervised Outlier Model Selection: A Review and Evaluation of Internal Evaluation StrategiesMartin Q. Ma, Yue Zhao, Xiaorong Zhang, Leman Akoglu. ACM SigKDD Explorations Newsletter, June 2023.
>
> ### **2. Discussion of the Term "Foundation Model"  in the Context of Tabular Outlier Detection**
>
> For key properties of FoMo-0D that sets it apart from all existing OD detectors, we continue to believe the term “foundation model” is suitable for it. First, it is pretrained on a very large number of OD tasks and is readily transferable to any downstream OD task which makes it a general purpose, “(pre)train once, apply many times” solution.  By lifting the burden of model training on new tasks, and even more importantly, the burden of HP tuning without labels, makes FoMo-0D a paradigm shift in OD, thanks to these foundational properties.
>
> Our title clearly states that it is a foundation model for OD (or similar tasks like OOD Detection, as shown in the Appendix G.3) and we make no further claims that it can address other tabular problems.
> Please note that similar papers in the literature for tabular data classification, e.g. recently published [1][2][3], use the same term while being casted only for tabular classification and time series forecasting, respectively.
>
> [1] N Hollmann te al. “Accurate predictions on small data with a tabular foundation model” Nature (2025)
>
> [2] Qu, Jingang, et al. "TabICL: A Tabular Foundation Model for In-Context Learning on Large Data." ICML (2025)
>
> [3] Rasul, Kashif, et al. "Lag-llama: Towards foundation models for probabilistic time series forecasting." arXiv preprint arXiv:2310.08278 (2023).
>
> ### **3. Explanation of Synthetic-Driven Zero-Shot Generalization**
> We agree with the reviewer that understanding why FoMo-0D succeeds as well as it does on real world tasks despite being trained purely on synthetic datasets deserves further investigation. On the other hand, understanding what transformers have learned mechanistically remains very challenging. Appendix G.2 is our attempt to characterize the goodness of fit that GMMs provide to real datasets in our benchmark, while Appendix G.1 is our attempt at stress-testing its generalization in various OOD scenarios.
>
> As further future work, one could systematically stress-test  FoMo-0D using synthetically generated test datasets that contain known outlier types and inlier distributions distinct from those used during pretraining, while varying the degree of out-of-distribution characteristics in a controlled manner. We also elaborate on these and other future directions in Appendix L (as referenced from Line 442 in main text). We expect that deriving other distinct priors and feeding them into FoMo-0D pretraining could help push its performance even further. As the first pretrained model for OD, our work plants the seeds of the key ideas, showcasing its prowess even based on a relatively simple prior from the GMM family, while paving the way to follow-up work, both theoretical and empirical.

---

> > ### Author Response · Authors · 2025-06-29
> > **Rebuttal to Weaknesses (2/2)**
> >
> > ### **4. Concerns on Varying Tolerance to False Positives/Negatives and Averaging Results**
> > These are good points, thank you for bringing them up.
> >
> > 1) Re: **varying tolerance to false positives/negatives**: That is why we stray from imposing a classification threshold and rather, evaluate ***ranking*** performance by metrics such as AUROC and AUPRC. Those quantify the area under the performance curves across multiple viable thresholds. In other words, when the receiver operating characteristics (ROC) are unknown (such as tolerance to false positives), it is most meaningful to evaluate overall ranking performance.
> >
> > 2) Re: **averaging results across 57 datasets**: Since tasks differ widely in their difficulty, averaging actual performance metrics is not meaningful. That is why we report average ***rank*** of each method across datasets (accounting for ties) (see rank in parentheses in Tables 16.1-18.2). Further, our statistical test Wilcoxon Signed Rank Test also compares methods w.r.t. rank statistics.
> >
> > ### **5. Removing Dashes**
> > We thank the reviewer for giving us the opportunity to make the presentation more succinct. Interestingly, we have not used *any* LLM support in the writing of our original manuscript. As suggested, we eliminated all (but two) — from the revision, by splitting those long sentences.

---

> > > ### Author Response · Authors · 2025-06-29
> > > **Reply to Requested Changes**
> > >
> > > ### **1. Clarification of the term “foundation model”**
> > > See **2. Discussion of the Term "Foundation Model" in the Context of Tabular Outlier Detection** above.
> > >
> > > ### **2. Explanation of zero-shot capabilities**
> > > See **3. Explanation of Synthetic-Driven Zero-Shot Generalization** above. Generalization is at the heart of machine learning and thus, also foundation models. Our paper has primarily focused on highlighting the remarkable prowess of in-context learning post-synthetic pretraining on *real* datasets. As such, our work aims to demonstrate the practical, real-world value of pretrained models for OD. On the other hand, theoretical analyses remain difficult as it is challenging to characterize the distributions that real-world datasets follow and the types of anomalies they contain. We have explored in Appendix G, empirically stress-testing FoMo-0D’s robustness to various OOD settings, as well as the goodness of fit of GMMs to the real-world tasks. Due to page limit, we dedicate a subsection 4.4 in the main text, pointing to those empirical analyses.
> > >
> > >
> > > ### **3. Evaluation strategy across 57 datasets**
> > > See **4. Concerns on Varying Tolerance to False Positives/Negatives and Averaging Results** above.
> > >
> > > ### **4. Improvement of writing style**
> > > See **5. Removing Dashes** above.
> > >
> > > ### **5. Discussion of limitations on synthetic data**
> > > See **3. Explanation of Synthetic-Driven Zero-Shot Generalization** above.
> > >
> > > ### **6. Broader Impact Concerns**
> > >
> > > Our original submission contained at the end, page 12, Impact Statement (following Section 6 Conclusion). We revised the title as Broader Impact Statement. We noticed that we fell short in emphasizing potential shortcomings and thank the reviewer for pointing out critical potential implications, which we included in the revised section.
> > > We note that FoMo-0D provides inlier/outlier probabilities, rather than binary decisions, which can be thresholded based on each application’s risk tolerance to false positives and misses. As pointed out, FoMo-0D does not factor into account metrics beyond detection performance, such as fairness, biases, or other potential blindspots, which should be taken into consideration for sensitive application domains.

---

### Review · Reviewer_w67h · 2025-06-08

**Summary Of Contributions:**

This manuscript propose a novel Zero-Shot Outlier Detection model. The model utilizes the Prior-data Fitted Networks (FPN) to model the posterior distribution $p(y\mid x,D)$ with Neural Network $q_\theta(y\mid x, D)$ for binary classification to separate inliers and outliers.

**Audience:**

Yes

**Claims And Evidence:**

Yes

**Requested Changes:**

I think this manuscript propose an interesting OD approach, it would be better if the author could:
1. Provide model size comparison table in the revision.
2. Provide a rough analysis of why synthesized training data can be effective, as this journal expect the accepted papers to be convincing.

**Strengths And Weaknesses:**

Pros:
1. Model trained entirely with synthesized data that are easy to sample.
2. Forward pass only in test time, no retraining.
3. Scalable to large models with the help of "router mechanism" in previous work
4. Demonstrated good performance in real world datasets, with extensive experiments carried out.
5. Code publicly available.

Cons:
1. Computational cost. Detecting each outlier $x$ with $q_\theta(y\mid x,D)$ that uses the entire dataset $D$ as input could be really expensive, especially for high dimensional data.
2. The reasons that synthesized training data can be effective is not explored.
3. The model size (No. of trainable parameters) should be somewhat compared. Since the proposed model introduces different mechanisms, a comparison of model size should be provided to help analyze the model performance.
4. [minor] Explanation of Fig 3. is a bit unclear, if it is showing rand distribution, some methods will have non-zero density at rank 1 right? Why does the figure showing non?

---

> ### Author Response · Authors · 2025-06-29
> **Rebuttal to Weaknesses (1/2)**
>
> ### **1. Computational Cost of FoMo-0D**
>
> Inference with Fomo-0D is very scalable and time efficient. Let us elaborate:
>
> W.r.t. (high) feature dimensionality (dim): Fomo-0D is pretrained on datasets with up to D features (set to D=100 in experiments). As outlined in Appendix C.2, we have a linear embedding layer of each sample’s feature vector at the input from D to h_dim (set to 256), after which token embedding size is constant. Following Hollmann et. al.’s TabPFN, we zero-pad those datasets with dim<D features, and subsample features for those with dim>D. Note that we do _not_ use feature-feature attention as in TabPFN v2.
>
> → We included this in Lines 214-215 in the main text, and also revised Appendix C.2 for better clarity.
>
> W.r.t. D_train size (n):  In Section 3.3, we describe the router architecture that scales quadratic attention to O(n) through a small number (set to 500) of routers.
>
> → We clarified in Line 243 that this scalability applies to both training and inference.
>
> Overall, this makes FoMo-0D scale linearly w.r.t. both feature and dataset sizes, with 7.7ms wallclock time at inference per test sample. Note that inference is easily parallelizable over test samples. Other tricks such as KV caching can also be used for further speed up, when D_train samples are reused across test samples; e.g. in scenarios where n is less than our maximum context size (5K).
>
> ### **2. Why Synthesized Data Can Be Effective**
>
> As motivated in lines 150-155 in our now (revised) paragraph in the main text, the core reason behind synthetic data for FoMo-0D pretraining is the scarcity of a large number of publicly available real-world OD datasets (in contrast to the massive amounts of text corpora used for pretraining language models). As it is understood by today’s scaling laws for foundation models, their capabilities grow with increasing pretraining data. We are also motivated by earlier such models for tabular classification [Müller et al., 2022; Hollmann et al., 2023], time series forecasting [Dooley et al., 2023]  (as cited in Line 140), as well as Amazon’s Chronos [Ansari et al. (2024)] that either pretrained purely on synthetic data or showed the benefit of augmenting their real time series data with synthesized datasets (as cited in Line 151).
>
> We are also inspired by various examples of synthetic data augmentation to improve reasoning of LLMs, as was done with DeepSeek-R1. Alpha-Go was also trained heavily by self-generated data via self-play.
>
> While why we used synthetic data is well-motivated due to real data scarcity, why we used GMMs in particular is worth further elaboration. As remarked in Lines 170-177, our preliminary GMM prior readily yielded strong results, outperforming numerous SOTA baselines. We conjecture that 1) Gaussian mixtures with multiple components can create nontrivial distributions, and 2) Subspace outliers offer nontrivial outliers to identify hidden in subset of features.
>
> → We added a new footnote #3 on page 5; referencing Appendix G.2 where we perform goodness of fit analysis of our 57 real world datasets with GMMs. As Figure 16 shows FoMo-0D does well on datasets for which GMM provides a good fit, while still generalizing to some others for which the fit is subpar. (Earlier we referenced this section from Experiments on Generalization analysis.)
>
> → Our work achieves strong results even with a simple prior. We leave as future work the exploration of other priors, impact of different priors on performance, as well as prior mixture composition, based on which we revised Lines 176-178.

---

> > ### Author Response · Authors · 2025-06-29
> > **Rebuttal to Weaknesses (2/2)**
> >
> > ### **3. Model Size Comparison**
> > We conduct analysis on how performance scales with model size in Appendix F.10, and provide the results in the table below. We vary the number of transformer layers L=1, 2, 3 (the default is L=4) and report the p-values, where a p-value >0.05 means no statistical evidence to suggest performance difference between FoMo-0D and the compared baseline.
> > We can observe that FoMo-0D is not comparable to the top-10 baselines with one layer (i.e., unable to learn meaningful decision boundaries with limited parameters), and shows improved performance (i.e., p-value increases and becomes larger than 0.05) as the number of layers increases, which suggests that scaling up the model size might help improve performance.
> >
> > On the other hand, we can see that the detection performance didn't increase a lot from L=3 to 4, which suggests there exist diminishing returns in scaling the model size, and to further improve performance, one has to consider other methods (e.g., increase prior complexity, more pre-training epochs) besides scaling up only the model size.
> >
> > | Method | #parameters | DTE-NP | kNN | ICL | DTE-C | LOF | CBLOF | Feat.Bag. | SLAD | DDPM | OCSVM | DTE-NP avg | kNN avg | ICL avg | DTE-C avg |
> > |--------|-------------|--------|-----|-----|--------|-----|--------|-----------|------|------|--------|-------------|---------|----------|-------------|
> > | L=1    | 1.34M       | 1.66 × 10⁻⁹ | 4.30 × 10⁻⁹| 1.25 × 10⁻⁶ | 1.34 × 10⁻⁷ | 5.08 × 10⁻⁶ | 5.44 × 10⁻⁸ | 1.31 × 10⁻⁴ | 1.07 × 10⁻⁶ | 1.30 × 10⁻⁶ | 1.46 × 10⁻⁷ | 2.68 × 10⁻⁹ | 1.53 × 10⁻⁸ | 2.13 × 10⁻⁷ | 0.395 |
> > | L=2    | 2.52M       | 0.006 | 0.036 | 0.157 | 0.259 | 0.442 | 0.557 | 0.703 | 0.431 | 0.759 | 0.805 | 0.021 | 0.134 | 0.333 | 1.000 |
> > | L=3    | 3.70M       | 0.016 | 0.098 | 0.372 | 0.579 | 0.572 | 0.871 | 0.808 | 0.652 | 0.921 | 0.961 | 0.085 | 0.335 | 0.652 | 1.000 |
> > | L=4    | 4.89M       | 0.016 | 0.106 | 0.462 | 0.454 | 0.585 | 0.750 | 0.823 | 0.759 | 0.901 | 0.895 | 0.112 | 0.315 | 0.670 | 1.000 |
> >
> > ### **4. Explanation of Figure 3**
> >
> > We appreciate the opportunity to improve Figure 3. The original figure only showed the 5-95% range. In the now-revised Figure 3 (AUROC score distribution, we provide the rank distribution in  appendix Figure 17, line 1057), the box shows the 25-75%, bars range 5-95%, and circles show the datasets at the tails, while the black vertical bar depicts the mean.

---

> > > ### Author Response · Authors · 2025-06-29
> > > **Reply to Requested Changes**
> > >
> > > ### **1. Provide model size comparison table in the revision**
> > > See **3. Model Size Comparison** above, and in particular Appendix F.10.
> > >
> > > ### **2. Provide a rough analysis of why synthesized training data can be effective**
> > >
> > > See **2. Why Synthesized Data Can Be Effective** above, and in particular Appendix G.2.

---

### Review · Reviewer_uvaG · 2025-06-17

**Summary Of Contributions:**

This paper tackles the challenge of learning a foundation anomaly/outlier detector for tabular data. Because unsupervised outlier detectors strongly depend on their hyperparameter configurations, learning a single model that is able to detect outliers on a variety of different scenarios is hard. This paper solves this problem by using PFN to approximate the posterior bayesian distribution, which is trained on synthetic data (no real-world), and only requires the normal training samples and a test example to compute predictions at test time. Following the benchmark ADBench, the paper compares the proposed approach to 27 OD models on 57 benchmark datasets.

**Audience:**

No

**Claims And Evidence:**

Yes

**Requested Changes:**

The main critical point concerns the text:

1. The confusing unsupervised/semi-supervised settings;

2. The phrasing of the experimental results;

In addition, I would expect to have more detailed and clearer evaluation in the experiments, by removing the image/NLP datasets and including the AUC values.

**Strengths And Weaknesses:**

# Main Strengths:

1. Although foundation models are widely spread nowadays, the field of unsupervised anomaly detection still lacks of pretrained models for zero shot inference; This paper fills the gap;

2. ADBench is well recognized by the community as a thorough repository for the evaluation of unsupervised outlier detection models, as it is comprehensive of a large variety of OD models and benchmarks;

3. The paper is clearly written. Especially, I appreciate the redundancy of stating multiple times that only synthetic datasets are used for training, and only benchmarks are used for testing. Being very specific for the experiments is helpful to understand how to reproduce the results.

4. Related work is sufficiently included, and recent papers are used for comparison in the experiments and in the story of the paper.


# Main Weaknesses

1. Although the related work at the end of the main paper describes that unsupervised and semi-supervised (in the sense of one-class) are two different settings, the paper confuses them and creates ambiguity of which of the two is used. For example, the abstract uses unsupervised, while the proposed method is semi-supervised as it requires normal training samples to be used for inference. Also, it is unclear how the compared models are trained (unsupervised or semi-supervised). This aspect is important and has to be cleared.

2. The difficulty of outlier/anomaly detection, as opposed to traditional binary classification tasks, is that the outlier class might not follow a specific distribution (or, equivalently but more technical, the training and test outlier class might have different distributions). Because of this, anomalies may be categorized in different types and might behave differently according to the specific scenario (see e.g. the Chandola survey on outlier detection). This makes it hard for OD models to be better than all the others over all the benchmarks, as one model is usually great at detecting a specific type of outlier - the one that follows the intuition behind the OD model. For this reason, the foundation model might work because the generated training datasets cover "all types of anomalies" present in the benchmarks, which is definitely not exhaustive for real-world settings.

3. The paper introduces the foundation model for tabular data, but the evaluation is inconsistent: ADBench does not only have tabular data, but also images and NLP datasets where the authors extract features using resnet-18. These do not conform to the definition of tabular, which (please let me know if you think of it differently) includes interpretable features and not embeddings. Otherwise, any type of data with the appropriate feature extractor is tabular.

4. The evaluation metrics used are the AUC, AUPR and F1 score as stated in the experimental section. However, the visuals in the main paper only report the rankings for the AUC (and the p-values), which makes me wonder why the other metrics are even included. Also, rankings and actual AUC values provide different information: the former how often baselines win against each other, the latter by how much they win. I feel that both pieces of information should go into the main paper, to give the correct overview to the readers.

5. The selling point of the experiments is, as far as I understand, that FoMo-0D is "not that bad", rather than being the best or being sufficiently complementary to the others. For example, table 2 claims that it "shows no statistically significant difference from the top 3rd baseline", which means it is statistically worse than the first two. This way to phrase the comparison tends to put too much emphasis on the proposed method's good properties, while ignoring completely the main limitations.

6. Although the proposed method looks interesting and has some value, methods like kNN and iForest have very little training time, are often interpretable, and are not stat sig different from the proposed one. Thus, the question is: why should one adopt FoMo-0D for real-world tasks?

---

> ### Author Response · Authors · 2025-06-29
> **Rebuttal to Weaknesses (1/2)**
>
> ### **1. Unsupervised vs. Semi-supervised**
>
> We see that these different verbiages can be confusing, and thus thank the reviewer for giving us the opportunity to consolidate the naming convention we choose in our paper.
>
> In our opinion, semi-supervised OD is a misnomer and can lead to confusion: the reason is, semi-supervised ML setting typically assumes some labeled data is available from ALL classes. On the other hand, some papers in OD literature refer to the semi-supervised setting as the one where inlier-only training data is available (no labeled outliers); while they refer to the unsupervised setting as the one where (unlabeled) training data is contaminated (i.e. mixed). This creates a mismatch and a source of confusion.
>
> First we clarify this in Section 2.1, where we commit to the phrasing “unsupervised OD with inlier-only training data” versus “unsupervised OD with contaminated training data” to differentiate the two settings. In both settings, there exist no labeled outliers, making model selection difficult. Therefore we believe the term “unsupervised” better represents the original task.
>
> Accordingly, we removed all references to semi-supervised OD throughout text, except in Section 2.1 to clarify the misnomer and mismatch in terminology; clearly specifying our choice of verbiage and the problem setting we consider in our paper.
>
>
> Re "**how the compared models are trained**": All baselines consume the same training data that FoMo-0D is fed with at inference, that is, "clean’’ inlier-only points. For example methods that do parameter training like autoencoders are trained on clean data, while nonparametric methods like kNN use the clean inlier-only data as their support set to find the nearest neighbors of test samples. As such, baselines and FoMo-0D ingest the same data for training. All splits are also the same.
>
>
> ### **2. Paradigm Shift From Model Bias to Data Priors**
>
> We agree with the reviewer on these statements. All models have inductive biases, and OD models are no exception in modeling an implicit definition of what makes a point an outlier. For a foundation model like FoMo-0D to excel well, it should thus be pretrained on diverse datasets reflective of inlier and outlier distributions and types present in the real world.
>
> Importantly, the notable advantage of a paradigm like supervised synthetic pretraining shines here: while typical OD models need to choose their inductive biases at ***modeling*** stage, FoMo-0D defines them at ***data prior*** stage. While it may be difficult to come up with a model that can detect all possible outlier types, it may be more feasible to design priors to simulate datasets with a diverse family of outlier types for the supervised pretraining to learn from.  As such, our work proposes a paradigm shift, shifting the focus from modeling to data synthesis, opening new ways of thinking to tackle the OD problem.
>
> ### **3. Removing NLP, CV Datasets in Evaluation**
>
> In the paper, we highlight the nature of 10 out of 57 datasets in the original ADBench, pointing out their key difference: that they are not “natural” tabular datasets.
> We repeat the significance tests, leaving out image/NLP datasets with embeddings, and we present the p-values of removed NLP, CV datasets in Table 1 (line 37).
>
> We agree with the reviewer that embedded image/text datasets are not truly tabular datasets. In fact, the results are better without these datasets where we have a p-value of 0.164 against the top-1 method DTE-NP,  showing no statistical evidence for a performance difference between FoMo-0D and any baseline. On the other hand, we leave the original results in the main text as is, primarily to follow the original ADBench, and to avoid the impression that we “proctored” the benchmark to make our results look better.
>
> ### **4. Replace Figure 3 with AUROC Score Distribution**
>
> Thanks for the advice. In the revision, we replace Figure 3 (line 352) to AUROC distribution. And due the page limitation, we put the rank plot w.r.t AUROC to appendix Figure 17 (line 1057).

---

> > ### Author Response · Authors · 2025-06-29
> > **Rebuttal to Weaknesses (2/2)**
> >
> > ### **5. Interpretation of FoMo-0D’s experiments**
> >
> > Let us clarify all of our individual results one by one, all of which can be derived from Table 1 (our main results table): (1) FoMo-0D is significantly worse (at 0.05) (p=0.016) than only DTE-NP when All (57) datasets are considered. We took baseline performance values directly from DTE-NP, which reported to be SOTA when it was published.
> > (2) However, DTE-NP reports results with their author-selected hyperparameters. When we randomized the HPs and considered average performances (see ^avg versions of top 4 baselines in last 4 cols of Table 1), p value becomes 0.112.
> > (3) FoMo-0D is pretrained on datasets with d<=100. When we compare it to baselines only on those real datasets with d<=100, none of p-values convey significance.
> > (4) In fact, even up to d<=500, FoMo-0D remains competitive.
> >
> >
> > We believe these results showcase both the strengths and weaknesses of FoMo-0D: (1) and (2) show hyperparameter sensitivity of the baselines. (3) showcases FoMo-0D’s limitation in being pretrained with limited dimensional datasets, suggesting further pretraining could help, while (4) showcases its out-of-distribution performance, suggesting in-context learning can be a powerful asset for generalization.
> >
> > We note that in this work our goal has not been to achieve SOTA results, but rather (1) propose a new paradigm for outlier detection (there is no existing work showing synthetic pretraining and in-context learning can be promising for OD), and (2) instantiate this paradigm with a simple version: GMMs and subspace outliers as data prior in up to 100 dimensions. We expect our work will trigger a shift in OD research, and it also paves the way for future work to build on it, improving it in various aspects, starting with the data prior.
> >
> > ### **6. Why choose FoMo-0D for real-world tasks**
> >
> > We disagree slightly with the reviewer: kNN can be expensive for large (both large n and large d) datasets, may not be meaningful in very high dimensions, has limited interpretability, and is sensitive to k–the choice of which is nontrivial (cf. p-values in Table 1 for kNN vs kNN^avg).
> >
> > As for iForest: FoMo-0D significantly outperforms iForest; see detailed results w.r.t. AUPRC in Tables 17.1 and 17.2: iForest avg rank is 18.798 and is outperformed by FoMo-0D significantly (all p-values are 1.0). Conclusions are similar w.r.t AUROC (Tables 16.1-16.2) and w.r.t. F1 (Tables 18.1-18.2).
> > iForest is indeed fast, but is not effective in high dimensions and FoMo-0D is comparably fast; with a single forward pass it takes mere 7.7ms per test point to be labeled.
> >
> > More importantly, our work paves the path toward more powerful foundation models for OD, that consume pretraining datasets from potentially more complex data priors. Such foundation models will also capitalize with any architectural improvements in scalability and in-context learning prowess. Those make our proposed paradigm a promising direction to shift toward for OD researchers.

---

> > > ### Author Response · Authors · 2025-06-29
> > > **Reply to Requested Changes**
> > >
> > > ### **1. The confusing unsupervised/semi-supervised settings**
> > >
> > > We thank the reviewer for pointing this out. We consolidated the terminology, clarified the setting and differences in Section 2.1. The rest of text is consistent with the established and clarified verbiage and problem setting in 2.1. Please also see **1. Unsupervised vs. Semi-supervised** above.
> > >
> > >
> > > ### **2. The phrasing of the experimental results**
> > >
> > > Please see **5. Interpretation of FoMo-0D’s experiments** above.
> > >
> > > ### **3. More detailed and clearer evaluation in the experiments**
> > >
> > > Please see **3. Removing NLP, CV Datasets in Evaluation** above.

---

### Author Response · Authors · 2025-06-29

To aid with the review and rebuttal process, we 1) added line numbers to the revision and 2) used blue text for changes in our revised submission.

We thank the reviewers for their questions and suggestions for clarity in our revision.

---

### Decision · Action_Editor_VNFR · 2025-08-01

**Recommendation:** Accept with minor revision

**Additional Comments:**

The submission received the comments of three reviewers. After rebuttal, two reviewers recommended "Leaning Accept" while one reviewer recommended "Leaning Reject". Despite the diverse the recommendations from the reviewers, AE has read the manuscript and found the remaining concerns could be reconsidered.

Remaining concerns:
- The explanation of effectiveness of the model provided by the authors are still unconvincing. Intuitive observations could only motivate the approach but can not justify it.
- As stated by the other reviewers and Fig. 3 in the paper, existing models like KNN exhibits comparable performance on the tested dataset. Then why bother using these model instead of the proposed models, when the proposed model requires extra training and test resources for the millions of trainable hyperparameters?
- [minor] I agree with the other reviewer that the phrase "foundation model" may not be the suitable for the proposed approach.

AE shared some points about the remaining concerns:
- The intuition is straightforward and clear, since model selection and many hyperparameters limit the practical use of OD. As compared in Fig. 1, we can find that it aims to avoid the repetitive training and test along with the scenarios, by means of the pretraining.
- About the performance in Fig. 3, we should notice that it avoids the downstream repetition to build a downstream specific OD model, which actually is the pursuit of the generalist foundation model in this era. And for the performance in the whole manuscript, the authors do well claim the performance gain with the statistic significance verification.
- Foundation model is proper when we consider the authors' setup and paradigm in the manuscript. It tries to build a zero-shot OD model for the downstream tasks, which is usually hard but proper as a first exploration.

However, some minors could be improved: 1) Some obvious grammar error could be corrected, e.g., Line 28 incorrect citation command (bracket missing). Line 56 bracket wrapped for 46 and 57 can be removed. There are similar issues in the manuscript. 2) It is not sure that the authors have not emphasized the "tabular data" in the title. For even foundation models, they still have the limited corresponding domains applied to. Will be more exact to emphasize this point in the title?

**Audience:**

Yes

**Audience Explanation:**

Outlier detection is a basic ability for foundation models, known as "know that do not know". It is critical to build this safegurad in current LLMs, which attracts many researchers in the trustworthy machine learning area.

**Claims And Evidence:**

Yes

**Claims Explanation:**

For three reviewers, two of them considered "Yes" to "Claim and Evidence" while one reviewer remains "No". AE has read the manuscript and considered the claims are well supported, such as the performance claim and the points of mechanism design.